

# Geometry of variational methods: dynamics of closed quantum systems

**Lucas Hackl[1,2,3]\*, Tommaso Guaita[2,3]†, Tao Shi[4,5],
Jutho Haegeman[6], Eugene Demler[7] and J. Ignacio Cirac[2,3]**

**1** QMATH, Department of Mathematical Sciences, University of Copenhagen,
Universitetsparken 5, 2100 Copenhagen, Denmark
**2** Max-Planck-Institut für Quantenoptik, Hans-Kopfermann-Str. 1, 85748 Garching, Germany
**3** Munich Center for Quantum Science and Technology,
Schellingstr. 4, 80799 München, Germany
**4** CAS Key Laboratory of Theoretical Physics, Institute of Theoretical Physics,
Chinese Academy of Sciences, Beijing 100190, China
**5** CAS Center for Excellence in Topological Quantum Computation,
University of Chinese Academy of Sciences, Beijing 100049, China
**6** Department of Physics and Astronomy, Ghent University,
Krijgslaan 281, 9000 Gent, Belgium
**7** Lyman Laboratory, Department of Physics, Harvard University,
17 Oxford St., Cambridge, MA 02138, USA

\* lucas.hackl@math.ku.dk, † tommaso.guaita@mpq.mpg.de

## Abstract

We present a systematic geometric framework to study closed quantum systems based on suitably chosen variational families. For the purpose of (A) real time evolution, (B) excitation spectra, (C) spectral functions and (D) imaginary time evolution, we show how the geometric approach highlights the necessity to distinguish between two classes of manifolds: Kähler and non-Kähler. Traditional variational methods typically require the variational family to be a Kähler manifold, where multiplication by the imaginary unit preserves the tangent spaces. This covers the vast majority of cases studied in the literature. However, recently proposed classes of generalized Gaussian states make it necessary to also include the non-Kähler case, which has already been encountered occasionally. We illustrate our approach in detail with a range of concrete examples where the geometric structures of the considered manifolds are particularly relevant. These go from Gaussian states and group theoretic coherent states to generalized Gaussian states.

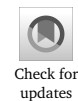
# 1  Introduction

Variational methods are of utmost importance in quantum physics. They have played a crucial role in the discovery and characterization of paradigmatic phenomena in many-body system, like Bose-Einstein condensation [1,2], superconductivity [3], superfluidity [4], the fractional quantum hall [5] or the Kondo effect [6]. They are the basis of Hartree-Fock methods [7], Bardeen-Cooper-Schrieffer theory [3], Gutzwiller [8] or Laughlin wavefunctions [9], the Gross-Pitaevskii equation [1,2], and the density matrix renormalization group [10], which are nowadays part of standard textbooks in quantum physics. Variational methods are particularly well suited to describe complex systems where exact or perturbative techniques cannot be applied. This is typically the case in many-body problems: on the one hand, the exponential growth of the Hilbert space dimension with the system size restricts exact computational methods to relatively small systems; on the other hand, as perturbations are generally extensive, they cannot be considered as small as the system size grows. Furthermore, variational methods are becoming especially relevant in recent times due to the continuous growth of computational power, as this enables to enlarge the number of variational parameters, for instance, to scale polynomially with the system size. Their power and scope can be further extended in combination with other methods, like Monte-Carlo, or even in the context of quantum computing.

A variational method parametrizes a family of states $|\psi(x)\rangle$ or, in case of mixed states, $\rho(x)$, in terms of so-called variational parameters $x = (x^1, \ldots, x^n)$. The choice of the family of states is crucial as it has to encompass the physical behavior we want to describe, as well as to be amenable of efficient computations, circumventing the exponential growth in com-

putational resources that appears in exact computations. A variety of variational principles can then be used, depending on the problem at hand. At thermal equilibrium, one can rely on the fact that the state minimizes the free energy, which reduces to the energy at zero temperature. In that case, for instance, one can just compute the expectation value $E(x)$ of the Hamiltonian in the state $|\psi(x)\rangle$, and find the $x_0$ that minimizes that quantity, yielding a state, $|\psi(x_0)\rangle$ that should resemble the real ground state of the system. For time-dependent problems, one can use Dirac's variational principle. There, one computes the action $S[x(t), \dot{x}(t)]$ for the state $|\psi(x(t))\rangle$ and extracts a set of differential equations for $x(t)$ requiring it to be stationary. Thus, the computational problem is reduced to solving this set of equations, which can usually be done even for very large systems. While the use of time-dependent variational methods is not so widespread as those for thermal equilibrium, the first have experienced a renewed interest thanks to the recent experimental progress in taming and studying the dynamics of many-body quantum systems in diverse setups. They include cold atoms in bulk or in optical lattices [11], trapped ions [12, 13], boson-fermion mixtures [14], quantum impurity problems [15] and pump and probe experiments in condensed matter systems [16–18]. Recently, such methods have been used in the context of matrix product states to analyze a variety of phenomena, or with Gaussian states in the study of impurity problems [19, 20], Holstein models [21], or Rydberg states in cold atomic systems [22, 23].

Time-dependent variational methods can also be formulated in geometric terms. Here, the family of states is seen as a manifold in Hilbert space, and the differential equations for the variational parameters are derived by projecting the infinitesimal change of the state onto the tangent space of the manifold. This approach offers a very intuitive understanding of the variational methods through geometry. The translation between the different formulations is straightforward in the case of complex parametrizations: that is, where the $x^\mu$ are complex variables, in which case the corresponding manifold is, from the geometric point of view, usually referred to as a Kähler manifold. If this is not the case, the geometric formalism has the advantage of highlighting several subtleties that appear and that have to be treated with care.

The main aim of the present paper is two-fold. First, to give a complete formulation of the geometric variational principle in the more general terms, not restricting ourselves to the case of complex parametrizations: that is, when the $x^\mu$ are taken to be real parameters[1]. For all the variational methods that will be introduced, we will provide a detailed analysis of the differences that emerge in the non-complex case, most importantly the existence of inequivalent time dependent variational principles. The motivation to address the case of real parametrization stems from the fact that, in some situations, one has to impose that some of the variational parameters are real since otherwise the variational problem becomes intractable. This occurs, for instance, when one deals with a family of the form $|\psi(x)\rangle = \mathcal{U}(x)|\phi\rangle$, where $\phi$ is some fiducial state and $\mathcal{U}(x)$ is unitary. By taking $x$ complex, $\mathcal{U}(x)$ ceases to be unitary, and thus even the computation of the normalization of the state may require unreasonable computational resources.

Second, even though there exists a vast literature on geometrical methods [24–28], it is mostly addressed to mathematicians and it may be hard to practitioners to extract from it readily applicable methods. Here, we present a comprehensive, but at the same time rigorous illustration of geometric methods that is accessible to readers ranging from mathematical physicists to condensed matter physicists. For this, we first give a simple and compact formulation, and then present the mathematical subtleties together with simple examples and illustrations. We will address some of the issues which are most important when it comes to

---

[1]Note that a complex parametrization (in terms of $z^\mu \in C$) can always be expressed in terms of a real parametrization just by replacing $z^\mu = x_1^\mu + i x_2^\mu$, with $x_{1,2}^\mu \in \mathbb{R}$.

the practical application of these methods: the conservation of physical quantities, the computation of excitations above the ground state, and the evaluation of spectral functions as suggested by the geometrical approach. For each of them, we will provide a motivation and derivation from physical considerations and, where we find inequivalent feasible approaches, give a detailed discussion of the differences and subtleties. Moreover, we discuss how the geometrical method can be naturally extended to imaginary time evolution, providing us with a very practical tool for analyzing systems at zero temperature.

To illustrate our results and to connect to applications, we discuss representative examples of variational classes, for which the presented methods are suitable. In particular, we will recast the prominent families of bosonic and fermionic Gaussian states in a geometric language, which makes their variational properties transparent. We will further show how the geometric structures discussed in this paper emerge in a natural way in the context of Gilmore-Perelomov group theoretic coherent states [29,30], of which traditional coherent and Gaussian states are examples. Finally, we will discuss possible generalizations going beyond *ansätze* of this type.

The paper is structured as follows: In section 2, we motivate our geometric approach and present its key ingredients without requiring any background in differential geometry. In section 3, we give a pedagogical introduction to the differential geometry of Kähler manifolds and fix conventions for the following sections. In section 4, we define our formalism in geometric terms and discuss various subtleties for the most important applications ranging from (A) real time evolution, (B) excitation spectra, (C) spectral functions to (D) imaginary time evolution. In section 5, we apply our formalism to the variational families of Gaussian states, group theoretic coherent states (Gilmore–Perelomov) and certain classes of non-Gaussian states. In section 6, we summarize and discuss our results.

## 2 Variational principle and its geometry

This section serves as a prelude and summary of the more technical sections 3 and 4. In section 3 we will give a rigorous definition of the most generic variational family as a differentiable manifold embedded in projective Hilbert space and define the structures that characterise it as Kähler manifold. In section 4, we illustrate the possible ways in which variational principles can be defined on such manifolds and highlight their differences. A reader familiar with the general approach, but interested in the technical details may skip directly to sections 3 and 4.

In the present section, on the other hand, we illustrate and summarize these results in a physical language that aims to be familiar also for readers who may not be accustomed to the more mathematical formulation of the following sections.

We consider closed quantum systems with Hamiltonian $\hat{H}$ acting on some Hilbert space $\mathcal{H}$. Here, we have a many-body quantum system in mind, where $\mathcal{H}$ is a tensor product of local Hilbert spaces, although we will not use this fact in the general description. We would like to find the evolution of an initial state, $|\psi(0)\rangle$, according to the real-time Schrödinger equation

$$\mathrm{i}\frac{d}{dt}|\psi\rangle = \hat{H}|\psi\rangle \,, \tag{1}$$

and also according to the imaginary-time evolution

$$\frac{d}{d\tau}|\psi\rangle = -(\hat{H} - \langle\hat{H}\rangle)|\psi\rangle \,. \tag{2}$$

The latter will converge to a ground state of $\hat{H}$ as long as $|\psi(0)\rangle$ possesses a non-vanishing overlap with the corresponding ground subspace.

## 2.1 Variational families

We will study variational families of states described as $|\psi(x)\rangle \in \mathcal{H}$, where $x^\mu \in \mathbb{R}^N$ is a set of real parameters. The goal is to approximate the time evolution within this class of states, *i.e.,* to find a set of differential equations for $x^\mu(t)$ so that, provided the variational family accounts well for the physically relevant properties of the states, we can approximate the exact evolution by $|\psi[x(t)]\rangle$. In the case of imaginary-time evolution, the goal will be to find $x_0$ such that $|\psi(x_0)\rangle$ minimizes the energy, *i.e.,* the expectation value of $\hat{H}$, within the variational family.

In principle, one could restrict oneself to variational families that admit a complex parametrization, *i.e.,* where $|\psi(z)\rangle \in \mathcal{H}$ is holomorphic in $z^\mu \in \mathbb{C}^M$ and thus independent of $z^*$. As we will see, this leads to enormous simplifications, as in the geometric language we are dealing with so-called Kähler manifolds, which have very friendly properties. However, in general, we want to use real parametrizations, which cover the complex case (taking the real and imaginary part of $z$ as independent real parameters), but apply to more general situations.

While in certain situations, it is easy to extend or map a real parametrization to a complex one, this is not always the case. This applies, in particular, to parametrizations of the form

$$|\psi(x)\rangle = \mathcal{U}(x)|\phi\rangle \,, \tag{3}$$

where $|\phi\rangle$ is a suitably chosen reference state and $\mathcal{U}(x)$ is a unitary operator that depends on $x^\mu \in \mathbb{R}^N$. Such parametrizations are often used to describe various many-body phenomena appearing in impurity models [19,22,23,31] electron-phonon interactions [21,32] and lattice gauge theory [33], and the fact that $\mathcal{U}(x)$ is unitary is crucial to compute physical properties efficiently. However, extending $x^\mu$ analytically to complexify our parametrizations, will break unitarity of $U$ and often make computations inefficient, thereby limiting the applicability of the variational class. We review such examples in section 5, including bosonic and fermionic Gaussian states and certain non-Gaussian generalizations.

The following example shows an important issue about different possibilities for parametrizations:

**Example 1.** *For a single bosonic degree of freedom (with creation operator $\hat{a}^\dagger$ and annihilation operator $\hat{a}$), we define normalized coherent states as*

$$|\psi(x)\rangle = e^{-|z|^2/2} e^{z\hat{a}^\dagger} |0\rangle \,, \tag{4}$$

*where $x = (\mathrm{Re}\,z, \mathrm{Im}\,z)$. This parametrization is complex but not holomorphic. We can define the extended family*

$$|\psi'(z)\rangle = z_1 e^{z_2 \hat{a}^\dagger} |0\rangle \,, \tag{5}$$

*whose parametrization is holomorphic in $z^\mu \equiv (z_1, z_2)$. The latter parametrization differs from the former as it allows the total phase and normalization of the state to vary freely.*

Given a family with a generic parametrization $|\psi(x)\rangle$ we can always include two other parameters, $(\kappa, \varphi)$, to allow for a variation of normalization and complex phase, so that the new family is

$$|\psi'(x')\rangle = e^{\kappa + \mathrm{i}\varphi} |\psi(x)\rangle \,, \tag{6}$$

where now the total set of variational parameters is $x' = (\kappa, \varphi, x)$. While the global phase does not have a physical meaning on its own, if we want to study the evolution of a superposition of one or several variational states, or quantities like spectral functions, the phase will be relevant and thus should be included in the computation.

This extension of the variational parameteres can always be done at little extra computational cost and the variational principle can be formulated most simply in terms of the extended variables $x'$. For this reason, in the rest of this section (except for subsection 2.2.1 where we add some extra observations on this issue) we will assume that this extension has been done and we will drop the primes, for the sake of an easier notation.

## 2.2 Time-dependent variational principle

One can get Schrödinger's equation (1) from the action

$$S = \int dt \, L \quad \text{with} \quad L = \text{Re} \, \langle \psi | (i\tfrac{d}{dt} - \hat{H}) | \psi \rangle \,, \tag{7}$$

as the Euler-Lagrange equation ensuring stationarity of $S$. This immediately yields a variational principle for the real-time evolution, the so-called Dirac principle. For this, we compute the Euler-Lagrange equations[2]. as

$$\omega_{\mu\nu} \dot{x}^\nu = -\partial_\mu \varepsilon(x) \,, \tag{8}$$

where $\varepsilon(x) = \langle \psi(x) | \hat{H} | \psi(x) \rangle$ is the expectation value of $\hat{H}$ on the unnormalized state $| \psi(x) \rangle$, $\omega_{\mu\nu} = 2 \, \text{Im} \, \langle v_\mu | v_\nu \rangle$ and $| v_\mu \rangle = \partial_\mu | \psi(x) \rangle$. We exploited the antisymmetry of $\omega$, resulting from the antilinearity of the Hermitian inner product. Here and in the following, we use Einstein's convention of summing over repeated indices and we omit to indicate the explicit dependence on $x$ of some quantities, such as $\omega$. Furthermore, we refer to the time derivative $\frac{d}{dt}$ by a dot, and to the partial derivative with respect to $x^\nu$ by $\partial_\nu$.

In cases, where $\omega$ is invertible, *i.e.*, where an $\Omega$ exists, such that $\Omega^{\mu\nu} \omega_{\nu\sigma} = \delta^\mu_{\ \sigma}$, we obtain the equations of motion

$$\dot{x}^\mu = -\Omega^{\mu\nu} \partial_\nu \varepsilon(x) =: \mathcal{X}^\mu(x) \,. \tag{9}$$

If $\omega$ is not invertible, this means that the evolution equations for $x^\mu$ are underdetermined. The reason may be an overparametrization, in which case one can simply drop some of the parameters. However, when we discuss the geometric approach, we will see there can be other reasons related to the fact that the parameters $x^\mu$ are real, in which case one has to proceed in a different way. In particular, it may even occur that (8) becomes ill-defined, so that we need to project out a part of its RHS. We will discuss this in section 4.1.1 and appendix E.

Let us also remark that if we have a complex representation of the state, *i.e.*, $| \psi(z) \rangle$ is holomorphic in $z \in \mathbb{C}^M$, we can get the equations directly for $z$, namely[3]

$$\dot{z}^\mu = -\tilde{\Omega}^{\mu\nu} \frac{\partial \varepsilon(z, z^*)}{\partial z^{*\nu}} \,, \tag{10}$$

---

[2] We find $L(x, \dot{x}) = \dot{x}^\nu \text{Re} \langle \psi(x) | i | v_\nu(x) \rangle - \varepsilon(x)$ with $\varepsilon$ and $| v_\mu \rangle$ defined after (8). The Euler-Lagrange equations $\frac{d}{dt} \frac{\partial L}{\partial \dot{x}^\mu} = \frac{\partial L}{\partial x^\mu}$ follow then directly from $\frac{d}{dt} \frac{\partial L}{\partial \dot{x}^\mu} = \dot{x}^\nu (\text{Re} \langle v_\nu | i | v_\mu \rangle + \text{Re} \langle \psi | i \partial_\nu | v_\mu \rangle)$, $\frac{\partial L}{\partial x^\mu} = \dot{x}^\nu (\text{Re} \langle v_\mu | i | v_\nu \rangle + \text{Re} \langle \psi | i \partial_\mu | v_\nu \rangle) - \partial_\mu \varepsilon$ and the definition of $\omega_{\mu\nu} = 2 \text{Im} \langle v_\mu | v_\nu \rangle = \text{Re} \langle v_\nu | i | v_\mu \rangle - \text{Re} \langle v_\mu | i | v_\nu \rangle$

[3] We have $L = \frac{i}{2} (\dot{z}^\mu \langle \psi | v_\mu \rangle - \dot{z}^{*\mu} \langle v_\mu | \psi \rangle) - \varepsilon(z, z^*)$. The Euler-Lagrange equations $\frac{d}{dt} \frac{\partial L}{\partial \dot{z}^{*\mu}} = \frac{\partial L}{\partial z^{*\mu}}$ are therefore given by the expression $\frac{d}{dt} \frac{\partial L}{\partial \dot{z}^{*\mu}} = -\frac{i}{2} (\dot{z}^\nu \langle v_\mu | v_\nu \rangle + \dot{z}^{*\nu} \langle \partial_\nu^* v_\mu | \psi \rangle)$ and also $\frac{\partial L}{\partial z^{*\mu}} = \frac{i}{2} (\dot{z}^\nu \langle v_\mu | v_\nu \rangle - \dot{z}^{*\nu} \langle \partial_\mu^* v_\nu | \psi \rangle) - \partial_\mu^* \varepsilon$. Notice that if the ket $| \psi \rangle$ is a function of $z$ only, then the corresponding bra $\langle \psi |$ is a function of $z^*$ only.

where $(\tilde{\Omega}^{-1})_{\mu\nu} = -i \langle v_\mu | v_\nu \rangle$ with $|v_\mu\rangle = \frac{d}{dz^\mu} |\psi\rangle$ and $\varepsilon(z, z^*) = \langle \psi(z^*) | \hat{H} | \psi(z) \rangle$. Notice that in this case, $\tilde{\Omega}$ is invertible unless there is some redundancy in the parametrization. This is a consequence of the fact that such variational families are, from the geometric point of view, what is known as a Kähler manifold (see definition Section 3).

In what follows, we will see that many desirable properties are naturally satisfied when dealing with Kähler manifolds, while we also point out various subtleties that arise otherwise.

### 2.2.1 Dynamics of phase and normalization

Let us now briefly consider some more details related to the inclusion of the normalization and phase $(\kappa, \varphi)$ as variational parameters. For this, we will temporarily reintroduce the distinction between $|\psi(x)\rangle$ and $|\psi'(x')\rangle$ as in equation (6). If we consider the Euler-Lagrange equations corresponding specifically to each of the parameters $(\kappa, \varphi, x)$, we have

$$0 = \frac{d}{dt} \left( e^\kappa \langle \psi(x) | \psi(x) \rangle^{1/2} \right), \tag{11}$$

$$\dot{\varphi} = -\frac{\varepsilon(x') + \dot{x}^\mu \operatorname{Im} \langle \psi'(x') | v_\mu \rangle}{e^{2\kappa} \langle \psi(x) | \psi(x) \rangle} \tag{12}$$

and equations for $x$ that do not depend on $\varphi$ and are proportional to $e^{2\kappa}$, so that one can replace the solution of (11) in those equations and solve them independently. If one is interested in the evolution of the phase, then one just has to plug the solutions for $x$ in (12) and integrate that differential equation separately.

It is important to note that using the Lagrangian $L$ from (7) *without* having introduced the extra parameters $(\kappa, \varphi)$ or, more precisely, without ensuring that *both* phase and normalisation can be freely varied, can lead to unexpected results. More specifically, it can produce equations of motion which leave some parameters undetermined or where the unwanted coupling between phases and physical degrees of freedom leads to artificial dynamics.

Nonetheless, one can also equivalently derive the equations for the $x$ directly from a Lagrangian formulation, without introducing the extra parameters $(\kappa, \varphi)$. It is sufficient to use the alternative Lagrangian

$$\mathcal{L}(x, \dot{x}) = \frac{\operatorname{Re} \langle \psi(x) | (i\frac{d}{dt} - \hat{H}) | \psi(x) \rangle}{\langle \psi(x) | \psi(x) \rangle}, \tag{13}$$

which is invariant under $|\psi(x)\rangle \to c(x) |\psi\rangle$ (up to a total derivative) and thus differs from (7). We discuss more in detail how these two definitions are related in Section 4.1.3.

### 2.2.2 Conserved quantities

An important feature of the time-dependent variational principle is that energy expectation value is conserved if the Hamiltonian is time-independent. This can be readily seen because from (11) we know that the states remain normalized during the evolution, so for an initially normalized state, the energy $E$ will always coincide with the function $\varepsilon$, for which we find

$$\frac{d}{dt} \varepsilon(x) = \dot{x}^\mu \partial_\mu \varepsilon = -\Omega^{\mu\nu} (\partial_\mu \varepsilon)(\partial_\nu \varepsilon) = 0, \tag{14}$$

as $\Omega$ is antisymmetric.

However, in general, other observables $\hat{A} = \hat{A}^\dagger$ that commute with the Hamiltonian may not be conserved by the time-dependent variational principle. Indeed, for every variational

family, one can find symmetry generators $\hat{A}$ with $[\hat{A}, \hat{H}] = 0$ which will not be preserved. The question is if those quantities are conserved, that are relevant for describing the physics of the problem at hand. In the special case where we have a complex parametrization, we will show in Section 4.1.2 what further conditions $\hat{A}$ has to satisfy for it to be conserved. More specifically, it must fulfil a compatibility requirement with the chosen variational family. Importantly, we will also discuss how this simple picture is no longer true in the case that no complex parametrization is available.

If the observables of interest happen not to be conserved, it may be wise to consider an alternative variational family, but one can also enforce conservation by hand at the expense of effectively reducing the number of parameters. There are indeed several possibilities to enforce the conservation of observables other than the energy. For instance, one may think of including time-dependent Lagrange multipliers in the Lagrangian action to ensure that property [34]. However, this can only work in a restricted number of cases, as can be already seen if one wants to conserve just a single observable $\hat{A}$. Denoting by $A(x) = \langle \psi(x)|\hat{A}|\psi(x)\rangle$, and adding to the Lagrangian $L$ the term $\lambda(t)A$, it is easy to see that both $\varepsilon[x(t)]$ and $A[x(t)]$ remain constant if

$$\dot{A}(t) = -\Omega^{\mu\nu}(\partial_\mu \varepsilon)(\partial_\nu A) = 0 \tag{15}$$

for all times. The function $\lambda(t)$ can be chosen such that $\ddot{A}(t) = 0$ for all times, namely taking $\lambda(t) = \zeta^\mu \partial_\mu \varepsilon / \zeta^\nu \partial_\nu A$, where $\zeta^\mu = \Omega^{\mu\nu}\partial_\nu \Omega^{\alpha\beta}\partial_\alpha \varepsilon \partial_\beta A$. On top of that, one has to choose an initial state and a parametrization such that at the initial time $\dot{A}(0) = 0$. Furthermore, the denominator in the definition of $\lambda(t)$ must not vanish and since the addition of a Lagrange multiplier modifies the Schrödinger equation, one has to compensate for that. In particular, at the final time $T$, one has to apply the operator $\exp(i \int_0^T \lambda(t) dt \hat{A})$ to the final state, which may be difficult in practice. This severely limits the range of applicability of the Lagrange multiplier method.

Another possibility is to solve $A(x) = A_0$ for one of the variables, e.g. leading to $x^N = f(x^1, \dots, x^{N-1})$, and choose a new reduced variational family with parameters $\tilde{x} = (x^1, \dots, x^{N-1})$ as $|\tilde{\psi}(\tilde{x})\rangle = |\psi(\tilde{x}, f(\tilde{x}))\rangle$. On this reduced family, $A$ will have the constant value $A_0$ by construction. However, this requires to find the function $f$ which may be difficult in practice. In Section 4.1.2 we will discuss how, thanks to the geometric understanding, this condition can be easily enforced locally without having to explicitly solve for $f$. In the same section we will also discuss how to deal with the fact that reducing the variational family by an odd number of real degrees of freedom, as proposed here, will inevitably make $\omega$ degenerate and thus non-invertible.

### 2.2.3 Excitation spectra

An approach often used systematically [35] for computing the energy of elementary excitations is to linearize the equations of motion (9) around the approximate ground state $\psi(x_0)$ to find

$$\delta \dot{x}^\mu = K^\mu{}_\nu \delta x^\nu \quad \text{with} \quad K^\mu{}_\nu = \frac{\partial \mathcal{X}^\mu}{\partial x^\nu}(x_0). \tag{16}$$

The spectrum of $K$ comes in conjugate imaginary pairs $\pm i\omega_\ell$. The underlying idea is that if we slightly perturb the state within the variational manifold and solve the linearized equations of motion, we can approximate the excitation energies as the resulting oscillation frequencies $\omega_\ell$ of the normal mode perturbations around the approximate ground state.

We will see how our geometric perspective provides us with another possibility to compute the excitation spectrum. Both methods have advantages and drawbacks which we carefully explain in 4.2.

### 2.2.4 Spectral functions

In the literature one can find several approaches [36–38] for estimating spectral functions by relying on a variational family. In section 4.3, we argue that the approach that at the same time is most in line with the spirit of variational principles and better adapts to being used with generic ansätze consists in performing linear response theory directly on the variational manifold. Furthermore, this approach leads to a simple closed formula for the spectral function based only on the generator $K^{\mu}{}_{\nu}$ of the linearized equations of motion introduced in (16). Let us decompose $K^{\mu}{}_{\nu}$ in terms of eigenvectors as

$$K^{\mu}{}_{\nu} = \sum_{\ell} \mathrm{i}\omega_{\ell} \mathcal{E}^{\mu}(\mathrm{i}\omega_{\ell}) \widetilde{\mathcal{E}}_{\nu}(\mathrm{i}\omega_{\ell}), \tag{17}$$

where $\widetilde{\mathcal{E}}_{\nu}(\lambda)$ refers to the dual basis of left eigenvectors[4], chosen such that $\mathcal{E}^{\mu}(\lambda)\widetilde{\mathcal{E}}_{\mu}(\lambda') = \delta_{\lambda\lambda'}$. Further, we use the normalization $\mathcal{E}^{\mu}(\mathrm{i}\omega_{\ell})^{*} \omega_{\mu\nu} \mathcal{E}^{\nu}(\mathrm{i}\omega_{\ell}) = \mathrm{i}\,\mathrm{sgn}(\omega_{\ell})$, where we apply complex conjugation component-wise. Then, the spectral function associated to a perturbation $\hat{V}$ is

$$\mathcal{A}_{V}(\omega) = \mathrm{sgn}(\omega) \sum_{\ell} \left| (\partial_{\mu}\langle \hat{V} \rangle) \mathcal{E}^{\mu}(\mathrm{i}\omega_{\ell}) \right|^{2} \delta(\omega - \omega_{\ell}). \tag{19}$$

## 2.3 Geometric approach

Let us now make the connection between the time-dependent variational method reviewed above with a differential geometry description. The basic idea is to consider the states $|\psi(x)\rangle$ as constituting a manifold $\mathcal{M}$ embedded in Hilbert space, and define a tangent space at each point. Then the evolution can be viewed as a projection on that tangent space after each infinitesimal time step. The main issue here is that, if our parametrization is real, the tangent space is not a complex vector space. Therefore, we cannot utilize projection operators in Hilbert space, but rather need to define them on the real tangent spaces. Before entering the general case, let us briefly analyze the one of complex parametrization.

In that case, the left hand side of Schrödinger's equation (1) would yield

$$\mathrm{i}\frac{d}{dt}|\psi(z)\rangle = \mathrm{i}\dot{z}^{\mu}|v_{\mu}\rangle, \tag{20}$$

where $|v_{\mu}\rangle = \partial_{\mu}|\psi(z)\rangle$. Thus, it lies in the tangent space, which is spanned by the $|v_{\mu}\rangle$. The right hand side of the equation, however, does not necessarily do so, as $\hat{H}|\psi(z)\rangle$ will have components outside that span. If we evolve infinitesimally and we want to remain in the manifold, we will have to project the change of $|\psi(z)\rangle$ onto the tangent space. In fact, in this way we get the optimal approximation to the real evolution within our manifold. In practice, this amounts to projecting the right hand side of (1) on that tangent space. This can be achieved by just taking the scalar product on both sides of the equation with $\langle v_{\nu}|$ which leads exactly to (10).

If we do *not* have a complex parametrization, this procedure needs to be modified. In the rest of this section, we will explain how this is done.

---

[4]As explained in section 4.2.2, $K^{\mu}{}_{\nu}$ is diagonalizable and has completely imaginary eigenvalues $\lambda = \pm\mathrm{i}\omega_{\ell}$ with a complete set of right-eigenvectors $\mathcal{E}^{\mu}(\lambda)$ and left-eigenvectors $\widetilde{\mathcal{E}}_{\mu}(\lambda)$ satisfying

$$K^{\mu}{}_{\nu}\mathcal{E}^{\nu}(\lambda) = \lambda\mathcal{E}^{\mu}(\lambda), \quad \widetilde{\mathcal{E}}_{\mu}(\lambda)K^{\mu}{}_{\nu} = \lambda\widetilde{\mathcal{E}}_{\nu}(\lambda). \tag{18}$$

Note that the eigenvectors will be complex with the relations $\mathcal{E}^{\mu}(\lambda^{*}) = \mathcal{E}^{*\mu}(\lambda)$ and $\widetilde{\mathcal{E}}_{\mu}(\lambda^{*}) = \widetilde{\mathcal{E}}_{\mu}^{*}(\lambda)$. We choose the normalizations $\mathcal{E}^{\mu}(\lambda)\widetilde{\mathcal{E}}_{\mu}(\lambda') = \delta_{\lambda\lambda'}$ and $\mathcal{E}^{*\mu}(\mathrm{i}\omega_{\ell})\omega_{\mu\nu}\mathcal{E}^{\nu}(\mathrm{i}\omega_{\ell}) = \mathrm{i}\,\mathrm{sgn}(\omega_{\ell})$.

### 2.3.1 Tangent space and Kähler structures

The tangent space $\mathcal{T}_\psi$ to the manifold $\mathcal{M}$ at its point $|\psi\rangle$ is the space of all possible linear variations on the manifold around $|\psi(x)\rangle$. We can write them as $\dot{x}^\mu \partial_\mu |\psi(x)\rangle$ and thus the tangent space can be defined as the span of the tangent vectors $|v_\mu\rangle = \partial_\mu |\psi(x)\rangle$. However, as our parameters $x$ are taken to be real to maintain generality, this span should only allow real coefficients. The tangent space should therefore be understood as a real linear space embedded in the Hilbert space $\mathcal{H}$. In particular, this implies that for $|v\rangle \in \mathcal{T}_\psi$, the direction $\mathrm{i}|v\rangle$ should be seen as linearly independent of $|v\rangle$ and therefore may itself not belong to the tangent space.

Note that if, on the other hand, $\mathcal{M}$ has a complex holomorphic parametrization then both $|v_\mu\rangle$ and $\mathrm{i}|v_\mu\rangle$ naturally belong to the tangent space as they correspond to $\frac{\partial}{\partial \mathrm{Re} z^\mu}|\psi\rangle$ and $\frac{\partial}{\partial \mathrm{Im} z^\mu}|\psi\rangle$, respectively. In this case, $\mathcal{T}_\psi$ is clearly a complex subspace of the Hilbert space.

From the Hilbert inner product we can derive the two real-valued bilinear forms

$$g_{\mu\nu} = 2\,\mathrm{Re}\langle v_\mu|v_\nu\rangle \quad \text{and} \quad \omega_{\mu\nu} = 2\,\mathrm{Im}\langle v_\mu|v_\nu\rangle \,. \tag{21}$$

We define their inverses respectively as $G^{\mu\nu}$ and $\Omega^{\mu\nu}$. As mentioned before, $\omega$ may not necessarily admit a regular inverse, in which case we can still define a meaningful pseudo-inverse as discussed in section 3.3 and in further detail in appendix E.

Given any Hilbert space vector $|\phi\rangle$, we define its projection on the tangent space $\mathcal{T}_\psi$ as the vector $P_\psi|\phi\rangle \in \mathcal{T}_\psi$ that minimizes the distance from $|\phi\rangle$ in state norm. As we are not dealing with a complex linear space, this will not be given by the standard Hermitian projection operator in Hilbert space. Rather, it takes the form

$$P_\psi|\phi\rangle = 2\,|v_\mu\rangle\, G^{\mu\nu}\mathrm{Re}\langle v_\nu|\phi\rangle \,. \tag{22}$$

Finally, let us introduce $J^\mu{}_\nu = -G^{\mu\sigma}\omega_{\sigma\nu}$, which represents the projection of the imaginary unit, as seen from

$$P_\psi \mathrm{i}|v_\nu\rangle = 2\,|v_\mu\rangle\, G^{\mu\sigma}\mathrm{Re}\,\langle v_\sigma|\mathrm{i}|v_\nu\rangle = J^\mu{}_\nu\,|v_\mu\rangle \,, \tag{23}$$

where we used (22) and (21). As highlighted previously, $\mathrm{i}|v_\nu\rangle$ may not lie in the tangent space, in which case the projection is non-trivial and we have $J^2 \neq -\mathbb{1}$ in contrast to $\mathrm{i}^2 = -1$. We will explain that $J$ satisfying $J^2 = -\mathbb{1}$ on every tangent space is equivalent to having a manifold that admits a complex holomorphic parametrization, in such case we will speak of a Kähler manifold. If, on the other hand, it is somewhere not satisfied, we speak of a non-Kähler manifold and in this case there exist tangent vectors $|v_\mu\rangle$, for which $\mathrm{i}|v_\mu\rangle$ will not belong to the tangent space. Moreover, the projection $P_\psi$ will not commute with multiplication by the imaginary unit.

**Example 2.** *Following example 1, normalized coherent states have tangent vectors*

$$\begin{aligned}
|v_1\rangle &= \frac{\partial}{\partial \mathrm{Re} z}|\psi(x)\rangle = (\hat{a}^\dagger - \mathrm{Re} z)|\Psi(x)\rangle \,, \\
|v_2\rangle &= \frac{\partial}{\partial \mathrm{Im} z}|\psi(x)\rangle = (\mathrm{i}\hat{a}^\dagger - \mathrm{Im} z)|\Psi(x)\rangle \,.
\end{aligned} \tag{24}$$

*For $z \neq 0$, $\mathrm{i}|v_\mu\rangle$ will not be a tangent vector, i.e., $\mathrm{i}|v_\mu\rangle \notin \mathrm{span}_\mathbb{R}(|v_1\rangle, |v_2\rangle)$. This changes if we allowed for a variation of phase and normalization (from the complex holomorphic parametrization), such that we had the additional basis vectors $|v_3\rangle = |\Psi(x)\rangle$ and $|v_4\rangle = \mathrm{i}|\Psi(x)\rangle$.*

As emphasized at the beginning of section 2, here we use a simplified notation, where the variational family $M$ is a subset of Hilbert space $\mathcal{H}$ with complex phase $\varphi$ and normalization $\kappa$ as free parameters. In the more technical treatments of sections 3 and 4.1.1, we will avoid this by defining variational families $\mathcal{M}$ as subsets of projective Hilbert space $\mathcal{P}(\mathcal{H})$, where we project out those tangent directions that correspond to changing phase or normalization of the state. To avoid confusion between these different definitions we use the symbols $\omega$, $g$, $J$, $P_\psi$ and $|v_\mu\rangle$ to indicate the quantities introduced here (including phase and normalization), while later we will use $\boldsymbol{\omega}$, $\boldsymbol{g}$, $\boldsymbol{J}$, $\mathbb{P}_\psi$ and $|V_\mu\rangle$ (with phase and normalization being removed).

### 2.3.2 Real time evolution

We already mentioned how the time dependent variational principle is equivalent to projecting infinitesimal time evolution steps onto the tangent space. In the general case of non-Kähler manifolds, there exist two inequivalent projections of Schrödinger's equation given by

$$P_\psi(\mathrm{i}\tfrac{d}{dt} - \hat{H})|\psi\rangle = 0 \quad \text{or} \quad P_\psi(\tfrac{d}{dt} + \mathrm{i}\hat{H})|\psi\rangle = 0 , \tag{25}$$

which are obviously equivalent on a complex vector space, as the two forms only differ by a factor of i. However, the defining property of a non-Kähler manifold is precisely that its tangent space is not a complex, but merely a real vector space and multiplication by i will not commute with the projection $P_\psi$.

In section 4 we will show that the first projection Schrödinger's equation in (25) is equivalent to the formulation in terms of a Lagrangian $L$ introduced earlier. It consequently leads to the equations of motion (9). The second choice of (25), often referred to as the *McLachlan variational principle,* corresponds to minimizing the local error $\|\tfrac{d}{dt}|\psi\rangle - (-\mathrm{i}\hat{H})|\psi\rangle\|$ made at every step of the approximation of the evolution and leads to the equations

$$\dot{x}^\mu = 2G^{\mu\nu}\mathrm{Im}\langle v_\nu|\hat{H}|\psi\rangle . \tag{26}$$

In section 4, we will argue that in most cases the Lagrangian action principle presents the more desirable properties. In particular, it leads to simple equations of motion that only depend on the gradient $\partial_\mu \varepsilon$ and whose dynamics necessarily preserve the energy itself. However, for some aspects, the McLachlan evolution still has some advantages, such as the conservation of observables that commute with the Hamiltonian and are compatible with the variational family, in the sense defined in Section 4.1.2. We will further explain, how one can construct a restricted evolution that maintains the desirable properties of both projections in (25) for non-Kähler families, but at the expense of locally reducing the number of free parameters.

Finally, our geometric formalism provides a simple notation to understand and describe the methods reviewed so far.

## 2.4 Imaginary time evolution

So far, our discussion was purely focused on real-time dynamics. In the context of excitations and spectral functions, we referred to an approximate ground state $|\psi_0\rangle$ in our variational family, that minimizes the energy function $E(x)$. While there are many numerical methods to finding minima, our geometric perspective leads to a natural approach based on approximating imaginary time evolution, which we defined in (2) for the full Hilbert space. We would like to approximate this evolution as it is known to converge to a true ground state of the Hamiltonian, provided one starts from a state with a non-vanishing overlap with such ground state. However, as this evolution does not derive from an action principle, one cannot naively

generalise for it Dirac's time dependent variational principle. On the other hand, the tangent space projection can be straightforwardly applied to equation (2), leading, as we prove in Section 4.4, to the time evolution

$$\frac{dx^\mu}{d\tau} = -G^{\mu\nu}\partial_\nu E(x)\,, \tag{27}$$

where $E(x) = \langle\psi(x)|\hat{H}|\psi(x)\rangle / \langle\psi(x)|\psi(x)\rangle$ is the energy expectation value function.

The evolution defined in this way always decreases the energy $E$ of the state, as can be seen from [39]

$$\frac{d}{d\tau}E = \frac{dx^\mu}{d\tau}\partial_\mu E = -G^{\mu\nu}\partial_\nu E\partial_\mu E < 0\,, \tag{28}$$

where we used that $G$ is positive definite.

Indeed, the dynamics defined by (27) can be simply recognised as a gradient descent on the manifold with respect to the energy function and the natural notion of distance given by the metric $g$. Consequently, this evolution will converge to a (possibly only local) minimum of the energy. In conclusion, we recognize imaginary time evolution projected onto the variational manifold as a natural method to find the approximate ground the state $|\psi_0\rangle = |\psi(x_0)\rangle$.

# 3 Geometry of variational families

In this section, we review the mathematical structures of variational families, assuming them to be defined by real parameters, which leads to a description that is more general than the complex case. First, we explain how a complex Hilbert space can be described as real vector space equipped with so called Kähler structures. Second, we describe the manifold of *all* pure quantum states as projective Hilbert space, which is a real differentiable manifold whose tangent spaces can be embedded as *complex* subspaces in Hilbert space and thereby inherit Kähler structures themselves. Third, we introduce general variational families as *real* submanifolds, whose tangent spaces may lose the Kähler property. Fourth, we study this potential violation and possible cures.

Starting with the present section, we define variational families $\mathcal{M}$ as sub manifolds of projective Hilbert space $\mathcal{P}(\mathcal{H})$, *i.e.*, we describe pure states $\psi$ rather than state vectors $|\psi\rangle$ as already foreshadowed after example 2.

## 3.1 Hilbert space as Kähler space

Given a separable Hilbert space $\mathcal{H}$ with inner product $\langle\cdot|\cdot\rangle$, we can always describe vectors by a set of complex number $\psi_n$ with respect to a basis $\{|n\rangle\}$, *i.e.*,

$$|\psi\rangle = \sum_n \psi_n |n\rangle\,. \tag{29}$$

We will see that the tangent space of a general variational manifold is a real subspace of Hilbert space. Given a set of vectors $\{|n\rangle\}$, we distinguish the real and complex span

$$\begin{aligned}
\text{span}_{\mathbb{C}}\{|n\rangle\} &= \left\{\textstyle\sum_n \psi_n |n\rangle \,\middle|\, \psi_n \in \mathbb{C}\right\}, \\
\text{span}_{\mathbb{R}}\{|n\rangle\} &= \left\{\textstyle\sum_n \psi_n |n\rangle \,\middle|\, \psi_n \in \mathbb{R}\right\}.
\end{aligned} \tag{30}$$

On a real vector space, $|\psi\rangle \neq 0$ and $i|\psi\rangle$ are linearly independent vectors, because one cannot be expressed as linear combination with real coefficients of the other. A real basis $\{|V_\mu\rangle\}$ of $\mathcal{H}$ has therefore twice as many elements as the complex basis $\{|n\rangle\}$, such as

$$\{|V_\mu\rangle\} \equiv \{|1\rangle, i|1\rangle, |2\rangle, i|2\rangle, \dots\} . \tag{31}$$

Given any real basis $\{|V_\mu\rangle\}$ of vectors, we can express every vector $|X\rangle$ as real linear combination

$$|X\rangle = X^\mu |V_\mu\rangle , \tag{32}$$

where we use Einstein's summation convention[5].

A general real linear map $\hat{A} : \mathcal{H} \to \mathcal{H}$ will satisfy $\hat{A}(\alpha|X\rangle) = \alpha\hat{A}|X\rangle$ only for real $\alpha$. If it also holds for complex $\alpha$, we refer to $\hat{A}$ as complex-linear. The imaginary unit $i$ becomes itself a linear map, which only commutes with complex-linear maps.

The Hermitian inner product $\langle \cdot | \cdot \rangle$ can be decomposed into its real and imaginary parts given by

$$\langle V_\mu | V_\nu \rangle = \frac{\mathcal{N}}{2}\big(g_{\mu\nu} + i\,\omega_{\mu\nu}\big), \tag{33}$$

with $g_{\mu\nu} = \frac{2}{\mathcal{N}}\operatorname{Re}\langle V_\mu | V_\nu \rangle$, $\omega_{\mu\nu} = \frac{2}{\mathcal{N}}\operatorname{Im}\langle V_\mu | V_\nu \rangle$ and $\mathcal{N}$ being a normalization which we will fix in (51). This gives rise to the following set of structures, illustrated in figure 1.

**Definition 1.** *A real vector space is called Kähler space if it is equipped with the following two bilinear forms*

- ***Metric**[6] $g_{\mu\nu}$ being symmetric and positive-definite with inverse $G^{\mu\nu}$, so that $G^{\mu\sigma}g_{\sigma\mu} = \delta^\mu{}_\nu$,*

- ***Symplectic form** $\omega_{\mu\nu}$ being antisymmetric and non-degenerate[7] with inverse $\Omega^{\mu\nu}$, so that $\Omega^{\mu\sigma}\omega_{\sigma\nu} = \delta^\mu{}_\nu$,*

*and such that the linear map $J^\mu{}_\nu := -G^{\mu\sigma}\omega_{\sigma\nu}$ is a*

- ***Complex structure** $J^\mu{}_\nu$ satisfying $J^2 = -\mathbb{1}$.*

*The last condition is also called compatibility between $g$ and $\omega$. We refer to $(g, \omega, J)$ as **Kähler structures**.*

Clearly, $g$ is a metric and $\omega$ is a symplectic form. Furthermore, we will see that they are indeed compatible and define a complex structure $J$. For this, it is useful to introduce the real dual vectors $\operatorname{Re}\langle X|$ and $\operatorname{Im}\langle X|$ that act on a vector $|Y\rangle$ via

$$\operatorname{Re}\langle X|Y\rangle = \frac{\mathcal{N}}{2}X^\mu g_{\mu\nu}Y^\nu, \ \operatorname{Im}\langle X|Y\rangle = \frac{\mathcal{N}}{2}X^\mu \omega_{\mu\nu}Y^\nu, \tag{34}$$

as one may expect. The identity $\mathbb{1} = \sum_n |n\rangle\langle n|$ is then

$$\mathbb{1} = \frac{2}{\mathcal{N}}G^{\mu\nu}|V_\mu\rangle\operatorname{Re}\langle V_\nu| . \tag{35}$$

---

[5]We will be careful to only write equations with indices that are truly independent of the choice of basis, such that the symbol $X^\mu$ may very well stand for the vector $|X\rangle$ itself. This notation is known as abstract index notation (see appendix A.2).

[6]Here, "metric" refers to a metric tensor, *i.e.*, an inner product on a vector space. It should not be confused with the notion of metric spaces in analysis and topology.

[7]A bilinear form $b_{\mu\nu}$ is called non-degenerate, if it is invertible. For this, we can check $\det(b) \neq 0$ in any basis of our choice.

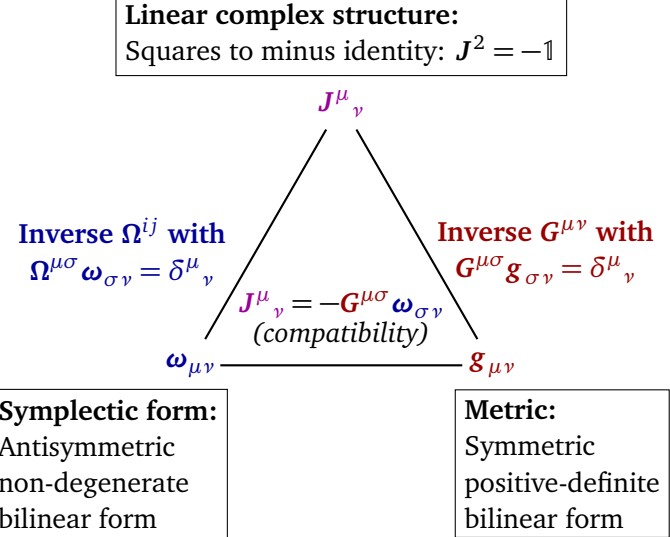

Figure 1: *Triangle of Kähler structures.* This sketch illustrates the triangle of Kähler structures, consisting of a symplectic form $\omega$, a positive definite metric $g$ and a linear complex structure $J$. We also define the inverse symplectic form $\Omega$ and the inverse metric $G$.

Similarly, the matrix representation of an operator $\hat{A}$ is

$$A^{\mu}{}_{\nu} = \tfrac{2}{\mathcal{N}} G^{\mu\sigma} \operatorname{Re}\langle V_{\sigma}|\hat{A}|V_{\nu}\rangle \,. \tag{36}$$

In particular, we compute the matrix representation of the imaginary unit i to be given by

$$J^{\mu}{}_{\nu} = \tfrac{2}{\mathcal{N}} G^{\mu\sigma} \operatorname{Re}\langle V_{\sigma}|\mathrm{i}|V_{\nu}\rangle = -G^{\mu\sigma}\omega_{\sigma\nu} \tag{37}$$

as anticipated in our definition. From $\mathrm{i}^2 = -1$, we conclude that the so defined $J$ indeed satisfies $J^2 = -\mathbb{1}$ and is thus a complex structures. Therefore, $g$ and $\omega$ as defined in (33) are compatible.

**Example 3.** *A qubit is described by the Hilbert space $\mathcal{H} = \mathbb{C}^2$ with complex basis $\{|0\rangle, |1\rangle\}$ and real basis*

$$|V_i\rangle \equiv \{|0\rangle, |1\rangle, \mathrm{i}\,|0\rangle, \mathrm{i}\,|1\rangle\}\,. \tag{38}$$

*With respect to this real basis $g_{\mu\nu}$, $\omega_{\mu\nu}$ and $J^{\mu}{}_{\nu}$ are*

$$g_{\mu\nu} \equiv \tfrac{2}{\mathcal{N}}\begin{pmatrix} \mathbb{1} & 0 \\ 0 & \mathbb{1} \end{pmatrix}, \ \omega_{\mu\nu} \equiv \tfrac{2}{\mathcal{N}}\begin{pmatrix} 0 & \mathbb{1} \\ -\mathbb{1} & 0 \end{pmatrix}, \ J^{\mu}{}_{\nu} \equiv \begin{pmatrix} 0 & -\mathbb{1} \\ \mathbb{1} & 0 \end{pmatrix}, \tag{39}$$

*where $\mathbb{1}$ is the $2 \times 2$ identity matrix. We can represent a complex-linear map $\hat{A} = \sum_{n,m} a_{nm}\,|n\rangle\,\langle m|$, i.e., with $[A, J] = 0$, as the matrix*

$$A^{\mu}{}_{\nu} \equiv \begin{pmatrix} \mathbb{A} & -\mathbb{B} \\ \mathbb{B} & \mathbb{A} \end{pmatrix}, \tag{40}$$

*where $\mathbb{A} = \operatorname{Re}(a)$ and $\mathbb{B} = \operatorname{Im}(a)$ in above basis.*

In summary, every Hilbert space is a real Kähler space with metric, symplectic form and complex structure.

## 3.2 Projective Hilbert space

Multiplying a Hilbert space vector $|\psi\rangle$ with a non-zero complex number does not change the quantum state it represents. Therefore, the manifold representing all physical states is given by the projective Hilbert space $\mathcal{P}(\mathcal{H})$, which we will define and analyze in this section. Variational families, which we will discuss in the following section, should then naturally be understood as submanifolds $\mathcal{M}$ of projective Hilbert space $\mathcal{P}(\mathcal{H})$.

The projective Hilbert space of $\mathcal{H}$

$$\mathcal{P}(\mathcal{H}) = (\mathcal{H} \setminus \{0\}) / \sim \tag{41}$$

is given by the equivalence classes of non-zero Hilbert space vectors with respect to the equivalence relation

$$|\psi\rangle \sim |\tilde{\psi}\rangle \quad \Longleftrightarrow \quad \exists c \in \mathbb{C} \quad \text{with} \quad |\tilde{\psi}\rangle = c\,|\psi\rangle\,. \tag{42}$$

Thus, a state $\psi \in \mathcal{P}(\mathcal{H})$ is a ray in Hilbert space consisting of all non-zero vectors that are related by multiplication with a non-zero complex number $c$.

The tangent space $\mathcal{T}_\psi \mathcal{P}(\mathcal{H})$ represents the space of changes $\delta\psi$ around an element $\psi \subset \mathcal{P}(\mathcal{H})$. Changing a representative $|\psi\rangle$ in the direction of itself, *i.e.*, $|\delta\psi\rangle \propto |\psi\rangle$, corresponds to changing $|\psi\rangle$ by a complex factor and thus does not change the underlying state $\psi$. Two Hilbert space vectors $|X\rangle, |\tilde{X}\rangle \in \mathcal{H}$ therefore represent the same change $|\delta\psi\rangle$ of the state $|\psi\rangle \in \psi$, if they only differ by some $\alpha\,|\psi\rangle$. We define tangent space as

$$\mathcal{T}_\psi \mathcal{P}(\mathcal{H}) = \mathcal{H}/\approx\,, \tag{43}$$

where we introduced the equivalence relation

$$|X\rangle \approx |\tilde{X}\rangle \quad \Longleftrightarrow \quad \exists c \in \mathbb{C} \quad \text{with} \quad |X\rangle - |\tilde{X}\rangle = c\,|\psi\rangle\,, \tag{44}$$

leading to a regular (not projective) vector space.

We can pick a unique representative $|X\rangle$ of the class $[|\delta\psi\rangle]$ at the state $|\psi\rangle$ by requiring $\langle\psi|X\rangle = 0$. Viceversa, two vectors $|X\rangle \neq |\tilde{X}\rangle$ both satisfying $\langle\psi|X\rangle = \langle\psi|\tilde{X}\rangle = 0$ belong to different equivalence classes. We thus identify $\mathcal{T}_\psi \mathcal{P}(\mathcal{H})$ with

$$\mathcal{H}_\psi^\perp = \left\{ |X\rangle \in \mathcal{H} \,\middle|\, \langle\psi|X\rangle = 0 \right\}\,. \tag{45}$$

Given a general representative $|\delta\psi\rangle \in [|\delta\psi\rangle]$, we compute the unique representative mentioned above as $|X\rangle = \mathbb{Q}_\psi\,|\delta\psi\rangle$ with

$$\mathbb{Q}_\psi\,|\delta\psi\rangle = |\delta\psi\rangle - \frac{\langle\psi|\delta\psi\rangle}{\langle\psi|\psi\rangle}\,|\psi\rangle\,. \tag{46}$$

There is a further subtlety: representing a change $\delta\psi$ of a state $\psi$ as vector $|\delta\psi\rangle$ will always be with respect to a representative $|\psi\rangle$. If we choose a different representative $|\tilde{\psi}\rangle = c\,|\psi\rangle \in \psi$, the same change $\delta\psi$ would be represented by a different Hilbert space vector $|\delta\tilde{\psi}\rangle = c\,|\delta\psi\rangle$. It therefore does not suffice to specify a Hilbert space vector $|\delta\psi\rangle$, but we always need to say with respect to which representative $|\psi\rangle$ it was chosen. This could be avoided when moving to density operators[8].

---

[8]We can equivalently define projective Hilbert space as the set of pure density operators, *i.e.*, Hermitian, positive operators $\rho$ with $\text{Tr}\rho = \text{Tr}\rho^2 = 1$. The state $\psi$ is then given by the density operator $\rho_\psi = \frac{|\psi\rangle\langle\psi|}{\langle\psi|\psi\rangle}$ and its change $\delta\psi$ by $\delta\rho_\psi = \frac{|\delta\psi\rangle\langle\psi| + |\psi\rangle\langle\delta\psi|}{\langle\psi|\psi\rangle}$.

The fact we can identify the tangent space at each point with a Hilbert space $\mathcal{H}_\psi^\perp$ enables us, given a *local* real basis $\{|V_\mu\rangle\}$ at $\psi$, such that $\mathcal{H}_\psi^\perp = \text{span}_\mathbb{R}\{|V_\mu\rangle\}$, to induce a canonical metric $g_{\mu\nu}$, symplectic form $\omega_{\mu\nu}$ and $J^\mu{}_\nu$ onto the tangent space, which thus is a Kähler space, as discussed previously. We see at this point that on the tangent space $\mathcal{T}_\psi\mathcal{P}(\mathcal{H})$, it is convenient to choose $\mathcal{N} = \langle\psi|\psi\rangle$ as normalization for the Kähler structures. The rescaled metric $\frac{1}{2}g_{\mu\nu}$ is well-known as the Fubini-Study metric [40, 41], while the symplectic form gives projective Hilbert space a natural phase space structure.

Manifolds such as $\mathcal{P}(\mathcal{H})$, whose tangent spaces are equipped with differentiable Kähler structures, are called almost-Hermitian manifolds. In appendix C, we show that $\mathcal{P}(\mathcal{H})$ satisfies even stronger conditions, which make it a so called Kähler manifold.

**Example 4.** *The projective Hilbert space of a Qubit is $\mathcal{P}(\mathbb{C}^2) = S^2$, equivalent to the Bloch sphere. Using spherical coordinates $x^\mu \equiv (\theta, \phi)$ and the complex Hilbert space basis $\{|0\rangle, |1\rangle\}$, we can parametrize the set of states as*

$$|\psi(x)\rangle = \cos\left(\tfrac{\theta}{2}\right)|0\rangle + e^{i\phi}\sin\left(\tfrac{\theta}{2}\right)|1\rangle. \tag{47}$$

*The elements of $\mathcal{P}(\mathcal{H})$ are the equivalence classes $\psi(x) = \left\{c\,|\psi(x)\rangle \,\middle|\, c \in \mathbb{C}, c \neq 0\right\}$. Consequently, the tangent space $\mathcal{T}_\psi\mathcal{P}(\mathbb{C}^2) = \mathcal{H}_\psi^\perp$ of the Bloch sphere at $x^\mu \equiv (\theta, \phi)$ can be spanned by the basis $|V_\mu\rangle = \mathbb{Q}_\psi\left(\frac{\partial}{\partial x^\mu}\right)|\psi(x)\rangle$ with*

$$
\begin{aligned}
|V_1\rangle &= -\tfrac{1}{2}\sin\left(\tfrac{\theta}{2}\right)|0\rangle + \tfrac{e^{i\phi}}{2}\cos\left(\tfrac{\theta}{2}\right)|1\rangle, \\
|V_2\rangle &= -\tfrac{i}{2}\sin\left(\tfrac{\theta}{2}\right)\sin\theta\,|0\rangle + \tfrac{ie^{i\phi}}{2}\cos\left(\tfrac{\theta}{2}\right)\sin\theta\,|1\rangle.
\end{aligned}
\tag{48}
$$

*Using the definition* (33) *of the metric and symplectic form from the Hilbert space inner product, we can compute the matrix representations*

$$g_{\mu\nu} \equiv 2\begin{pmatrix} 1 & 0 \\ 0 & \sin^2\theta \end{pmatrix} \text{ and } \omega_{\mu\nu} \equiv 2\begin{pmatrix} 0 & \sin\theta \\ -\sin\theta & 0 \end{pmatrix}. \tag{49}$$

*We recognize $g_{\mu\nu}dx^\mu dx^\nu = \frac{1}{2}(d\theta^2 + \sin^2(\theta)d\phi^2)$ to be the standard metric of a sphere with radius $1/\sqrt{2}$. Similarly, we recognize $\omega_{\mu\nu}dx^\mu dx^\nu = \frac{1}{2}\sin\theta\,d\theta \wedge d\phi$ to be the standard volume form on this sphere. Finally, it is easy to verify that $J^2 = -\mathbb{1}$ everywhere.*

In summary, a given pure state can be represented by the equivalence class $\psi \in \mathcal{P}(\mathcal{H})$ of all states related by multiplication with a non-zero complex number. Similarly, a tangent vector $[|X\rangle] \in \mathcal{T}_\psi\mathcal{P}(\mathcal{H})$ at a state $\psi$ is initially defined as the affine space $[|X\rangle]$ of all vectors $|X\rangle$ differing by a complex multiple of $|\psi\rangle$. A unique representative $|\tilde{X}\rangle$ can be chosen requiring $\langle\psi|\tilde{X}\rangle = 0$. This leads to the identification $\mathcal{T}_\psi\mathcal{P}(\mathcal{H}) \simeq \mathcal{H}_\psi^\perp$, such that the Hilbert space inner product $\langle\cdot|\cdot\rangle$ induces local Kähler structures onto $\mathcal{T}_\psi\mathcal{P}(\mathcal{H})$.

### 3.3 General variational manifold

The most general variational family is a real differentiable submanifold $\mathcal{M} \subset \mathcal{P}(\mathcal{H})$. At every point $\psi \in \mathcal{M}$, we have the tangent space $\mathcal{T}_\psi\mathcal{M}$ of tangent vectors $|X\rangle_\psi$. $\mathcal{T}_\psi\mathcal{M}$ can be embedded into Hilbert space by defining the local frame $|V_\mu\rangle_\psi \in \mathcal{H}$, such that $|X\rangle = X^\mu|V_\mu\rangle$, as before. Note, however, that in general the tangent space $\mathcal{T}_\psi\mathcal{M} = \text{span}_\mathbb{R}\{|V_\mu\rangle\}$ is only a *real*, but not necessarily a complex subspace of $\mathcal{H}$. Thus, we will encounter families, for which $|X\rangle$ is a tangent vector, but not $i|X\rangle$.

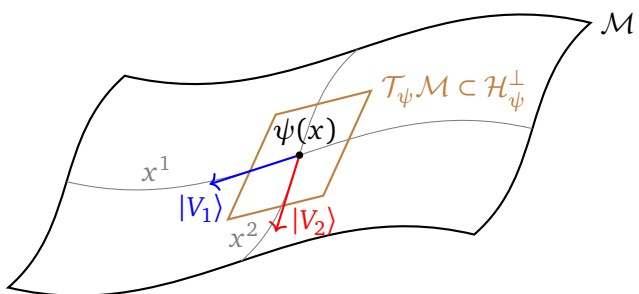

Figure 2: *Tangent vectors.* We sketch the basis vectors $|V_\mu\rangle$ of tangent space $\mathcal{T}_\psi \mathcal{M}$ for a manifold $\mathcal{M}$ parametrized by two coordinates $(x^1, x^2)$.

In practice, we often parametrize $\psi(x) \in \mathcal{M}$ by choosing a representative $|\psi(x)\rangle \in \mathcal{H}$. This allows us to construct the local basis $|V_\mu(x)\rangle$ of tangent space $\mathcal{T}_\psi \mathcal{M}$

$$|V_\mu(x)\rangle = \mathbb{Q}_{\psi(x)} \, \partial_\mu |\psi(x)\rangle \, , \tag{50}$$

at the state $|\psi(x)\rangle$, where $\mathbb{Q}_\psi$ was defined[9] in (46). To simplify notation, we will usually drop the reference to $\psi(x)$ or $x$ and only write $|V_\mu\rangle$, whenever it is clear at which state we are. The schematic idea behind tangent space is sketched in figure 2.

Similar to projective Hilbert space, we define *restricted* Kähler structures on tangent space $\mathcal{T}_\psi \mathcal{M} \subset \mathcal{T}_\psi \mathcal{P}(\mathcal{H})$ as

$$\boldsymbol{g}_{\mu\nu} = \frac{2 \, \mathrm{Re}\langle V_\mu | V_\nu \rangle}{\langle \psi | \psi \rangle} \quad \text{and} \quad \boldsymbol{\omega}_{\mu\nu} = \frac{2 \, \mathrm{Im}\langle V_\mu | V_\nu \rangle}{\langle \psi | \psi \rangle} \, . \tag{51}$$

There are two important differences to the corresponding definition (33) in full Hilbert space. First, with a slight abuse of notation, $|V_\mu\rangle$ here does not span the Hilbert space, but rather the typically much smaller tangent space. Second, we set $\mathcal{N} = \langle \psi | \psi \rangle$ just like for $\mathcal{P}(\mathcal{H})$, such that

$$\langle V_\mu | V_\nu \rangle = \frac{\langle \psi | \psi \rangle}{2} \left( \boldsymbol{g}_{\mu\nu} + \mathrm{i}\boldsymbol{\omega}_{\mu\nu} \right) . \tag{52}$$

This has the important consequence that the restricted Kähler structures are invariant under the change of representative $|\psi\rangle$ of the physical state. Namely, under the transformation $|\psi\rangle \to c \, |\tilde{\psi}\rangle$ with $|V_\mu\rangle \to c \, |V_\mu\rangle$, our Kähler structures will not change. This ensures that equations involving restricted Kähler structures are manifestly defined on projective Hilbert space and thus independent of the representative $|\psi(x)\rangle \in \mathcal{H}$, we use to represent the abstract state $\psi(x) \in \mathcal{M} \subset \mathcal{P}(\mathcal{H})$.

We have $\mathcal{T}_\psi \mathcal{M} \subset \mathcal{H}$ and for $|X\rangle, |Y\rangle \in \mathcal{H}$, we have the real inner product $\mathrm{Re}\langle X | Y \rangle$ on $\mathcal{H}$ inducing the norm $\||X\rangle\| = \sqrt{\mathrm{Re}\langle X | X \rangle} = \sqrt{\langle X | X \rangle}$, which is nothing more than the regular Hilbert space norm. We then define the orthogonal projector $\mathbb{P}_\psi$ from $\mathcal{H}$ onto $\mathcal{T}_\psi \mathcal{M}$ with respect to $\mathrm{Re}\langle \cdot | \cdot \rangle$, *i.e.*, for each vector $|X\rangle \in \mathcal{H}$ we find the vector $\mathbb{P}_\psi |X\rangle \in \mathcal{T}_\psi \mathcal{M}$ minimizing the norm $\||X\rangle - \mathbb{P}_\psi |X\rangle\|$.

---

[9] The projector $\mathbb{Q}_\psi$ is important to ensure that $|V_\mu\rangle$ can be identified with an element of $\mathcal{H}_\psi^\perp \simeq \mathcal{T}_\psi \mathcal{P}(\mathcal{H})$ as discussed in Section 3.2, *i.e.*, $\langle \psi | V_\mu \rangle = 0$. For derivations, it can be useful to go into a local coordinate system of $x^\mu$, in which $|V_\mu\rangle = \partial_\mu |\psi\rangle$, *i.e.*, the action of $\mathbb{Q}_\psi$ can be ignored. This can always be achieved locally at a point and any invariant expressions derived this way, will be valid in any coordinate system.

Table 1: *Comparison: Kähler vs. Non-Kähler.* We review the properties of restricted Kähler structures in each case. See appendix C for a review of the conditions for a general manifold to be Kähler.

| | Kähler | Non-Kähler | |
| --- | --- | --- | --- |
| | | (non-degenerate) | (degenerate) |
| **Restricted metric $g$** <br> inverse $G$ ($Gg = \mathbb{1}$) | symmetric, positive definite, invertible | symmetric, positive definite, invertible | |
| **Restricted symplectic form $\omega$** <br> inverse $\Omega$ ($\Omega\omega = \mathbb{1}$) or pseudo-inverse $\Omega$ | antisymmetric, closed ($d\omega = 0$), non-degenerate | antisymmetric, may not be closed | |
| | | non-degenerate | degenerate |
| **Restricted complex structure $J$** <br> inverse or pseudo-inverse $J^{-1} = -\Omega g$ | $J^2 = -\mathbb{1}$, invertible with $J^{-1} = -J$ | $0 \geq J^2 \geq -\mathbb{1}$, | |
| | | invertible | pseudo-invertible |

We can write this orthogonal projector in two ways:

$$\mathbb{P}_\psi = \frac{2\,|V_\mu\rangle\,\boldsymbol{G}^{\mu\nu}\mathrm{Re}\langle V_\nu|}{\langle\psi|\psi\rangle}\,, \quad \mathbb{P}_\psi^\mu = \frac{2\boldsymbol{G}^{\mu\nu}\mathrm{Re}\langle V_\nu|}{\langle\psi|\psi\rangle}\,, \tag{53}$$

such that we have $\mathbb{P}_\psi = |V_\mu\rangle\,\mathbb{P}_\psi^\mu$. The difference lies in the co-domain: While $\mathbb{P}_\psi : \mathcal{H} \to \mathcal{H}$ maps back onto Hilbert space, *e.g.*, to compute $\mathbb{P}_\psi^2 = \mathbb{P}_\psi$, we have that $\mathbb{P}_\psi^\mu : \mathcal{H} \to \mathcal{T}_\psi\mathcal{M}$ is a map from Hilbert space into tangent space. Due to $\mathcal{T}_\psi\mathcal{M} \subset \mathcal{T}_\psi\mathcal{P}(\mathcal{H})$, we have

$$\mathbb{P}_\psi = \mathbb{P}_\psi\mathbb{Q}_\psi = \mathbb{Q}_\psi\mathbb{P}_\psi \quad \text{and} \quad \mathbb{P}_\psi^\mu = \mathbb{P}_\psi^\mu\mathbb{Q}_\psi\,, \tag{54}$$

which follows from $\mathbb{Q}_\psi\,|V_\mu\rangle = |V_\mu\rangle$ and $\mathbb{Q}_\psi^\dagger = \mathbb{Q}_\psi$. In contrast to $\mathbb{Q}_\psi$, the projector $\mathbb{P}_\psi$ is in general not Hermitian.

Provided that there are no redundancies or gauge directions (only changing phase or normalization) in our choice of parameters, $\boldsymbol{g}_{\mu\nu}$ will still be positive-definite and invertible with inverse $\boldsymbol{G}^{\mu\nu}$. We find that

$$\boldsymbol{J}^\mu{}_\nu = -\boldsymbol{G}^{\mu\sigma}\boldsymbol{\omega}_{\sigma\nu} = \frac{2\boldsymbol{G}^{\mu\sigma}\mathrm{Re}\langle V_\sigma|\mathrm{i}|V_\nu\rangle}{\langle\psi|\psi\rangle} = \mathbb{P}_\psi^\mu\mathrm{i}\,|V_\nu\rangle \tag{55}$$

is the projection of the multiplication by the imaginary unit (as real-linear map) onto $\mathcal{T}_\psi\mathcal{M}$. It will not square to minus identity if multiplication by i in full Hilbert space does not preserve the tangent space.

If $\boldsymbol{g}_{\mu\nu}$ is not invertible, it means that there exists a set of coefficients $X^\mu$ such that $X^\mu\boldsymbol{g}_{\mu\nu}X^\nu = 0$, that is $\|X^\mu\,|V_\mu\rangle\| = 0$ and therefore $X^\mu\,|V_\mu\rangle = 0$. In other words, not all vectors $|V_\mu\rangle$ are linearly independent and thus also not all parameters are independent. If this is the case, it is not a real problem as the formalism introduced can still be used with little modifications. More precisely, the projectors (53), as well as all other objects we will introduce, are meaningfully defined if we indicate with $\boldsymbol{G}^{\mu\nu}$ the Moore-Penrose pseudo-inverse of $\boldsymbol{g}_{\mu\nu}$, *i.e.*, we invert $\boldsymbol{g}_{\mu\nu}$ only on the orthogonal complement to its kernel (orthogonal with respect

of the flat metric $\delta_{\mu\nu}$ in our coordinates[10]). Indeed, all directions in the kernel correspond to a vanishing vector in the tangent space and therefore do not matter. In this case, also $\Omega^{\mu\nu}$, should be defined as the inverse of $\omega_{\mu\nu}$ on the orthogonal complement to the kernel of $g_{\mu\nu}$.[11] However, it is still possible that $\omega$ and $J$ are not invertible even on this reduced subspace.

In this case, in order to define $\Omega$ one has to reduce oneself to working on an even smaller subspace, that is one that does not contain the kernel of $\omega$ and $J$. Here, however, the way in which we reduce these extra dimensions is not equivalent, as these directions are not anymore just redundant gauge choices of our parametrization. The reduction here effectively corresponds to working on a physically smaller manifold, as we will explain better in the next section. For what follows we will always suppose that $\Omega$ is defined by inverting $\omega$ on the tangent subspace orthogonal, with respect to the metric $g_{\mu\nu}$, to the kernel of $J$. That is, $\Omega$ is the Moore-Penrose pseudo-inverse of $\omega$ with respect to $g$, *i.e.*, the pseudo-inverse is evaluated in an orthonormal basis. In appendix E, we elaborate further on the definition and evaluation of this pseudo-inverse.

In conclusion, we see that we are able to define the *restricted* structures $(g, \omega, J)$ which, however, do not necessarily satisfy the Kähler property. This is due to the fact that the tangent space, as we have pointed out, is a real, but not necessarily complex subspace of $\mathcal{H}$. Note that these objects are locally defined in each tangent space $\mathcal{T}_\psi \mathcal{M}$ for $\psi \in \mathcal{M}$.

**Example 5.** *For the Hilbert space $\mathcal{H} = (\mathbb{C}^2)^{\otimes 2}$ of two Qubits, we can choose the variational manifold $\mathcal{M}$ of symmetric product states represented by*

$$|\varphi(x)\rangle = |\psi(x)\rangle \otimes |\psi(x)\rangle, \tag{56}$$

*with $x^\mu \equiv (\theta, \phi)$, where $|\psi(x)\rangle$ is a single Qubit state as parametrized in (47). The tangent space is spanned by*

$$|W_\mu\rangle = |V_\mu\rangle \otimes |\psi(x)\rangle + |\psi(x)\rangle \otimes |V_\mu\rangle, \tag{57}$$

*where $|V_\mu\rangle$ are the single Qubit tangent vectors defined in (48). With this, we find*

$$g_{\mu\nu} \equiv \begin{pmatrix} 1 & 0 \\ 0 & \sin^2(\theta) \end{pmatrix} \quad and \quad \omega_{\mu\nu} \equiv \begin{pmatrix} 0 & \sin\theta \\ -\sin\theta & 0 \end{pmatrix} \tag{58}$$

*leading to $J^2 = -\mathbb{1}$ everywhere. We therefore conclude that the tangent space $\mathcal{T}_\psi \mathcal{M}$ satisfies the Kähler property everywhere.*

**Example 6.** *For the single Qubit Hilbert space $\mathcal{H} = \mathbb{C}^2$, we can choose the equator of the Bloch sphere as our variational manifold $\mathcal{M}$. This amounts to fixing $\theta = \pi/2$ in the single Qubit state (47) leading to the representatives*

$$|\psi(\phi)\rangle = \frac{1}{\sqrt{2}}|0\rangle + \frac{e^{i\phi}}{\sqrt{2}}|1\rangle, \tag{59}$$

*with a single variational parameter $\phi$. We have the single tangent vector $|V\rangle = |V_1\rangle$ as defined in (48). From the inner product $\langle V|V\rangle = \frac{1}{4}$, we find $g = \frac{1}{2}$ and $\omega = 0$ implying $J = 0$. Consequently and not surprising due to the odd dimension, the tangent spaces of our variational manifold $\mathcal{M}$ are not Kähler spaces. Moreover, neither $\omega$ nor $J$ are invertible.*

---

[10]In the specific case of the manifold of matrix product states, there exists a different, more natural definition of orthogonality [42].

[11]Note that the kernel of $\omega_{\mu\nu}$ itself does not necessarily correspond to redundant directions of the parametrization as $X^\mu \omega_{\mu\nu} = 0$ does not imply $X^\mu |V_\mu\rangle = 0$.

**Example 7.** *We consider a bosonic system with two degrees of freedom associated with annihilation operators $\hat{a}_1$ and $\hat{a}_2$. The vacuum state $|0,0\rangle$ satisfies $\hat{a}_m|0,0\rangle = 0$, $\hat{a}_1^\dagger|0,0\rangle = |1,0\rangle$ and $\hat{a}_2^\dagger|0,0\rangle = |0,1\rangle$. We introduce*

$$\hat{b} = \cosh r\,\hat{a}_1 + \sinh r\,\hat{a}_2^\dagger, \tag{60}$$

*with canonical commutation relations $[\hat{b},\hat{b}^\dagger] = 1$ and $r$ being a fixed constant (not a variational parameter). We then define the states of our variational manifold as*

$$|\psi(\alpha)\rangle = e^{\alpha\hat{b}^\dagger - \alpha^*\hat{b}}|0\rangle\,, \tag{61}$$

*parametrized by a single complex number $\alpha$. $|\psi(\alpha)\rangle$ is not the one-mode coherent state $|\alpha\rangle = e^{\alpha\hat{a}^\dagger - \alpha^*\hat{a}}|0\rangle$, because $\hat{b}|0\rangle \neq 0$. Our variational manifold has two independent real parameters given by $x^\mu \equiv (\mathrm{Re}\,\alpha, \mathrm{Im}\,\alpha)$. After some algebra taking $[\hat{b},\hat{b}^\dagger] = 1$ into account, we find*

$$
\begin{aligned}
|V_1\rangle &= e^{\alpha\hat{b}^\dagger - \alpha^*\hat{b}}\left(\cosh r\,|1,0\rangle - \sinh r\,|0,1\rangle\right),\\
|V_2\rangle &= e^{\alpha\hat{b}^\dagger - \alpha^*\hat{b}}\,\mathrm{i}\left(\cosh r\,|1,0\rangle + \sinh r\,|0,1\rangle\right).
\end{aligned}
\tag{62}
$$

*Metric and symplectic form take the forms*

$$g_{\mu\nu} \equiv \cosh 2r\begin{pmatrix} 2 & 0 \\ 0 & 2 \end{pmatrix} \quad and \quad \omega_{\mu\nu} \equiv \begin{pmatrix} 0 & 2 \\ -2 & 0 \end{pmatrix}. \tag{63}$$

*This gives rise to the restricted complex structure*

$$J^\mu{}_\nu \equiv \mathrm{sech}\,2r\begin{pmatrix} 0 & -1 \\ 1 & 0 \end{pmatrix}, \tag{64}$$

*which only satisfies $J^2 = -\mathbb{1}$ for $r = 0$.*

In summary, we introduced general variational manifolds as real differentiable submanifolds $\mathcal{M}$ of projective Hilbert space $\mathcal{P}(\mathcal{H})$. By embedding the tangent spaces $\mathcal{T}_\psi\mathcal{M}$ into Hilbert space, the Hilbert space inner product defines restricted Kähler structures on the tangent spaces, whose properties we will explore next.

## 3.4 Kähler and non-Kähler manifolds

We categorize variational manifolds depending on whether their tangent spaces are Kähler spaces or not. We will see in the following sections that this distinction has some important consequences for the application of variational methods on the given family.

**Definition 2.** *We classify general variational families $\mathcal{M} \subset \mathcal{P}(\mathcal{H})$ based on their restricted Kähler structures. We refer to a variational family $\mathcal{M}$ as*

- ***Kähler**[12], if all tangent spaces $\mathcal{T}_\psi\mathcal{M}$ are a Kähler spaces, i.e., $J^2 = -\mathbb{1}$ everywhere on the manifold,*

---

[12]A general manifold $\mathcal{M}$, whose tangent spaces are equipped with compatible Kähler structures, is known as an almost Hermitian manifold. However, if an almost Hermitian manifold is the submanifold of a Kähler manifold (as defined in appendix C), then it is also a Kähler manifold itself. Thus, due to the fact that $\mathcal{P}(\mathcal{H})$ is a Kähler manifold, all almost Hermitian submanifolds $\mathcal{M} \subset \mathcal{P}(\mathcal{H})$ are also Kähler manifolds, which is why we use the term Kähler in this context.

- **Non-Kähler**, *if it is not Kähler. If $\boldsymbol{\omega}$ is degenerate, we define $\boldsymbol{\Omega}$ as the pseudo-inverse.*

*This classification refers to the manifold as a whole. In table 1 we summarize the properties of each class of manifolds.*

Many well-known variational families, such as Gaussian states [43], coherent states [29, 30, 44], matrix product states [45] and projected entangled pair states [46], are Kähler. On the other hand, one naturally encounters non-Kähler manifolds when one parametrizes states through a family of general unitaries $\mathcal{U}(x)$ applied to a reference state $|\phi\rangle$, *i.e.*,

$$|\psi(x)\rangle = \mathcal{U}(x)|\phi\rangle . \tag{65}$$

For example, this issue arises for the classes of generalized Gaussian states introduced in [32], for the Multi-scale Entanglement Renormalisation Ansatz states [47] or if one applies Gaussian transformations $\mathcal{U}(x)$ to general non-Gaussian states.

**Example 8.** *We already reviewed examples for these three cases in the previous section. More precisely, example 5 is Kähler, example 6 is non-Kähler with degenerate $\boldsymbol{\omega}$ and example 7 is non-Kähler with non-degenerate $\boldsymbol{\omega}$.*

Given a submanifold $\mathcal{M} \subset \mathcal{P}(\mathcal{H})$, we can use the embedding in the manifold $\mathcal{P}(\mathcal{H})$ to constrain the form that the restricted complex structure $J$ can take on $\mathcal{M}$.

**Proposition 1.** *On a tangent space $\mathcal{T}_\psi \mathcal{M} \subset \mathcal{H}$ of a submanifold $\mathcal{M} \subset \mathcal{P}(\mathcal{H})$ we can always find an orthonormal basis $\{|V_\mu\rangle\}$, such that $\boldsymbol{g}_{\mu\nu} \equiv \mathbb{1}$ and the restricted complex structure is represented by the block matrix*

$$
\boldsymbol{J}^\mu{}_\nu \equiv
\begin{pmatrix}
\begin{smallmatrix} & 1 & \\ -1 & & \\ & & \ddots \end{smallmatrix} & & \\
\hline
 & \begin{smallmatrix} & c_1 & & \\ -c_1 & & & \\ & & & c_2 \\ & & -c_2 & \\ & & & & \ddots \end{smallmatrix} & \\
\hline
 & & \begin{smallmatrix} 0 & \\ & \ddots \end{smallmatrix}
\end{pmatrix}
\tag{66}
$$

*with $0 < c_i < 1$. This standard form induces the decomposition of $\mathcal{T}_\psi \mathcal{M}$ into the three orthogonal parts*

$$\mathcal{T}_\psi \mathcal{M} = \underbrace{\overline{\overline{\mathcal{T}_\psi \mathcal{M}}} \oplus \mathcal{I}_\psi \mathcal{M}}_{\overline{\mathcal{T}_\psi \mathcal{M}}} \oplus \mathcal{D}_\psi \mathcal{M} , \tag{67}$$

*where $\overline{\overline{\mathcal{T}_\psi \mathcal{M}}}$ is the largest Kähler subspace and $\overline{\mathcal{T}_\psi \mathcal{M}}$ is the largest space on which $J$ and $\omega$ are invertible.*

*Proof.* We present a constrictive proof in appendix B. □

Proposition 1 is also relevant for classifying real subspaces of complex Hilbert spaces. Interestingly, it is linked to the entanglement structure of fermionic Gaussian states, as made explicit in [48].

The manifold $\mathcal{M}$ is Kähler if there is only the first block in (66) everywhere. The symplectic form $\boldsymbol{\omega}$ is non-degenerate if we only have the first two diagonal blocks. The next proposition provides some further intuition for the non-Kähler case, which is also known in mathematics in the context of sub manifolds of Kähler manifolds [49].

**Proposition 2.** *The Kähler property is equivalent to requiring that $\mathcal{T}_\psi \mathcal{M}$ is not just a real, but also a complex subspace, i.e., for all $|X\rangle \in \mathcal{T}_\psi \mathcal{M}$, we also have $i|X\rangle \in \mathcal{T}_\psi \mathcal{M}$. Therefore, the multiplication by $i$ commutes with the projector $\mathbb{P}_\psi$, i.e., $\mathbb{P}_\psi i = i \mathbb{P}_\psi$ and $\mathbb{P}_\psi$ is complex-linear.*

*Proof.* We present a proof in appendix B. $\qquad\square$

If a manifold admits a complex holomorphic parametrization, *i.e.*, a parametrization that depends on the complex parameters $z^\mu$, but not on $z^{*\mu}$, then the manifold will be Kähler. Indeed, taking $\mathrm{Re}z^\mu$ and $\mathrm{Im}z^\mu$ as real parameters gives the tangent vectors

$$|v_\mu\rangle = \frac{\partial}{\partial \mathrm{Re}z^\mu} |\psi(z)\rangle, \quad i|v_\mu\rangle = \frac{\partial}{\partial \mathrm{Im}z^\mu} |\psi(z)\rangle . \tag{68}$$

It is actually also possible to show that, viceversa, a Kähler manifold is also a complex manifold, that is it admits, at least locally, a complex holomorphic parametrization.

As mentioned before, in order to define the inverse of $\boldsymbol{\omega}$ it is necessary to restrict ourselves to work only in a subspace of $\mathcal{T}_\psi \mathcal{M}$. We now see that the definition we gave previously of always defining $\boldsymbol{\Omega}$ as the pseudo-inverse with respect to $\boldsymbol{g}$ coincides with always choosing to consider only the tangent directions in

$$\overline{\mathcal{T}_\psi \mathcal{M}} = \mathrm{span}_{\mathbb{R}}\{|\overline{V}_i\rangle\} . \tag{69}$$

In order to apply variational methods as explained in the following sections, it may be necessary to at least locally restore the Kähler property. We can achieve this by locally further restricting ourselves to

$$\overline{\overline{\mathcal{T}_\psi \mathcal{M}}} = \mathrm{span}_{\mathbb{R}}\{|\overline{\overline{V}}_i\rangle\} . \tag{70}$$

Using the bases $\{|\overline{\overline{V}}_\mu\rangle\}$ and $\{|\overline{V}_\mu\rangle\}$, we can define the restricted Kähler structures $(\overline{\overline{\boldsymbol{g}}}, \overline{\overline{\boldsymbol{\omega}}}, \overline{\overline{\boldsymbol{J}}})$, which are compatible, and $(\overline{\boldsymbol{g}}, \overline{\boldsymbol{\omega}}, \overline{\boldsymbol{J}})$, where $\overline{\boldsymbol{\omega}}$ and $\overline{\boldsymbol{J}}$ are non-degenerate.

Our assumption on the definition of $\boldsymbol{\Omega}$ can be understood as taking $\boldsymbol{\Omega}$ to be zero on the subspace $\mathcal{D}_\psi \mathcal{M}$, where $\boldsymbol{\omega}$ is not invertible, and equal to the inverse of $\overline{\boldsymbol{\omega}}$ on $\overline{\mathcal{T}_\psi \mathcal{M}}$. Note that this definition is only possible because the tangent space is also equipped with a metric $\boldsymbol{g}$, which makes the orthogonal decomposition $\mathcal{T}_\psi \mathcal{M} = \overline{\mathcal{T}_\psi \mathcal{M}} \oplus \mathcal{D}_\psi \mathcal{M}$ well-defined.

In summary, a general variational family $\mathcal{M} \subset \mathcal{P}(\mathcal{H})$ is not necessarily a Kähler manifold. We can check locally, if the restricted Kähler structures fail to satisfy the Kähler condition. If this happens, we can always choose local subspaces

$$\overline{\overline{\mathcal{T}_\psi \mathcal{M}}} \subset \overline{\mathcal{T}_\psi \mathcal{M}} \subset \mathcal{T}_\psi \mathcal{M} \tag{71}$$

on which the restricted Kähler structures satisfy the Kähler properties or are at least invertible. Defining $\boldsymbol{\Omega}$ as the pseudo-inverse with respect to $\boldsymbol{g}$ is equivalent to inverting $\boldsymbol{\omega}$ only on $\overline{\mathcal{T}_\psi \mathcal{M}}$. In what follows, we therefore do not need to distinguish between the non-Kähler cases with degenerate or non-degenerate structures, as we will always be able to apply the same variational techniques based on $\boldsymbol{\Omega}$.

### 3.5 Observables and Poisson bracket

Any Hermitian operator $\hat{A}$ defines a real-valued function $\langle\hat{A}\rangle$ on the manifold $\mathcal{M}$ and in fact on the whole projective Hilbert space. The function is given by the expectation value

$$A(x) = \langle\hat{A}\rangle(x) = \frac{\langle\psi(x)|\hat{A}|\psi(x)\rangle}{\langle\psi(x)|\psi(x)\rangle}. \tag{72}$$

It is invariant under rescalings of $|\psi\rangle$ by complex factors and is thus a well-defined map on projective Hilbert space $\mathcal{P}(\mathcal{H})$. We will use the notation $\langle\hat{A}\rangle$ and $A(x)$ interchangeably. For the function deriving from the Hamiltonian operator $\hat{H}$, we use the symbol $E = \langle\hat{H}\rangle$ and call it the *energy*.

Given a Hermitian operator $\hat{A}$ and the representative $|\psi(x)\rangle$, we have the important relation

$$\mathbb{P}^{\mu}_{\psi}\hat{A}|\psi\rangle = G^{\mu\nu}(\partial_{\nu}A), \tag{73}$$

which is invariant under the change of representative $|\psi\rangle \to c\,|\psi\rangle$ and $|V_{\mu}\rangle \to c\,|V_{\mu}\rangle$. It follows from

$$\partial_{\mu}A = \frac{2\operatorname{Re}\langle V_{\mu}|\hat{A}|\psi\rangle}{\langle\psi|\psi\rangle} = g_{\mu\nu}\mathbb{P}^{\nu}_{\psi}\hat{A}|\psi\rangle, \tag{74}$$

where we used product rule and (50).

The following definition will play an important role in the context of Poisson brackets and conserved quantities. Every operator $\hat{A}$ defines a vector field given by $\mathbb{Q}_{\psi}\hat{A}|\psi\rangle$. If this vector field is tangent to $\mathcal{M}$ for all $\psi \in \mathcal{M}$, the following definition applies.

**Definition 3.** *Given a general operator $\hat{A}$ and a variational family $\mathcal{M} \subset \mathcal{P}(\mathcal{H})$, we say $\hat{A}$ preserves $\mathcal{M}$ if*

$$\mathbb{Q}_{\psi}\hat{A}|\psi\rangle = (\hat{A} - \langle\hat{A}\rangle)|\psi\rangle \quad \textit{for all} \quad \psi \in \mathcal{M} \tag{75}$$

*lies in the tangent space $\mathcal{T}_{\psi}\mathcal{M}$, i.e., $\mathbb{Q}_{\psi}\hat{A}|\psi\rangle = \mathbb{P}_{\psi}\hat{A}|\psi\rangle$.*

The symplectic structure of the manifold naturally induces a Poisson bracket on the space of differentiable functions, which is given by

$$\{A, B\} := (\partial_{\mu}A)\,\Omega^{\mu\nu}(\partial_{\nu}B). \tag{76}$$

In some special cases this can be related to the commutator of the related operators.

**Proposition 3.** *Given two Hermitian operators $\hat{A}$ and $\hat{B}$ of which one preserves the Kähler manifold $\mathcal{M}$, i.e.,*

$$(\hat{A} - \langle\hat{A}\rangle)|\psi\rangle \in \mathcal{T}_{\psi}\mathcal{M} \textit{ or } (\hat{B} - \langle\hat{B}\rangle)|\psi\rangle \in \mathcal{T}_{\psi}\mathcal{M}, \tag{77}$$

*the Poisson bracket is related to the commutator via*

$$\{A, B\} = \mathrm{i}\frac{\langle\psi|[\hat{A}, \hat{B}]|\psi\rangle}{\langle\psi|\psi\rangle}. \tag{78}$$

Table 2: *Action principles.* We review the different action principles and how they relate to the respective manifolds.

| | Lagrangian | McLachlan | Dirac-Frenkel |
|---|---|---|---|
| **Definition** | $\mathbb{P}_\psi(\mathrm{i}\frac{d}{dt} - \hat{H})\lvert\psi\rangle = 0$ | $\mathbb{P}_\psi(\frac{d}{dt} + \mathrm{i}\hat{H})\lvert\psi\rangle = 0$ | both |
| **Kähler manifold** | always defined and all equivalent | | |
| **Non-Kähler manifold** | defined for chosen inverse $\Omega$ (see proposition 4) | always defined (see proposition 5) | not defined |
| **Advantage** | energy conservation (see proposition 4) | conservation of symmetries (see proposition 7) | both |
| **Linearization around ground state** | possible (see section 4.2.2) | not possible (see section 4.2.2) | possible |

*Proof.* We compute

$$\mathrm{i}\frac{\langle\psi\lvert[\hat{A},\hat{B}]\rvert\psi\rangle}{\langle\psi\rvert\psi\rangle} = \frac{2\,\mathrm{Re}\langle\psi\lvert(\hat{A}-\langle\hat{A}\rangle)\mathrm{i}(\hat{B}-\langle\hat{B}\rangle)\rvert\psi\rangle}{\langle\psi\rvert\psi\rangle}\,. \tag{79}$$

As one of the vectors $(\hat{A}-\langle\hat{A}\rangle)\lvert\psi\rangle$ or $(\hat{B}-\langle\hat{B}\rangle)\lvert\psi\rangle$ lies in the tangent space $\mathcal{T}_\psi\mathcal{M}$, (34) applies, giving

$$\mathrm{i}\frac{\langle\psi\lvert[\hat{A},\hat{B}]\rvert\psi\rangle}{\langle\psi\rvert\psi\rangle} = \mathbb{P}_\psi^\mu\hat{A}\lvert\psi\rangle\,\boldsymbol{g}_{\mu\nu}\,\mathbb{P}_\psi^\nu\mathrm{i}\hat{B}\lvert\psi\rangle = \partial_\nu A\,\boldsymbol{J}^\nu{}_\rho\boldsymbol{G}^{\rho\sigma}\partial_\sigma B = \partial_\nu A\,\boldsymbol{\Omega}^{\nu\sigma}\partial_\sigma B\,, \tag{80}$$

where we used (74) and $\boldsymbol{J} = -\boldsymbol{J}^{-1} = \boldsymbol{\Omega}\boldsymbol{g}$ for a Kähler manifold. $\square$

For $\mathcal{M} = \mathcal{P}(\mathcal{H})$, above conditions are clearly met for any Hermitian operators $\hat{A}$ and $\hat{B}$. For a general Kähler submanifold $\mathcal{M} \subset \mathcal{P}(\mathcal{H})$, however, the validity of (78) depends on the choice of operators considered. On a submanifold which is not Kähler the statement is in general no longer valid.

# 4 Variational methods

Having introduced the mathematical background in the previous section, we can now study how variational methods allow us to describe closed quantum systems approximately. Given a system defined by a Hilbert space $\mathcal{H}$ and a Hamiltonian $\hat{H}$, we assume that a choice of a variational manifold $\mathcal{M} \subset \mathcal{P}(\mathcal{H})$, as defined in the previous section, has been made and show how to (A) describe real time dynamics, (B) approximate excitation energies, (C) compute spectral functions, (D) search for approximate ground states. In doing so, we will emphasize the differences that arise between the cases where the chosen variational manifold is or is not of the Kähler type.

Following the conventions introduced in section 3, we present a systematic and rigorous treatment of variational methods, for which section 2 served as prelude with some simplifications as discussed after example 2.

## 4.1 Real time evolution

For what concerns real time evolution, we would like to approximate the Schrödinger equation (1) on our variational manifold $\mathcal{M}$. There are different principles, used extensively in the literature, according to which this approximation can be performed. We will see that only in the case of Kähler manifolds they are all equivalent.

### 4.1.1 Variational principles

Following the literature, we can define the following variational principles for $|\psi\rangle := |\psi(t)\rangle$.

**Lagrangian action principle [25].** The most commonly used variational principle relies on the Lagrangian action, already introduced in (13),

$$\mathcal{S} = \int_{t_i}^{t_f} \mathcal{L} \, dt = \int_{t_i}^{t_f} dt \, \text{Re} \, \frac{\langle \psi | (\mathrm{i}\frac{d}{dt} - \hat{H}) | \psi \rangle}{\langle \psi | \psi \rangle} \, , \tag{81}$$

whose stationary solution satisfies

$$0 = \text{Re} \, \langle \mathbb{Q}_\psi \delta\psi | (\mathrm{i}\tfrac{d}{dt} - \hat{H}) | \psi \rangle \tag{82}$$

for all times and all allowed variations $|\delta\psi(t)\rangle$ with $\mathbb{Q}_\psi |\delta\psi\rangle = |\delta\psi\rangle - \frac{\langle \psi | \delta\psi \rangle}{\langle \psi | \psi \rangle} |\psi\rangle$ from (46). This is equivalent to Schrödinger's equation on projective Hilbert space[13]. On a variational manifold $\mathcal{M} \subset \mathcal{P}(\mathcal{H})$, i.e., where we require $\mathbb{Q}_\psi |\delta\psi(t)\rangle \in \mathcal{T}_{\psi(t)}\mathcal{M}$ in (82), we instead have

$$\mathbb{P}_\psi \mathrm{i} \tfrac{d}{dt} |\psi\rangle = \mathbb{P}_\psi \hat{H} |\psi\rangle \, . \tag{83}$$

This gives rise to the equations of motion (9) anticipated in Section 2, which we derive in Proposition 4. For a time-independent Hamiltonian, they always preserve the energy expectation value.

**McLachlan minimal error principle [50].** Alternatively, we can try to minimize the error between the approximate trajectory and the true solution. As we do not know the latter, we cannot compute the total error, but at least we can quantify the local error in state norm

$$\left\| \tfrac{d}{dt} |\psi\rangle - (-\mathrm{i}\hat{H}) |\psi\rangle \right\| \, , \tag{84}$$

due to imposing that $\frac{d}{dt} |\psi(x)\rangle$ represents a variation tangent to the manifold, i.e., $\mathbb{Q}_\psi \frac{d}{dt} |\psi(x)\rangle \in \mathcal{T}_\psi \mathcal{M}$. It is minimized by the projection

$$\mathbb{Q}_\psi \tfrac{d}{dt} |\psi\rangle = -\mathbb{P}_\psi \mathrm{i}\hat{H} |\psi\rangle \, . \tag{85}$$

This gives rise to the equations of motion (26) anticipated in Section 2, which we derive in Proposition 5. The resulting equations of motion only agree with the Lagrangian action if $\mathcal{M}$ is a Kähler manifold. Otherwise, they may not preserve the energy expectation value.

**Dirac-Frenkel variational principle [51, 52].** Another variational principle requires

$$\langle \delta\psi | (\mathrm{i}\tfrac{d}{dt} - \hat{H}) | \psi \rangle = 0 \tag{86}$$

for all allowed variations $|\delta\psi(t)\rangle$. It is easy to see that the real and imaginary parts of (86) are equivalent to (83) and (85) respectively. Therefore, this principle is well-defined (and

---

[13]The fact that the projector $\mathbb{Q}_\psi$ onto projective tangent space $\mathcal{H}_\psi^\perp$ appears, shows that the resulting dynamics is defined on projective Hilbert space, while global phase and normalization are left undetermined. We will explain how to recover the dynamics of phase and normalization in Section 4.1.3.

equivalent to the other two) only in the cases in which they are equivalent between themselves, that is, as we will see, if and only if $\mathcal{M}$ is a Kähler manifold. Otherwise, the resulting equations will be overdetermined.

Expressing equations (83) and (85) in coordinates leads to flow equations for the manifold parameters $x(t)$. We can then define a real time evolution vector field $\mathcal{X}^\mu$ everywhere on $\mathcal{M}$, such that

$$\frac{dx^\mu}{dt} = \mathcal{X}^\mu(x).\tag{87}$$

Integrating such equations defines the flow map $\Phi_t$ that maps an initial set of coordinates $x^\mu(0)$ to the values $x^\mu(t)$ that they assume after evolving for a time $t$.

In the case of the Lagrangian action principle, the vector field $\mathcal{X}$ takes the form given in the following proposition. A similar derivation was also considered in [25].

**Proposition 4.** *The real time evolution projected according to the **Lagrangian action principle** (83) is*

$$\frac{dx^\mu}{dt} \equiv \mathcal{X}^\mu = -\mathbf{\Omega}^{\mu\nu}(\partial_\nu E), \quad \textbf{(Lagrangian)}\tag{88}$$

*where $E(x)$ is the energy function, defined in the context of equation (72). Such evolution always conserves the energy expectation value.*

*Proof.* From the definition (50) of the tangent space basis, we have

$$\frac{d}{dt}|\psi\rangle = \dot{x}^\mu \partial_\mu |\psi\rangle = \dot{x}^\mu |V_\mu\rangle + \frac{\langle\psi|\frac{d}{dt}\psi\rangle}{\langle\psi|\psi\rangle}|\psi\rangle.\tag{89}$$

We substitute this in (83) and then expand the projectors using the relations (53), (55) and $\mathbb{P}_\psi \mathrm{i}|\psi\rangle = 0$ to obtain

$$\mathbf{J}^\mu{}_\nu \mathcal{X}^\nu = \mathbf{G}^{\mu\rho}\frac{2\mathrm{Re}\langle V_\rho|\hat{H}|\psi\rangle}{\langle\psi|\psi\rangle}.\tag{90}$$

We further simplify the expression by using (74) and $(\mathbf{J}^{-1})^\mu{}_\nu = -\mathbf{\Omega}^{\mu\rho}\mathbf{g}_{\rho\nu}$ from (55). This leads to

$$\mathcal{X}^\mu = (\mathbf{J}^{-1})^\mu{}_\nu \mathbf{G}^{\nu\sigma}\partial_\sigma E = -\mathbf{\Omega}^{\mu\nu}\partial_\nu E.\tag{91}$$

To obtain the variation of the energy expectation value $E$ we compute directly

$$\frac{dE}{dt} = (\partial_\mu E)\frac{dx^\mu}{dt} = -(\partial_\mu E)\mathbf{\Omega}^{\mu\nu}(\partial_\nu E) = 0,\tag{92}$$

where we used the antisymmetry of $\mathbf{\Omega}^{\mu\nu}$. If $\mathbf{J}$ (and thus also $\mathbf{\Omega}$) is not invertible, one needs to restrict to an appropriate subspace. $\qquad\square$

The most important lesson of (88) is that projected time evolution on a Kähler manifold is equivalent to Hamiltonian evolution with respect to energy function $E(x)$. As was pointed out in [24], already the time evolution in full projective Hilbert space, *i.e.*, $\mathcal{M} = \mathcal{P}(\mathcal{H})$, follows the classical Hamilton equations if we use the natural symplectic form $\mathbf{\Omega}^{\mu\nu}$. Let us point out that the sign in equation (88) depends on the convention chosen for the symplectic form, which in classical mechanics differs from the one adopted here. One further consequence of equation (88) is that the real time evolution vector field $\mathcal{X}(x)$ vanishes in stationary points

of the energy, that is points $x_0$ such that $\partial_\mu E(x_0) = 0$. These points will therefore also be stationary points of the evolution governed by $\mathcal{X}$ as illustrated in figure 3.

Let us here recall that, if $\boldsymbol{\omega}_{\mu\nu}$ is not invertible, $\boldsymbol{\Omega}^{\mu\nu}$ refers to the pseudo-inverse, as discussed in sections 3.3 and 3.4. This convention means that in practice the Lagrangian evolution will always take place in the submanifold of $\mathcal{M}$ on which $\boldsymbol{\omega}$ is invertible. There may be pathological cases where $\boldsymbol{\omega}$ vanishes on the whole tangent space and therefore the Lagrangian principle does not lead to any evolution. In appendix E, we present a method to efficiently compute the pseudo-inverse in practical applications.

In the case of the McLachlan minimal error principle, the evolution equations take the form given in the following proposition, which cannot be simplified further. It is also in general not true that this evolution conserves the energy or that has a stationary point in energy minima.

**Proposition 5.** *The real time evolution projected based on the **McLachlan minimal error principle** (85) is*

$$\frac{dx^\mu}{dt} \equiv \mathcal{X}^\mu = -\frac{2\boldsymbol{G}^{\mu\nu}\mathrm{Re}\langle V_\nu | \mathrm{i}\hat{H} | \psi \rangle}{\langle \psi | \psi \rangle} \ . \quad \textbf{(McLachlan)} \tag{93}$$

*Proof.* By substituting (50) in (85), analogously to what was done in (89), we have

$$\dot{x}^\mu = \mathbb{P}^\mu_\psi(-\mathrm{i}\hat{H} | \psi \rangle), \tag{94}$$

from which the proposition follows by expanding the projector according to (53). □

To perform real time evolution in practice, either based on (88) for Lagrangian evolution or based on (93) for McLachlan evolution, we will typically employ a numerical integration scheme [53, 54] to evolve individual steps. It is generally hard to get rigorous bounds on the resulting error that increases over time, but in certain settings there still exist meaningful estimates [55]. Let us now relate the different variational principles, which has also been discussed in [56].

**Proposition 6.** *The Lagrangian, the McLachlan and the Dirac-Frenkel variational principle are equivalent if the variational family is Kähler.*

*Proof.* To prove the statement, it is sufficient to see that equations (83) and (85) can be written simply as applying the tangent space projector $\mathbb{P}_\psi$ to two different forms of the Schrödinger equation, *i.e.*,

$$\textbf{Lagrangian:} \qquad \mathbb{P}_\psi(\mathrm{i}\tfrac{d}{dt} - \hat{H}) | \psi \rangle = 0 \tag{95}$$

$$\textbf{McLachlan:} \qquad \mathbb{P}_\psi(\tfrac{d}{dt} + \mathrm{i}\hat{H}) | \psi \rangle = 0. \tag{96}$$

These two forms only differ by a factor of i. However, as we discussed in proposition 2, one equivalent way to define the Kähler property of our manifold is that multiplication by i commutes with the projector $\mathbb{P}_\psi$. Therefore, if the manifold is Kähler, an imaginary unit can be factored out of equations (95) and (96) making them coincide. If, on the other hand, the manifold is non-Kähler, this operation is forbidden and they are in general not equivalent. □

As discussed in Section 3.4, if the chosen manifold does not respect the Kähler condition, we always have the choice to locally restrict ourselves to consider only a subset of tangent directions with respect to which the manifold is again Kähler, *i.e.*, $\overline{\mathcal{T}_\psi\mathcal{M}}$. Then both principles will again give the same equation of motion, which will have the same form as (88) where we

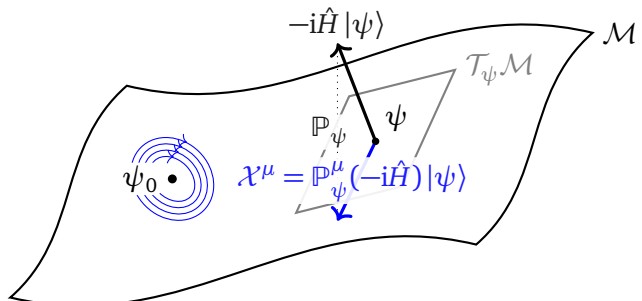

Figure 3: *Real time evolution.* We illustrate real time evolution on a variational manifold $\mathcal{M}$ according to the Dirac-Frenkel variational principle (where Lagrangian and McLachlan principles coincide). The time evolution vector $-\mathrm{i}\hat{H}\,|\psi\rangle$ at a state $\psi$ is orthogonally projected through $\mathbb{P}_\psi$ onto the variational manifold $\mathcal{M}$ to define the vector field $\mathcal{X}^\mu$. We further indicate how real time evolution near a fixed point $\psi_0$ follows approximately circles or ellipses.

just replace $\Omega^{\mu\nu}$ with $\overline{\overline{\Omega}}^{\mu\nu}$, which will conserve the energy and have stationary points in the minima of the energy. We will refer to this procedure as *Kählerization*.

We can compute explicitly how the vector fields of the Lagrangian and McLachlan variational principles differ. For this, we only consider the subspaces, defined in Proposition 1, in which the complex structure fails to be Kähler, *i.e.*, where we have

$$J \equiv \bigoplus_i \begin{pmatrix} & c_i \\ -c_i & \end{pmatrix}, \tag{97}$$

as in (66). On the enlarged tangent space including all vectors $\mathrm{i}\,|V_\mu\rangle$, the enlarged complex structure

$$\breve{J} \equiv \bigoplus_i \begin{pmatrix} & c_i & & \sqrt{1-c_i^2} \\ -c_i & & \sqrt{1-c_i^2} & \\ & -\sqrt{1-c_i^2} & & c_i \\ -\sqrt{1-c_i^2} & & -c_i & \end{pmatrix} \tag{98}$$

clearly satisfies $\breve{J}^2 = -\mathbb{1}$. For the time evolution vector field $\breve{\mathcal{X}} \equiv \oplus_i(a_i, b_i, \alpha_i, \beta_i)$ on the enlarged space, we find the two distinct restrictions

$$\mathcal{X}_{\text{Lagrangian}} = J^{-1}\mathbb{P}_\psi \breve{J}\breve{\mathcal{X}} \equiv \oplus_i\left(a_i - \frac{\sqrt{1-c_i^2}}{c_i}\alpha_i, b_i + \frac{\sqrt{1-c_i^2}}{c_i}\beta_i\right) \tag{99}$$

$$\mathcal{X}_{\text{McLachlan}} = \mathbb{P}_\psi \breve{\mathcal{X}} \equiv \oplus_i(a_i, b_i), \tag{100}$$

associated to the Lagrangian and the McLachlan principle, respectively. We see explicitly that they agree for $c_i = 1$, but also when $\alpha_i = \beta_i = 0$.

**Example 9.** *We consider the variational family from example 7 for a system with two bosonic degrees of freedom. We choose the Hamiltonian*

$$\hat{H} = \frac{\epsilon_+(\hat{n}_1+\hat{n}_2)+\epsilon_-[(\hat{n}_1-\hat{n}_2)\cos\phi+(\hat{a}_1^\dagger\hat{a}_2+\hat{a}_1\hat{a}_2^\dagger)\sin\phi]}{2}, \tag{101}$$

*where $\epsilon_1$ and $\epsilon_2$ are the excitation energies with $\epsilon_\pm = \epsilon_1 \pm \epsilon_2$, while $\phi$ is a coupling constant, such that $\hat{H} = \epsilon_1\hat{n}_1 + \epsilon_2\hat{n}_2$ for $\phi = 0$. Figure 4 shows the time evolution of the expectation values*

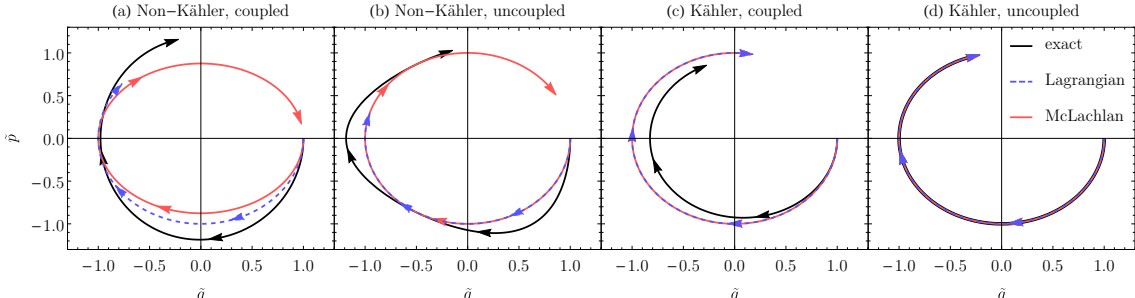

Figure 4: *Comparison of variational principles.* We illustrate how exact, Lagrangian and McLachlan evolution differ in example 9. We choose $\epsilon_1 = 1$, $\epsilon_2 = 2$, initial conditions $\tilde{z} \equiv (1,0)$ and $r = 0.3$ for non-Kähler, $r = 0$ for Kähler, $\phi = 0.3$ for coupled and $\phi = 0$ for uncoupled. To indicate speed, we place an arrow at $t \in \{1.5, 3, 4.5\}$. (a) All trajectories differ, (b) Lagrangian and McLachlan give the same trajectories with different speed, (c) Lagrangian and McLachlan agree, (d) Langrangian and McLachlan become exact.

$\tilde{z}^\alpha \equiv (\tilde{q}, \tilde{p})$ *for the two operators*

$$\hat{\tilde{q}} = \tfrac{1}{\sqrt{2}}(\hat{b}^\dagger + \hat{b}), \qquad\qquad \hat{\tilde{p}} = \tfrac{\mathrm{i}}{\sqrt{2}}(\hat{b}^\dagger - \hat{b}), \qquad\qquad (102)$$

*where $\hat{b}$ was defined in (60). For $r = 0$, the variational family is Kähler and the two variational principles give rise to the same evolution. For $r \neq 0$, the two principles generally disagree. The explicit formulas can be efficiently derived using the framework of Gaussian states reviewed in section 5.1, where we reconsider the present scenario in example 20.*

**Kähler vs. non-Kähler.** On a Kähler manifold all three variational principles are well-defined and equivalent. They all give the same energy conserving equations of motion (88). On a non-Kähler manifold, only the Lagrangian and McLachlan variational principles are well-defined, but they give in general inequivalent equations of motion given by (88) and (93). Only the Lagrangian ones will manifestly conserve the energy and have stationary points in the minima of the energy. In table 2, we review advantages and drawbacks discussed in the following. While in most cases, the Lagrangian principle appears to be a natural choice, the McLachlan principle is often preferable if $\boldsymbol{\omega}$ is highly degenerate—in particular, if $\boldsymbol{\omega} = 0$ its pseudo-inverse is $\boldsymbol{\Omega} = 0$ and the evolution would vanish everywhere independent of $\hat{H}$, such that the McLachlan principle appears to be the better choice.

### 4.1.2 Conserved quantities

Given the generator $\hat{A}$ of a symmetry of the Hamiltonian $\hat{H}$, i.e., $[\hat{H}, \hat{A}] = 0$, the expectation value $A(t) = \langle \psi_t | \hat{A} | \psi_t \rangle$ is necessarily preserved by unitary time evolution on the full Hilbert space

$$|\psi_t\rangle = U(t)|\psi_0\rangle = e^{-\mathrm{i}\hat{H}t} |\psi_0\rangle . \qquad\qquad (103)$$

We now consider if this continues to be true for projected time evolution on a manifold.

For a time-independent Hamiltonian $\hat{H}$, we have seen that the energy expectation value $E$ is always conserved by Lagrangian projected real time evolution. However, projected time evolution will not in general preserve expectation values of an operator $\hat{A}$ with $[\hat{H}, \hat{A}] = 0$. To guarantee this, one has to further require that $\hat{A}$ preserves the manifold.

**Proposition 7.** *Given a variational manifold $\mathcal{M}$ and a Hermitian operator $\hat{A}$, such that $[\hat{H}, \hat{A}] = 0$ and $\hat{A}$ preserves the manifold in the sense of Definition 3, i.e.,*

$$\mathbb{Q}_\psi \hat{A} |\psi\rangle = (\hat{A} - \langle \hat{A} \rangle) |\psi\rangle \in \mathcal{T}_\psi \mathcal{M} \quad \forall \psi \in \mathcal{M}, \tag{104}$$

*the expectation value $A(x(t))$, defined as in equation (72), is preserved under real time evolution projected according to the **McLachlan** variational principle. It is also true for **Lagrangian** variational principle, if the two principles agree, i.e., if the manifold is Kähler.*

*Proof.* We compute

$$\frac{d}{dt} A(t) = (\partial_\mu A) \frac{dx^\mu}{dt} = \mathbb{P}_\psi^\nu \hat{A} |\psi\rangle \, g_{\nu\mu} \mathbb{P}_\psi^\mu (-i\hat{H} |\psi\rangle) = \frac{2\mathrm{Re}\langle \psi|(\hat{A}-\langle\hat{A}\rangle)(-i\hat{H})|\psi\rangle}{\langle\psi|\psi\rangle} = \frac{i\langle\psi|[\hat{H},\hat{A}]|\psi\rangle}{\langle\psi|\psi\rangle} = 0, \tag{105}$$

where in the first line we used relation (74) for the gradient of $A$, the definition of McLachlan evolution (85) and that $\mathbb{P}_\psi^\mu \langle \hat{A} \rangle |\psi\rangle = 0$, in the second step we used that, thanks to the condition (104), the restricted bilinear form $g$ in the first line coincides with the full Hilbert space one in the second line of (105). □

This result only applies the McLachlan projected real time evolution, for which the equation of motion (94) holds. In the Lagrangian case, we would have

$$\dot{A} = (\partial_\mu A) \mathcal{X}^\mu = -(\partial_\mu A) \Omega^{\mu\nu} \partial_\nu E = \{E, A\}, \tag{106}$$

which is in general not equal to $i\langle\psi|[\hat{H},\hat{A}]|\psi\rangle \langle\psi|\psi\rangle^{-1}$ on a non-Kähler manifold[14] and thus not necessarily zero.

We see here the main advantage of the *Kählerization* procedure described in the previous section. Indeed, through *Kählerization* we are able to define, even for general non-Kähler manifolds, a projected real time evolution that shares the desirable properties of both, the Lagrangian and the McLachlan projections, *i.e.*, it is a symplectic, energy preserving evolution with stationary points in the energy minima and at the same time preserves the expectation value of symmetry generators satisfying (104). Note that Kählerization may spoil the conservation laws of observables $\hat{A}$, for which $\mathbb{Q}_\psi \hat{A} |\psi\rangle$ does not lie in the Kähler subspace $\overline{\mathcal{T}_\psi \mathcal{M}}$, in which case we will need to enforce conservation by hand, discussed next.

For operators $\hat{A}$ where (104) is not satisfied, we have two options to correct for this:

(a) Enlarge the variational manifold $\mathcal{M}$, such that condition (104) is satisfied.

(b) Enforce conservation by hand, for which we modify the projected time evolution vector field $\mathcal{X}^\mu$.

While option (a) is typically more desirable, it requires creativity to find a suitable extension of a given family $\mathcal{M}$. Of course, if $\hat{A}$ is an important physical observable that is relevant to the problem, a manifold that does not preserve it may not be a good choice to approximate the system's behavior. In practice, however, it may still be worthwhile to check the predictions of an approximated time-evolution adopting (b).

---

[14]For Kähler manifolds, as discussed in the context of Proposition 3, $\{F, G\} = i\langle\psi|[\hat{F}, \hat{G}]|\psi\rangle \langle\psi|\psi\rangle^{-1}$ only holds if either $\hat{F}$ or $\hat{G}$ preserves the manifold.

This is done by adding a further projection of the real time evolution flow onto the subspace of the tangent space orthogonal (with respect to $\boldsymbol{g}$) to the direction $\mathbb{P}_\psi^\mu \hat{A} |\psi\rangle = \boldsymbol{G}^{\mu\nu} \partial_\nu A$. This is equivalent to restricting ourselves to the submanifold

$$\widetilde{\mathcal{M}} = \left\{ |\psi\rangle \in \mathcal{M} \,\Big|\, \frac{\langle \psi|\hat{A}|\psi\rangle}{\langle \psi|\psi\rangle} = A_0 \right\} \subset \mathcal{M}, \tag{107}$$

where $A_0$ is the initial value $\langle \psi(0)|\hat{A}|\psi(0)\rangle \langle \psi(0)|\psi(0)\rangle^{-1}$. Note that this modified evolution may spoil other conservation laws (*e.g.*, energy) that were previously intact.

To preserve several quantities $\hat{A}_I$, we can project onto the subspace orthogonal to the span of $X_I = \mathbb{P}_\psi \hat{A}_I |\psi\rangle$. If we also want to preserve the Kähler property, we should choose $X_I = (\mathbb{P}_\psi \hat{A}_I |\psi\rangle, i\mathbb{P}_\psi \hat{A}_I |\psi\rangle)$. We can then define $\widetilde{g}_{IJ} = X_I^\mu \boldsymbol{g}_{\mu\nu} X_J^\nu$ to define the projector

$$\widetilde{P}^\mu{}_\nu = \delta^\mu{}_\nu - X_I^\mu \widetilde{G}^{IJ} X_J^\rho \boldsymbol{g}_{\rho\nu}, \tag{108}$$

where $\widetilde{G}^{IJ}$ is the inverse (or pseudo inverse, if not all vectors $X_I$ are linearly independent) of $\widetilde{g}_{IJ}$.

The modified Lagrangian evolution vector field $\widetilde{\mathcal{X}}^\mu$ is

$$\widetilde{\mathcal{X}}^\mu = -\widetilde{\boldsymbol{\Omega}}^{\mu\nu}(\partial_\nu E) \quad \text{with} \quad \widetilde{\boldsymbol{\Omega}}^{\mu\nu} = \widetilde{P}^\mu{}_\sigma \widetilde{P}^\nu{}_\rho \boldsymbol{\Omega}^{\sigma\rho}, \tag{109}$$

while for the McLachlan evolution, we find

$$\widetilde{\mathcal{X}}^\mu = \widetilde{P}^\mu{}_\nu \mathcal{X}^\nu, \tag{110}$$

where $\mathcal{X}^\mu$ represents the unmodified evolution vector field in the McLachlan case. It will conserve all expectation values $A_I(t)$ by construction. In the Lagrangian case also the energy will continue to be conserved by construction, which would need to be enforced by hand for the McLachlan case.

**Kähler vs. non-Kähler.** On a non-Kähler manifold, where we have two inequivalent definitions of the evolution, only the one coming from the McLachlan principle preserves the expectation value of symmetry generators satisfying (104). Thus a key reason to Kählerize a non-Kähler manifold is to conserve these expectation values also in the Lagrangian evolution.

### 4.1.3 Dynamics of global phase

Up to now we have always considered our variational manifolds $\mathcal{M}$ as submanifolds of projective Hilbert space and thus the tangent space $\mathcal{T}_\psi \mathcal{M}$ as a subspace of $\mathcal{H}_\psi^\perp$. This means all states are only defined up to a complex factor. In practice, our family $\psi(x) \in \mathcal{P}(\mathcal{H})$ will be described by a choice $|\psi(x)\rangle \in \mathcal{H}$ *i.e.*, for every set of parameters $x^\mu$, we will have a Hilbert space vector $|\psi(x)\rangle$ representing the quantum state $\psi(x) \in \mathcal{P}(\mathcal{H})$.

If the parametrization $x^\mu$ happens to contain the global phase or normalization of the state as an independent parameter, we are overparametrizing our family and the evolution equations (88) or (93) will keep the evolution of some parameters undetermined leading to some gauge redundancy. This is due to the fact that normalization and phase do not change the quantum state $|\psi(x)\rangle$ represents and our equations of motion only determine the physical evolution of the quantum state and not of its Hilbert space representative.

We can include the time evolution of the global phase and the state normalization by extending our parametrization by defining

$$|\Psi(x, \kappa, \varphi)\rangle = e^{\kappa + i\varphi} |\psi(x)\rangle, \tag{111}$$

where $\kappa$ and $\varphi$ are two additional real parameters. If phase or normalization were already contained in $x^\mu$ this will lead to an overparametrization, but we have already explained how to take care of this in Section 3.3.

Then, on top of the real time evolution equations (83) or (85), we obtain equations for these extra parameters by projecting Schrödinger's equation on the corresponding tangent space directions, *i.e.*, $|V_\kappa\rangle = |\Psi\rangle$ and $|V_\varphi\rangle = i|\Psi\rangle$, to find the two equations

$$\mathrm{Re}\,\langle\Psi|(-i\tfrac{d}{dt}+\hat{H})|\Psi\rangle = 0, \quad \mathrm{Re}\,\langle\Psi|\tfrac{d}{dt}+i\hat{H}|\Psi\rangle = 0. \tag{112}$$

Equivalently, as anticipated in Section 2, we can use the Lagrangian action principle to find the same equations by extremizing the alternative action

$$S = \int_{t_i}^{t_f} dt\,\mathrm{Re}\,\langle\Psi(t)|(i\tfrac{d}{dt}-\hat{H})|\Psi(t)\rangle \tag{113}$$

for the full set of parameters $(x,\kappa,\varphi)$ rather than $\mathcal{S}$ from (81) for only $x$.

In both cases, the time evolution of $x^\mu(t)$ is unchanged, but we find the additional equations

$$\dot{\varphi} = \frac{\mathrm{Re}\,\langle\psi|i\tfrac{d}{dt}|\psi\rangle}{\langle\psi|\psi\rangle} - E(t) \quad\text{and}\quad \dot{\kappa} = -\frac{\mathrm{Re}\,\langle\psi|\tfrac{d}{dt}|\psi\rangle}{\langle\psi|\psi\rangle} \tag{114}$$

relating the evolution of phase and normalization with $|\psi(x(t))\rangle$. Interestingly, the time evolution of $\kappa$ will ensure that $|\Psi(x,\kappa,\varphi)\rangle$ does not change normalization.

The procedure can be understood as follows. Global phase and normalization are conjugate parameters when considering Hilbert space as Kähler space, as can be seen from $|V_\varphi\rangle = i|V_\kappa\rangle$. When considering a variational manifold $\mathcal{M} \subset \mathcal{P}(\mathcal{H})$, we have the following options:

1. When we are only interested in the time evolution of physical states $\psi$, we must project out the information about global phase and normalization using $\mathbb{P}^\mu_{\psi}$. Consequently, our evolution equations will not determine how to change global phase or normalization as this information is pure gauge. We followed this philosophy until the current section.

2. When we are also interested in the time evolution of global phase and normalization, we can always extend $\mathcal{M}$ to include both phase and normalization as independent parameters. Given a generic parametrization $|\psi(x)\rangle$, we can extend it to $|\Psi(x,\kappa,\varphi)\rangle$ to ensure that it satisfies the Kähler property in the phase/normalization subspace. We can then find evolution equations for $\varphi$ and $\kappa$. This is what we explained in the current subsection.

Finally, let us emphasize that using equations (112) or extremizing action (113) without first ensuring both phase and normalization are included as independent parameters may lead to unphysical results.

**Example 10.** *We consider coherent states parametrized as* $|\psi(x)\rangle = e^{i\varphi(x_1,x_2)}e^{(x_1+ix_2)\hat{a}^\dagger}|0\rangle$, *where the states are not normalized due to* $\langle\psi(x)|\psi(x)\rangle = e^{x_1^2+x_2^2}$ *and we chose intentionally a phase* $\varphi(x_1,x_2)$. *We further consider the Hamiltonian* $\hat{H} = \omega\hat{a}^\dagger\hat{a}$. *The equation of motion on projective Hilbert space based on the action* (7) *are*

$$\dot{x}_1 = \omega x_2 \quad\text{and}\quad \dot{x}_2 = -\omega x_1. \tag{115}$$

*However, if we use* (113)*, we find the action*

$$S = \int dt \left( \dot{x}_1 x_2 - \dot{x}_2 x_1 - \frac{\partial \varphi}{\partial x_1} \dot{x}_1 - \frac{\partial \varphi}{\partial x_2} \dot{x}_2 \right) e^{x_1^2 + y_1^2}, \tag{116}$$

*which leads to the equations of motion given by*

$$(1 + x_1^2 + x_2^2)(\omega x_1 + \dot{x}_2) = (\frac{\partial \varphi}{\partial x_2} x_1 - \frac{\partial \varphi}{\partial x_1} x_2)\dot{x}_2, \tag{117}$$

$$(1 + x_1^2 + y_2^2)(\omega x_2 - \dot{x}_1) = (\frac{\partial \varphi}{\partial x_1} x_2 - \frac{\partial \varphi}{\partial x_2} x_1)\dot{x}_1. \tag{118}$$

*They only agree with* (115) *if* $\frac{\partial \varphi}{\partial x_2} x_1 - \frac{\partial \varphi}{\partial x_1} x_2 = 0$.

### 4.2 Excitation spectra

We would like to use a variational family $\mathcal{M}$ to approximate the excitation energies $E_i$ of some eigenstates $|E_i\rangle$ of the Hamiltonian. Typically, we are interested in low energy eigenstates, that is eigenstates close to the groundstate of the Hamiltonian. Suppose then that on $\mathcal{M}$ we are able to find an approximate ground state $|\psi_0\rangle$, that is the state with energy $\omega_0$ that represents the global energy minimum on $\mathcal{M}$ (we will describe a method for finding such state in Section 4.4). Then there are two distinct approaches of deriving a spectrum: the projection of the Hamiltonian and the linearization of the equations of motion.

#### 4.2.1 Projected Hamiltonian

Given a tangent space $\mathcal{T}_{\psi_0}\mathcal{M}$ at a state $\psi_0 \in \mathcal{M}$, we can approximate the excitation spectrum of the Hamiltonian $\hat{H}$ from its projection onto $\mathcal{T}_{\psi_0}\mathcal{M}$.

Based on the two action principles (Lagrangian vs. McLachlan), we define two different projections given by

$$\begin{aligned} H^\mu{}_\nu &= \mathbb{P}^\mu_{\psi_0} \hat{H} |V_\nu\rangle, && \textbf{(Lagrangian)} \\ R^\mu{}_\nu &= -\mathbb{P}^\mu_{\psi_0} i\hat{H} |V_\nu\rangle. && \textbf{(McLachlan)} \end{aligned} \tag{119}$$

On a Kähler manifold $\mathcal{M}$, we will have $R^\mu{}_\nu = -J^\mu{}_\sigma H^\sigma{}_\nu$ and $[J, H] = [R, H] = 0$. In this case, $H$ represents a Hermitian operator on tangent space (which is complex sub Hilbert space) and $R$ is anti-Hermitian. In this case, the eigenvalues of $H$ are real and come in pairs $(\omega_\ell, \omega_\ell)$, while the ones of $R$ come are purely imaginary and come in conjugate pairs $(i\omega_\ell, -i\omega_\ell)$. The two associated eigenvectors of $R$ are related by multiplication of $J$ and also span the respective eigenspace of $H$.

On a non-Kähler manifold $\mathcal{M}$, the relation between $H$ and $R$ as well as their respective spectra is non-trivial. The eigenvalues $\omega_\ell$ of $H^\mu{}_\nu$ will still be real, while the ones of $R^\mu{}_\nu$ continue to appear in conjugate pairs. The latter also implies that for an odd-dimensional manifold $R^\mu{}_\nu$ must have a vanishing eigenvalue, which is a pure artefact of the projection.

The projected Hamiltonian $H^\mu{}_\nu$ represents the full Hamiltonian restricted to the tangent space. The Courant–Fischer–Weyl min-max principle states that the eigenvalues $E_\ell$ of $\hat{H}$ and the eigenvalues $\omega_\ell$ of $H^\mu{}_\nu$ satisfy

$$E_\ell \le \omega_\ell \le E_{N-n+\ell}, \tag{120}$$

with $N = \dim_\mathbb{R} \mathcal{H}$ and $n = \dim_\mathbb{R} \mathcal{T}_{\psi_0}\mathcal{M}$, where we assume that all eigenvalues are sorted and appear with their multiplicity. Therefore, every approximate eigenvalue $\omega_\ell$ bounds a

corresponding true eigenvalue $E_i$ from above. How good this approximation is will highly depend on the choice of variational manifold. Note that the energy differences $\omega_\ell - \omega_0$ instead do not necessarily bound $E_\ell - E_0$, because the ground state energy $\omega_0$ might not be exact, *i.e.*, $\omega_0 > E_0$.

Furthermore, the eigenvalues $\omega_\ell$ are variational in the sense that if $X_\ell^\mu$ is an eigenvector of $\boldsymbol{H}^\mu{}_\nu$ such that $\boldsymbol{H}^\mu{}_\nu X_\ell^\nu = \omega_\ell X_\ell^\mu$, then the corresponding Hilbert space vector $|X_\ell\rangle = X_\ell^\mu |V_\mu\rangle$ satisfies

$$\frac{\langle X_\ell | \hat{H} | X_\ell \rangle}{\langle X_\ell | X_\ell \rangle} = \omega_\ell \,. \tag{121}$$

**Kähler vs. non-Kähler.** On a Kähler manifold, $\boldsymbol{H}^\mu{}_\nu$ and $\boldsymbol{R}^\mu{}_\nu$ are related via $\boldsymbol{R} = -\boldsymbol{J}\boldsymbol{H}$ and they will be the representations of a complex Hermitian and anti-Hermitian operators, respectively. Real eigenvalue pairs $(\omega_\ell, \omega_\ell)$ of $\boldsymbol{H}$ will be related to imaginary eigenvalue pairs $(\mathrm{i}\omega_\ell, -\mathrm{i}\omega_\ell)$ of $\boldsymbol{R}$. On a non-Kähler manifold, the eigenvalues $\omega_\ell$ of $\boldsymbol{H}$ could all be different and unrelated to the ones $\boldsymbol{R}$, which are still imaginary appearing in conjugate pairs.

### 4.2.2 Linearized equations of motion

A common alternative to projecting the Hamiltionian is to linearize the equations of motion around a fixed point $x_0$ such that $\mathcal{X}(x_0) = 0$

$$\frac{dx^\mu}{dt} = \mathcal{X}^\mu \quad \Rightarrow \quad \frac{d}{dt}\delta x^\mu = K^\mu{}_\nu \delta x^\nu \,, \tag{122}$$

with $\delta x^\mu = x^\mu - x_0^\mu$ and $K^\mu{}_\nu = \partial_\nu \mathcal{X}^\mu|_{x=x_0}$. Here, $\delta x^\mu$ represents a small perturbation around the approximate ground state. The frequencies $\omega_\ell$ appearing in conjugate pairs $\pm\mathrm{i}\omega_\ell$ in the spectrum of $K^\mu{}_\nu$ represent the frequencies with which such perturbations oscillate around the ground state and thus provide an approximation to the excitation energies $E_\ell - E_0$ of the Hamiltonian.

As pointed out in Section 4.1.1, the fixed point $x_0$ only coincides with the approximate ground state $\psi_0$ if the real time evolution is defined in terms of the Lagrangian action principle. We thus assume the equations of motion (88) based on Lagrangian action principle. In this case, we find

$$K^\mu{}_\nu = \partial_\nu \mathcal{X}^\mu = -\partial_\nu(\Omega^{\mu\rho}\partial_\rho E) = -\Omega^{\mu\rho}(\partial_\rho\partial_\nu E) \,, \tag{123}$$

where everything is evaluated at $x_0$ after taking the derivatives. We used that $\partial_\rho E = 0$ at the fixed point[15].

By construction, $K^\mu{}_\nu$ is a symplectic generator, because it satisfies $K\Omega + \Omega K^\intercal = 0$, which implies that $M = e^K$ preserves the symplectic form, *i.e.*, $M\Omega M^\intercal = \Omega$. Provided that $\psi_0$ is an energy minimum, the bilinear form $h_{\mu\nu} = \partial_\nu\partial_\mu E$ is positive definite. By Williamson's theorem [57], $K^\mu{}_\nu$ is diagonalizable and the resulting eigenvalues appear in conjugate pairs $(\mathrm{i}\omega_\ell, -\mathrm{i}\omega_\ell)$.

From a geometric point of view, $\delta x^\mu$ represents a tangent vector $|\delta\psi\rangle = \delta x^\mu |V_\mu\rangle$ living in the tangent space $\mathcal{T}_{\psi_0}\mathcal{M}$ at the approximate ground state $|\psi_0\rangle$. The time evolution of a

---

[15]Usually, defining a derivative of a vector field $\mathcal{X}^\mu$ requires a way to relate tangent spaces at adjacent points via a so-called connection. The resulting covariant derivative $\nabla_\nu \mathcal{X}^\mu = \partial_\nu \mathcal{X}^\mu + \Gamma^\mu_{\nu\rho}\mathcal{X}^\rho$ will depend on $\Gamma^\mu_{\nu\rho}$ that encodes the connection. In our case $\mathcal{X}^\mu$ vanishes at the fixed point, so that the dependence of $\Gamma^\mu_{\nu\rho}$ drops out and the spectrum of $K^\mu{}_\nu = \nabla_\nu \mathcal{X}^\mu$ at $|\psi_0\rangle$ is canonically defined.

tangent vector at a fixed point $|\psi_0\rangle$ is described by the linearized evolution flow[16]

$$d\Phi_t : \mathcal{T}_{\psi_0}\mathcal{M} \to \mathcal{T}_{\psi_0}\mathcal{M}. \tag{124}$$

$K$ is the generator of the flow $d\Phi_t$ leading to the important relation

$$d\Phi_t = e^{tK}, \tag{125}$$

which shows that $d\Phi_t$ is symplectic.

Unitary evolution on Hilbert space leads to a flow on projective Hilbert space that preserves all three Kähler structures. However, when we project this flow onto a variational manifold to find $\mathcal{X}^\mu$, we will project out the part of the vector field orthogonal to tangent space. When using the Lagrangian action principle, the projected flow will continue to be symplectic, *i.e.*, preserve $\boldsymbol{\Omega}$, but none of the other two Kähler structures[17]. Geometrically, this implies that the trajectories of states near the fixed point $\psi_0$ will be elliptic rather than circular, when measured with respect to $\boldsymbol{G}$.

Therefore, even if $\mathcal{M}$ is a Kähler manifold, $\boldsymbol{K}^\mu{}_\nu$ will in general neither commute with $\boldsymbol{J}$ nor be antisymmetric with respect to $G$, *i.e.*, satisfy $\boldsymbol{KG} = -\boldsymbol{GK}^\intercal$. This has the following consequences:

- Right-eigenvectors $\mathcal{E}^\mu(\lambda)$ with $\boldsymbol{K}^\mu{}_\nu \mathcal{E}^\nu(\lambda) = \lambda \mathcal{E}^\mu(\lambda)$ and left-eigenvectors $\widetilde{\mathcal{E}}_\mu(\lambda)$ with $\boldsymbol{K}^\mu{}_\nu \widetilde{\mathcal{E}}_\mu(\lambda) = \lambda \widetilde{\mathcal{E}}_\nu(\lambda)$ are not related via $\mathcal{E}^\mu(\lambda) = G^{\mu\nu}\widetilde{\mathcal{E}}_\nu(\lambda)$, but need to be computed independently. This is important when computing spectral functions in section 4.3.

- There does in general not exist a Hilbert space operator $\hat{K}$, such that $\boldsymbol{K}^\mu{}_\nu$ is its restriction in the sense of $\boldsymbol{K}^\mu{}_\nu = \mathbb{P}^\mu_{\psi_0}\hat{K}|V_\nu\rangle$ or $\boldsymbol{K}^\mu{}_\nu = \mathbb{P}^\mu_{\psi_0}\mathrm{i}\hat{K}|V_\nu\rangle$. Thus, $\boldsymbol{K}$ is not a restriction of a Hamiltonian.

**Kähler vs. non-Kähler.** On a non-Kähler manifold, where we have two inequivalent definitions of the equations of motion, it only makes sense to linearize the ones coming from the Lagrangian action principle, as their fixed point coincides with the approximate ground state. The resulting generator $K^\mu{}_\nu$ will in generally not commute with $\boldsymbol{J}$, even for Kähler manifolds, which has important consequences for its eigenvectors relevant for spectral functions.

### 4.2.3 Comparison: projection vs. linearization

In the following, we will compare the previously introduced approaches of approximating excitation energy. This comparison is particularly illuminating in the case of Kähler manifold.

At a stationary point, *i.e.*, $\partial_\mu E = 2\mathrm{Re}\,\langle V_\mu|\hat{H}|\psi\rangle = 0$, we consider the symplectic generator $K^\mu{}_\nu$ defined as

$$\boldsymbol{K}^\mu{}_\nu = -\boldsymbol{\Omega}^{\mu\sigma}(\partial_\sigma\partial_\nu E) = (\boldsymbol{J}^{-1})^\mu{}_\sigma\,\partial_\nu\!\left(\mathbb{P}^\sigma_\psi\hat{H}|\psi\rangle\right), \tag{126}$$

---

[16]Mathematically, the linearized flow $d\Phi_t$ is defined as the differential (also known as push-forward) of the flow map $\Phi_t$ defined after equation (87). In general it is a map from the tangent space $\mathcal{T}_{\psi(0)}\mathcal{M}$ to the tangent space $\mathcal{T}_{\psi(t)}\mathcal{M}$. In the special case of $|\psi_0\rangle$ being a fixed point of the time evolution, it reduces to a linear map from $\mathcal{T}_{\psi_0}\mathcal{M}$ onto itself. One can then show that this map is generated by the linearization $K^\mu{}_\nu$ of the vector field $\mathcal{X}^\mu$ that defines the evolution flow.

[17]Note that due to the 2-out-of-3 principle, any linear map $M$ satisfying two out of the three conditions $M\Omega M^\intercal = \Omega$, $MGM^\intercal = G$ and $MJM^{-1} = J$ will satisfy all three. Thus, any violation will necessarily affect at least two Kähler structures.

where we only have $J^{-1} = -J$ for Kähler manifolds. Evaluating the derivative in (126) gives the two pieces

$$\partial_\nu \big( \mathbb{P}_\psi^\sigma \hat{H} |\psi\rangle \big) = \mathbb{P}_\psi^\sigma \hat{H} |V_\nu\rangle + (\partial_\nu \mathbb{P}_\psi^\sigma) \hat{H} |\psi\rangle \,, \tag{127}$$

where we evaluate everything at $\psi_0$ after computing the derivatives. We recognize $H^\sigma{}_\nu = \mathbb{P}_\psi^\sigma \hat{H} |V_\nu\rangle$ and define $F^\sigma{}_\nu = (\partial_\nu \mathbb{P}_\psi^\sigma) \hat{H} |\psi\rangle = \frac{2}{\langle\psi|\psi\rangle} G^{\sigma\rho} \langle \partial_\nu V_\rho | \hat{H} | \psi_0 \rangle$ leading to

$$K^\mu{}_\nu = (J^{-1})^\mu{}_\sigma \Big( H^\sigma{}_\nu + F^\sigma{}_\nu \Big). \tag{128}$$

In summary, we see that the linearization $K^\mu{}_\nu$ consists of the two pieces: First, the projected Hamiltonian $H$ and second, the derivative of the projector. These terms are multiplied with the inverse complex structure $J^{-1}$.

In the case of a Kähler manifold there is a further way to understand these two term that make up $K^\mu{}_\nu$. In this case, we can use $J^2 = -\mathbb{1}$ to decompose any linear operator $K$ on $\mathcal{T}_{\psi_0}\mathcal{M}$ as $K = K_+ + K_-$ with

$$K_\pm = \tfrac{1}{2}(K \pm JKJ), \ \{K_+, J\} = 0, \ [K_-, J] = 0. \tag{129}$$

We will see that this decomposition coincides exactly with the one of $K^\mu{}_\nu$ in (128). To do this, we use the fact that a Kähler manifold of dimension $2n$ always admits[18] a parametrization $x^\mu = (x_1, \cdots, x_{2n})$ satisfying for $1 \le j \le n$

$$|V_j\rangle = \mathrm{i} |V_{n+j}\rangle \,, \tag{130}$$

i.e., the coordinate $x_j$ is conjugate to $x_{n+j}$. In this basis, $J$ and $\Omega$ are

$$J \equiv \begin{pmatrix} 0 & -\mathbb{1} \\ \mathbb{1} & 0 \end{pmatrix}, \quad \Omega \equiv \frac{1}{2} \begin{pmatrix} -\mathrm{Im}\,\eta^{-1} & -\mathrm{Re}\,\eta^{-1} \\ \mathrm{Re}\,\eta^{-1} & -\mathrm{Im}\,\eta^{-1} \end{pmatrix}, \tag{131}$$

where $\eta_{jk} = \langle V_j | V_k \rangle$. Then the structure of matrices that commute or anti-commute with $J$ is

$$K_- = \begin{pmatrix} a & b \\ -b & a \end{pmatrix}, \quad K_+ = \begin{pmatrix} a & b \\ b & -a \end{pmatrix}. \tag{132}$$

We can evaluate $K^\mu{}_\nu$ to find exactly this form

$$K^\mu{}_\nu = -\Omega^{\mu\rho} \partial_\rho \partial_\nu E = (K_+)^\mu{}_\nu + (K_-)^\mu{}_\nu, \tag{133}$$

where its two pieces are explicitly given by

$$K_+ \equiv \begin{pmatrix} \mathrm{Im}(\eta^{-1}h) & \mathrm{Re}(\eta^{-1}h) \\ -\mathrm{Re}(\eta^{-1}h) & \mathrm{Im}(\eta^{-1}h) \end{pmatrix}, \tag{134}$$

$$K_- \equiv \begin{pmatrix} \mathrm{Im}(\eta^{-1}f) & \mathrm{Re}(\eta^{-1}f) \\ \mathrm{Re}(\eta^{-1}f) & -\mathrm{Im}(\eta^{-1}f) \end{pmatrix}, \tag{135}$$

where $h_{jk} = \langle V_j | \hat{H} | V_k \rangle$ and $f_{jk} = \langle \partial_j V_k | \hat{H} | \psi_0 \rangle$. This clearly shows that the two pieces are given by $K_- = JH$ and $K_+ = JF$ as defined before (128).

In conclusion, from the decomposition (133) we immediately see again that $K^\mu{}_\nu$ has two contributions. One is related to the projected Hamiltonian $H^\mu{}_\nu$ and commutes with $J$. The

---

[18]This ultimately coincides with showing that a Kähler manifold is also a complex manifold, that is it admits a holomorphic parametrisation in terms of complex parameters $z^\alpha = \mathrm{x}^\alpha + \mathrm{i}\mathrm{y}^\alpha$.

other is related to the overlap of $\hat{H}|\psi_0\rangle$ with the *double tangent vectors* $|\partial_\alpha V_\beta\rangle$, which coincides with the one we previously described in terms of the derivative of the projector and anti-commutes with $J$. Thus $K_-$ is a complex linear map, while $K_+$ is a contribution that makes $K^\mu{}_\nu$ non-complex linear.

Finally, if we complexify tangent space, *i.e.*, treat complex linear combinations of $|V_\mu\rangle$ as linearly independent, there exists a basis transformation that makes $J$ diagonal and brings $K_+$ and $K_-$ respectively, into block diagonal and block off-diagonal form, given by

$$
J \equiv \mathrm{i} \begin{pmatrix} \mathbb{1} & 0 \\ 0 & -\mathbb{1} \end{pmatrix}, \quad K_- = \mathrm{i} \begin{pmatrix} -\eta^{-1}h & 0 \\ 0 & (\eta^{-1}h)^* \end{pmatrix}, \quad K_+ = \mathrm{i} \begin{pmatrix} 0 & (\eta^{-1}f)^* \\ -\eta^{-1}f & 0 \end{pmatrix}, \quad (136)
$$

*i.e.*, for Kähler manifolds the terms in (128) decouple nicely. For a non-Kähler manifold, neither $H$ nor $F$ may commute with $J$, but even the decomposition (129) will not work for $J^2 \neq -\mathbb{1}$.

In the next section, we will see that the term $K_-$ can be a blessing and a curse: on the one hand, it can ensure that in systems with spontaneously broken symmetry the eigenvalues of $K^\mu{}_\nu$ contain a Goldstone mode. On the other hand, for unfortunate choices of the variational family $\mathcal{M}$ we may encounter such massless modes even if there is no spontaneously broken symmetry (spurious Goldstone mode).

**Kähler vs. non-Kähler.** We can relate the linearization $K^\mu{}_\nu$ with the projected Hamiltonian $H^\mu{}_\nu$ via (128). For Kähler manifolds, this decomposition becomes particularly geometric, as the two pieces correspond to its complex linear and complex anti-linear part.

### 4.2.4 Spurious Goldstone mode

The spectrum of $K^\mu{}_\nu$ is not variational. In contrast to a variational approximation of an eigenstate, our eigenvector $|\mathcal{E}^\mu(\lambda)\rangle$ of $K^\mu{}_\nu$ with

$$
K^\mu{}_\nu \mathcal{E}^\nu(\lambda) = \lambda \mathcal{E}^\mu(\lambda), \tag{137}
$$

and $|\mathcal{E}(\lambda)\rangle = \mathcal{E}^\mu(\lambda)|V_\mu\rangle$, does not satisfy

$$
\lambda = \pm \langle \mathcal{E}(\lambda)|\mathrm{i}\hat{H}|\mathcal{E}(\lambda)\rangle. \tag{138}
$$

The expectation value of the full Hamiltonian with respect to $|\mathcal{E}(\lambda)\rangle$ is in general not easily related to $\lambda$, as it would be for a variational state. It is also not true that for every eigenvalue pair $\pm\mathrm{i}\omega_\ell$, there exists a true eigenstate $|E_\ell\rangle$ of $\hat{H}$ with excitation energy $E_\ell - E_0 \leq \omega_\ell$.

In fact, there are situations, where the true ground state $|E_0\rangle$ is non-degenerate, but $K^\mu{}_\nu$ still has a zero eigenvalue associated to a massless Goldstone mode. This typically occurs if we have a conserved quantity $\hat{A}$ with $[\hat{A}, \hat{H}] = 0$, such that $-\mathrm{i}\hat{A}|\psi\rangle \in \mathcal{T}_\psi \mathcal{M}$ everywhere as discussed in the context of Proposition 7. At this point, the question is if the global energy minimum $|\psi_0\rangle$ on $\mathcal{M}$ is invariant under $e^{-\mathrm{i}\hat{A}}$ or not. Whenever the global minimum on $\psi_0$ on $\mathcal{M}$ is not invariant, *i.e.*, there is a whole family $|\psi_0(\varphi)\rangle = e^{-\mathrm{i}\varphi\hat{A}}|\psi_0\rangle$ of approximate ground states, the generator $K^\mu{}_\nu$ will have a massless Goldstone mode

$$
\mathcal{E}^\mu_G = \mathbb{P}^\mu_{\psi_0}(-\mathrm{i}\hat{A})|\psi_0\rangle \quad \text{with} \quad K^\mu{}_\nu \mathcal{E}^\nu_G = 0. \tag{139}
$$

Whenever the true ground state $|E_0\rangle$ of the system is invariant under $e^{-\mathrm{i}\hat{A}}$, this Goldstone mode is spurious and merely an artefact of a spontaneous symmetry breaking on $\mathcal{M}$, but not on full $\mathcal{P}(\mathcal{H})$.

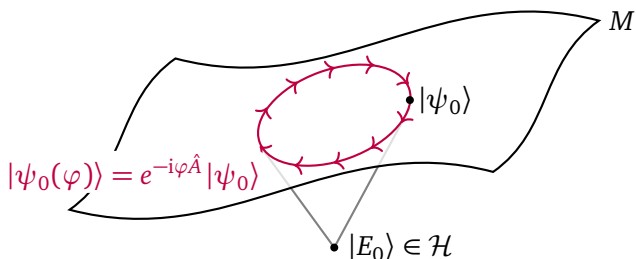

Figure 5: *Spurious Goldstone mode.* Even if the Hamiltonian has a unique ground state $|E_0\rangle$ without a spontaneously broken symmetry in full Hilbert space, on a chosen sub manifold $M$ there may be inequivalent states $|\psi_0(\varphi)\rangle = e^{-\mathrm{i}\varphi\hat{A}}|\psi_0\rangle$ that all minimize the energy. This leads to a spontaneous breaking of the symmetry generated by $\hat{A}$ and the appearance of a spurious Goldstone mode.

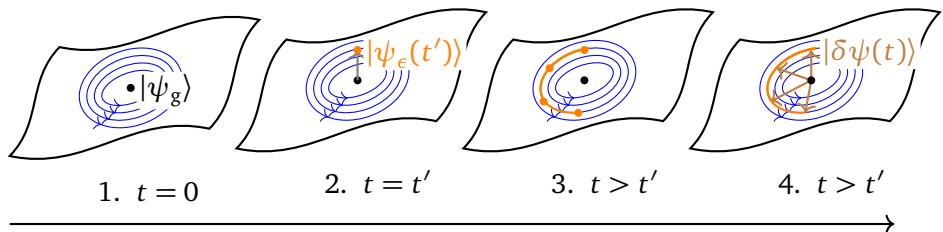

Figure 6: *Linear response theory.* We consider an approximate ground state $\psi_0 \in \mathcal{M}$. While $|\psi_0\rangle$ does not evolve in time, certain trajectories of nearby states are approximately elliptic. A finite perturbation at time $t'$ changes the state to $|\psi_\epsilon(t')\rangle = e^{\mathrm{i}\epsilon\hat{V}}|\psi_0\rangle$. This state will then evolve according to the equations of motion. We linearize by taking the limit $\epsilon \to 0$, where we find that the tangent vector $|\delta\psi(t)\rangle = \frac{d}{d\epsilon}|\psi_\epsilon(t)\rangle|_{\epsilon=0}$ can be decomposed into eigenvectors of $K^\mu{}_\nu$ that rotate on ellipses.

This was pointed out in [58] as an important problem of approximating the spectrum via linearized equations of motion rather than using the projected Hamiltonian. However, in [59] we found that this can also be desirable to capture features of the thermodynamic limit for finite system size. In particular, the gapless Bogoliubov excitation spectrum of the Bose-Hubbard model can be shown to result from the diagonalization of the generator (123) of the linearized equations of motion on the manifold of coherent states. The Bogoliubov spectrum is gapless independent from the system size, even though the true ground state $|E_0\rangle$ becomes only degenerate in the thermodynamic limit. This also extends from Bogoliubov theory to the full Gaussian time dependent variational principle, as discussed in [59].

We illustrate this issue in figure 5, where the Hamiltonian is spontaneously broken only on the variational manifold $\mathcal{M}$, but not in the full Hilbert space, where it has a unique ground state $|E_0\rangle$.

## 4.3 Spectral functions

Next, we would like to use the variational manifold $\mathcal{M}$ to estimate the spectral function of a system with respect to the perturbation operator $\hat{V}$.

Given a Hermitian operator $\hat{V}$, the spectral function is

$$\mathcal{A}(\omega) = -\frac{1}{\pi} \operatorname{Im} G^R(\omega), \tag{140}$$

where $G^R$ is the retarded Green's function

$$G^R(\omega) = -\mathrm{i} \int dt \, e^{\mathrm{i}\omega t} \Theta(t) \frac{\langle \psi_0 | [\hat{V}(t), \hat{V}] | \psi_0 \rangle}{\langle \psi_0 | \psi_0 \rangle}, \tag{141}$$

with $\Theta(t)$ being the step function and $\hat{V}(t)$ the Heisenberg evolved operator under the system Hamiltonian $\hat{H}$.

The definition in terms of the retarded Green's function stems from linear response theory. Indeed, let us suppose that a small external perturbing probe field $\epsilon\varphi(t)$ couples to our system through the operator $\hat{V}$. That is, the system state $|\psi_\epsilon(t)\rangle$ evolves under the perturbed Hamiltonian $\hat{H} + \epsilon\varphi(t)\hat{V}$. Then, let us measure the response of the system through the expectation value of the same observable $\hat{V}$. As the perturbation is ideally infinitesimally small, we only consider such response up to linear order in $\epsilon$. Consequently, we define the time-domain linear response as

$$\delta V(t) = \frac{d}{d\epsilon}\bigg|_{\epsilon=0} \frac{\langle \psi_\epsilon(t) | \hat{V} | \psi_\epsilon(t) \rangle}{\langle \psi_\epsilon(t) | \psi_\epsilon(t) \rangle}. \tag{142}$$

Then in frequency domain we have that

$$\delta\tilde{V}(\omega) \equiv \int dt \, e^{\mathrm{i}\omega t} \delta V(t) = \tilde{\varphi}(\omega) G^R(\omega). \tag{143}$$

That is, $G^R$ is exactly the so-called linear susceptibility of the system, which is an experimentally accessible quantity.

Given a variational manifold there are two possible paths to trying to approximate $\mathcal{A}(\omega)$.

1. We can calculate the quantity (142) after having projected the evolution of $|\psi_\epsilon(t)\rangle$ on the manifold. In other words, we perform linear response theory directly on the variational manifold. This leads us to express $\mathcal{A}(\omega)$ in terms of the eigendecomposition of the generator of linearized real time evolution $K^\mu{}_\nu$ introduced in (123).

2. Alternatively, one can try to approximate on the manifold the quantity

$$e^{-\mathrm{i}\hat{H}t} \hat{V} | \psi_0 \rangle, \tag{144}$$

   that appears in equation (141). In this case one should note that in general $\hat{V} | \psi_0 \rangle$ does not belong to the variational manifold, so one has to perform some truncation even before applying the time evolution operator $e^{-\mathrm{i}\hat{H}t}$. The other subtlety here is that one must make sure that the quantity (144) is calculated with the correct global phase, as we explained in Section 4.1.3.

It seems to us that method 2 captures less the spirit of variational manifolds. Indeed one has that the quantity $\hat{V} | \psi_0 \rangle$ would morally represent a small perturbation around the ground-state $|\psi_0\rangle$ and would thus naturally live in the tangent space to the manifold of states at $|\psi_0\rangle$. Representing $\hat{V} | \psi_0 \rangle$ as a vector of $\mathcal{M}$ therefore is only meaningful if the manifold itself is a good representation of its own tangent space. But this is not true for general manifolds and indeed there is no uniquely defined method for representing $\hat{V} | \psi_0 \rangle$ on $\mathcal{M}$. The first method, on the other hand, can alternatively be thought of precisely as representing the perturbations

generated by $\hat{V}$ on $\mathcal{T}_{\psi_0}\mathcal{M}$. Furthermore, we will show that method 1 leads to a closed expression for the spectral function from which it is immediate to see that $\text{sgn}\mathcal{A}(\omega) = \text{sgn}\,\omega$ (as it is in the full Hilbert space), while this cannot be shown in general for method 2.

For these reasons in the next subsections we will focus on the details of the first method, giving a final expression for the spectral function estimated in this way in Proposition 10.

### 4.3.1 Linear response theory

As mentioned, a possible way of calculating spectral functions is to perform linear response theory directly on the variational manifold. In this subsection we will then briefly explain how this can be done. The idea is illustrated in figure 6.

Let us consider a possibly time-dependent perturbation $\hat{A}(t)$ of our unperturbed Hamiltonian $\hat{H}_0$, such that $\hat{H}_\epsilon(t) = \hat{H}_0 + \epsilon\hat{A}(t)$, and an observable $\hat{B}$, whose response we are interested in. For spectral functions, we will be interested in the particular case where $\hat{A}(t) = \varphi(t)\hat{V}$ for arbitrary functions $\varphi(t)$ and $\hat{B} = \hat{V}$, but we will for the moment keep our treatment general.

Our perturbed Hamiltonian gives rise to the time dependent real time evolution vector field $\mathcal{X}_\epsilon^\mu(t)$, which is

$$\mathcal{X}_\epsilon(t) = \mathcal{X}_0 + \epsilon\,\mathcal{X}_A(t), \tag{145}$$

where $\mathcal{X}_0$ and $\mathcal{X}_A(t)$ are the evolution vector fields associated to the Hamiltonians $\hat{H}_0$ and $\hat{A}(t)$ respectively. The solution of this perturbed evolution is $|\psi_\epsilon(t)\rangle$ satisfying

$$\mathbb{Q}_\psi \frac{d}{dt}|\psi_\epsilon(t)\rangle = \mathcal{X}_\epsilon^\mu(t)|V_\mu\rangle. \tag{146}$$

For the following Proposition 8 it would not be important whether the evolution vector field is defined according to the Lagrangian or McLachlan variational principles, as long as it has the form (145). However, later we will be interested in the case in which the perturbed evolution happens around the approximate ground state $|\psi_0\rangle$ and it will be important that this state is also a fixed point of the time evolution. So, as was the case in Section 4.2.2, from now on we will suppose that the evolution vector fields are defined according to the Lagrangian evolution (88).

We are interested in the response in expectation value of the observable $\hat{B}$ at linear order in $\epsilon$, that is

$$\delta B(t) = \frac{d}{d\epsilon}\left.\frac{\langle\psi_\epsilon(t)|\hat{B}|\psi_\epsilon(t)\rangle}{\langle\psi_\epsilon(t)|\psi_\epsilon(t)\rangle}\right|_{\epsilon=0} = \delta x^\mu(t)\,\partial_\mu B(x(t)), \tag{147}$$

where we defined the propagated perturbation

$$\delta x^\mu(t)|V_\mu\rangle = \mathbb{Q}_\psi \frac{d}{d\epsilon}|\psi_\epsilon(t)\rangle\bigg|_{\epsilon=0} \in \mathcal{T}_{\psi(t)}\mathcal{M}, \tag{148}$$

which can be evaluated as follows.

**Proposition 8.** *Given a variational manifold $\mathcal{M}$ we define (according to the Lagrangian action principle) the free projected real time evolution $|\psi(t)\rangle$ as governed by the free Hamiltonian $\hat{H}_0$ and the perturbed projected real time evolution $|\psi_\epsilon(t)\rangle$ as governed by the perturbed Hamiltonian $\hat{H}_\epsilon(t) = \hat{H}_0 + \epsilon\hat{A}(t)$, both with the same initial state $|\psi(0)\rangle$. Then, the propagated perturbation, defined according to (148), is given by*

$$\delta x^\mu(t) = -\int_{-\infty}^{t} dt'\,(d\Phi_{t-t'})^\mu{}_\nu\,\mathbf{\Omega}^{\nu\rho}\,\partial_\rho A(t')\big|_{\psi(t')}, \tag{149}$$

*where $d\Phi_t$ is the linearized free evolution flow[19].*

*Proof.* This can be shown in a standard way by using the interaction representation. We sketch a proof in Appendix B. □

Put simply, $\delta x^\mu$ is the superposition of all propagated perturbations, *i.e.*, a perturbation

$$-\boldsymbol{J}^\nu{}_\rho \mathbb{P}^\rho_{\psi(t')} \hat{A}(t') |\psi(t')\rangle = -\boldsymbol{\Omega}^{\nu\rho} \left. \partial_\rho A \right|_{\psi(t')} \tag{150}$$

at time $t'$ is evolved with the linearized free evolution $d\Phi_{t-t'}$ to time $t$ where it contributes towards $\delta x^\mu(t)$.

If we now take as initial state $|\psi(0)\rangle$ the approximate ground state $|\psi_0\rangle$, that is a fixed point of the projected evolution, we have that the free evolution is trivial $|\psi(t)\rangle = |\psi_0\rangle$. It also follows that $d\Phi_t$ is a linear map from $\mathcal{T}_{\psi_0}\mathcal{M}$ onto itself given by

$$d\Phi_t = e^{Kt}, \tag{151}$$

where $K^\mu{}_\nu$ is the generator of the linearized flow introduced in (123). The map $d\Phi_t$ can therefore be evaluated in terms of the spectral decomposition of $K^\mu{}_\nu$.

**Proposition 9.** *The linear response to a perturbation $\hat{A}(t)$, measured in terms of the observable $\hat{B}$, for a system initially in the state $\psi_0 \in \mathcal{M}$ is given by*

$$\delta B(t) = -i \sum_\ell \mathrm{sgn}(i\lambda_\ell) [\mathcal{E}^\mu(\lambda_\ell)\partial_\mu B \int_{-\infty}^t dt' \, e^{\lambda_\ell(t-t')} [\mathcal{E}^\nu(\lambda_\ell)\partial_\nu A(t')]^*, \tag{152}$$

*where all derivatives are evaluated at $|\psi_0\rangle$ and $\mathcal{E}^\mu(\lambda_\ell)$ is an eigenvector of $K^\mu{}_\nu$ such that*

$$K^\mu{}_\nu \mathcal{E}^\nu(\lambda_\ell) = \lambda_\ell \mathcal{E}^\mu(\lambda_\ell), \tag{153}$$

*and normalised so that $\mathcal{E}^\mu(\lambda_\ell)\boldsymbol{\omega}_{\mu\nu}\mathcal{E}^\nu(\lambda_\ell)^* = i\,\mathrm{sgn}(i\lambda_\ell)$.*

*Proof.* We can always decompose $K^\mu{}_\nu$ in terms eigenvectors $\mathcal{E}^\mu(\lambda)$ with eigenvalues $\lambda$ and dual eigenvectors[20] $\widetilde{\mathcal{E}}_\mu(\lambda)$, such that

$$K^\mu{}_\nu = \sum_\ell \lambda_\ell \, \mathcal{E}^\mu(\lambda_\ell) \widetilde{\mathcal{E}}_\nu(\lambda_\ell). \tag{154}$$

The eigenvalues $\lambda_\ell$ will come in conjugate pairs $\pm i\omega_\ell$, which implies that the associated eigenvectors and dual eigenvectors are complex and mathematically speaking lie the complexified tangent space. However, as $K^\mu{}_\nu$ is a real map, we must have $\mathcal{E}^\mu(i\omega) = \mathcal{E}^\mu(-i\omega)^*$.
We then notice that $\Omega^{\mu\nu}\widetilde{\mathcal{E}}_\nu(-i\omega)$ is an eigenvector of $K$ with eigenvalue $i\omega$. To see this it is sufficient to apply $K$ to it and use the symplectic property $K\Omega = -\Omega K^\mathsf{T}$. It is then always possible to normalize the eigenvectors $\mathcal{E}^\mu$ such that the relation $\Omega^{\mu\nu}\widetilde{\mathcal{E}}_\nu(-i\omega) = -i\,\mathrm{sgn}(\omega)\mathcal{E}^\mu(i\omega) = -i\,\mathrm{sgn}(\omega)\mathcal{E}^\mu(-i\omega)^*$ holds.[21] From this, inverting $\Omega$ and exploiting its antisymmetry, we have $\widetilde{\mathcal{E}}_\mu(-i\omega) = i\,\mathrm{sgn}(\omega)\mathcal{E}^\nu(-i\omega)^*\boldsymbol{\omega}_{\nu\mu}$.
Using this and (154), we can rewrite (151) as

$$(d\Phi_t)^\mu{}_\nu = i \sum_\ell \mathrm{sgn}(i\lambda_\ell) \, e^{\lambda_\ell t} \, \mathcal{E}^\mu(\lambda_\ell) \mathcal{E}^\rho(\lambda_\ell)^* \, \boldsymbol{\omega}_{\rho\nu}. \tag{155}$$

Combining this with (147) and (149) we have (152). □

---

[19] See footnote 16.

[20] The dual vector $\widetilde{\mathcal{E}}_\mu(\lambda)$ is defined by $\widetilde{\mathcal{E}}_\mu(\lambda)\mathcal{E}^\mu(\lambda') = \delta_{\lambda,\lambda'}$

[21] Doing this rescaling while maintaining the property $\mathcal{E}^\mu(i\omega) = \mathcal{E}^\mu(-i\omega)^*$ is actually only possible if $\mathcal{E}^\mu(-i\omega)\boldsymbol{\omega}_{\mu\nu}\mathcal{E}^\nu(i\omega) = ia$ with $a > 0$, $\forall \omega > 0$. But this is always true because by definition $K = -\Omega h$, where $h_{\mu\nu} = \partial_\mu\partial_\nu E$ is positive definite (Hessian at a local minimum). It follows that $-\boldsymbol{\omega}K > 0$ and therefore $0 < -\mathcal{E}^\mu(i\omega)^*\boldsymbol{\omega}_{\mu\rho}K^\rho{}_\nu\mathcal{E}^\nu(i\omega) = -i\omega\mathcal{E}^\mu(-i\omega)\boldsymbol{\omega}_{\mu\nu}\mathcal{E}^\nu(i\omega) = \omega a$.

### 4.3.2 Spectral response

To calculate spectral functions we now just need to evaluate the result (152) for $\hat{A}(t) = \varphi(t)\hat{V}$ and $\hat{B} = \hat{V}$ and then take the Fourier transform.

**Proposition 10.** *The spectral function with respect to the perturbation operator $\hat{V}$, estimated by performing linear response theory on the variational manifold $\mathcal{M}$, is*

$$\mathcal{A}(\omega) = \text{sgn}(\omega) \sum_\ell \left| \mathcal{E}^\mu(\mathrm{i}\omega_\ell) \, \partial_\mu V \right|^2 \delta(\omega - \omega_\ell), \tag{156}$$

*where $\mathcal{E}^i(\mathrm{i}\omega_\ell)$ are the eigenvectors of $K^\mu{}_\nu$, normalized such that $\mathcal{E}^\mu(\mathrm{i}\omega_\ell)^* \, \boldsymbol{\omega}_{\mu\nu} \mathcal{E}^\nu(\mathrm{i}\omega_\ell) = \mathrm{i}\,\text{sgn}(\omega_\ell)$, and the sum runs over all possible values of $\omega_\ell$ (appearing in pairs of opposite signs).*

*Proof.* Evaluating the Fourier transform of (152) and comparing with (143) leads us to the estimate for the retarded Green's function

$$
\begin{aligned}
G^R(\omega) &= -\mathrm{i} \sum_\ell \text{sgn}(\omega_\ell) \left| \mathcal{E}^\mu(\mathrm{i}\omega_\ell) \, \partial_\mu V \right|^2 \int dt \, e^{\mathrm{i}(\omega - \omega_\ell)t} \Theta(t) \\
&= \sum_\ell \text{sgn}(\omega_\ell) \left| \mathcal{E}^\mu(\mathrm{i}\omega_\ell) \, \partial_\mu V \right|^2 \left[ P \frac{1}{\omega - \omega_\ell} - \mathrm{i}\pi\delta(\omega - \omega_\ell) \right],
\end{aligned}
\tag{157}
$$

where the Sokhotski-Plemelj formula has been used. The imaginary part of this expression can be then be inserted into the definition of the spectral function (140), leading to the result (156). $\qquad\square$

Spectral functions calculated in this way have the desirable property $\text{sgn}\,\mathcal{A}(\omega) = \text{sgn}\,\omega$.

**Kähler vs. non-Kähler.** On a non-Kähler manifold, where we have two inequivalent definitions of the equations of motion, it only makes sense to perform linear response theory with the ones coming from the Lagrangian action principle, as their fixed point coincides with the approximate ground state.

## 4.4 Imaginary time evolution

In the previous sections we have assumed we knew the state $|\psi_0\rangle$ that minimizes the energy on the variational manifold $\mathcal{M}$. Solving this optimization problem is often non-trivial and different methods may be more appropriate in different situations. However, we would like here to present a method, known as projected imaginary time evolution, that makes use of the same geometric notions introduced in Section 4.1 for real time evolution.

On full Hilbert space, imaginary time evolution is

$$\frac{d}{d\tau} |\psi(\tau)\rangle = -(\hat{H} - E(\tau)) |\psi(\tau)\rangle, \tag{158}$$

which can be integrated to the solution

$$|\psi(\tau)\rangle = \frac{e^{-\hat{H}\tau} |\psi(0)\rangle}{\sqrt{\langle\psi(0)|e^{-2\hat{H}\tau}|\psi(0)\rangle}}. \tag{159}$$

This will converge in the limit $\tau \to \infty$ to a true ground state if and only if the initial state $|\psi(0)\rangle$ had some non-zero overlap with the ground state space.

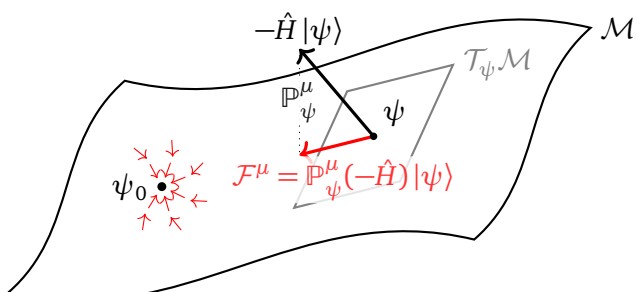

Figure 7: *Imaginary time evolution.* We illustrate the imaginary time evolution vector field $\mathcal{F}^\mu$ on the variational family $\mathcal{M}$ analogously to figure 3, which is given by the orthogonal projection of $-\hat{H}\,|\psi\rangle$ through $\mathbb{P}_\psi$ onto $\mathcal{M}$. This vector field flows towards the global minimum $\psi_0$ of the energy function.

Given a variational manifold $\mathcal{M}$, we can approximate imaginary time evolution on it and hope that it will also converge to the approximate ground state $|\psi_0\rangle$. This can be done by projecting (158) onto tangent space. Contrary to real time evolution, there does not exist a formulation of imaginary time evolution in terms of an action principle, so the projection can only be done according to the McLachlan minimal error principle.

We would like to minimize the local projection error

$$\left\| \tfrac{d}{d\tau}\,|\psi(\tau)\rangle - (E - \hat{H})\,|\psi(\tau)\rangle \right\|\,, \tag{160}$$

imposing that $\tfrac{d}{d\tau}\,|\psi(\tau)\rangle \in \mathcal{T}_\psi \mathcal{M}$, which leads to

$$\tfrac{d}{d\tau}\,|\psi(\tau)\rangle = \mathbb{P}_{\psi(\tau)}(E - \hat{H})\,|\psi(\tau)\rangle = -\mathbb{P}_{\psi(\tau)}\hat{H}\,|\psi(\tau)\rangle\,, \tag{161}$$

where we used $\mathbb{P}_\psi\,|\psi\rangle = 0$.

This leads to the projected evolution equation

$$\frac{dx^\mu}{d\tau}\,|V_\mu\rangle = -\mathbb{P}_{\psi(\tau)}\hat{H}\,|\psi(\tau)\rangle\,, \tag{162}$$

from which we can define the imaginary time evolution vector field $\mathcal{F}^\mu$ everywhere on $\mathcal{M}$, such that

$$\frac{dx^\mu}{d\tau} = \mathcal{F}^\mu(x) = -\mathbb{P}^\mu_{\psi(x)}\hat{H}\,|\psi(x)\rangle\,. \tag{163}$$

This vector field can be understood as follows.

**Proposition 11.** *Given a manifold $\mathcal{M}$, the projected imaginary time evolution is given by*

$$\frac{dx^\mu}{d\tau} = \mathcal{F}^\mu(x) = -\boldsymbol{G}^{\mu\nu}(\partial_\nu E)\,, \tag{164}$$

*where $E(x)$ is the energy function, defined in the context of equation (72). Its solution $x(\tau)$ monotonically decreases the energy.*

*Proof.* We apply the projector $\mathbb{P}^\mu_\psi$ in (163) to find

$$\mathcal{F}^\mu = -\mathbb{P}^\mu\hat{H}\,|\psi\rangle = -\frac{2}{\langle\psi|\psi\rangle}\boldsymbol{G}^{\mu\nu}\,\mathrm{Re}\,\langle V_\nu|\hat{H}\,|\psi\rangle\,. \tag{165}$$

We simplify this by using (74). Plugging this back into the previous equations, we arrive at (164). To show that the energy monotonically decreases, we find

$$\frac{dE}{d\tau} = (\partial_\mu E)\frac{dx^\mu}{d\tau} = -(\partial_\mu E)\,\boldsymbol{G}^{\mu\nu}\,(\partial_\nu E) \leq 0\,, \tag{166}$$

which follows from the positivity of $\boldsymbol{G}^{\mu\nu}$. $\qquad\square$

We thus recognize projected imaginary time evolution (164) as gradient descent of the energy function $E(x)$ with respect to the natural geometry encoded in the metric $\boldsymbol{G}^{\mu\nu}$ on the manifold $\mathcal{M}$, as illustrated in figure 7. It is our experience that solving (164) numerically has better convergence properties than performing a naive gradient descent, where we just try to minimize the energy $E(x)$ as a function of $x^\mu$ assuming a flat metric. Hence, when replacing the fixed time step by a line search, imaginary time evolution becomes equivalent to Riemannian gradient descent. Indeed, the literature on Riemannian optimization [60–63] describes how the Riemannian geometry (*i.e.*, the metric) of a manifold can be taken into account in each of the standard optimization algorithms such as the gradient descent method, Newton's method, the conjugate gradient method, and quasi-Newton methods such as the (limited memory) Broyden–Fletcher–Goldfarb–Shanno scheme [64–67].

**Kähler vs. non-Kähler.** The results discussed in this section do not rely on the manifold $\mathcal{M}$ being a Kähler manifold. The McLachlan projection principle is the only one that can be resonably defined for imaginary time evolution and leads to the desirable gradient descent result for any real differentiable manifold, independently of the Kähler property.

### 4.4.1 Conserved quantities

In many situations, one would like to further constrain our variational manifold by requiring that certain operators $\hat{A}_I$ have fixed expectation values $A_I$. Geometrically, this amounts to restricting the search to the submanifold

$$\widetilde{\mathcal{M}} = \left\{ \psi \in \mathcal{M} \,\big|\, \tfrac{\langle\psi|\hat{A}_I|\psi\rangle}{\langle\psi|\psi\rangle} = A_I \,\forall\, I \right\} \subset \mathcal{M}\,. \tag{167}$$

For example, for Hamiltonians commuting with the total particle number operator $\hat{N}$, one often wants to find lowest energy state within an eigenspace of $\hat{N}$ with $\hat{N}\,|\psi\rangle = N\,|\psi\rangle$. To approximate such a state on a variational manifold $\mathcal{M}$, we can search for minimal energy state on the submanifold of states with $\langle\hat{N}\rangle = N$.

In general, this manifold $\widetilde{\mathcal{M}}$ will not satisfy the Kähler property anymore. In particular, if we only fix a single expectation value, we will generically reduce the dimension of a Kähler $\mathcal{M}$ to an odd dimension, which cannot be again a Kähler manifold. However, we have seen that for the purpose of finding the state of minimal energy, we can apply formula (158) on the reduced manifold, regardless of the Kähler property.

Instead of finding a new parametrization of the reduced variational manifold, as long as we choose an initial state for the imaginary time evolution that satisfies the desired constraints, we can then just implement them locally. We can indeed modify the imaginary time evolution vector field $\mathcal{F}$ by further projecting it onto the restricted tangent space $\mathcal{T}\widetilde{\mathcal{M}}$. In this way the respective expectation values are preserved by construction.

If there are several quantities $\hat{A}_I$ that we wish to fix, $\mathcal{T}_\psi\widetilde{\mathcal{M}}$ is given by the sub tangent space orthogonal to the span of $X_I^\mu = \mathbb{P}_\psi^\mu \hat{A}_I\,|\psi\rangle$. To project onto it, we define

$$\widetilde{g}_{IJ} = X_I^\mu\,\boldsymbol{g}_{\mu\nu}X_J^\nu\,, \tag{168}$$

which gives rise to the projector

$$\widetilde{P}^{\mu}{}_{\nu} = \delta^{\mu}{}_{\nu} - X^{\mu}_I \, \widetilde{G}^{IJ} X^{\rho}_J \, g_{\rho\nu}, \tag{169}$$

where $\widetilde{G}^{IJ}$ is the inverse of $\widetilde{g}_{IJ}$ (or pseudo inverse, if not all constraints are independent). The modified imaginary time evolution vector field is then

$$\widetilde{\mathcal{F}}^{\mu} = \widetilde{P}^{\mu}{}_{\nu} \mathcal{F}^{\nu}, \tag{170}$$

which will conserve all the expectation values $A_I(\tau)$. In analogy to (109), this is equivalent to

$$\widetilde{\mathcal{F}}^{\mu} = -\widetilde{G}^{\mu\nu}(\partial_\nu E) \quad \text{with} \quad \widetilde{G}^{\mu\nu} = \widetilde{P}^{\mu}{}_{\sigma}\widetilde{P}^{\nu}{}_{\rho}G^{\sigma\rho}\,. \tag{171}$$

If we want to fix the expectation value of the number operator $\hat{N}$, we have the scalar function $N(x) = \langle \hat{N} \rangle$ with $X^{\mu} = \mathbb{P}^{\mu}_{\psi}\hat{N}\,|\psi\rangle = G^{\mu\nu}\partial_\nu N$, such that

$$\widetilde{\mathcal{F}}^{\mu} = \mathcal{F}^{\mu} - \frac{G^{\mu\nu}(\partial_\nu N)}{(\partial_\sigma N)G^{\sigma\rho}(\partial_\rho N)}(\partial_\lambda N)\mathcal{F}^{\lambda}\,, \tag{172}$$

which clearly satisfies $\frac{dN}{d\tau} = (\partial_\mu N)\widetilde{\mathcal{F}}^{\mu} = 0$.

# 5 Applications

In this section we will apply the formalism developed in this paper to some examples of variational manifolds. The aim is to illustrate how different Kähler and non-Kähler structures can arise in practice.

In section 5.1 we will discuss a very common and natural example of Kähler manifolds, namely bosonic and fermionic Gaussian states. Indeed, in the case of Gaussian states, Kähler structures arise naturally directly in the way these states can be parametrized. We will introduce them using a formalism that stresses this aspect. In sections 5.2 and 5.3 we will focus on states of the form $|\psi(x)\rangle = \mathcal{U}(x)|\phi\rangle$ where $\mathcal{U}(x)$ is subset of unitary transformations and $|\phi\rangle$ an appropriately chosen reference state. We will show how this class of states can give rise to manifolds that violate the Kähler conditions to different degrees.

## 5.1 Gaussian states

We consider pure bosonic and fermionic Gaussian states, which are also known as quasi-free states. Bosonic Gaussian states (squeezed coherent states) form a prominent variational family in the study of bosonic systems, such as Bose-Einstein condensates [68], cold atoms in optical lattices [59] and photonic systems [69]. Fermionic Gaussian states (generalized Hartree-Fock states, including Slater determinants) are equally important for the study of fermionic systems, including Bardeen–Cooper–Schrieffer theory [3] and the Hartree-Fock method [7]. Other applications range from field theory [70], continuous variable quantum information [71], relativistic quantum information [72] and quantum fields in curved spacetime [73]. Most importantly, they are completely determined by their one- and two-point function and they satisfy Wick's theorem. Interestingly, pure Gaussian states are in one-to-one correspondence to compatible Kähler structures $(g_{ab}, \omega_{ab}, J^a{}_b)$ on the classical phase space. These Kähler structures are distinct from those $(g_{\mu\nu}, \omega_{\mu\nu}, J^{\mu}{}_{\nu})$ on tangent space, but we will derive relations between them.

While the relationship between pure Gaussian states and Kähler structures has been pointed out before in the context of quantum fields in curved spacetime [73–76], the linear complex structure $J$ was only recently used as a convenient parametrization of states. This allowed for a unified formulation of bosonic and fermionic Gaussian states [77,78], which enabled new results in the context of their entanglement dynamics [79–81], the typicality of energy eigenstate entanglement [82–84] and in the study of circuit complexity in quantum field theory [85,86].

### 5.1.1 Definition

The theory of $N$ bosonic or fermionic degrees of freedom can be constructed on the classical phase space $V \simeq \mathbb{R}^{2N}$. We denote a phase space vector by $\xi^a \in V$ and a dual vector by $w_a \in V^*$, where we use the Latin indices $a, b, c, d, e$ for phase space to distinguish from tangent space indices $\mu, \nu, \sigma, \lambda$. We will see in a moment how this relates to standard bosonic and fermionic creation and annihilation operators.

The classical bosonic phase space $V$ is always equipped with symplectic form $\Omega^{ab}$ and its inverse $\omega_{ab}$ satisfying $\Omega^{ac}\omega_{cb} = \delta^a{}_b$, which define the classical Poisson brackets and then give rise to the canonical commutation relations (CCR). Similarly, one can define classical fermionic phase space $V$ by equipping it with a positive definite metric $G^{ab}$ and its inverse $g_{ab}$ satisfying $G^{ac}g_{cb} = \delta^a{}_b$ giving rise to the canonical anti-commutation relations (CAR).

We can always choose a basis of classical linear observables $\xi^a \stackrel{q,p}{\equiv} (q_1, \cdots, q_N, p_1, \cdots, p_N)$, which can be bosonic or fermionic, such that $(\omega, \Omega)$ or $(g, G)$, respectively, take their standard form given by

$$
\begin{aligned}
\omega_{ab} &\stackrel{q,p}{\equiv} \begin{pmatrix} 0 & -\mathbb{1} \\ \mathbb{1} & 0 \end{pmatrix}, \quad \Omega^{ab} \stackrel{q,p}{\equiv} \begin{pmatrix} 0 & \mathbb{1} \\ -\mathbb{1} & 0 \end{pmatrix}, \quad \textbf{(bosons)} \\
g_{ab} &\stackrel{q,p}{\equiv} \begin{pmatrix} \mathbb{1} & 0 \\ 0 & \mathbb{1} \end{pmatrix}, \quad G^{ab} \stackrel{q,p}{\equiv} \begin{pmatrix} \mathbb{1} & 0 \\ 0 & \mathbb{1} \end{pmatrix}, \quad \textbf{(fermions)}
\end{aligned}
\tag{173}
$$

where we will use the symbol $\stackrel{q,p}{\equiv}$ throughout this manuscript to indicate that the RHS of the equation show the vector or matrix representation with respect to the above standard basis.

Quantization promotes these linear observables $\xi^a$ to Hermitian quantum operators

$$
\hat{\xi}^a \stackrel{q,p}{\equiv} (\hat{q}_1, \cdots, \hat{q}_N, \hat{p}_1, \cdots, \hat{p}_N)
\tag{174}
$$

satisfying the commutation or anti-commutation relations

$$
\begin{aligned}
[\hat{\xi}^a, \hat{\xi}^b] &= i\Omega^{ab}, \quad \textbf{(bosons)} \\
\{\hat{\xi}^a, \hat{\xi}^b\} &= G^{ab}. \quad \textbf{(fermions)}
\end{aligned}
\tag{175}
$$

We refer to $\hat{\xi}^a$ as quadratures for bosons and as Majorana modes for fermions. These relations ensure that the bosonic or fermionic creation and annihilation operators $\hat{a}_i = (\hat{q}_i + i\hat{p}_i)/\sqrt{2}$ and $\hat{a}_i^\dagger = (\hat{q}_i - i\hat{p}_i)/\sqrt{2}$ will satisfy their standard commutation or anti-commutation relations, respectively.

Given a normalized quantum state $|\psi\rangle$, we define its one and two point correlation functions as

$$
z^a = \langle\psi|\hat{\xi}^a|\psi\rangle ,
\tag{176}
$$

$$
C_2^{ab} = \langle\psi|(\hat{\xi} - z)^a(\hat{\xi} - z)^b|\psi\rangle .
\tag{177}
$$

The fact that $\hat{\xi}^a$ is Hermitian implies that $z^a$ is real. This does not apply to $C_2^{ab}$, which we therefore decompose into its real and imaginary part given by

$$C_2^{ab} = \frac{1}{2}\left(G^{ab} + i\Omega^{ab}\right). \tag{178}$$

It is easy to verify that $G^{ab}$ is real, symmetric and positive definite with inverse $g_{ab}$ and $\Omega^{ab}$ is anti-symmetric. For bosons, $\Omega^{ab}$ is fixed by the commutation relations, while for fermions, $G^{ab}$ is fixed by the anti-commutation relations. With respect to our basis $\hat{\xi}^a \equiv (\hat{q}_1, \cdots, \hat{q}_N, \hat{p}_1, \cdots, \hat{p}_N)$, they are given by (173). We define the covariance matrix

$$\Gamma^{ab} = \begin{cases} G^{ab} = C_2^{ab} + C_2^{ba} & \textbf{(bosons)} \\ \Omega^{ab} = -i(C_2^{ab} - C_2^{ba}) & \textbf{(fermions)} \end{cases}, \tag{179}$$

which is the only part of $C_2^{ab}$ that depends on $|\psi\rangle$. Later on, we will use $\Gamma^{ab}$ as parameters of our variational manifold.

We define the linear complex structure $J$ of $|\psi\rangle$ as

$$J^a{}_b = -G^{ac}\omega_{cb}, \tag{180}$$

as before. Note, however, that for fermions $\Omega^{ab}$ may not be invertible, in which case we define $\omega_{ab}$ as its unique pseudoinverse with respect to $G^{ab}$, *i.e.*, we invert $\Omega^{ab}$ on the orthogonal complement of its kernel. We define

$$|\psi\rangle \text{ is Gaussian} \quad \Leftrightarrow \quad J^2 = -\mathbb{1}. \tag{181}$$

Put differently, pure bosonic and fermionic Gaussian states are those states, for which $(g_{ab}, \omega_{ab}, J^a{}_b)$ are compatible Kähler structures as defined in section 3. For fermions, this also implies[22] $z^a = 0$, while for bosons $z^a$ is a free parameter, which we call the displacement. We denote a pure Gaussian state with displacement $z^a$ and complex structure $J$ by $|J, z\rangle$, which is uniquely determined (up to normalization and phase) by requiring

$$\frac{1}{2}(\delta^a{}_b + iJ^a{}_b)(\hat{\xi}^b - z^b)|J, z\rangle = 0. \tag{182}$$

Given a Gaussian state $|J, z\rangle$, we define the operators

$$\hat{\xi}_\pm^a = \frac{1}{2}(\delta^a{}_b \mp iJ^a{}_b)(\hat{\xi}^b - z^b), \tag{183}$$

which satisfy $\hat{\xi}^a = \hat{\xi}_+^a + \hat{\xi}_-^a + z^a$ and $\hat{\xi}_-|J, z\rangle = 0$. Put differently, we project onto the eigenspaces of $J$ with eigenvalues $\pm i$ and displaced by $z$. These spaces, *i.e.*, complex linear combinations of $\hat{\xi}_+^a$ or $\hat{\xi}_-^a$, correspond to the spaces of creation and annihilation operators, respectively. Most importantly, $\hat{\xi}_\pm^a$ satisfy the commutation and anti-commutation relations given by

$$[\hat{\xi}_\pm^a, \hat{\xi}_\pm^b] = 0, \qquad\qquad [\hat{\xi}_\mp^a, \hat{\xi}_\pm^b] = C_2^{ab}, \quad \textbf{(bosons)} \tag{184}$$

$$\{\hat{\xi}_\pm^a, \hat{\xi}_\pm^b\} = 0, \qquad\qquad \{\hat{\xi}_\mp^a, \hat{\xi}_\pm^b\} = C_2^{ab}. \quad \textbf{(fermions)} \tag{185}$$

Using these relations together with $\hat{\xi}_-|J, z\rangle = 0$, one can show the equivalence of (177) and (182). Moreover, one can show that any higher order correlation function

$$C_n^{a_1 \cdots a_n} = \langle J, z|(\hat{\xi} - z)^{a_1} \ldots (\hat{\xi} - z)^{a_n}|J, z\rangle, \tag{186}$$

with $n > 2$ can be computed purely from $C_2^{ab}$ via Wick's theorem, which states the following:

---

[22]There exist fermionic coherent states [87], where $z^a$ is a Grassmann number, which we do not consider here.

(a) Odd correlation functions vanish, *i.e.*, $C_{2n+1} = 0$.

(b) Even correlation functions are given by the sum over all two-contractions

$$C_{2n}^{a_1 \cdots a_{2n}} = \sum_{\sigma} \frac{|\sigma|}{n!} C_2^{a_{\sigma(1)} a_{\sigma(2)}} \cdots C_2^{a_{\sigma(2n-1)} a_{\sigma(2n)}}, \tag{187}$$

where the permutations $\sigma$ satisfy $\sigma(2i-1) < \sigma(2i)$ and $|\sigma| = 1$ for bosons and $|\sigma| = \operatorname{sgn}(\sigma)$, called parity, for fermions.

This enables us to compute arbitrary expectation values of observables $\mathcal{O}$ written as polynomials in $\hat{\xi}^a$ efficiently.

**Example 11.** *The harmonic oscillator $\hat{H} = \frac{1}{2}(\hat{p}^2 + \omega^2 \hat{q}^2)$ with standard position and momentum operators $[\hat{q}, \hat{p}] = i$ has ground state wave function $\psi(q) = \omega^{1/4} e^{-\omega q^2 / 2}$ from which we find the one- and two point functions*

$$z^a = 0, \quad C_2^{ab} = \frac{1}{2}(G^{ab} + i\Omega^{ab}) \equiv \frac{1}{2} \begin{pmatrix} 1/\omega & i \\ -i & \omega \end{pmatrix}. \tag{188}$$

*This leads to the linear complex structure*

$$J^a{}_b = -G^{ac} \omega_{cb} \equiv \begin{pmatrix} 0 & 1/\omega \\ -\omega & 0 \end{pmatrix}. \tag{189}$$

*We can compute $\hat{\xi}_-^a = \frac{1}{2}(\mathbb{1} + iJ)^a{}_b (\hat{\xi} - z)^b$ as*

$$\hat{\xi}_-^a = \frac{1}{2} \begin{pmatrix} 1 & i/\omega \\ -i\omega & 1 \end{pmatrix} \begin{pmatrix} \hat{q} \\ \hat{p} \end{pmatrix} = \frac{1}{\sqrt{2\omega}} \begin{pmatrix} \hat{a} \\ -i\omega \hat{a} \end{pmatrix}, \tag{190}$$

*where $\hat{a} = \frac{1}{2\sqrt{\omega}}(\omega \hat{q} + i\hat{p})$.*

**Example 12.** *The fermionic oscillator of a single degree of freedom is given by $\hat{H} = \omega(\hat{n} - \frac{1}{2})$, where $\hat{n} = \hat{a}^\dagger \hat{a}$ with $\hat{a}^\dagger$ and $\hat{a}$ being fermionic creation and annihilation operators. We can go to Majorana modes $q = \frac{1}{\sqrt{2}}(\hat{a}^\dagger + \hat{a})$ and $\hat{p} = \frac{i}{\sqrt{2}}(\hat{a}^\dagger - \hat{a})$ which satisfy $q^2 = p^2 = \frac{1}{2}$. There are only two distinct fermionic Gaussian states given by $|0\rangle$ and $|1\rangle$. The associated complex structures are*

$$J_+ \overset{q,p}{\equiv} \begin{pmatrix} 0 & 1 \\ -1 & 0 \end{pmatrix} \quad \text{and} \quad J_- \overset{q,p}{\equiv} \begin{pmatrix} 0 & -1 \\ 1 & 0 \end{pmatrix}, \tag{191}$$

*respectively. We have the annihilation operators $\hat{\xi}_-^a \equiv \frac{1}{\sqrt{2}}(\hat{a}, i\hat{a})$ for the state $|0\rangle$ and $\hat{\xi}_-^a \equiv \frac{1}{\sqrt{2}}(\hat{a}^\dagger, -i\hat{a}^\dagger)$ for $|1\rangle$.*

In summary, we showed how Kähler structures can be used as unifying framework to treat bosonic and fermionic Gaussian states on an equal footing. In particular, we can label pure Gaussian states $|J, z\rangle$ by their associated complex structure $J$, rather than using their covariance matrix. In praxis, we can quickly switch between the different Kähler structures using their relations as reviewed in appendix A.4.

### 5.1.2 Gaussian transformations

There is a subgroup of unitary transformations that map pure Gaussian states onto themselves. We refer to them as Gaussian transformations and we will see that they are given by the exponential of linear and quadratic operators in terms of $\hat{\xi}^a$.

In this context, we will encounter the symplectic and orthogonal Lie groups and algebras. We identify them with linear maps $M^a{}_b$ and $K^a{}_b$ on the classical phase space $V$. Using the symplectic form $\Omega^{ab}$ (bosons) and the metric $G^{ab}$ (fermions), we define the symplectic and orthogonal group $\mathcal{G}$ as

$$\begin{aligned}
\mathrm{Sp}(2N,\mathbb{R}) &= \{M^a{}_b \in \mathrm{GL}(2N,\mathbb{R}) | M\Omega M^\mathsf{T} = \Omega\}, \\
\mathrm{O}(2N,\mathbb{R}) &= \{M^a{}_b \in \mathrm{GL}(2N,\mathbb{R}) | MGM^\mathsf{T} = G\},
\end{aligned} \tag{192}$$

respectively, with their associated Lie algebras $\mathfrak{g}$

$$\begin{aligned}
\mathfrak{sp}(2N,\mathbb{R}) &= \{K^a{}_b \in \mathfrak{gl}(2N,\mathbb{R}) | K\Omega + \Omega K^\mathsf{T} = 0\}, \\
\mathfrak{so}(2N,\mathbb{R}) &= \{K^a{}_b \in \mathfrak{gl}(2N,\mathbb{R}) | KG + GK^\mathsf{T} = 0\}.
\end{aligned} \tag{193}$$

Group elements $M \in \mathcal{G}$ are equivalent to the well-known Bogoliubov transformations. To see this relation, we can pick a basis $\hat{\xi}^a \overset{q,p}{\equiv} (\hat{q}_1,\ldots,\hat{q}_N,\hat{p}_1,\ldots\hat{q}_N)$ of Hermitian operators satisfying (175) and transform it to a new basis $\hat{\xi}'^a \equiv M^a{}_b \hat{\xi}^b$. This fixes a transformation $\hat{a}'_i = \sum_{j=1}^N (\alpha_{ij}\hat{a}_j + \beta_{ij}\hat{a}_j^\dagger)$, which now takes the known standard form of a Bogoliubov transformation.

We can represent Lie algebra elements $K$ faithfully as anti-Hermitian quadratic operators $\widehat{K}$ given by

$$K^a{}_b \quad \Longleftrightarrow \quad \widehat{K} = \begin{cases} -\frac{\mathrm{i}}{2}\omega_{ac}K^c{}_b\hat{\xi}^a\hat{\xi}^b & \textbf{(bosons)} \\ \frac{1}{2}g_{ac}K^c{}_b\hat{\xi}^a\hat{\xi}^b & \textbf{(fermions)} \end{cases}. \tag{194}$$

Using the canonical commutation, we can verify

$$\widehat{[K_1,K_2]} = [\widehat{K_1},\widehat{K_2}], \tag{195}$$

*i.e.*, our quadratic operators $\widehat{K}$ form a representation of the Lie algebra $\mathfrak{g}$. The respective exponentials of $\widehat{K}$ give rise to a projective representation of the associated Lie group $\mathcal{G}$, *i.e.*, we have the identification

$$M = e^K \quad \Longleftrightarrow \quad \mathcal{S}(M) = e^{\widehat{K}}, \tag{196}$$

between Lie group elements $M$ and unitary operators $\mathcal{S}(M)$, which we refer to as squeezing transformations.

For bosons, we also have displacement transformations

$$z^a \quad \Longleftrightarrow \quad \mathcal{D}(z) = \exp\left(\mathrm{i}z^a\omega_{ab}\hat{\xi}^b\right), \tag{197}$$

*i.e.*, we identify a phase space vector $z \in V$ with the respective unitary operator $\mathcal{D}(z)$.

For fermions, products of $M = e^K$ for $K \in \mathfrak{so}(2N,\mathbb{R})$ will only generate the subgroup $\mathrm{SO}(2N,\mathbb{R})$, whose group elements satisfy $\det M = 1$. To generate other group elements $M \in \mathrm{O}(2N,\mathbb{R})$ with $\det M = -1$, we can take any dual vector $v_a \in V^*$ satisfying $v_a G^{ab} v_b = 2$ to define

$$\mathcal{S}(M_v) = v_a\hat{\xi}^a, \qquad \textbf{(fermions)} \tag{198}$$

representing $(M_v)^a{}_b = v_c G^{ca} v_b - \delta^a{}_b \in \mathrm{O}(2N, \mathbb{R})$ with $\det M_v = -1$. We can further check that $\mathcal{S}(M_v)$ is unitary. Moreover, we have $\mathcal{S}^\dagger(M_v)\hat{\xi}^a \mathcal{S}(M_v) = (M_v)^a{}_b \hat{\xi}^b$. Consequently, together $\mathcal{S}(e^K)$ and $\mathcal{S}(M_v)$ for a single chosen $v_a$ generate the full orthogonal group $\mathrm{O}(2N, \mathbb{R})$, *i.e.*, every element $M \in \mathrm{O}(2N, \mathbb{R})$ with $\det M = -1$ can be represented as a $\mathcal{S}(M) \simeq \mathcal{S}(e^K)\mathcal{S}(M_v)$ for a fixed $v_a$ and $K = \log M M_v^{-1}$.

Displacement and squeezing transformations form projective representations of vector addition in $V$ and group multiplication in $\mathcal{G}$. We define Gaussian transformations

$$\mathcal{U}(M, z) \cong \mathcal{D}(z)\mathcal{S}(M), \tag{199}$$

where "$\cong$" refers to equality up to a complex phase. Using Baker-Campbell-Hausdorff, it is not difficult to show

$$\mathcal{U}^\dagger(M, z)\hat{\xi}^a \mathcal{U}(M, z) = M^a{}_b \hat{\xi}^b + z^a. \tag{200}$$

Therefore, it is a projective representation with

$$\mathcal{U}(M, z)\mathcal{U}(\tilde{M}, \tilde{z}) \cong \mathcal{U}(M\tilde{M}, z + M\tilde{z}), \tag{201}$$

where $(M\tilde{z})^a = M^a{}_b \tilde{z}^b$. The action of $\mathcal{U}(M, z)$ onto a Gaussian state is particularly simple and given by

$$\mathcal{U}(M, z)\,|J_0, z_0\rangle \cong |MJ_0 M^{-1}, Mz_0 + z\rangle. \tag{202}$$

Given a linear complex structure $J_0$, there is a unique group of transformations $M$ preserving $J_0$, *i.e.*,

$$\mathrm{GL}(N, \mathbb{C}) = \{M^a{}_b \in \mathrm{GL}(2N, \mathbb{R}) | MJ_0 M^{-1} = J_0\}. \tag{203}$$

This group is indeed the complex linear group $\mathrm{GL}(N, \mathbb{C})$, because the complex structure $J_0$ can turn $V \simeq \mathbb{R}^{2N}$ into a complex vector space $V \simeq \mathbb{C}^N$ with scalar multiplication $\odot : (\mathbb{C}, V) \to V$ satisfying $(\alpha + \mathrm{i}\beta) \odot v = \alpha v + \beta J_0 v$. With this in mind, a transformation $M \in \mathrm{GL}(N, \mathbb{C})$ can be seen as a linear map on $V \simeq \mathbb{C}^N$ that commute with the representation $J_0$ of the imaginary unit, *i.e.*, $MJ_0 = J_0 M$.

Consequently, the subgroup of $\mathcal{G}$ which also preserves $J_0$ is the intersection

$$\mathcal{G} \cap \mathrm{GL}(N, \mathbb{C}) = \mathrm{U}(N). \tag{204}$$

This turns out to be the group of transformations that preserve all Kähler structures $(G, \Omega, J_0)$ on $V$, which are just the unitary transformations preserving the Kähler induced Hermitian inner product on $V$ given by

$$\langle z, \tilde{z}\rangle = \frac{1}{2}(g_{ab} + \mathrm{i}\omega_{ab})z^a \tilde{z}^b. \tag{205}$$

Given a Gaussian state $|J_0, 0\rangle$, we can reach another[23] Gaussian state $|J, z\rangle$ by applying the transformation

$$\mathcal{U}(e^K, z) \cong \mathcal{D}(z)\,e^{\hat{K}} \quad \text{with} \quad K = \frac{1}{2}\log \Delta, \tag{206}$$

where we introduced the relative covariance matrix

$$\Delta^a{}_b = -J^a{}_c (J_0)^c{}_b = \Gamma^{ac}(\Gamma_0)^{-1}_{cb}. \tag{207}$$

Above transformation follows from $J = e^K J_0 e^{-K}$.

---

[23]For bosons, all Gaussian states are connected and can be reached. For fermions, there exist two disconnected sets of Gaussian states, which cannot be continuously connected.

**Example 13.** *The squeezing transformations of a single bosonic mode $\hat{\xi}^a \stackrel{q,p}{\equiv} (\hat{q},\hat{p})$ are $\mathcal{G} = \mathrm{Sp}(2,\mathbb{R})$, generated by*

$$X \stackrel{q,p}{\equiv} \begin{pmatrix} 1 & 0 \\ 0 & -1 \end{pmatrix}, \quad Y \stackrel{q,p}{\equiv} \begin{pmatrix} 0 & 1 \\ 1 & 0 \end{pmatrix}, \quad Z \stackrel{q,p}{\equiv} \begin{pmatrix} 0 & 1 \\ -1 & 0 \end{pmatrix}, \tag{208}$$

*with their associated quadratic operators*

$$\widehat{X} = -\mathrm{i}\frac{\hat{q}\hat{p}+\hat{p}\hat{q}}{2}, \quad \widehat{Y} = \mathrm{i}\frac{\hat{q}^2-\hat{p}^2}{2}, \quad \widehat{Z} = -\mathrm{i}\frac{\hat{q}^2+\hat{p}^2}{2}. \tag{209}$$

*The Gaussian state $|J_0,0\rangle$ is preserved by $u \in \mathrm{U}(1)$ with*

$$J_0 \stackrel{q,p}{\equiv} \begin{pmatrix} 0 & 1 \\ -1 & 0 \end{pmatrix} \quad and \quad u \stackrel{q,p}{\equiv} \begin{pmatrix} \cos\varphi & \sin\varphi \\ -\sin\varphi & \cos\varphi \end{pmatrix}, \tag{210}$$

*where $\mathrm{U}(1) = \mathrm{Sp}(2,\mathbb{R}) \cap \mathrm{GL}(1,\mathbb{C})$ and $u = e^{\varphi Z}$ with $[Z,J_0] = 0$. The most general Gaussian state of one bosonic mode has complex structure*

$$J \stackrel{q,p}{\equiv} \begin{pmatrix} -\cos\phi\sinh\rho & \sin\phi\sinh\rho + \cosh\rho \\ \sin\phi\sinh\rho - \cosh\rho & \cos\phi\sinh\rho \end{pmatrix}, \tag{211}$$

*for which we can verify $J^2 = -\mathbb{1}$. From the relative complex structure $\Delta = -JJ_0$, we compute the generator*

$$K = \frac{1}{2}\log\Delta \stackrel{q,p}{\equiv} \frac{\rho}{2} \begin{pmatrix} \sin\phi & \cos\phi \\ \cos\phi & -\sin\phi \end{pmatrix}, \tag{212}$$

*such that $e^{\widehat{K}}|J_0,0\rangle \cong |J,0\rangle$.*

**Example 14.** *The squeezing transformations of a single fermionic mode $\hat{\xi}^a \stackrel{q,p}{\equiv} (\hat{q},\hat{p})$ are $\mathcal{G} = \mathrm{O}(2,\mathbb{R})$, generated by*

$$X \stackrel{q,p}{\equiv} \begin{pmatrix} 0 & 1 \\ -1 & 0 \end{pmatrix} \quad \Longleftrightarrow \quad \widehat{X} = \frac{\hat{q}\hat{p}-\hat{p}\hat{q}}{2} \tag{213}$$

*and our choice of the additional group element*

$$M_v \stackrel{q,p}{\equiv} \begin{pmatrix} 1 & 0 \\ 0 & -1 \end{pmatrix} \quad \Longleftrightarrow \quad \mathcal{S}(M_v) = \sqrt{2}\,\hat{q} \tag{214}$$

*with $v_a \equiv (\sqrt{2},0)$. As seen in example 12, there are exactly two distinct complex structures given by $J_+$ and $J_-$ from (191) which are related by $J_+ = M_v J_- M_v^{-1}$ and which both satisfy $uJu^{-1} = J$ for $u = e^X$. This is because $u$ is the generator $\mathrm{SO}(2,\mathbb{R})$ which is identical to the subgroup $\mathrm{U}(1)$ preserving $J_\pm$.*

**Example 15.** *The squeezing transformations of two fermionic modes $\hat{\xi}^a \stackrel{q,p}{\equiv} (\hat{q}_1,\hat{q}_2,\hat{p}_1,\hat{p}_2)$ are $\mathcal{G} = \mathrm{O}(4,\mathbb{R})$ with six linearly independent generators $K_i$ satisfying $KG + GK^\mathsf{T} = 0$. Given the linear complex structure*

$$J_0 \stackrel{q,p}{\equiv} \begin{pmatrix} & & 1 & \\ & & & 1 \\ -1 & & & \\ & -1 & & \end{pmatrix}, \tag{215}$$

there is a 4-dimensional subspace of these generators also satisfying $[K, J_0]$, which generates $\mathrm{U}(2) \subset \mathrm{O}(4, \mathbb{R})$. The most general fermionic complex structure is

$$
J_\pm \overset{q,p}{\equiv} \begin{pmatrix} 0 & \mp \sin\theta \sin\phi & \pm \cos\theta & \pm \sin\theta \cos\phi \\ \pm \sin\theta \sin\phi & 0 & -\sin\theta \cos\phi & \cos\theta \\ \mp \cos\theta & \sin\theta \cos\phi & 0 & \sin\theta \sin\phi \\ \mp \sin\theta \cos\phi & -\cos\theta & -\sin\theta \sin\phi & 0 \end{pmatrix}. \tag{216}
$$

We have $J_+ = M J_0 M^{-1}$ and $J_- = M_v M J_0 M^{-1} M_v$ for $v_a \equiv (\sqrt{2}, 0, 0, 0)$ and $M = e^K$ with

$$
K = \frac{1}{2} \log \Delta \overset{q,p}{\equiv} \frac{\theta}{2} \begin{pmatrix} 0 & \cos\phi & 0 & \sin\phi \\ -\cos\phi & 0 & -\sin\phi & 0 \\ 0 & \sin\phi & 0 & -\cos\phi \\ -\sin\phi & 0 & \cos\phi & 0 \end{pmatrix} \tag{217}
$$

for $\Delta = J_+ J_0$. From the explicit form of $J_\pm$, we see that $(\theta, \phi)$ behave as spherical coordinates. This agrees with the fact that the manifold of fermionic Gaussian states with two modes consists of two disjoint spheres.

In summary, Gaussian transformations consist of squeezing (bosons and fermions) and displacements (only bosons). The former changes both the complex structure (equivalently: covariance matrices) and displacements, while the latter only displaces the state. Whenever we choose a Gaussian state $|J_0, z_0\rangle$, it defines with the respective background structure (symplectic form for bosons or metric for fermions) a subgroup $\mathrm{U}(N)$ of transformations in $\mathcal{G}$ that preserve $J_0$.

### 5.1.3 Geometry of variational family

We will now shift gears. Rather than looking at the Kähler structures $(g_{ab}, \omega_{ab}, J^a{}_b)$ associated to an individual Gaussian state $|J, z\rangle$, we now consider the full variational family $\mathcal{M}$ of pure bosonic or fermionic Gaussian states to study the Kähler structures $(\boldsymbol{g}_{\mu\nu}, \boldsymbol{\omega}_{\mu\nu}, \boldsymbol{J}^\mu{}_\nu)$ on tangent space $\mathcal{T}_\psi \mathcal{M}$. For this, we will need to compute the tangent basis vectors $|V_\mu\rangle$, which we will split into two types $|V_a\rangle$ and $|V_{ab}\rangle$.

We introduced Gaussian states $|J, z\rangle$ as being uniquely (up to a complex phase) characterized by their complex structure $J$ and their displacement vector $z$. Consequently, a general tangent vector is characterized by the pair $(\delta J^a{}_b, \delta z^a)$ describing the respective changes of $J^a{}_b$ and $z^a$. However, the resulting expressions simplify if we express everything in terms of the (bosonic or fermionic) covariance matrix $\Gamma^{ab}$ as defined in (179). We will therefore label states by $|\Gamma, z\rangle$ and tangent vectors by $(\delta\Gamma, \delta z)$ with

$$
|\delta\Gamma, \delta z\rangle = \delta\Gamma^{ab} |V_{ab}\rangle + \delta z^a |V_a\rangle \in \mathcal{H}^\perp_{|\Gamma, z\rangle}. \tag{218}
$$

The tangent space $\mathcal{T}_{(\Gamma, z)} \mathcal{M}$ thus decomposes as

$$
\mathcal{T}_{(\Gamma, z)} = \mathcal{S}_{(\Gamma, z)} \oplus \mathcal{D}_{(\Gamma, z)}, \tag{219}
$$

where $\mathcal{S}_{(\Gamma, z)}$ corresponds to the changes of $J$ and $\mathcal{D}_{(\Gamma, z)}$ refers to the changes of $z$ (only for bosons). We will see that these spaces can be naturally identified with the space of one- and two-particle excitations in Hilbert space (constructed as Fock space).

**Squeezing tangent space $\mathcal{S}_{(\Gamma, z)}$.** The tangent space of squeezings is described by variations $\delta\Gamma^{ab}$ of the covariance matrix. Note that such changes are constrained to preserve the complex

structure property $J^2 = -\mathbb{1}$ and reflect the symmetry/antisymmetry of $\Gamma$ for bosons/fermions, respectively. We therefore have

$$
\begin{aligned}
\delta\Gamma^{ab} = \delta\Gamma^{ba}, && \delta\Gamma J^\intercal = J\delta\Gamma, && \textbf{(bosons)} \\
\delta\Gamma^{ab} = -\delta\Gamma^{ba}, && \delta\Gamma J^\intercal = J\delta\Gamma. && \textbf{(fermions)}
\end{aligned}
\tag{220}
$$

The tangent vectors $|V_{ab}\rangle = \mathbb{Q}_{(\Gamma,z)} \frac{\partial}{\partial\Gamma^{ab}} |\Gamma,z\rangle$ are then[24]

$$
|V_{ab}\rangle = \begin{cases} \frac{i}{4} g_{ac}\omega_{bd}\hat{\xi}_+^c\hat{\xi}_+^d |\Gamma,z\rangle & \textbf{(bosons)} \\ \frac{1}{4} g_{ac}\omega_{bd}\hat{\xi}_+^c\hat{\xi}_+^d |\Gamma,z\rangle & \textbf{(fermions)} \end{cases} .
\tag{221}
$$

The set of tangent vectors $|V_{ab}\rangle$ is overcomplete, but this does not cause any problems as for any change $\delta\Gamma^{ab}$, the associated tangent vector is given by $|\delta\Gamma\rangle = \delta\Gamma^{ab} |V_{ab}\rangle$. Using Wick's theorem, we can compute the inner product

$$
\langle\delta\Gamma|\delta\tilde{\Gamma}\rangle = \frac{1}{16}(g_{ac}g_{bd} + i\omega_{ac}g_{bd})\delta\Gamma^{ab}\delta\tilde{\Gamma}^{cd} .
\tag{222}
$$

We thus see that the inner product between squeezing tangent vectors can be computed from the Kähler structure $(g, \omega, J)$ on the classical phase space $V$.

**Displacement tangent space** $\mathcal{D}_{(\Gamma,z)}$. The tangent space of displacements is described by the variations $\delta z^a$ of $z^a$, which can be changed freely. Therefore, we can identify the tangent space $\mathcal{T}_{|\Gamma,z\rangle}$ with the classical phase space $V$, i.e., $\delta z^a \in V$. The local frame $|V_a\rangle$ is

$$
|V_a\rangle = \mathbb{Q}_{(\Gamma,z)} \frac{\partial}{\partial z^a} |\Gamma,z\rangle = i\omega_{ab}\hat{\xi}_+^b |\Gamma,z\rangle .
\tag{223}
$$

We find the inner product

$$
\langle\delta z|\delta\tilde{z}\rangle = \frac{1}{2}(g_{ab} - i\omega_{ab})z^a\tilde{z}^b ,
\tag{224}
$$

which implies that the tangent space is isomorphic to $V$ embedded with metric $g_{ab}$ and symplectic form $-\omega_{ab}$.[25] The spaces $\mathcal{S}_{(\Gamma,z)}$ and $\mathcal{D}_{(\Gamma,z)}$ are orthogonal, because $\langle V_a|V_{bc}\rangle = 0$ follows from $C_3^{abc} = 0$ in Wick's theorem.

**Kähler structures on tangent space.** The tangent space of bosonic Gaussian states can be decomposed into the direct sum of displacements and squeezings, which are orthogonal due to $\langle V_{ab}|V_c\rangle = 0$. The tangent space of fermionic Gaussian states, only consists of the squeezings. Evaluating the respective inner products gives

$$
\begin{aligned}
\langle\delta\Gamma, \delta z|\delta\tilde{\Gamma}, \delta\tilde{z}\rangle = &\frac{1}{16}(g_{ce}g_{df} - i\omega_{ce}g_{df})\delta\Gamma^{cd}\delta\tilde{\Gamma}^{ef} \\
&+ \frac{1}{2}(g_{ab} - i\omega_{ab})\delta z^a\delta\tilde{z}^b ,
\end{aligned}
\tag{225}
$$

giving rise to the Kähler structures

$$
\boldsymbol{g}_{\mu\nu} \equiv (+g_{ab}) \oplus (+\tfrac{1}{8}g_{ce} \otimes g_{df}),
\tag{226}
$$

$$
\boldsymbol{\omega}_{\mu\nu} \equiv (-\omega_{ab}) \oplus (-\tfrac{1}{8}\omega_{ce} \otimes g_{df}).
\tag{227}
$$

---

[24] The most general tangent vector of the squeezing manifold is given by $|V_K\rangle = \mathbb{Q}_{(\Gamma,z)}\widehat{K}|\Gamma,z\rangle$, where $\widehat{K}$ is a general quadratic operator from (194). We can then relate $|V_K\rangle$ to the change $\delta\Gamma = 2\text{Re}\langle\Gamma,z|(\hat{\xi}^a\hat{\xi}^b + \hat{\xi}^b\hat{\xi}^a - 2z^az^b)|V_K\rangle$ for bosons and $\delta\Gamma = 2\text{Im}\langle\Gamma,0|(\hat{\xi}^a\hat{\xi}^b - \hat{\xi}^b\hat{\xi}^a)|V_K\rangle$ for fermions to find (221). For bosonic states $|\Gamma,z\rangle$ with $z^a \neq 0$, $|V_K\rangle$ also has a component $|\delta z\rangle = \delta z^a |V_a\rangle$ with $\delta z^a = K^a{}_b z^b$ in the displacement tangent space $\mathcal{D}_{(\Gamma,z)}$, discussed next.

[25] The sign difference is due to our chosen conventions and related to the issue discussed in footnote 26.

Here, we do *not* have $\langle V_{ab}|V_{cd}\rangle = \frac{1}{16}(g_{ac}g_{bd} - i\omega_{ac}g_{bd})$, but only $\langle\delta\Gamma|\delta\tilde{\Gamma}\rangle = \frac{1}{16}(g_{ac}g_{bd} - i\omega_{ac}g_{bd})\delta\Gamma^{ab}\delta\tilde{\Gamma}^{cd}$, where $\delta\Gamma^{ab}$ needs to satisfy (220). Inverting them on this subspace leads to the dual Kähler structures

$$
\begin{aligned}
\boldsymbol{G}^{\mu\nu} &\equiv (+G^{ab}) \oplus (+8G^{ce} \otimes G^{df}), \\
\boldsymbol{\Omega}^{\mu\nu} &\equiv (-\Omega^{ab}) \oplus (-8\Omega^{ce} \otimes G^{df}), \\
\boldsymbol{J}^{\mu}{}_{\nu} &\equiv (-J^{a}{}_{b}) \oplus (-J^{c}{}_{e} \otimes \delta^{d}{}_{f}).
\end{aligned}
\tag{228}
$$

They are defined on the dual subspace spanned by $(w_a, W_{ab})$ with $W_{ab}$ satisfying

$$
\begin{aligned}
W_{ab} &= W_{ba}, & WJ &= J^{\mathsf{T}}W, & \textbf{(bosons)} \\
W_{ab} &= -W_{ba}, & WJ &= J^{\mathsf{T}}W, & \textbf{(fermions)}
\end{aligned}
\tag{229}
$$

dual to (220). Given a general $W_{ab}$, we can define its projection $\lfloor W \rfloor$ onto this subspace as

$$
\lfloor W \rfloor_{ab} = \begin{cases} \frac{1}{2}(W - J^{\mathsf{T}}WJ)_{(ab)} & \textbf{(bosons)} \\ \frac{1}{2}(W - J^{\mathsf{T}}WJ)_{[ab]} & \textbf{(fermions)} \end{cases},
\tag{230}
$$

with $W_{(ab)} = \frac{1}{2}(W_{ab} + W_{ba})$ and $W_{[ab]} = \frac{1}{2}(W_{ab} - W_{ba})$ referring to symmetrization and anti-symmetrization, respectively. The resulting projection $W_{ab}$ satisfies $\lfloor W \rfloor_{ab}\delta\Gamma^{ab} = W_{ab}\delta\Gamma^{ab}$ for $\delta\Gamma^{ab}$ satisfying (229).

We find the action of i onto $|\delta\Gamma, \delta z\rangle$ to be

$$
\mathrm{i}|\delta\Gamma, \delta z\rangle = -J^{a}{}_{c}\delta\Gamma^{cb}|V_{ab}\rangle - J^{c}{}_{d}\delta z^{d}|V_c\rangle = |-J\delta\Gamma, -J\delta z\rangle.
\tag{231}
$$

**Example 16.** *We consider the Gaussian state $|J_0, 0\rangle$ of a single bosonic mode $\hat{\xi}^a \equiv (\hat{q}, \hat{p})$ with Kähler structures*

$$
J_0 \stackrel{q,p}{\equiv} \begin{pmatrix} 0 & 1 \\ -1 & 0 \end{pmatrix}, \quad G \stackrel{q,p}{\equiv} \begin{pmatrix} 1 & 0 \\ 0 & 1 \end{pmatrix}, \quad \Omega \stackrel{q,p}{\equiv} \begin{pmatrix} 0 & 1 \\ -1 & 0 \end{pmatrix}.
\tag{232}
$$

*A change $\delta\Gamma$ of the covariance matrix $\Gamma = G$ is constrained to satisfy (220), such that we have*

$$
\delta\Gamma \stackrel{q,p}{\equiv} \begin{pmatrix} a & b \\ b & -a \end{pmatrix} \quad \text{and} \quad \delta z \stackrel{q,p}{\equiv} \begin{pmatrix} c \\ d \end{pmatrix}.
\tag{233}
$$

*The associated tangent vectors are given by*

$$
\begin{aligned}
|V_{11}\rangle &\equiv -|V_{22}\rangle \equiv \frac{(\hat{a}^\dagger)^2}{8}|J_0, 0\rangle, & |V_1\rangle &= \frac{\hat{a}^\dagger}{\sqrt{2}}|J_0, 0\rangle, \\
|V_{12}\rangle &\equiv |V_{21}\rangle \equiv \frac{\mathrm{i}(\hat{a}^\dagger)^2}{8}|J_0, 0\rangle, & |V_2\rangle &= \frac{\mathrm{i}\hat{a}^\dagger}{\sqrt{2}}|J_0, 0\rangle,
\end{aligned}
\tag{234}
$$

*which leads to a general tangent vector*

$$
|\delta\Gamma, \delta z\rangle = \left[\frac{1}{4}(a + \mathrm{i}b)(\hat{a}^\dagger)^2 + \frac{1}{\sqrt{2}}(c + \mathrm{i}d)\hat{a}^\dagger\right]|J_0, 0\rangle.
\tag{235}
$$

**Example 17.** *We saw in example 14 that fermionic Gaussian states for $N = 1$ consists of two points and thus is zero dimensional. Instead, we consider for $N = 2$ the state $|\Gamma, 0\rangle$ and allowed change $\delta\Gamma$ with*

$$
\Gamma \stackrel{q,p}{\equiv} \begin{pmatrix} & & 1 & \\ & & & 1 \\ -1 & & & \\ & -1 & & \end{pmatrix}, \quad \delta\Gamma \stackrel{q,p}{\equiv} \begin{pmatrix} & a & & b \\ -a & & -b & \\ & b & & -a \\ -b & & a & \end{pmatrix}.
\tag{236}
$$

*The non-zero tangent vectors are then*

$$|V_{13}\rangle \equiv |V_{42}\rangle \equiv -|V_{24}\rangle \equiv -|V_{31}\rangle \equiv \frac{i\hat{a}_1^\dagger \hat{a}_2^\dagger}{8}|\Gamma, 0\rangle \,, \tag{237}$$

$$|V_{14}\rangle \equiv |V_{23}\rangle \equiv -|V_{32}\rangle \equiv -|V_{41}\rangle \equiv -\frac{\hat{a}_1^\dagger \hat{a}_2^\dagger}{8}|\Gamma, 0\rangle \,, \tag{238}$$

*where we recall* $|V_{ab}\rangle \equiv -|V_{ba}\rangle$. *We thus find*

$$|\delta\Gamma, 0\rangle = -\tfrac{1}{2}(a + ib)i\hat{a}_1^\dagger \hat{a}_2^\dagger |\Gamma, 0\rangle \,. \tag{239}$$

In summary, the Kähler structure $(g_{ab}, \omega_{ab}, J^a{}_b)$ on the classical phase space $V$ are intimately linked to the ones $(\boldsymbol{g}_{\mu\nu}, \boldsymbol{\omega}_{\mu\nu}, \boldsymbol{J}^\mu{}_\nu)$ on tangent space. For bosons, we saw that the displacement tangent space $\mathcal{D}_{|J,z\rangle}$ can be naturally identified with phase space $V$ as Kähler space.

### 5.1.4 Variational methods

After having clarified the Kähler geometry of bosonic and fermionic Gaussian states, we can use them to illustrate the variational methods discussed in section 4. Applications include (A) real time evolution, (B) excitation spectra, (D) spectral functions and (D) imaginary time evolution.

**Real time evolution.** Based on (88), we have the Lagrangian evolution equation $\frac{dx^\mu}{dt} = -\Omega^{\mu\nu}\frac{\partial E}{\partial x^\nu}$. As Gaussian states are parametrized by $x^\mu = (\Gamma, z)$, we need to compute $\frac{\partial E}{\partial \Gamma^{ab}}$ and $\frac{\partial E}{\partial z^a}$. As $\Omega^{\mu\nu}$ is only defined on the subspace satisfying (229), we need to project $\frac{\partial E}{\partial \Gamma}$ onto this subspace to find $\frac{d}{dt}\Gamma^{ab} = 8\Omega^{ac}G^{bd}\lfloor\frac{\partial E}{\partial \Gamma}\rfloor_{cd}$ and thus[26]

$$\begin{aligned}
\frac{d}{dt}z^a &= X^a = \Omega^{ab}\frac{\partial E}{\partial z^b}\,, \\
\frac{d}{dt}\Gamma^{ab} &= X^{ab} = 4\Omega^{ac}\left(\frac{\partial E}{\partial \Gamma} - J^\intercal\frac{\partial E}{\partial \Gamma}J\right)_{cd}G^{db} = 4\left(\Omega\frac{\partial E}{\partial \Gamma}G - G\frac{\partial E}{\partial \Gamma}\Omega\right)^{ab}
\end{aligned} \tag{240}$$

in agreement with the respective equations[27] of [32]. We introduced the evolution vector field $\mathcal{X}^\mu = (X^a, X^{ab})$. We can also define the instantaneous Hamiltonian

$$\hat{H} = \begin{cases} \frac{1}{2}k_{ab}(\hat{\xi} - z)^a(\hat{\xi} - z)^b + l_a\hat{\xi}^a - E & \textbf{(bosons)} \\ \frac{i}{2}k_{ab}\hat{\xi}^a\hat{\xi}^b - E & \textbf{(fermions)} \end{cases} \tag{241}$$

with $l = \frac{\partial E}{\partial z}$ and $k = \pm 2\lfloor\frac{\partial E}{\partial \Gamma}\rfloor$, where $(+)$ applies to bosons and $(-)$ to fermions. This is the quadratic Hamiltonian whose time evolution on the Gaussian family agrees with the projection $\mathcal{X}^\mu$, i.e., $-i\hat{H}|\Gamma, z\rangle = \mathcal{X}^\mu|V_\mu\rangle$. We further define the instantaneous Lie algebra element

$$K^a{}_b = \begin{cases} \Omega^{ac}k_{cb} & \textbf{(bosons)} \\ G^{ac}k_{cb} & \textbf{(fermions)} \end{cases} \,, \tag{242}$$

which allows us to write the time evolution equation for the linear complex structure and displacement simply as

$$\dot{z} = \Omega^{ab}l_b \quad \text{and} \quad \dot{J} = [K, J]\,. \tag{243}$$

---

[26]By construction the energy will only depend on the symmetric or antisymmetric part of $\Gamma^{ab}$ for bosons and fermions, respectively, such that $\frac{\partial E}{\partial \Gamma}$ will be automatically symmetric or antisymmetric.
Note the sign difference in $\dot{z} = \Omega\partial E$ compared to $\dot{\mathcal{X}} = -\Omega\partial E$, which is due to the chosen conventions mentioned in 25.

[27]See equations (31) of [32], where $z = \Delta_R/\sqrt{2}$, $\Omega = \sigma$ and $\Gamma = G = \Gamma_b$ for bosons and $G = \mathbb{1}$ and $\Gamma = \Omega = -\Gamma_m$ for fermions.

Table 3: *Dimension counting.* We count the dimensions of the Gaussian manifold for $N$ bosonic or fermionic modes.

|  | Displacements (bosons) | Squeezings (bosons) | Squeezings (fermions) |
|---|---|---|---|
| Change | $\delta z^a$ | $\delta \Gamma^{ab}$ | $\delta \Gamma^{ab}$ |
| Constraints | none | $\Gamma^{ab} = \Gamma^{ba}$, $\Delta\Gamma J^{\intercal} = J\delta\Gamma$ | $\Gamma^{ab} = -\Gamma^{ba}$, $\Delta\Gamma J^{\intercal} = J\delta\Gamma$ |
| Dimensions | $2N$ | $N(N+1)$ | $N(N-1)$ |

Note that these equations are in general non-linear, as $K$ and $l$ depend on the state and thus on $J$ and $z$.

**Imaginary time evolution.** We recall that on a general manifold, we have the the imaginary time evolution vector field $\frac{dx^\mu}{d\tau} = -\mathbf{G}^{\mu\nu}\frac{\partial E}{\partial x^\nu}$. This translates to $\frac{d}{d\tau}\Gamma^{ab} = 8\Omega^{ac}G^{bd}\lfloor\frac{\partial E}{\partial\Gamma}\rfloor_{cd}$ and thus to[28]

$$\frac{d}{d\tau}z^a = -G^{ab}\frac{\partial E}{\partial z^b} \tag{244}$$

$$\frac{d}{d\tau}\Gamma^{ab} = -4G^{ab}\left(\frac{\partial E}{\partial\Gamma} - J^{\intercal}\frac{\partial E}{\partial\Gamma}J\right)_{cd}G^{db} = -4\left(G\frac{\partial E}{\partial\Gamma}G + \Omega\frac{\partial E}{\partial\Gamma}\Omega\right)^{ab} \tag{245}$$

which agrees with the respective equations[29] in [32].

**Excitation spectrum.** We recall that we can approximate the excitation spectrum either by linearizing the equations of motion or by projecting the Hamiltonian onto tangent space. The latter can be straightforwardly done in the number basis by expressing $\hat{H}$ in terms of creation and annihilation operators associated to the approximate ground state $|\psi_0\rangle$. We therefore focus here on the former case, where the spectrum is encoded in the eigenvalues of $K^\mu{}_\nu = -\Omega^{\mu\sigma}(\partial_\nu\partial_\sigma E)$. We evaluate $K$ for Gaussian states at a stationary point, *i.e.*, at $x_0$ with $(\partial_\mu E)(x_0) = 0$. We use the real time evolution vector field $\mathcal{X}^\mu = (X^a, X^{ab})$ from (240) to compute

$$K \equiv \left(\begin{array}{c|c} \frac{\partial X^a}{\partial z^c} & \frac{\partial X^{ab}}{\partial z^c} \\ \hline \lfloor\frac{\partial X}{\partial\Gamma}\rfloor^a_{cd} & \lfloor\frac{\partial X}{\partial\Gamma}\rfloor^{ab}_{cd} \end{array}\right), \tag{246}$$

which can be explicitly computed as[30]

$$\frac{\partial X^a}{\partial z^c} = \Omega^{ae}\frac{\partial^2 E}{\partial z^c\partial z^e}, \tag{247}$$

$$\frac{\partial X^{ab}}{\partial z^c} = 4\left(\Omega^{ae}\frac{\partial^2 E}{\partial z^c\partial\Gamma^{ef}}\Gamma^{fb} - \Gamma^{ae}\frac{\partial^2 E}{\partial z^c\partial\Gamma^{ef}}\Omega^{fb}\right), \tag{248}$$

$$\left\lfloor\frac{\partial X}{\partial\Gamma}\right\rfloor^a_{cd} = \frac{\Omega^{ae}}{2}\left(\frac{\partial^2 E}{\partial\Gamma^{cd}\partial z^e} - J^f_c\frac{\partial^2 E}{\partial\Gamma^{fg}\partial z^e}J^g_d\right), \tag{249}$$

$$\left\lfloor\frac{\partial X}{\partial\Gamma}\right\rfloor^{ab}_{cd} = 4\Big(\Omega^{ae}\frac{\partial^2 E}{\partial\Gamma^{cd}\partial\Gamma^{ef}}\Gamma^{fb} - \Gamma^{ae}\frac{\partial^2 E}{\partial\Gamma^{cd}\partial\Gamma^{ef}}\Omega^{fb} + \Omega^{ae}\frac{\partial E}{\partial\Gamma^{ec}}\delta^b{}_d - \delta^a{}_c\frac{\partial E}{\partial\Gamma^{df}}\Omega^{fb}$$

$$+ J^g_c\Gamma^{ae}\frac{\partial^2 E}{\partial\Gamma^{gh}\partial\Gamma^{ef}}\Omega^{fb}J^h_d - J^g_c\Omega^{ae}\frac{\partial^2 E}{\partial\Gamma^{gs}\partial\Gamma^{ef}}\Gamma^{fb}J^h_d \tag{250}$$

$$+ J^a_c\frac{\partial E}{\partial\Gamma^{gf}}\Omega^{fb}J^g_d - (J^{\intercal})_c{}^g\Omega^{ae}\frac{\partial E}{\partial\Gamma^{eg}}J^b_d\Big).$$

---

[28] See footnote 26.

[29] See equations (30) of [32].

[30] We make the assumptions discussed in footnote 26.

Note that for fermions, we only have the last block $\lfloor \frac{\partial X}{\partial \Gamma} \rfloor$, as the others are related to the displacement $z^a$, which vanishes for fermions. For bosons, the spectrum of the first block $\frac{\partial X^a}{\partial z^c}$ can be related to Bogoliubov mean field theory, as discussed in [59], *i.e.*, it captures the one-particle spectrum.

**Spectral functions.** We can then evaluate spectral functions based on formula (156) for any operator $\hat{V}$. In practice, all eigenvectors $\mathcal{E}^\mu(\lambda_\ell)$ will be represented as

$$\mathcal{E}^\mu(\lambda_\ell) \equiv (\delta z^a, \delta \Gamma^{ab}), \tag{251}$$

such that we can compute the dual vectors

$$\partial_\mu V \equiv (\tfrac{\partial V}{\partial z^a}, \tfrac{\partial V}{\partial \Gamma^{ab}}). \tag{252}$$

Note that we will need to remove unphysical eigenvectors and dual eigenvectors, as discussed in the next paragraph. The spectral function is then directly computed from (156). This was done explicitly in [59] to study the excitation spectrum and spectral functions of the paradigmatic Bose-Hubbard model, where $\hat{V}$ was chosen to either describe density variations or lattice modulations.

As explained in table 3, the manifold of bosonic states is $N(N+3)$-dimensional, while the manifold of fermionic states is $N(N-1)$-dimensional. In contrast, the matrix representation (246) has a much larger dimension, due to the fact that we do not implement the constraints on $\Gamma$ directly in the basis of $K$, but rather by applying the projection (230) in the definition of $K$. By construction, any forbidden change $\delta\Gamma$ violating the constraints (220) will be projected out and thus contributes the eigenvalue zero to the spectrum of $K$. In practice, we therefore have two options:

(a) We can just compute the spectrum of $K$ as written in (246) and drop $N(N \mp 1)$ vanishing eigenvalues for bosons and fermions, respectively. If there are more zero eigenvalues, the spectrum will have one or more zero modes. To identify the corresponding (physical) eigenvector $(z^c, \delta\Gamma^{cd})$, one would need identify those eigenvectors which do not violate the constraints (220). All eigenvectors with non-vanishing eigenvalues are physical and necessarily satisfy the constraint.

(b) We can also reduce the dimension of $K$ by constructing an orthonormal basis $\delta\Gamma_i^{cd}$ explicitly. This allows us restrict/project $K$ onto this subspace. This is particulary important when

There are large classes of examples where variational methods have been successfully applied to the family of Gaussian states to describe physical systems. Given a single fixed unitary transformation $U_0$, we can define the variational family of transformed Gaussian states $\mathcal{M}' = \{U_0 | \Gamma, z\rangle\}$. Using this variational family for a given Hamiltonian $\hat{H}$ is equivalent to applying our methods directly to the transformed Hamiltonian $\hat{H}' = U_0^\dagger \hat{H} U_0$. This approach has been successfully applied to various systems, such as the Kondo problem [19].

**Example 18.** *We consider a free bosonic system with one degree of freedom. Let its quadratic Hamiltonian be*

$$\hat{H} = \frac{1}{2} h_{ab} \hat{\xi}^a \hat{\xi}^b \quad with \quad h \overset{q,p}{\equiv} \begin{pmatrix} \omega & 0 \\ 0 & \omega \end{pmatrix}. \tag{253}$$

*The manifold of Gaussian states was explicitly parametrized in (211), such that we find the*

$$E = \frac{1}{4} h_{ab} \Gamma^{ab} + \frac{1}{2} h_{ab} z^a z^b = \frac{\omega}{2} (\cosh\rho + z_1^2 + z_2^2). \tag{254}$$

The change of $z^a$ and $\Gamma^{ab}$ under time evolution are

$$\frac{dz}{dt} qp\omega \begin{pmatrix} z_2 \\ -z_1 \end{pmatrix}, \quad \frac{d\Gamma}{dt} \overset{q,p}{\equiv} -2\omega \sinh \rho \begin{pmatrix} \cos \phi & -\sin \phi \\ -\sin \phi & -\cos \phi \end{pmatrix} \tag{255}$$

leading to $\dot\rho = 0$ and $\dot\phi = 2\omega$. Similarly, imaginary time evolution is given by

$$\frac{dz}{d\tau} \overset{q,p}{\equiv} \begin{pmatrix} -\omega z_1 \\ -\omega z_2 \end{pmatrix}, \quad \frac{d\Gamma}{d\tau} \overset{q,p}{\equiv} -\omega \begin{pmatrix} 2\sinh^2 \rho + \sin \phi \sinh 2\rho & \cos \phi \sinh 2\rho \\ \cos \phi \sinh 2\rho & 2\sinh^2 \rho - \sin \phi \sinh 2\rho \end{pmatrix} \tag{256}$$

leading to $\rho' = -2\sinh \rho$ and $\phi' = 0$. Finally, we find

$$K \equiv \begin{pmatrix} & \omega & & \\ -\omega & & & \\ \hline & & & 2\omega \\ & & -2\omega & \end{pmatrix}, \tag{257}$$

from which we can read off the 1- and 2-particle excitation energies of the free Hamiltonian. For the matrix representation of $K$, we chose the orthonormal basis $(z^a, \Gamma_i)$

$$\delta\Gamma_1 \overset{q,p}{\equiv} \begin{pmatrix} 1 & \\ & -1 \end{pmatrix}, \quad \delta\Gamma_2 \overset{q,p}{\equiv} \begin{pmatrix} & 1 \\ 1 & \end{pmatrix}. \tag{258}$$

**Example 19.** *We consider a free fermionic system with two degrees of freedom. Let its quadratic Hamiltonian be*

$$\hat{H} = \frac{1}{2} h_{ab} \hat\xi^a \hat\xi^b \quad \text{with} \quad h \overset{q,p}{\equiv} \begin{pmatrix} & & \omega_1 & \\ & & & \omega_2 \\ -\omega_1 & & & \\ & -\omega_2 & & \end{pmatrix}. \tag{259}$$

*This manifold of Gaussian states was explicitly parametrized in (216) with two coordinates $x^\mu \equiv (\theta, \phi)$ leading to the energy of the state with $J_\pm$ given by*

$$E = -\frac{1}{4} h_{ab} \Gamma^{ab} = -(\omega_2 \pm \omega_1) \cos \theta. \tag{260}$$

*The resulting equations for real time evolution are*

$$\frac{d\Gamma}{dt} \overset{q,p}{\equiv} (\omega_2 \pm \omega_1) \sin \theta \begin{pmatrix} & & \cos \phi & \sin \phi \\ -\cos \phi & & & -\sin \phi \\ & \sin \phi & & -\cos \phi \\ -\sin \phi & & \cos \phi & \end{pmatrix} \tag{261}$$

*leading to the equations $\dot\theta = 0$ and $\dot\phi = (\omega_2 \pm \omega_1)$. Similarly, imaginary time evolution is described by*

$$\frac{d\Gamma}{d\tau} \overset{q,p}{\equiv} \frac{(\omega_2 \pm \omega_1) \sin 2\theta}{2} \begin{pmatrix} & -\sin \phi & -\tan \theta & \cos \phi \\ \sin \phi & & -\cos \phi & -\tan \theta \\ \tan \theta & \cos \phi & & \sin \phi \\ -\cos \phi & \tan \theta & -\sin \phi & \end{pmatrix} \tag{262}$$

*leading to $\theta' = (\omega_2 \pm \omega_1) \sin \theta$ and $\phi' = 0$. We can compute the matrix representation of $K^\mu{}_\nu$ as*

$$K \equiv \begin{pmatrix} 0 & \omega_1 + \omega_2 \\ -\omega_1 - \omega_2 & 0 \end{pmatrix}, \tag{263}$$

*whose eigenvalues correspond to the 2-particle spectrum of the free Hamiltonian $\hat{H}$. For the matrix representation of $K^\mu{}_\nu$, we chose the orthonormal basis of variations*

$$\delta\Gamma_1 \equiv \begin{pmatrix} & 1 & & \\ -1 & & & \\ & & & -1 \\ & & 1 & \end{pmatrix}, \delta\Gamma_2 \equiv \begin{pmatrix} & & & 1 \\ & & -1 & \\ & 1 & & \\ -1 & & & \end{pmatrix}. \tag{264}$$

**Example 20.** *We now derive the time evolution for example 9 of a free bosonic system with two degrees of freedom. We fix a basis $\hat{\xi}^a \overset{q,p}{\equiv} (\hat{q}_1, \hat{q}_2, \hat{p}_1, \hat{p}_2)$, with respect to which symplectic form and covariance matrix of $|\Gamma, z\rangle$ are*

$$
\Omega \overset{q,p}{\equiv} \begin{pmatrix} & & 1 & \\ & & & 1 \\ -1 & & & \\ & -1 & & \end{pmatrix}, \quad \Gamma \overset{q,p}{\equiv} \begin{pmatrix} 1 & & & \\ & 1 & & \\ & & 1 & \\ & & & 1 \end{pmatrix}. \tag{265}
$$

*We now choose the 2-dimensional variational families of coherent states $|\Gamma, z\rangle$, where $z^a \overset{q,p}{\equiv} (q_1, p_1, q_2, p_2)$ depends on the two variational parameters $\tilde{z}^\alpha \overset{q,p}{\equiv} (\tilde{q}, \tilde{p})$ via*

$$
z^a = T^a{}_\beta \tilde{z}^\beta \quad \text{with} \quad T \overset{q,p}{\equiv} \begin{pmatrix} \cosh r & 0 \\ \sinh r & 0 \\ 0 & \cosh r \\ 0 & -\sinh r \end{pmatrix}, \tag{266}
$$

*which agrees with $|\psi(\alpha)\rangle$ from example 7. The $\tilde{z}^\alpha$ are the expectation values of the canonically conjugated operators*

$$
\hat{\tilde{q}} = \hat{q}_1 \cosh r - \hat{q}_2 \sinh r, \tag{267}
$$
$$
\hat{\tilde{p}} = \hat{p}_1 \cosh r - \hat{p}_2 \sinh r. \tag{268}
$$

*We represent the orthogonal projector $\mathbb{P}_{(\Gamma, z)}$ as 2-by-4 matrix $P^\alpha{}_b = \mathbb{P}^\alpha_{(\Gamma, z)} |V_b\rangle$ with $P^\alpha{}_b T^b{}_\gamma = \delta^\alpha{}_\gamma$ given by*

$$
P \overset{q,p}{\equiv} \begin{pmatrix} \cosh r & -\sinh r & 0 & 0 \\ 0 & 0 & \cosh r & \sinh r \end{pmatrix}, \tag{269}
$$

*where $P$ acts on the tangent space of all displacements $\delta z$ from (223) and orthogonally projects onto the tangent space of our family described by $\delta \tilde{z}$. The Hamiltonian (101) can be rewritten as $\hat{H} = \frac{1}{2} h_{ab} \hat{\xi}^a \hat{\xi}^b$ with*

$$
h \overset{q,p}{\equiv} \begin{pmatrix} c_1 & c_3 & & \\ c_3 & c_2 & & \\ & & c_1 & c_3 \\ & & c_3 & c_2 \end{pmatrix}, \tag{270}
$$

*where $c_1 = \epsilon_1 \cos^2 \phi + \epsilon_2 \sin^2 \phi$, $c_1 = \epsilon_1 \sin^2 \phi + \epsilon_2 \cos^2 \phi$ and $c_3 = \frac{\epsilon_1 - \epsilon_2}{2} \sin 2\phi$. We consider the time evolution under $\hat{H}$ of the expectation values $\tilde{z} \equiv (\tilde{q}, \tilde{p})$ for the following scenarios.*
***True evolution.*** *The time evolution equation $\dot{z}^a = \mathcal{X}^a$ follows from (240) and is given by*

$$
\mathcal{X}^a = K^a{}_b z^b \quad \text{with} \quad K^a{}_b = \Omega^{ac} h_{cb} \tag{271}
$$

*and solved by $z(t) = M(t) z(0)$ with $M(t) = e^{tK}$.*
***Lagrangian vs. McLachlan evolution.*** *The Lagrangian and McLachlan evolution are based on projecting the equations of motion (240) in the two ways*

$$
\mathcal{X}^\alpha_{\text{Lagrangian}} = P^\alpha{}_b \mathcal{X}^b = (K_1)^\alpha{}_\gamma \tilde{z}^\gamma, \tag{272}
$$
$$
\mathcal{X}^\alpha_{\text{McLachlan}} = (\tilde{J}^{-1})^\alpha{}_\beta P^\beta{}_c J^c{}_d \mathcal{X}^d = (K_2)^\alpha{}_\gamma \tilde{z}^\beta, \tag{273}
$$

*where we have $\mathcal{X}^a = K^a{}_b T^b{}_\gamma \tilde{z}^\gamma$. We compute*

$$
\tilde{J} = PJT \overset{q,p}{\equiv} \begin{pmatrix} 0 & -\text{sech} \, 2r \\ \text{sech} \, 2r & 0 \end{pmatrix}, \quad K_i \overset{q,p}{\equiv} \begin{pmatrix} 0 & a_i^+ \\ a_i^- & 0 \end{pmatrix}, \tag{274}
$$

*where we introduced the constants given by*

$$
\begin{aligned}
a_1^\pm &= \pm \frac{(\epsilon_1 - \epsilon_2)\cos 2\phi + (\epsilon_1 + \epsilon_2)\operatorname{sech}(2r)}{2}, \\
a_2^\pm &= \pm \frac{(\epsilon_1 + \epsilon_2)\cosh 2r + (\epsilon_1 - \epsilon_2)(\cos 2\phi \mp \sin 2\phi \sinh 2\phi)}{2}.
\end{aligned}
\tag{275}
$$

*We can compare the time evolution of the variational parameters $\tilde{z}^a \equiv (\tilde{q}, \tilde{p})$ for the two variational principles with the exact evolution of the expectation values of $(\hat{q}, \hat{\dot{q}})$ for the same initial state $|\Gamma, z\rangle$, as shown in figure 4.*

### 5.1.5  Approximating expectation values

In this section, we compare how the expectation value of observables changes depending on if we evolve in the full Hilbert space with $-i\hat{H}$ or on our variational manifold with $\mathbb{P}_\psi(-i\hat{H})$. Clearly, any observable $\mathcal{O}$ that can be expanded in powers of $\hat{\xi}^a$ can, in principle, be computed exactly from the covariance matrix $\Gamma^{ab}$ and displacement vector $z^a$ (for bosons) using Wick's theorem.

As it turns out, the derivative $\frac{d}{dt}\langle \hat{A}\rangle$ for linear-quadratic observables $\hat{A}$ agrees in the two cases on the Gaussian manifold, *i.e.*, linear-quadratic observables are insensitive to non-Gaussianities to linear order on the Gaussian manifold.

Put differently, the variation $\delta\langle \hat{A}\rangle$ is insensitive if we perturb the Gaussian state $|\Gamma, z\rangle$ either into the direction $|\delta\psi\rangle$, which will generally leave the class of Gaussian states, or into the projected direction $\mathbb{P}_{|\Gamma,z\rangle}|\psi\rangle$, which is projected onto the Gaussian manifold. Here, we have $\delta\langle \hat{A}\rangle = \frac{d}{dt}\langle \hat{A}\rangle$ if we choose $|\delta\psi\rangle = \frac{d}{dt}|\psi\rangle = -i\hat{H}|\psi\rangle$.

**Proposition 12.** *Given a Gaussian state $|\Gamma, z\rangle$ and an arbitrary tangent vector $|\phi\rangle$ with $\langle \Gamma, z|\phi\rangle = 0$, the change of linear-quadratic observables $\hat{A} = f_a \hat{\xi}^a + \frac{1}{2}h_{ab}\hat{\xi}^a\hat{\xi}^b$ is*

$$
\delta\langle \hat{A}\rangle = 2\operatorname{Re}\langle \Gamma, z|\hat{A}|\phi\rangle = 2\operatorname{Re}\langle \Gamma, z|\hat{A}\mathbb{P}_{(\Gamma,z)}|\phi\rangle.
\tag{276}
$$

*Proof.* The proof is rather simple and goes in two steps: First, we note that a linear-quadratic operator $\hat{A}$ acting on $|\Gamma, z\rangle$ allows for the decomposition

$$
\hat{A}|\Gamma, z\rangle = C|\Gamma, z\rangle + X^a|V_a\rangle + X^{ab}|V_{ab}\rangle.
\tag{277}
$$

Second, the inner product between $\hat{A}|\Gamma, z\rangle$ and the tangent vector $|\phi\rangle$ only happen in this subspace. Consequently, equation (276) follows. $\square$

**Corollary 1.** *The time derivatives of displacement vector $\dot{z}^a$ and covariance matrix $\dot{\Gamma}^{ab}$ at a Gaussian state $|\Gamma, z\rangle$ are the same for the true time evolution with some interacting Hamiltonian $\hat{H}$ and for its projection onto the Gaussian manifold. In formulas, this means*

$$
\begin{aligned}
\dot{z}^a &= 2\operatorname{Re}\langle \Gamma, z|\hat{\xi}^a \mathbb{P}_{(\Gamma,z)}(-i\hat{H})|\Gamma, z\rangle \\
&= \langle \Gamma, z|[-i\hat{H}, \hat{\xi}^a]|\Gamma, z\rangle,
\end{aligned}
\tag{278}
$$

$$
\begin{aligned}
\dot{G}^{ab} &= 2\operatorname{Re}\langle \Gamma, z|\{\hat{\xi}^a, \hat{\xi}^b\}\mathbb{P}_{(\Gamma,z)}(-i\hat{H})|\Gamma, z\rangle - \frac{d(z^a z^b)}{dt} \\
&= \langle \Gamma, z|[-i\hat{H}, \{\hat{\xi}^a, \hat{\xi}^b\}]|\Gamma, z\rangle - \dot{z}^a z^b - z^a \dot{z}^b,
\end{aligned}
\tag{279}
$$

$$
\begin{aligned}
\dot{\Omega}^{ab} &= 2\operatorname{Re}\langle \Gamma, z|[\hat{\xi}^a, \hat{\xi}^b]\mathbb{P}_{(\Gamma,z)}(-i\hat{H})|\Gamma, z\rangle \\
&= \langle \Gamma, z|[-i\hat{H}, [\hat{\xi}^a, \hat{\xi}^b]]|\Gamma, z\rangle,
\end{aligned}
\tag{280}
$$

*where $\Gamma = G$ for bosons and $\Gamma = \Omega$ for fermions.*

In summary, projecting onto the Gaussian manifold is equivalent to truncating the equations of motion of the $n$-point functions at second order, *i.e.*, we integrate equations (278) to (280) and ignore non-Gaussian evolution of higher $n$-point functions $C_n$ for $n > 2$.

## 5.2 Group theoretic coherent states

Standard coherent states of the form $|\alpha\rangle = e^{\alpha \hat{a}^\dagger - \alpha^* \hat{a}}|0\rangle$ of a single bosonic degree of freedom were introduced by Glauber [44]. From the perspective of Gaussian states, as presented in the previous section, coherent states correspond to a submanifold of bosonic Gaussian states with fixed complex structure $J^a{}_b$ (or covariance matrix $\Gamma^{ab}$), but variable displacement vector $z^a$. Given a fixed Gaussian state $|J, 0\rangle$, we can reach the set of coherent states $|J, z\rangle$ with all possible $z^a \in V$ by applying the displacement transformation $|J, z\rangle = \mathcal{D}(z)|J, 0\rangle$. So, here, the main structure is $\mathcal{D}(z)$, which gives a projective representation of the group of vector addition $V$.

In this section, we generalize this construction by considering, instead of $\mathcal{D}(z)$, generic unitary representations of arbitrary semi-simple Lie groups. This leads to construct the so-called *group theoretic coherent states*. They were independently introduced by Gilmore [29, 88] and Perelomov [30, 89]. While we follow the excellent review [90], we will particularly emphasize the geometric structure of the resulting variational families and their advantages. In particular, we will show that group theoretic coherent states constructed from certain vectors, called highest weight vectors, always give rise to Kähler manifolds. Furthermore, we will connect to the previous section to show explicitly how the full families of bosonic and fermionic Gaussian states can be naturally understood as group theoretic coherent states with respect to groups $\mathcal{G} = \mathrm{Sp}(2N, \mathbb{R})$ and $\mathcal{G} = \mathrm{O}(2N, \mathbb{R})$ for bosons and fermions, respectively.

### 5.2.1 Definition

We consider a separable Hilbert space $\mathcal{H}$, a real Lie group $\mathcal{G}$ with real Lie algebra $\mathfrak{g}$ and a possibly projective unitary representation $\mathcal{U}$ of $\mathcal{G}$ on $\mathcal{H}$, *i.e.*, we have

$$\mathcal{U}(M) : \mathcal{H} \to \mathcal{H} \quad \text{with} \quad \mathcal{U}(M_1)\mathcal{U}(M_2) \simeq \mathcal{U}(M_1 M_2), \tag{281}$$

where $\simeq$ indicates equality up to a complex phase. We may later impose conditions, such as requiring that the Lie group is compact or semi-simple. Given a basis $\Xi_i$ of the Lie algebra $\mathfrak{g}$, we have the Lie brackets[31]

$$[\Xi_i, \Xi_j] = c_{ij}^k \Xi_k, \tag{282}$$

where $c_{ij}^k$ are called structure constants. Our group representation induces a representation of $\Xi_i$ as operators

$$\hat{\Xi}_i = \frac{d}{ds}\mathcal{U}(e^{s\Xi_i})\Big|_{s=0}, \tag{283}$$

which are anti-Hermitian[32] due to $\mathcal{U}^\dagger = \mathcal{U}^{-1}$. A general Lie algebra element $A \in \mathfrak{g}$ is then represented as anti-Hermitian operator $\hat{A} = A^i \hat{\Xi}_i$.

We can always represent group and Lie algebra through their action on the Lie algebra itself. This is called the adjoint representation. Here, a Lie group element $M$ is represented as

---

[31]In physics, some authors [91] choose the basis $X_i = -\mathrm{i}\Xi_i$, such that $A = A^i \Xi_i = \mathrm{i}A^i X_i$ and $[X_i, X_j] = \mathrm{i}c_{ij}^k \Xi_k$.

[32]A basis $X_i = -\Xi_i$ would lead to Hermitian operators $\hat{X}_i = -\mathrm{i}\hat{\Xi}_i$.

the linear map $\mathrm{Ad}_M : \mathfrak{g} \to \mathfrak{g}$ with

$$\mathrm{Ad}_M(\Xi_i) = \frac{d}{ds} M e^{s\Xi_i} M^{-1}\big|_{s=0}, \tag{284}$$

which reduces for matrices to $\mathrm{Ad}_M(\Xi_i) = M\Xi_i M^{-1}$. Similarly, the adjoint representation of a Lie algebra element $\Xi_i$ is given by the linear map $\mathrm{ad}_i : \mathfrak{g} \to \mathfrak{g}$ with

$$\mathrm{ad}_i(\Xi_j) = [\Xi_i, \Xi_j] = c_{ij}^k \Xi_k, \tag{285}$$

which implies that the adjoint representation of $\Xi_i$ always takes the matrix form $(\mathrm{ad}_i)^k{}_j = c_{ij}^k$ with respect to $\Xi_j$. The adjoint representation defines the Killing form

$$\mathcal{K}_{ij} = \mathrm{Tr}(\mathrm{ad}_i \mathrm{ad}_j) = (\mathrm{ad}_i)^k{}_l (\mathrm{ad}_i)^l{}_k = c_{il}^k c_{jk}^l, \tag{286}$$

which is non-degenerate for semi-simple Lie groups.

**Example 21.** *We consider the Lie group* $\mathrm{SU}(2)$ *consisting of complex, unitary 2-by-2 matrices with unit determinant. Its algebra is well known to be spanned by the matrices* $\Xi_i \equiv \frac{i}{2}\sigma_i$, *where* $\sigma_i$ *are the Pauli matrices, with structure constants* $c_{ij}^k \equiv \epsilon^k{}_{ij}$ *given by the totally antisymmetric tensor with* $\epsilon^1_{11} = 1$. *The Killing form is thus* $\mathcal{K}_{ij} = \epsilon^k{}_{il}\epsilon^l{}_{jk} = -2\delta_{ij}$. *We consider the following two representations:*

**Spin-$\frac{1}{2}$.** *This is the fundamental representation and thus coincides with the definition. We represent the group on the Hilbert space* $\mathcal{H} = \mathbb{C}^2$ *and it is generated by the anti-Hermitian operators* $\hat{\Xi}_i \equiv -\frac{i}{2}\sigma_i$ *with*

$$\hat{\Xi}_1 \equiv -\frac{1}{2}\begin{pmatrix} & i \\ i & \end{pmatrix}, \hat{\Xi}_2 \equiv \frac{1}{2}\begin{pmatrix} & -1 \\ 1 & \end{pmatrix}, \hat{\Xi}_3 \equiv \frac{1}{2}\begin{pmatrix} -i & \\ & i \end{pmatrix}, \tag{287}$$

*which coincides with the definition* $\Xi_i$.

**Spin-1.** *This representation is also the adjoint representation and the matrices can be chosen to be real. It is then given by real 3-by-3 rotation matrices acting on the Hilbert space* $\mathcal{H} = \mathbb{C}^3$. *It is generated by*

$$\hat{\Xi}_1 \equiv \begin{pmatrix} 0 & & \\ & & -1 \\ & 1 & \end{pmatrix}, \hat{\Xi}_2 \equiv \begin{pmatrix} & & -1 \\ & 0 & \\ 1 & & \end{pmatrix}, \hat{\Xi}_3 \equiv \begin{pmatrix} & 1 & \\ -1 & & \\ & & 0 \end{pmatrix}, \tag{288}$$

*which are infinitesimal rotations in the three planes. As matrices, they correspond to* $(\hat{\Xi}_i)^k{}_j = \epsilon^k{}_{ij}$, *which confirms that this is the adjoint representation.*

We will now explain how the choice of a reference state $\phi$ together with a projective representation $\mathcal{U}(M)$ on some Hilbert space $\mathcal{H}$ defines a variational family $\mathcal{M}_\phi \subset \mathcal{P}(\mathcal{H})$.

**Definition 4.** *Choosing a non-zero state* $|\phi\rangle \in \mathcal{H}$ *defines the associated manifold of group theoretic coherent states*

$$\mathcal{M}_\phi = \{\mathcal{U}(M)|\phi\rangle \mid M \in \mathcal{G}\}/\sim \, \subset \mathcal{P}(\mathcal{H}), \tag{289}$$

*where* $|\psi\rangle \sim |\tilde{\psi}\rangle \Longleftrightarrow |\psi\rangle = c|\tilde{\psi}\rangle$ *for some* $c \in \mathbb{C}$.

When applying variational methods to $\mathcal{M}_\phi$, it is useful to parametrize states by group elements, *i.e.*, $|\psi(M)\rangle = \mathcal{U}(M)|\phi\rangle$. Rather than taking derivatives with respect to some artificial coordinates, we define local coordinates

$$|\psi(M,x)\rangle = \mathcal{U}(M e^{x^i \Xi_i})|\phi\rangle = \mathcal{U}(M)e^{x^i \hat{\Xi}_i}|\phi\rangle \tag{290}$$

around every state $|\psi(M)\rangle$ with tangent vectors at $x = 0$

$$|\mathcal{V}_i\rangle = \mathbb{Q}_{\psi(M)} \frac{\partial}{\partial x^i} |\psi(M, x)\rangle \big|_{x=0} = \mathbb{Q}_{\psi(M)} \mathcal{U}(M) \hat{\Xi}_i |\phi\rangle . \tag{291}$$

This allows us to introduce a map between the Lie algebra $\mathfrak{g}$ and the tangent spaces of the manifold $\mathcal{M}_\phi$ as follows.

**Definition 5.** *For any $M \in \mathcal{G}$, we associate to every Lie algebra element $A = A^i \Xi_i$ the tangent vector*

$$A^i |\mathcal{V}_i\rangle = A^i \mathbb{Q}_{\psi(M)} \mathcal{U}(M) \hat{\Xi}_i |\phi\rangle \in \mathcal{T}_{\psi(M)} \mathcal{M}_\phi . \tag{292}$$

This map is only an isomorphism if there are no Lie algebra elements $A$ that only generate a change of complex phase and are thus mapped to $A^i |\mathcal{V}_i\rangle = 0$. Such Lie algebra elements define a subalgebra $\mathfrak{h}_\phi$ generating the stabilizer group $H_\phi$, defined as

$$H_\phi = \left\{ M \in \mathcal{G} \, \big| \, \mathcal{U}(M) |\phi\rangle \sim |\phi\rangle \right\}, \tag{293}$$

$$\mathfrak{h}_\phi = \{ A = A^i \Xi_i \, | \, A^i |\mathcal{V}_i\rangle = A^i \mathbb{Q}_\phi \hat{\Xi}_i |\phi\rangle = 0 \}. \tag{294}$$

As a manifold, we thus have $\mathcal{M}_\phi = \mathcal{G}/H_\phi$. Consequently, all tangent spaces $\mathcal{T}_{\psi(M)} \mathcal{M}_\phi$ are isomorphic to $\mathfrak{g}/\mathfrak{h}_\phi = \mathfrak{g}/\approx$ with $A \approx B$ if $A - B \in \mathfrak{h}_\phi$. If the Lie algebra is semi-simple, we can use the then non-degenerate Killing form to uniquely represent $\mathfrak{g}/\mathfrak{h}_\phi$ as

$$\mathfrak{h}_\phi^\perp = \{ B = B^i \Xi_i \, | \, \mathcal{K}_{ij} A^i B^j = 0 \, \forall A \in \mathfrak{h}_\phi \}. \tag{295}$$

We can now proceed to calculate the restricted Kähler strucures for the manifold $\mathcal{M}_\phi$. We can use the Lie algebra induced tangent space vectors $|\mathcal{V}_i\rangle$, introduced in (291), to derive simple expressions of the restricted Kähler structures independent of $M$.

**Proposition 13.** *The restricted Kähler structures of the manifold $\mathcal{M}_\phi$ are*

$$g_{ij} = \frac{2\mathrm{Re}\langle\mathcal{V}_i(M)|\mathcal{V}_j(M)\rangle}{\langle\psi(M)|\psi(M)\rangle} = -\frac{\langle\phi|\hat{\Xi}_i\mathbb{Q}_\phi\hat{\Xi}_j + \hat{\Xi}_j\mathbb{Q}_\phi\hat{\Xi}_i|\phi\rangle}{\langle\phi|\phi\rangle}, \tag{296}$$

$$\omega_{ij} = \frac{2\mathrm{Im}\langle\mathcal{V}_i(M)|\mathcal{V}_j(M)\rangle}{\langle\psi(M)|\psi(M)\rangle} = \frac{\langle\phi|\hat{\Xi}_j\mathbb{Q}_\phi\hat{\Xi}_i - \hat{\Xi}_i\mathbb{Q}_\phi\hat{\Xi}_j|\phi\rangle}{\langle\phi|\phi\rangle}, \tag{297}$$

*which are independent of $M$ and thus everywhere the same.*

*Proof.* We can straightforwardly compute

$$\langle\mathcal{V}_i|\mathcal{V}_j\rangle = \langle\phi|\hat{\Xi}_i^\dagger \mathcal{U}^\dagger(M)\mathbb{Q}_{\psi(M)}\mathcal{U}(M)\hat{\Xi}_j|\phi\rangle = -\langle\phi|\hat{\Xi}_i\mathbb{Q}_\phi\hat{\Xi}_j|\phi\rangle , \tag{298}$$

where we used $\mathcal{U}^\dagger(M)\mathbb{Q}_{\psi(M)}\mathcal{U}(M) = \mathbb{Q}_\phi$ and $\hat{\Xi}_i^\dagger = -\hat{\Xi}_i$. $\qquad \square$

Proposition 13 implies a dramatic simplification of the variational manifold $\mathcal{M}_\phi$, because it suffices to choose a single basis $\Xi_i$ of generators to bring the Kähler structures into a standard form which extends to all tangent spaces via the map of definition 5. This is particularly important for numerical implementations as discussed in section 5.2.5.

**Example 22.** *We will reconsider the Lie group* SU(2) *from example 21. As we are interested in the possible families $\mathcal{M}_\phi$, we will need to understand which $\phi$ are inequalivant, i.e., do not give rise to the same family $\mathcal{M}_\phi$. We can compute the tangent vectors $|\mathcal{V}_i\rangle$ based on definition 5 for the following representations.*

**Spin-$\frac{1}{2}$.** Up to a multiplication with a complex number $c$, we can use our fundamental representation $\mathcal{U}(M)$ to transform any non-zero vector into any other complex vector. As our definition of $\mathcal{M}_\phi$ ignores complex rescalings, we therefore must have $\mathcal{M}_\phi = \mathcal{P}(\mathbb{C}^2)$, i.e., for any non-zero state $\phi$, the resulting family is the full projective Hilbert space as discussed in example 4. To compute the tangent space vectors, we take $|\phi\rangle \equiv (1,0)$ and find

$$|\mathcal{V}_1\rangle \equiv \begin{pmatrix} 0 \\ \frac{i}{2} \end{pmatrix}, \quad |\mathcal{V}_2\rangle \equiv \begin{pmatrix} 0 \\ \frac{1}{2} \end{pmatrix}, \quad |\mathcal{V}_3\rangle \equiv \begin{pmatrix} 0 \\ 0 \end{pmatrix}. \tag{299}$$

As $|\mathcal{V}_3\rangle = 0$, the tangent space is $\mathcal{T}_\phi \mathcal{M}_\phi = \mathrm{span}_{\mathbb{R}}(|\mathcal{V}_1\rangle, |\mathcal{V}_2\rangle)$ leading to the restricted Kähler structures

$$\boldsymbol{g} \equiv \begin{pmatrix} 2 & 0 \\ 0 & 2 \end{pmatrix}, \boldsymbol{\omega} \equiv \begin{pmatrix} 0 & 2 \\ -2 & 0 \end{pmatrix}, \boldsymbol{J} \equiv \begin{pmatrix} 0 & -1 \\ 1 & 0 \end{pmatrix}, \tag{300}$$

which clearly satisfy $\boldsymbol{J}^2 = -\mathbb{1}$.

**Spin-1.** Our representation $\mathcal{U}(M)$ consists of standard 3-by-3 rotation matrices acting on $\mathbb{C}^3$. In the basis of these matrices, we can thus represent any vector as column $|\phi\rangle \equiv \vec{a} + i\vec{b}$, with $\vec{a}, \vec{b} \in \mathbb{R}^3$. We choose this vector to be normalized, such that $\vec{a}^2 + \vec{b}^2 = 1$. Multiplying with a complex phase $e^{i\varphi}$ corresponds to a rotation $\vec{a} \to \cos(\varphi)\vec{a} + \sin(\varphi)\vec{b}$ and $\vec{b} \to \cos(\varphi)\vec{b} - \sin(\varphi)\vec{a}$, which we can use to ensure $\vec{a} \cdot \vec{b} = 0$ and $\vec{a}^2 \geq \vec{b}^2$, such that we can always choose $0 \leq \theta \leq \frac{\pi}{4}$ with $|\vec{a}| = \cos\theta$ and $|\vec{b}| = \sin\theta$. We can then apply the rotation matrices $\mathcal{U}(M)$, such that $\vec{a} = (\cos\theta, 0, 0)$ and $\vec{b} = (0, \sin\theta, 0)$. Furthermore, we find the tangent vectors

$$|\mathcal{V}_1\rangle \equiv \begin{pmatrix} 0 \\ 0 \\ -i\sin\theta \end{pmatrix}, |\mathcal{V}_2\rangle \equiv \begin{pmatrix} 0 \\ 0 \\ \cos\theta \end{pmatrix}, |\mathcal{V}_3\rangle \equiv \begin{pmatrix} -i\cos 2\theta \sin\theta \\ -\cos 2\theta \cos\theta \\ 0 \end{pmatrix}. \tag{301}$$

For this choice, we can compute the Kähler structures

$$\boldsymbol{g} \equiv 2\begin{pmatrix} \sin^2\theta & 0 & 0 \\ 0 & \cos^2\theta & 0 \\ 0 & 0 & \cos^2\frac{\theta}{2} \end{pmatrix}, \boldsymbol{\omega} \equiv 2\begin{pmatrix} 0 & \sin 2\theta & 0 \\ -\sin 2\theta & 0 & 0 \\ 0 & 0 & \cos^2\frac{\theta}{2} \end{pmatrix} \tag{302}$$

leading to the linear complex structure

$$\boldsymbol{J} \equiv \begin{pmatrix} 0 & -\cot\theta & 0 \\ \tan\theta & 0 & 0 \\ 0 & 0 & 0 \end{pmatrix}. \tag{303}$$

For $0 < \theta < \frac{\pi}{4}$, we have $\mathfrak{h}_\phi = \mathrm{span}(|\mathcal{V}_1\rangle, |\mathcal{V}_2\rangle, |\mathcal{V}_3\rangle)$ and $\mathcal{M}_\phi$ is degenerate non-Kähler. For $\theta = 0$, we have $\mathfrak{h}_\phi = \mathrm{span}(|\mathcal{V}_2\rangle, |\mathcal{V}_3\rangle)$, on which $\omega$ vanishes and $\mathcal{M}_\phi$ is again degenerate non-Kähler. Only for $\theta = \frac{\pi}{4}$, we have $\mathfrak{h}_\phi = \mathrm{span}(|\mathcal{V}_1\rangle, |\mathcal{V}_2\rangle)$, on which $\boldsymbol{J}^2 = -\mathbb{1}$ holds, and $\mathcal{M}_\phi$ is Kähler. The families $\mathcal{M}_\phi \subset \mathcal{P}(\mathbb{C}^3)$ are 3-dimensional copies of $\mathrm{SO}(3, \mathbb{R})$ for $0 < \theta < \frac{\pi}{4}$ and spheres $S^2$, otherwise. Together these orbits foliate the projective Hilbert space $\mathcal{P}(\mathbb{C}^3)$ just like a sphere can be foliated in circles of latitudes with single points at the poles.

### 5.2.2  Compact Lie groups

We are interested in geometry of the variational family $\mathcal{M}_\phi \subset \mathcal{P}(\mathcal{H})$, namely its restricted Kähler structures (296) and (297). In particular, we would like to find a simple criterion when such a family of group theoretic coherent states is a Kähler manifold. In this section, we will focus on the representation theory of compact semi-simple Lie algebras, as discussed in [91]. Later, we will also consider Lie algebras that are not compact or not semi-simple, as discussed in [92].

For readers familiar with the theory of Lie algebras, let us emphasize that we need to carefully distinguish between the theory of real and complex Lie algebras. In our case, we have a real Lie algebra $\mathfrak{g}$, because we started from a real Lie group $\mathcal{G}$ and its unitary representation $\mathcal{U}$. We will be lead to also consider the complexification

$$\mathfrak{g}^{\mathbb{C}} = \{A^i \Xi_i \,|\, A^i \in \mathbb{C}\}, \tag{304}$$

but only real elements $A \in \mathfrak{g} \subset \mathfrak{g}^{\mathbb{C}}$ will be represented by anti-Hermitian operators $\hat{A}$. For a general element $A = A^i \Xi_i \in \mathfrak{g}^{\mathbb{C}}$, we define its conjugation as $A^* = A^{*i} \Xi_i$, such that $A$ is only real if $A = A^*$.

We consider a compact semi-simple Lie group $\mathcal{G}$ with real Lie algebra $\mathfrak{g}$, *i.e.*, the Killing form $\mathcal{K}$ is negative definite, such that $\mathcal{K}(A, B) < 0$ for all non-zero $A, B \in \mathfrak{g}$. Our construction relies on first choosing a Cartan subalgebra $\mathfrak{h} \subset \mathfrak{g}$ (not to be confused with $\mathfrak{h}_\phi$), characterized by the property that if $[X, Y] \in \mathfrak{h}$ for all $X \in \mathfrak{h}$ also $Y \in \mathfrak{h}$. For semi-simple Lie algebras, this implies that $\mathfrak{h}$ is Abelian, *i.e.*, $[X, Y] = 0$ for all $X, Y \in \mathfrak{h}$. We choose a basis of $\mathfrak{h}$ as $H_I = H_I^i \Xi_i$, which is smaller than $\Xi_i$ that spans $\mathfrak{g}$. While the choice of $\mathfrak{h}$ is not unique, they are all isomorphic for compact semi-simple Lie algebras and they give rise to the following structures[33]:

- The adjoint representation $\mathrm{ad}_I = H_I^i \, \mathrm{ad}_i$ of the Cartan basis $H_I$ has joint eigenspaces $\mathfrak{v}_\alpha \subset \mathfrak{g}^{\mathbb{C}}$ with

$$\mathrm{ad}_I(E_\alpha) = [H_I, E_\alpha] = \alpha_I E_\alpha \ \forall \ E_\alpha \in \mathfrak{v}_\alpha, \tag{305}$$

  where the set of eigenvalues $\alpha_I$ is called a root[34], which always come in pairs $(\alpha, -\alpha)$.

- The root system $\Delta$ is the set of all non-zero roots $\alpha$. It can be split into the two disjoint sets of positive roots $\Delta^+$ and negative roots $\Delta^-$, such that for every $\alpha \in \Delta^+$, we have $-\alpha \in \Delta^-$.

- We have the root space decomposition given by

$$\mathfrak{g}^{\mathbb{C}} = \mathfrak{h}^{\mathbb{C}} \oplus \bigoplus_{\alpha \in \Delta^+} \mathfrak{v}_\alpha \oplus \mathfrak{v}_{-\alpha}, \tag{306}$$

  where all $\mathfrak{v}_\alpha$ are complex and one-dimensional.

- For every root $\alpha \in \Delta$, we have the generator[35]

$$H_\alpha = \alpha_I (\mathcal{K}^{-1})^{IJ} H_J \in \mathfrak{h}^{\mathbb{C}}, \tag{307}$$

  such that $[E_\alpha, E_{-\alpha}] = \mathcal{K}(E_\alpha, E_{-\alpha}) H_\alpha$.

- We can choose for each eigenspace $\mathfrak{v}_\alpha$ an eigenvector $E_\alpha$, such that $\mathcal{K}(E_\alpha, E_{-\alpha}) > 0$ and such that

$$[E_\alpha, E_\beta] = N_{\alpha\beta} E_{\alpha+\beta} \quad \text{if} \quad \alpha + \beta \in \Delta, \tag{308}$$

  where $N_{\alpha\beta}$ are real and satisfy $N_{\alpha,\beta} = -N_{-\alpha,-\beta}$, while all other brackets $[E_\alpha, E_\beta]$ vanish.

---

[33]The same structures arise for non-compact semi-simple Lie algebras, but they are not necessarily isomorphic anymore.

[34]Each root is a linear map $\alpha : \mathfrak{h} \to \mathbb{C}$ with $\alpha(A^I H_I) = A^I \alpha_I$.

[35]There is a slight abuse of notation: $H_I$ refers to a real basis of $\mathfrak{h}$ and is distinct from $H_\alpha \in \mathfrak{h}^{\mathbb{C}}$ defined in (307).

One can use the property of $\mathfrak{g}$ being compact to show that above choices of $E_\alpha$ satisfy $E_\alpha^* = -E_{-\alpha}$ and that all $\alpha_I$ are imaginary. Thus, we have the real basis

$$\Xi_i \equiv (H_I, Q_\alpha, P_\alpha), \tag{309}$$

where $\alpha \in \Delta^+$ and we introduced the real generators

$$Q_\alpha = E_\alpha - E_{-\alpha} \quad \text{and} \quad P_\alpha = \mathrm{i}(E_\alpha + E_{-\alpha}). \tag{310}$$

When we represent $\Xi_i$ by anti-Hermitian operators $\hat{\Xi}_i$ on a Hilbert space $\mathcal{H}$, all $\hat{H}_I = H_I^i \hat{\Xi}_i$ commute with each other. A common eigenvector $|\mu\rangle \in \mathcal{H}$ of all $\hat{H}_I$ with

$$\hat{H}_I |\mu\rangle = \mu_I |\mu\rangle \tag{311}$$

is called a weight vector and the joint eigenvalues $\mu_I$ are called its weight[36]. A weight vector $|\mu\rangle$ is called highest weight vector[37] if it is annihilated by all positive root operators $\hat{E}_\alpha$ with $\alpha \in \Delta^+$, i.e.,

$$|\mu\rangle \text{ highest weight} \quad \Longleftrightarrow \quad \hat{E}_\alpha |\mu\rangle = 0 \,\forall\, \alpha \in \Delta^+. \tag{312}$$

Different choices of $\mathfrak{h}$ and different assignments of which roots are positive will lead to different highest weight vectors. We refer to all vectors $|\mu\rangle$ as highest weight vectors, for which there exists a choice of Cartan subalgebra $\mathfrak{h}$ and an assignment of positive roots, such that $|\mu\rangle$ is a highest weight vector in the above sense. For compact Lie groups, all such highest weight vectors are related by applying $\mathcal{U}(M)$ for some $M \in \mathcal{G}$, i.e., the family $\mathcal{M}_\phi$ for $\phi$ being a highest weight vector is actually the set of *all* highest weight vectors with respect to all possible choices of Cartan subalgebras and positivity of roots. The following proposition shows that such $\mathcal{M}_\phi$ is Kähler, which was also recognized in [90].

**Proposition 14.** *If $\phi$ is a highest weight vector and $\mathcal{G}$ a semi-simple compact group, the manifold $\mathcal{M}_\phi$ is Kähler.*

*Proof.* As the group is compact, because of the discussion in section 5.2.2, a basis of the corresponding algebra is given by (309). Let $|\phi\rangle = |\mu\rangle$ the highest weight vector with respect to the Cartan subalgebra $\mathfrak{h}^{\mathbb{C}}$ spanned by $H_\alpha$. We split our set $\Delta^+$ of positive root into those $\tilde{\alpha} \in \Delta^+$ with $\hat{E}_{-\tilde{\alpha}} |\mu\rangle = 0$ and those $\alpha \in \Delta^+$ with $\hat{E}_{-\alpha} |\mu\rangle \neq 0$. Note that $\hat{E}_\alpha |\mu\rangle = 0$ for all $\alpha \in \Delta^+$ due to $|\mu\rangle$ being highest weight. With this, we can split our basis $\Xi_i$ further into the three parts $\Xi_i \equiv (H_I, Q_{\tilde{\alpha}}, P_{\tilde{\alpha}}, Q_\alpha, P_\alpha)$. We can construct the induced tangent space basis $|\mathcal{V}_i\rangle$ from their definition (5) to find

$$|\mathcal{V}_I\rangle = \mathbb{Q}_\mu \hat{H}_I |\mu\rangle = 0, \tag{313}$$

$$|\mathcal{V}_{\tilde{\alpha}}^1\rangle = \mathbb{Q}_\mu \hat{Q}_{\tilde{\alpha}} |\mu\rangle = 0, \tag{314}$$

$$|\mathcal{V}_{\tilde{\alpha}}^2\rangle = \mathbb{Q}_\mu \hat{P}_{\tilde{\alpha}} |\mu\rangle = 0, \tag{315}$$

$$|\mathcal{V}_\alpha^1\rangle = \mathbb{Q}_\mu \hat{Q}_\alpha |\mu\rangle = \hat{E}_{-\alpha} |\mu\rangle \propto |\mu - \alpha\rangle \neq 0, \tag{316}$$

$$|\mathcal{V}_\alpha^2\rangle = \mathbb{Q}_\mu \hat{P}_\alpha |\mu\rangle = \mathrm{i}\hat{E}_{-\alpha} |\mu\rangle \propto \mathrm{i}|\mu - \alpha\rangle \neq 0. \tag{317}$$

Here, $(|\mathcal{V}_\alpha^1\rangle, |\mathcal{V}_\alpha^2\rangle)$ forms a basis of tangent space. Indeed $\hat{E}_{-\alpha} |\mu\rangle$ is an eigenvectors of $\hat{H}_I$ with eigenvalue $\mu_I - \alpha_I$, such that $\hat{H}_I \hat{E}_{-\alpha} |\mu\rangle = (\mu_I - \alpha_I) \hat{E}_{-\alpha} |\mu\rangle$ and as such are orthonormal. Clearly, we have the pairs $|\mathcal{V}_\alpha^1\rangle$ and $|\mathcal{V}_\alpha^2\rangle = \mathrm{i} |\mathcal{V}_\alpha^1\rangle$ ensuring that tangent space satisfies the Kähler property. $\qquad\square$

---

[36]Weights are linear maps $\mu : \mathfrak{h} \to \mathbb{C}$ with $\mu(A^I H_I) = A^I \mu_I$.

[37]Technically, we could also call it lowest weight vector, if we switched the roles of positive and negative roots, which is why some authors use the term 'extremal weight'.

Assuming $|\mu\rangle$ is a weight vector, *i.e.*, an eigenvector of the Cartan subalgebra operators, we can explicitly compute the matrix representations

$$\boldsymbol{g} \equiv 2 \bigoplus_\alpha \frac{\langle\mu|[\hat{E}_{-\alpha},\hat{E}_\alpha]+2\hat{E}_\alpha\hat{E}_{-\alpha}|\mu\rangle}{\langle\mu|\mu\rangle} \begin{pmatrix} 1 & 0 \\ 0 & 1 \end{pmatrix}, \tag{318}$$

$$\boldsymbol{\omega} \equiv 2 \bigoplus_\alpha \frac{\langle\mu|[\hat{E}_{-\alpha},\hat{E}_\alpha]|\mu\rangle}{\langle\mu|\mu\rangle} \begin{pmatrix} 0 & 1 \\ -1 & 0 \end{pmatrix}, \tag{319}$$

$$\boldsymbol{J} \equiv \bigoplus_\alpha \frac{\langle\mu|[\hat{E}_{-\alpha},\hat{E}_\alpha]|\mu\rangle}{\langle\mu|[\hat{E}_{-\alpha},\hat{E}_\alpha]+2\hat{E}_\alpha\hat{E}_{-\alpha}|\mu\rangle} \begin{pmatrix} 0 & -1 \\ 1 & 0 \end{pmatrix}. \tag{320}$$

with respect to $(|\mathcal{V}_\alpha^1\rangle, |\mathcal{V}_\alpha^2\rangle)$. We see immediately that such structures then take the canonical Kähler form ($\boldsymbol{J}^2 = -\mathbb{1}$) only if $|\mu\rangle$ is the highest weight, that is $\langle\mu|\hat{E}_\alpha\hat{E}_{-\alpha}|\mu\rangle = 0$. In this case, they only depend on the following factor which we evaluate explicitly using (307):

$$\frac{\langle\mu|[\hat{E}_{-\alpha},\hat{E}_\alpha]|\mu\rangle}{\langle\mu|\mu\rangle} = -\frac{\langle\mu|\mathcal{K}(E_\alpha,E_{-\alpha})\hat{H}_\alpha|\mu\rangle}{\langle\mu|\mu\rangle} = -\frac{\mathcal{K}(E_\alpha,E_{-\alpha})\mu(H_\alpha)}{\langle\mu|\mu\rangle}. \tag{321}$$

This can be explicitly checked in the following example.

**Example 23.** *We reconsider example 22 of* SU(2) *for the spin-$\frac{1}{2}$ and spin-1 representation.* SU(2) *is a compact group. We choose as real Cartan subalgebra $\mathfrak{h} = \mathrm{span}_\mathbb{R}(\Xi_3)$, which gives rise to $E_\pm = \frac{i}{2}(\Xi_1 \pm i\Xi_2)$, where we $E_+$ corresponds to the only positive root.*
**Spin-$\frac{1}{2}$.** *There is only one $\mathcal{M}_\phi$, which is the full space $\mathcal{P}(\mathbb{C}^2)$. Therefore every state in the representation is a highest weight state with respect to some choice of $\mathfrak{h}^\mathbb{C}$ and selection of positive roots. For our choice, the highest weight state is $|\phi\rangle = (1,0)$, for which we already computed the tangent vector in example 22 and verified that $\mathcal{M}_\phi$ is Kähler.*
**Spin-1.** *Not every state is a highest weight state anymore. With respect to our choice of positive root vector $E_+$, the highest weight state is $|\phi\rangle = (\frac{1}{\sqrt{2}}, \frac{i}{\sqrt{2}}, 0)$, which corresponds to the boundary case $\theta = \pi/4$ from our previous considerations. This agrees with our finding that $\mathcal{M}_\phi$ is only Kähler for $\theta = \frac{\pi}{4}$. Note that we can always include more generators to construct a larger group (here:* SU(3)*), so that also states for $\frac{\pi}{4} \neq \theta \neq 0$ are highest weight states (with respect to the larger state) and $\mathcal{M}_\phi$ will be Kähler (here: full $\mathcal{P}(\mathcal{H})$).*

### 5.2.3 General Lie groups

Proposition 14 shows that $\mathcal{M}_\phi$ is a Kähler manifold if $\mathcal{G}$ is a compact semisimple Lie grup and $\phi$ a highest weight vector. How much of this analysis can be carried out for non-compact or non-semi-simple Lie groups/algebras?

For a semi-simple, non-compact real Lie algebra $\mathfrak{g}$, *i.e.*, $\mathcal{K}$ is not negative definite anymore (but still non-degenerate), we can still choose a Cartan subalgebra $\mathfrak{h} \subset \mathfrak{g}$ and use the same root space decomposition (306). Not all choices of $\mathfrak{h}$ are isomorphic anymore, as $\mathcal{K}$ restricted to $\mathfrak{h}$ may have different signatures. This may lead to additional requirements for a highest weight vector $|\mu\rangle$ to give rise to Kähler manifolds. For any choice of Cartan subalgebra $\mathfrak{h}$ and a positive root system $\Delta^+$, we consider representations[38] with unique highest weight vector $|\mu\rangle$ annihilated by all positive root operators $\hat{E}_\alpha$. We can distinguish the following cases:

**Compact Cartan subalgebra.** We refer to a chosen Cartan subalgebra $\mathfrak{h}$ as compact if the Killing form $\mathcal{K}$ restricted to $\mathfrak{h}$ is negative definite. In this case, all roots $\alpha \in \Delta$ are imaginary,

---

[38]For non-compact Lie groups, any non-trivial unitary representation will be infinite dimensional, in which case there are also representations without highest weight vectors. We do not consider those.

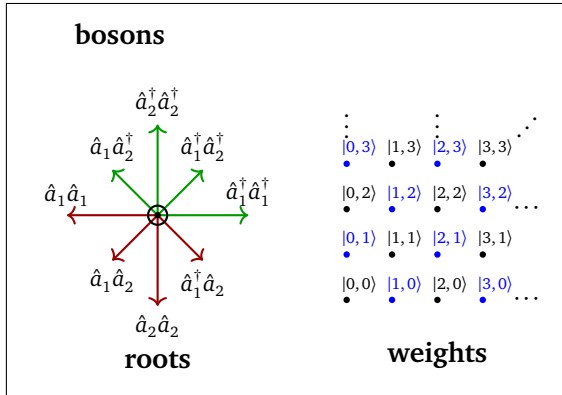
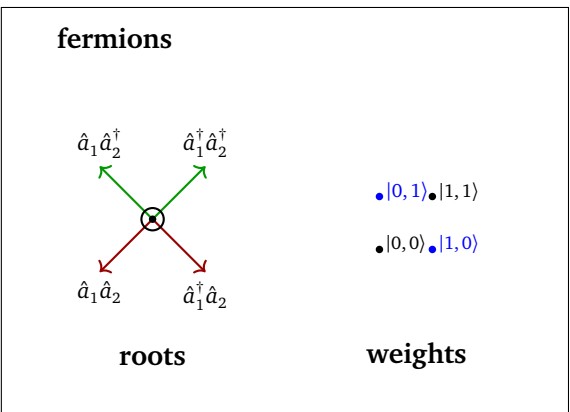

Figure 8: *Root and weight diagrams for* $\mathfrak{sp}(4,\mathbb{R})$ *and* $\mathfrak{so}(4,\mathbb{R})$. We illustrate the roots and weights for the squeezing representation $\mathcal{S}(M)$ in the case of two bosonic (left) and two fermionic (right) modes. The corresponding Lie algebras $\mathfrak{g}$ are $\mathfrak{sp}(4,\mathbb{R})$ and $\mathfrak{so}(4,\mathbb{R})$, respectively. As Cartan subalgebra we choose $\hat{H}_I \equiv (i\hat{n}_1 \pm \frac{i}{2}, i\hat{n}_2 \pm \frac{i}{2})$. There are eight bosonic and four fermionic roots. The root vectors are purely imaginary, allowing us to represent $\text{Im}(\alpha_I)$ by arrows in two dimensions (red = positive roots, green = negative roots). The weight vectors are the number eigenstates $|n_1, n_2\rangle$ with highest weight vector $|0,0\rangle$. They can also be represented in two dimensions by plotting on each axis their eigenvalue with respect to one of the Cartan algebra elements (black dots = even sector, blue dots = odd sector).

*i.e.,* all $\alpha(H_I) \in i\mathbb{R}$, and the positive roots $\Delta^+$ can be divided into two sets[39]: the set $\Delta_{\mathfrak{k}}$ of compact roots and the set $\Delta_{\mathfrak{p}}$ of non-compact roots. The associated root operators $E_\alpha$ satisfy $E_\alpha^* = \mp E_{-\alpha}$ with $(-)$ for compact roots and $(+)$ for non-compact roots. A real basis of the real algebra $\mathfrak{g}$ is then

$$\Xi_i \equiv (H_I, Q_{\alpha_{\mathfrak{k}}}, P_{\alpha_{\mathfrak{k}}}, iQ_{\alpha_{\mathfrak{p}}}, iP_{\alpha_{\mathfrak{p}}}), \tag{322}$$

where $\alpha_{\mathfrak{k}} \in \Delta_{\mathfrak{k}}, \alpha_{\mathfrak{p}} \in \Delta_{\mathfrak{p}}$. At this stage, the proof of proposition 14 can be directly applied to above basis $\Xi_i$ and $\mathcal{M}_\phi$ is Kähler.

**Non-compact Cartan subalgebra.** Whenever the Killing form restricted to our chosen Cartan subalgebra is not negative definite, some roots $\alpha \in \Delta$ may be non-imaginary, in which case $\mathcal{M}_\mu$ constructed from the associated highest weight vector $|\mu\rangle$ may not be Kähler. Only if *all* the non-imaginary root operators $\hat{E}_\alpha$ annihilate the highest weight state $|\mu\rangle$, *i.e.,* $E_\alpha |\mu\rangle = E_{-\alpha} |\mu\rangle = 0$ for all non-imaginary roots $\alpha$, we can once again apply the proof of proposition 14. Indeed, in this case the basis of the real Lie algebra $\mathfrak{g}$ will be given by (322) plus some real basis vectors constructed from the non-imaginary root spaces. Those additional basis vectors, however, annihilate the reference state and thus do not play a role in the properties of the tangent space and $\mathcal{M}_\phi$ will again be Kähler. In summary, a highest weight vector $|\mu\rangle$ will give rise to a Kähler manifold $\mathcal{M}_\mu$ if it is not only annihilated by the positive imaginary root operators $\hat{E}_\alpha$, but also by all non-imaginary root operators $\hat{E}_{\pm\alpha}$.

Much less is known for general Lie groups that are not semi-simple, because their Lie algebras are still not classified. Instead one can attempt to apply the presented analysis case by case. Most importantly, this analysis works for the prominent family of regular coherent

---

[39]If the algebra is not compact, it can be divided into the two subspaces $\mathfrak{g} = \mathfrak{k} \oplus \mathfrak{p}$, such that the Killing form is negative definite on $\mathfrak{k}$ and positive definite on $\mathfrak{p}$. Having a compact Cartan subalgebra means $\mathfrak{h} \subset \mathfrak{k}$ and is equivalent to having only imaginary roots. An imaginary root $\alpha$ is respectively compact or non-compact if $\mathfrak{v}_\alpha$ and $\mathfrak{v}_{-\alpha}$ are contained in $\mathfrak{k}^{\mathbb{C}}$ or $\mathfrak{p}^{\mathbb{C}}$.

states, constructed from the Heisenberg algebra that is not semi-simple, as we will explain in example 25.

**Example 24.** *Bosonic and fermionic Gaussian states $|J, 0\rangle$ as defined in section 5.1 are group theoretic coherent states with respect to the groups $\mathrm{Sp}(2N, \mathbb{R})$ and $\mathrm{O}(2N, \mathbb{R})$ respectively, as defined in (192). The corresponding algebras $\mathfrak{sp}(2N, \mathbb{R})$ and $\mathfrak{so}(2N, \mathbb{R})$ are represented by anti-Hermitian quadratic combinations of the linear phase space operators $\hat{\xi}^a$, as in (194). For bosons this representation is reducible and decomposes over the even and odd part of the Hilbert space, i.e., $\mathcal{H} = \mathcal{H}_+ \oplus \mathcal{H}_-$ where $\mathcal{H}_\pm$ are the eigenspace of the parity operator $P = e^{\mathrm{i}\pi\hat{N}}$ with $\hat{N}$ being a total number operator. For fermions, the group representation is irreducible due $\mathcal{S}(M_v)$ defined in (198), which mixes the two sectors, while the algebra representation still splits over the even and odd sector, as the bosonic case. The algebra is $N(2N + 1)$ dimensional for bosons and $N(2N - 1)$ dimensional for fermions.*

*We can always select creation and annihilation operators $\hat{a}_i^\dagger$ and $\hat{a}_i$ with number operators $\hat{n}_i = \hat{a}_i^\dagger \hat{a}_i$. This allows us to choose the N-dimensional Cartan subalgebra, whose basis is represented as $H_I \equiv (\mathrm{i}\hat{n}_1 \pm \frac{\mathrm{i}}{2}, \ldots, \mathrm{i}\hat{n}_N \pm \frac{\mathrm{i}}{2})$, where (+) applies to bosons and (−) to fermions. We divide the associated $N(2N - 1 \pm 1)$ roots into positive and negative roots and pick the corresponding basis vectors as*

$$\hat{E}_\alpha \in \begin{cases} \{\mathrm{i}\hat{a}_i\hat{a}_j, \hat{a}_k^\dagger \hat{a}_l | i \le j, k < l\} & \textbf{(bosons)} \\ \{\hat{a}_i\hat{a}_j, \hat{a}_k^\dagger \hat{a}_l | i < j, k < l\} & \textbf{(fermions)} \end{cases}, \tag{323}$$

$$\hat{E}_{-\alpha} \in \begin{cases} \{\mathrm{i}\hat{a}_i^\dagger \hat{a}_j^\dagger, \hat{a}_l^\dagger \hat{a}_k | i \le j, k < l\} & \textbf{(bosons)} \\ \{\hat{a}_j^\dagger \hat{a}_i^\dagger, \hat{a}_l^\dagger \hat{a}_k | i < j, k < l\} & \textbf{(fermions)} \end{cases}. \tag{324}$$

*With this choice of real Cartan subalgebra all roots are imaginary. Only the roots associated to $\hat{a}_i\hat{a}_j$ and $\hat{a}_i^\dagger \hat{a}_j^\dagger$ for bosons are non-compact, while all others are compact. We verify $\hat{E}_\alpha^\dagger = \pm \hat{E}_{-\alpha}$, with (+) for compact and (−) for non-compact roots. Vice versa $E_\alpha^* = \mp E_{-\alpha}$, with (−) for compact and (+) for non-compact roots. With this choice, the weight vectors of the representation on $\mathcal{H}_+$ are the states $|n_1, \cdots, n_N\rangle$ with fixed excitation numbers $n_i$ for all $\hat{n}_i$ and $\sum_i n_i$ even. The highest weight vector is $|0, \ldots, 0\rangle$. The representation on $\mathcal{H}_-$ is the same, except that we require $\sum_i n_i$ to be odd, such that the highest weight vector is $|1, 0, \ldots, 0\rangle$. For fermions, the highest weight families $\mathcal{M}_\phi$ of the two sectors are actually a single family (consisting of two disconnected components), because we also represent group elements $M$ with $\det M = -1$ in our representation. For bosons, only the family constructed from the highest weight, $|0, \ldots, 0\rangle$, in the even representation is called Gaussian. Let us highlight that the family $\mathcal{M}_\phi$ for $|\phi\rangle = |1, 0, \ldots, 0\rangle$ is an interesting Kähler family whose properties are not fully explored. For $N = 2$, the Cartan subalgebra is 2-dimensional, which allows us to plot roots and weights in the plane, as done in figure 8.*

**Example 25.** *Standard bosonic coherent states are probably the most well-known type of coherent states used in physics. These are generated from the displacement transformations $\mathcal{D}(z)$ introduced in (197). While $\mathcal{D}(z)$ is a projective representation of the Abelian group of phase space translations $(V, +)$ with $\mathcal{D}(z)\mathcal{D}(\tilde{z}) = e^{\mathrm{i}z^a \omega_{ab}\tilde{z}^b}\mathcal{D}(z + \tilde{z})$, this is actually not true for its Lie algebra. Instead, one can consider $e^{\mathrm{i}\varphi}\mathcal{D}(z)$ as a representation of the Heisenberg group with $(2N + 1)$-dimensional Lie algebra represented by the anti-Hermitian operators $\hat{\Xi}_i \equiv (\mathrm{i}\hat{q}_1, \mathrm{i}\hat{p}_1, \ldots, \mathrm{i}\hat{q}_N, \mathrm{i}\hat{p}_N, \mathrm{i}\mathbb{1})$. This Lie algebra is not semi-simple and the standard approach of computing roots fails. However, if we extend the Lie algebra even further to also include the N elements represented as number operators $\mathrm{i}\hat{n}_i = \mathrm{i}\hat{a}_i^\dagger \hat{a}_i = \frac{\mathrm{i}}{2}(\hat{q}_i^2 + \hat{p}_i^2 - 1)$, we can construct a root diagram. The Cartan subalgebra is then represented by $\hat{H}_I \equiv (\mathrm{i}\hat{n}_1, \ldots, \mathrm{i}\hat{n}_N, \mathrm{i}\mathbb{1})$ with root vectors $\hat{a}_i$ and $\hat{a}_i^\dagger$. Consequently, the roots form an orthonormal basis, the eigenstates $|n_1, \ldots, n_N\rangle$ of the number operators $\hat{n}$ are the weight states and $|0, \ldots, 0\rangle$ is the highest weight state, as for bosonic Gaussian states.*

To summarize we have the following cases:

- **Semi-simple, compact algebra.** Any compact group $\mathcal{G}$ has a compact Lie algebra $\mathfrak{g}$, for which the family $\mathcal{M}_\phi$ is Kähler if $|\phi\rangle$ is the highest weight state of the representation. *See example 24 of fermionic Gaussian states.*

- **Semi-simple, non-compact algebra.** If the group is non-compact, not all highest weight vectors give rise to Kähler manifolds. For every highest weight vectors associated to the real Cartan subalgebra $\mathfrak{h} \subset \mathfrak{g}$, we need to split the corresponding roots into imaginary and non-imaginary roots. Only highest weight vectors that are also annihilated by all (not just the positive) non-imaginary root vectors $E_\alpha$ will give rise to Kähler manifolds. This includes in particular highest weight vectors, whose Cartan subalgebra is compact and thus has only imaginary roots. *See example 24 of bosonic Gaussian states.*

- **Not semi-simple algebra.** The Kähler properties of the manifold have to be checked case by case, as there is no general classification theory. In the physical important case of the Heisenberg algebra (regular coherent states), the presented construction still works. *See example 25 of coherent states.*

Our proofs and discussions only showed one direction, namely that $\phi$ being a highest weight vector (and annihilated by non-imaginary roots, if there are any) implies that $\mathcal{M}_\phi$ is Kähler. According to [90, 93], the opposite is also true, *i.e.*, group theoretic coherent states are Kähler if and only if they constructed from such a state $\phi$. According to [90], variational families $\mathcal{M}_\phi$ that are Kähler also satisfy the conditions of so called symmetric spaces [94].

### 5.2.4 Co-adjoint orbits

In the context of group theoretic coherent states, one often transforms the problem of describing the geometry of $\mathcal{M}_\phi$ to a real submanifold $\mathcal{O}_\phi$ of the dual Lie algebra $\mathfrak{g}^*$, which is called co-adjoint orbit. Using this language is particularly useful when we can establish an isomorphism $\mathcal{M}_\phi \simeq \mathcal{O}_\phi$, *i.e.*, when they are equivalent manifolds.

**Definition 6.** *Given a non-zero state $|\phi\rangle \in \mathcal{H}$ and $M \in \mathcal{G}$, we define the dual Lie algebra element $\beta_M \in \mathfrak{g}^*$ as*

$$\beta_M : \mathfrak{g} \to \mathbb{R}; A \mapsto \mathrm{i} \frac{\langle \phi | \mathcal{U}^\dagger(M) \hat{A} \mathcal{U}(M) | \phi \rangle}{\langle \phi | \phi \rangle} . \tag{325}$$

*This gives rise to its coadjoint orbit as the submanifold*

$$\mathcal{O}_\phi = \{\beta_M | M \in \mathcal{G}\} \subset \mathfrak{g}^* , \tag{326}$$

*where we can compute* $(\beta_M)_i = (\beta_\mathbb{1})_j (\mathrm{Ad}_M)^j{}_i$.

The motivation for introducing the coadjoint orbit is that $\mathcal{O}_\phi$ and $\mathcal{M}_\phi$ will turn out to coincide under certain conditions, including the important case where $\mathcal{M}_\phi$ is of Kähler type. Analogous to $H_\phi$ and $\mathfrak{h}_\phi$, we define

$$\begin{aligned} S_\phi &= \{M \in G \,|\, \beta_M = \beta_\mathbb{1}\}, \\ \mathfrak{s}_\phi &= \{A \in \mathfrak{g} \,|\, (\beta_\mathbb{1})_k (\mathrm{ad}_A)^k{}_j = (\beta_\mathbb{1})_k (A^i c^k_{ij}) = 0\}, \end{aligned} \tag{327}$$

which are the stabilizer subgroup of the dual vector $\beta_\mathbb{1} = \mathrm{i} \langle \phi | \hat{\Xi}_i | \phi \rangle / \langle \phi | \phi \rangle$ and the associated Lie algebra. In other words, $S_\phi$ behaves to the orbit $\mathcal{O}_\phi$ just like $H_\phi$ to $\mathcal{M}_\phi$.

**Proposition 15.** *We always have $\mathfrak{h}_\phi \subset \mathfrak{s}_\phi$ and $H_\phi \subset S_\phi$. If they are equal, $\mathcal{O}_\phi$ and $\mathcal{M}_\phi$ are isomorphic, i.e.,*

$$\mathfrak{h}_\phi = \mathfrak{s}_\phi \quad \Leftrightarrow \quad \mathcal{M}_\phi \simeq \mathcal{G}/H_\phi = \mathcal{G}/S_\phi \simeq \mathcal{O}_\phi. \tag{328}$$

*Proof.* We have $H_\phi \subset S_\phi$, because for $M \in H_\psi$ we have

$$\beta_M(A) = \mathrm{i}\frac{\langle\psi|\mathcal{U}^\dagger(M)\hat{A}\mathcal{U}(M)|\psi\rangle}{\langle\phi|\phi\rangle} = \mathrm{i}\langle\psi|\hat{A}|\psi\rangle = \beta_{\mathbb{1}}(A). \tag{329}$$

Consequently, we also have $\mathfrak{h}_\phi \subset \mathfrak{s}_\phi$. $\qquad\square$

Only if $H_\phi = S_\phi$, we will have the equality while $\mathcal{M}_\phi$ is in general larger than $\mathcal{O}_\phi$, *i.e.*, there may be distinct states $\mathcal{U}(M)|\psi\rangle \in \mathcal{M}_\phi$ for which $\beta_M$ are the same. The following proposition allows the efficient computation of $\boldsymbol{\omega}_{ij}$ from $\beta_{\mathbb{1}}$.

**Proposition 16.** *The symplectic form $\boldsymbol{\omega}_{ij}$ on $\mathcal{M}_\phi$ is*

$$\boldsymbol{\omega}_{ij} = \beta_{\mathbb{1}}([\Xi_i, \Xi_j]) = (\beta_{\mathbb{1}})_k c_{ij}^k. \tag{330}$$

*It is non-degenerate if and only if $\mathfrak{h}_\phi = \mathfrak{s}_\phi$ or, equivalently, $\mathcal{M}_\phi = \mathcal{O}_\phi$. This directly implies*

$$\mathcal{M}_\phi \text{ is Kähler} \quad \Rightarrow \quad \mathcal{M}_\phi = \mathcal{O}_\phi. \tag{331}$$

*Proof.* We compute explicitly

$$\begin{aligned}
\boldsymbol{\omega}_{ij} &= \frac{\langle\phi|\hat{\Xi}_j\mathbb{Q}_\phi\hat{\Xi}_i - \hat{\Xi}_i\mathbb{Q}_\phi\hat{\Xi}_j|\phi\rangle}{\mathrm{i}\langle\phi|\phi\rangle} \\
&= \frac{\langle\phi|[\hat{\Xi}_j,\hat{\Xi}_i]|\phi\rangle}{\langle\phi|\phi\rangle} + \frac{2\mathrm{Im}(\langle\phi|\Xi_i|\phi\rangle\langle\phi|\Xi_j|\phi\rangle)}{\langle\phi|\phi\rangle^2} \\
&= \frac{\mathrm{i}\langle\phi|c_{ij}^k\hat{\Xi}_k|\phi\rangle}{\langle\phi|\phi\rangle} + 0 = c_{ij}^k(\beta_{\mathbb{1}})_k.
\end{aligned} \tag{332}$$

Degeneracy of $\boldsymbol{\omega}_{ij}$ on tangent space means that there is a non-zero $A^i|V_i\rangle \neq 0$ with $A^i\boldsymbol{\omega}_{ij} = 0$. Recalling

$$\mathfrak{h}_\phi = \{A^i\Xi_i \,|\, A^i|V_i\rangle = 0\}, \tag{333}$$
$$\mathfrak{s}_\phi = \{A^i\Xi_i \,|\, (\beta_{\mathbb{1}})_k(A^ic_{ij}^k) = A^i\boldsymbol{\omega}_{ij} = 0\}, \tag{334}$$

implies such $A^i$ cannot exist if and only if $\mathfrak{h}_\phi = \mathfrak{s}_\phi$. $\qquad\square$

The conclusion of this proposition is that $\boldsymbol{\omega}_{ij}$ is only non-degenerate if $\mathcal{M}_\phi = \mathcal{O}_\phi$ and vice versa. It is well-known in the theory of Lie groups [95] that any coadjoint orbit comes naturally equipped with a non-degenerate symplectic form and here we see that it agrees with the one on $\mathcal{M}_\phi$ if $\mathcal{M}_\phi = \mathcal{O}_\phi$.

**Example 26.** *Let us consider the example of* SU(2) *for the spin-$\frac{1}{2}$ and spin-1 representation a final time. The co-adjoint representation is isomorphic to the spin-1 representation (just like the adjoint) and can be understood as real 3-by-3 rotation matrices acting on real dual vectors $(\beta_{\mathbb{1}})_k$. Therefore, all co-adjoint orbits for $\beta_{\mathbb{1}} \neq 0$ are 2-spheres, while the orbit $\beta_{\mathbb{1}} = 0$ is the single point $\{0\}$.*

*Spin-$\frac{1}{2}$. We recall that there is only a single $\mathcal{M}_\phi = \mathcal{P}(\mathcal{C}^2)$, so that we can just compute $\beta_{\mathbb{1}}$ for the representative $|\phi\rangle \equiv (1, 0)$. This gives the dual vector $\beta_{\mathbb{1}} \equiv (0, 0, \frac{1}{2})$. Unsurprisingly, we have the orbit $\mathcal{O}_\phi \simeq \mathcal{M}_\phi \simeq S^2$.*

*Spin-1. We were able to parametrize all possible families $\mathcal{M}_\phi$ by a representative $|\phi\rangle \equiv (\cos\theta, \mathrm{i}\sin\theta, 0)$ for $0 \leq \theta \leq \frac{\pi}{4}$. From here, we find $\beta_{\mathbb{1}} \equiv (0, 0, \sin 2\theta)$. Consequently, the orbit $\mathcal{O}_\phi$ will be a sphere for $\theta > 0$. However, only for $\theta = \frac{\theta}{4}$, i.e., for $\phi$ being of highest weight, we have $\mathcal{M}_\phi \simeq \mathcal{O}_\phi$.*

### 5.2.5 Numerical implementation

The goal of this section is to explain how any class of coherent states, be it Kähler or non-Kähler, allows for an efficient implementation of real and imaginary time evolution. Here, we use the Lagrangian action for real time dynamics. The key advantage of coherent states lies in the fact that we can use the Lie algebra to identify the different tangent spaces, such that symplectic form $\mathbf{\Omega}^{ij}$ and metric $\mathbf{G}^{ij}$ does not need to be evaluated at every step. This provides a significant computational advantage compared to a naive implementation without taking the natural group structure into account.

In practice, this reduces the calculation of real and imaginary time evolution tremendously. Instead of needing to compute metric and symplectic form at every step with respect to given coordinates, we can parametrize states by matrices $M$ and use the identification

$$\mathcal{T}_{\psi(M)}\mathcal{M}_\phi \simeq \mathfrak{h}_\phi^\perp \tag{335}$$

from definition 5 to evaluate $\mathbf{G}^{ij}$ or $\mathbf{\Omega}^{ij}$ once based on proposition 13. This gives the following dynamics.

**Proposition 17.** *The equations of motion for $M$ are*

$$\frac{dM}{dt}(t) = M\,\Xi_i\,\mathcal{X}^i\,, \tag{336}$$

$$\frac{dM}{d\tau}(\tau) = -M\,\Xi_i\,\mathbf{G}^{ij}(\partial_i E) \tag{337}$$

*for real time and imaginary time evolution respectively, where $\mathcal{X}^i$ is given by (88) for Lagrangian and by (93) for McLachlan evolution.*

*Proof.* At each point $M(s)$, with respect to the local coordinates introduced in (290) the time evolution is

$$\frac{d}{ds}M = \frac{d}{ds}M(s) \equiv \frac{d}{ds}(M e^{x^i(s)\Xi_i}) = M\Xi_i \dot{x}^i, \tag{338}$$

where $s = t$ and $\dot{x}^i$ is given by $\mathcal{X}^i$ from (88) or (93) for real time evolution and $s = \tau$ and $\dot{x}^i$ is given by $\mathcal{F}^i$ from (164) for imaginary time evolution. $\qquad\square$

Consequently, a numerical implementation can be based on the following algorithm.

1. Choose a basis $\Xi_i$ of $\mathfrak{h}_\phi^\perp$ represented as matrices and compute the required geometric structure, such as $\mathbf{g}_{ij}$ or $\boldsymbol{\omega}_{ij}$ and their inverses $\mathbf{G}^{ij}$ or $\mathbf{\Omega}^{ij}$. In many situations, it is convenient to choose $\Xi_i$, such that the associated geometric structures take some standard forms.

2. Compute the gradient of $E(M)$ at $M$ as

$$\partial_i E(M) := \frac{d}{ds}f(M e^{s\Xi_i})\Big|_{s=0}. \tag{339}$$

   The evaluation of this derivative is the only problem specific piece to be implemented. For Gaussian states $|\psi(M)\rangle = |MJ_0M^{-1}, z\rangle$, the energy function $E(M) = \langle MJ_0M^{-1}, z|\hat{H}|MJ_0M^{-1}, z\rangle$ can be evaluated analytically using Wick's theorem,

such that also its derivative can be found as formal expression in terms of $M$ and $\Xi_i$. The evolution is then

$$
X^i(M) = \begin{cases} -\boldsymbol{\Omega}^{ij}(\partial_j E)(M) & \textbf{(Lagrangian)} \\ -\boldsymbol{G}^{ij}\frac{2\mathrm{Re}\langle\mathcal{V}_j|\mathrm{i}\hat{H}|\psi\rangle}{\langle\psi|\psi\rangle} & \textbf{(McLachlan)} \\ -\boldsymbol{G}^{ij}(\partial_j E)(M) & \textbf{(gradient)} \end{cases} \tag{340}
$$

for real time (Lagrangian or McLachlan) or imaginary time evolution, respectively. Let us emphasize that $\boldsymbol{G}^{ij}$ or $\boldsymbol{\Omega}^{ij}$ does not need to be evaluated again, as it does not depend on $M$ when parametrized in this way.

3. The evolution equation is then solved by performing discrete steps. Starting with some initial matrix $M_0$, we compute

$$
M_{n+1} = M_n \, e^{\epsilon \delta M_n} \approx M_n \left( \frac{\mathbb{1} + \frac{\epsilon}{2}\,\delta M_n}{\mathbb{1} - \frac{\epsilon}{2}\,\delta M_n} \right), \tag{341}
$$

where $\delta M_n = X^i(M_n)\Xi_i$. Here, we approximate the exponential for small $\epsilon$, such that $M_{n+1} \in \mathcal{G}$.

Note that we need to choose $\epsilon$ sufficiently small to ensure that the denominator has eigenvalues close to 1, which can be achieved by choosing $\epsilon^{-1}$ larger than the largest eigenvalue of $\delta M_n$. For imaginary time evolution, we need further to choose $\epsilon$ sufficiently small, such that the energy decreases in each step. For real time Langrangian evolution, the energy is only preserved for infinitesimal steps, while for finite $\epsilon$ some deviation may occur. There exist various numerical schemes, including symplectic integration, to deal with this issue more efficiently [53, 54]. For real time evolution with varying step sizes, it is important to keep track of the passed time to capture the correct dynamics of observables. This is less important for imaginary time evolution where we are mostly interested in the convergence to the energy minimum. The algorithm can be further enhanced by approximating the exponential function in (341) to higher order, similar to higher order Runge-Kutta methods.

In [59], this method is used to study the ground state properties of the Bose-Hubbard model in the superfluid phase. Apart from real and imaginary time evolution, the prescribed method can be used for any functional optimization on the constructed manifold $\mathcal{M}$. In [96], these methods are used to evaluate entanglement and complexity of purification for Gaussian states, which requires a minimization over all Gaussian purifications of a given mixed state.

## 5.3 Generalized Gaussian states

In the previous section we have considered manifolds made up of states of the form $\mathcal{U}(M)|\phi\rangle$, where, as in definition 4, $\mathcal{U}$ is a unitary representation of the Lie group $\mathcal{G}$, $M$ is an element of $\mathcal{G}$ and $|\phi\rangle$ is a chosen reference state. We have seen that the geometric properties of the manifold defined in this way crucially depend on the choice of $|\phi\rangle$. Indeed, if $|\phi\rangle$ is chosen as a highest weight vector of the representation then the Kähler property of the manifold follows naturally from the group structure. On the other hand if $|\phi\rangle$ is not a highest weight vector, then the Kähler property is not guaranteed.

We can schematically distinguish the following cases:

- **Highest weight state** $|\phi\rangle$
  The family $|\psi(M)\rangle = \mathcal{U}(M)|\phi\rangle$ is defined taking $|\phi\rangle$ as a highest weight state with

respect to the representation $\mathcal{U}(M)$. If $\mathcal{G}$ is a compact semi-simple Lie group, the resulting variational family is always Kähler[40]. As discussed in examples 24 and 25, several commonly used variational fall into this class, in particular coherent states and general bosonic or fermionic Gaussian states. We have reviewed in detail the Kähler properties of bosonic and fermionic Gaussian states in section 5.1.

- **Fixed non-highest weight state $|\phi\rangle$**
  The family $|\psi(M)\rangle = \mathcal{U}(M)|\phi\rangle$ is defined taking a reference state $|\phi\rangle$ which is not a highest weight state with respect to the representation $\mathcal{U}(M)$. The resulting variational family is typically non-Kähler. Nonetheless, these families maintain the advantages discussed in section 5.2, namely a natural basis in every tangent space, with respect to which the restricted Kähler structures have the same matrix representations and need not to be evaluated at every step.

- **Family of reference states $|\phi(x)\rangle$**
  The family $|\psi(M)\rangle = \mathcal{U}(M)|\phi(x)\rangle$ is defined taking a reference state $|\phi(x)\rangle$ that depends on additional parameters. As $|\phi(x)\rangle$ can include highest weight states, the full manifold will in general consist of individual sheets labelled by $x$, which may be partially Kähler and partially non-Kähler (making the full family non-Kähler). Such families were recently [32] used to construct non-Gaussian states with correlations between bosonic and fermionic degrees of freedom, as we discuss in section 5.3.2.

We will now illustrate this scheme in more detail by focusing on a specific choice of Lie group and representation, *i.e.*, the group of Gaussian unitaries $\mathcal{U}(M, z)$ introduced in (199). These are unitaries that can be written as the exponential of anti-Hermitian operators at most quadratic in the linear operators $\hat{\xi}^a$ and, as seen in example 24, they give a representation of the Lie groups $\mathrm{ISp}(2N, \mathbb{R})$ for bosons and $\mathrm{O}(2N, \mathbb{R})$ for fermions.

As warm up, we will take the simple case of a single bosonic mode. Here, all the above cases emerge depending on the choice of reference state $|\phi\rangle$. Then we will move to the general case of an arbitrary number of bosonic and fermionic modes, where we will use the formalism of the previous sections to define the variational family of generalized Gaussian states [32] in the group-theoretic language and study their Kähler properties.

### 5.3.1 Warm up examples (single bosonic mode)

Let us consider a single bosonic mode defined by the creation and annihilation operators $\hat{b}^\dagger$ and $\hat{b}$ or the quadratures $\hat{q} = \frac{1}{\sqrt{2}}(\hat{b}^\dagger + \hat{b})$ and $\hat{p} = \frac{i}{\sqrt{2}}(\hat{b}^\dagger - \hat{b})$. As discussed in example 24, in this case the Gaussian algebra is spanned by the Cartan subalgebra element $\hat{H} = i(1 + 2\hat{b}^\dagger\hat{b})$ and by the elements $E_+ = \hat{b}\hat{b}$ and $E_- = \hat{b}^\dagger\hat{b}^\dagger$, corresponding to the one positive and one negative root of the algebra. Then, the real basis (309) of the algebra is represented by the operators

$$\hat{\Xi}_1 \equiv \widehat{X} = \hat{b}^{\dagger 2} - \hat{b}^2 \tag{342}$$

$$\hat{\Xi}_2 \equiv \widehat{Y} = i(\hat{b}^{\dagger 2} + \hat{b}^2) \tag{343}$$

$$\hat{\Xi}_3 \equiv \widehat{Z} = i(\hat{n} + \tfrac{1}{2}), \tag{344}$$

---

[40]If the group is not compact or not semi-simple $|\phi\rangle$ may have to satisfy further conditions, as we will discuss in section 5.2.3.

where we followed the conventions of example 13. Indeed, the most general Gaussian squeezing operator in such system can be then written as

$$\mathcal{U}(x, y, z) = e^{x\hat{X} + y\hat{Y} + z\hat{Z}} \,. \tag{345}$$

We then consider the manifold of group theoretic coherent states of the form

$$|\psi(x, y, z)\rangle = \mathcal{U}(x, y, z)|\phi\rangle \,. \tag{346}$$

As discussed in the context of defintion 5, for states of this form there exists a natural isomorphism between the tangent spaces at different points, which are all equivalent to a subspace of the associated Lie algebra. It is thus sufficient to consider the tangent space at the point $|\psi(0, 0, 0)\rangle = |\phi\rangle$. At this point, tangent space is spanned by the vectors

$$
\begin{aligned}
|\mathcal{V}_1\rangle &\equiv \mathbb{Q}_\phi (\hat{b}^{\dagger 2} - \hat{b}^2)|\phi\rangle \,, \\
|V_2\rangle &\equiv i\mathbb{Q}_\phi (\hat{b}^{\dagger 2} + \hat{b}^2)|\phi\rangle \,, \\
|V_3\rangle &\equiv i\mathbb{Q}_\phi (1 + 2\hat{n})|\phi\rangle \,,
\end{aligned}
\tag{347}
$$

which also coincide with the ones introduced in (291). The form of these states, after applying the projector $\mathbb{Q}$, depends on the specific choice of $|\phi\rangle$.

We will now review some possible choices, showing that if we choose a highest weight state of the representation (first case) we will obtain a Kähler manifold, while if we do not (subsequent cases) we can construct non-Kähler manifolds of the different types discussed in the previous sections.

**Kähler (Gaussian) case $|\phi\rangle = |0\rangle$.** As already discussed extensively, choosing $|\phi\rangle$ as the Fock vacuum leads to the manifold of one-mode bosonic squeezed states. We have seen that these form a Kähler manifold, as can be expected in the light of what discussed in Section 5.2. Indeed, $|0\rangle$ is the highest weight state of the representation of $\mathrm{Sp}(2, \mathbb{R})$ we are using, so the resulting manifold is Kähler by Proposition 14. More concretely, we see that the last vector in (347) will be proportional to $|\phi\rangle$ and thus will vanish once the projector $\mathbb{Q}_\phi$ is applied. We are then left with a two dimensional tangent space, spanned by the first two which form a Kähler pair. Overall we indeed have

$$|\mathcal{V}_1\rangle \equiv \sqrt{2}|2\rangle \,, \quad |\mathcal{V}_2\rangle \equiv i\sqrt{2}|2\rangle \,, \quad |\mathcal{V}_3\rangle \equiv 0 \,. \tag{348}$$

Note that, as we are only considering squeezing and not displacements, a similar behaviour will appear also for $|\phi\rangle = |1\rangle$. This is indeed the the highest weight state of the odd sector of the representation, as discussed in example 24.

**Non-Kähler non-degenerate case $|\phi\rangle = |2\rangle$.** If we choose $|\phi\rangle$ as a Fock state $|n\rangle$ with $n \geq 2$ then we will similarly have that the the last vector in (347) vanishes after applying $\mathbb{Q}_\phi$. The remaining two vectors will however not form a conjugate pair. For $n = 2$, we find

$$|\mathcal{V}_1\rangle \equiv \sqrt{12}|4\rangle - \sqrt{2}|0\rangle \,, \tag{349}$$

$$|\mathcal{V}_2\rangle \equiv i(\sqrt{12}|4\rangle + \sqrt{2}|0\rangle) \,, \tag{350}$$

$$|\mathcal{V}_3\rangle \equiv 0 \,. \tag{351}$$

Through simple calculations one can see that in this case the symplectic form $\boldsymbol{\omega}$ will be invertible, but the complex structure $\boldsymbol{J}$ will, in the basis $\{|\mathcal{V}_1\rangle, |\mathcal{V}_2\rangle\}$, have the form

$$
\boldsymbol{J} = \begin{pmatrix} 0 & -5/7 \\ 5/7 & 0 \end{pmatrix} \,,
\tag{352}
$$

which clearly does not square to $-\mathbb{1}$.

**Non-Kähler degenerate case** $|\phi\rangle = \frac{1}{\sqrt{2}}(|0\rangle + |2\rangle)$**.** If we choose a $|\phi\rangle$ that is not an eigenstate of $\hat{n}$, for example a superposition of two different Fock states, then it will not be a weight state of the representation. In this case none of the vectors in (347) will vanish after applying $\mathbb{Q}_\phi$. Therefore, we will have a three dimensional tangent space. Being odd-dimensional, it cannot admit an invertible symplectic form $\boldsymbol{\omega}$. In particular, for $|\phi\rangle = \frac{1}{\sqrt{2}}(|0\rangle + |2\rangle)$ we have

$$|\mathcal{V}_1\rangle \equiv \sqrt{6}\,|4\rangle + |2\rangle - |0\rangle \,, \tag{353}$$

$$|\mathcal{V}_2\rangle \equiv \mathrm{i}\sqrt{6}\,|4\rangle \,, \tag{354}$$

$$|\mathcal{V}_3\rangle \equiv \mathrm{i}\sqrt{2}(\sqrt{2} - \sqrt{0})\,. \tag{355}$$

This leads to the complex structure

$$J = \frac{1}{4}\begin{pmatrix} 0 & -3 & -\sqrt{2} \\ 4 & 0 & 0 \\ 2\sqrt{2} & 0 & 0 \end{pmatrix}, \tag{356}$$

from which we have that $J^2$ has eigenvalues $0$, $-1$ and $-1$. We see therefore that the manifold is not naturally Kähler, however if we invert $\boldsymbol{\omega}$ only on the two-dimensional subspace where this is possible (*i.e.*, with the pseudo-inverse), as discussed in section 3.4, then the resulting manifold is Kähler.

**Variable reference state** $|\phi(\alpha,\Gamma,z)\rangle = e^{\mathrm{i}\alpha\hat{n}^2}|\Gamma,z\rangle$**.** Finally, we can also consider the case where $|\phi\rangle$ is not a fixed state, but rather depends itself on some parameters that will then be part of the total parameter set of the manifold. In this case, different instances of the possibilities discussed above may occur at different points of the manifold. In particular, let us consider for $|\phi\rangle$ the states obtained by applying the unitary $e^{\mathrm{i}\alpha\hat{n}^2}$, depending on the single real parameter $\alpha$, to the family of Gaussian states, parametrized by $|\Gamma,z\rangle$ according to the conventions of section 5.1. Here, at a generic point of the manifold, varying the parameter $\alpha$ will lead to a single independent unpaired tangent vector $\mathbb{Q}_\phi\partial_\alpha e^{\mathrm{i}\alpha\hat{n}^2}|\Gamma,z\rangle$, which will necessarily make the tangent space non-Kähler. However at the special points where $J = \mathbb{1}$ and $z = 0$, that is $|\phi\rangle = |0\rangle$ for any $\alpha$, we have that the tangent space will be isomorphic to the one of Gaussian states, that is Kähler.

### 5.3.2 Definition of a class of generalized Gaussian states

As seen in the previous section, choosing variable reference state $|\phi(x)\rangle$ can lead to manifolds with rather elaborate geometric structures. In this section, we will introduce a set of states that can be considered as a generalization of such example to the case of an arbitrary number of bosonic and fermionic modes. These states were first introduced in [32] as a variational *ansatz* in many-body physics that extends Gaussian states. We will define this family in group-theoretic terms, study their Kähler properties and thereby show that they do not form Kähler manifolds. Thus, they are an important example to apply the methods introduced in the previous sections.

We consider a Hilbert space containing both bosonic and fermionic degrees of freedom, *i.e.*, $\mathcal{H} = \mathcal{H}_{\mathrm{b}} \otimes \mathcal{H}_{\mathrm{f}}$ where $\mathcal{H}_{\mathrm{b}}$ is the bosonic Fock space of $N_{\mathrm{b}}$ bosonic modes and $\mathcal{H}_{\mathrm{f}}$ is the fermionic Fock space of $N_{\mathrm{f}}$ fermionic modes. We refer to the classical phase spaces $V_{\mathrm{b}} \simeq \mathbb{R}^{2N_{\mathrm{b}}}$ and $V_{\mathrm{f}} \simeq \mathbb{R}^{2N_{\mathrm{f}}}$. On this space, we fix a basis of bosonic and fermionic linear operators

$$\hat{\xi}_{\mathrm{b}}^{a} \equiv (\hat{q}_1^{\mathrm{b}}, \cdots, \hat{q}_{N_{\mathrm{b}}}^{\mathrm{b}}, \hat{p}_1^{\mathrm{b}}, \cdots, \hat{p}_{N_{\mathrm{b}}}^{\mathrm{b}}), \tag{357}$$

$$\hat{\xi}_{\mathrm{f}}^{a} \equiv (\hat{q}_1^{\mathrm{f}}, \cdots, \hat{q}_{N_{\mathrm{f}}}^{\mathrm{f}}, \hat{p}_1^{\mathrm{f}}, \cdots, \hat{p}_{N_{\mathrm{f}}}^{\mathrm{f}}), \tag{358}$$

which also determine the number operators

$$\hat{n}_{\mathrm{b/f}}^i = \frac{1}{2}\left(\left(\hat{q}_i^{\mathrm{b/f}}\right)^2 + \left(\hat{p}_i^{\mathrm{b/f}}\right)^2 - 1\right) \tag{359}$$

for all $i = 1, \ldots, N_{\mathrm{b/f}}$.

Gaussian states of systems with bosonic and fermionic degrees of freedom are defined as the tensor product

$$|\Gamma_{\mathrm{b}}, z_{\mathrm{b}}, \Gamma_{\mathrm{f}}\rangle \equiv |\Gamma_{\mathrm{b}}, z_{\mathrm{b}}\rangle \otimes |\Gamma_{\mathrm{f}}, 0\rangle \tag{360}$$

and are thus unable to capture correlations between bosons and fermions. This is an important drawback when studying correlated boson-fermion mixtures, which generalized Gaussian states are able to overcome. Gaussian transformations on the mixed Hilbert space $\mathcal{H} = \mathcal{H}_{\mathrm{b}} \otimes \mathcal{H}_{\mathrm{f}}$ are defined as the representation of the group $\mathrm{ISp}(2N_{\mathrm{b}}, \mathbb{R}) \times \mathrm{O}(2N_{\mathrm{f}}, \mathbb{R})$

$$\mathcal{U}(M_{\mathrm{b}}, M_{\mathrm{f}}) = \mathcal{U}_{\mathrm{b}}(M_{\mathrm{b}}, z_{\mathrm{b}}) \otimes \mathcal{U}_{\mathrm{f}}(M_{\mathrm{f}}), \tag{361}$$

with $M_{\mathrm{b}} \in \mathrm{Sp}(2N_{\mathrm{b}}, \mathbb{R})$, $z_{\mathrm{b}} \in V_{\mathrm{b}}$ and $_{\mathrm{f}} \in \mathrm{O}(2N_{\mathrm{f}}, \mathbb{R})$, such that $\mathcal{U}_{\mathrm{b}}$ and $\mathcal{U}_{\mathrm{f}}$ are the representations of respectively bosonic and fermionic Gaussian unitaries defined in section 5.1.

We now consider states of the form $\mathcal{U}(M_{\mathrm{b}}, M_{\mathrm{f}})|\phi(x)\rangle$ for variable choices of the state $|\phi(x)\rangle$. More precisely, we consider the non-Gaussian reference states

$$|\phi(\alpha, \Gamma_{\mathrm{b}}, z_{\mathrm{b}}, \Gamma_{\mathrm{f}})\rangle = \mathcal{U}_{\mathrm{NG}}(\alpha)|\Gamma_{\mathrm{b}}, z_{\mathrm{b}}, \Gamma_{\mathrm{f}}\rangle \, . \tag{362}$$

Here, $\mathcal{U}_{\mathrm{NG}}$ is a non-Gaussian unitary, acting on the full Hilbert space $\mathcal{H} = \mathcal{H}_{\mathrm{b}} \otimes \mathcal{H}_{\mathrm{f}}$, parametrized by the real parameters $\alpha$ and $|\Gamma_{\mathrm{b}}, z_{\mathrm{b}}, \Gamma_{\mathrm{f}}\rangle$ is Gaussian.

There are two important choices for the unitary $\mathcal{U}_{\mathrm{NG}}$, depending on the type of correlation between the bosonic and fermionic sector one wishes to introduce,

$$\mathcal{U}_{\mathrm{NG}}^{(1)}(\alpha^{\mathrm{b}}, \alpha^{\mathrm{f}}, \alpha^{\mathrm{bf}}) = e^{\mathrm{i}(\alpha_{ij}^{\mathrm{b}}\hat{n}_{\mathrm{b}}^i\hat{n}_{\mathrm{b}}^j + \alpha_{\tilde{i}\tilde{j}}^{\mathrm{f}}\hat{n}_{\mathrm{f}}^{\tilde{i}}\hat{n}_{\mathrm{f}}^{\tilde{j}} + \alpha_{i\tilde{j}}^{\mathrm{bf}}\hat{n}_{\mathrm{b}}^i\hat{n}_{\mathrm{f}}^{\tilde{j}})}, \tag{363}$$

$$\mathcal{U}_{\mathrm{NG}}^{(2)}(\alpha^{\mathrm{b}}, \alpha^{\mathrm{f}}, \alpha^{\mathrm{bf}}) = e^{\mathrm{i}(\alpha_{ij}^{\mathrm{b}}\hat{n}_{\mathrm{b}}^i\hat{n}_{\mathrm{b}}^j + \alpha_{i\tilde{j}}^{\mathrm{f}}\hat{n}_{\mathrm{f}}^i\hat{n}_{\mathrm{f}}^{\tilde{j}} + \alpha_{a\tilde{j}}^{\mathrm{bf}}\hat{\xi}_{\mathrm{b}}^a\hat{n}_{\mathrm{f}}^{\tilde{j}})}, \tag{364}$$

where the indices $i, j$ run over the bosonic modes and $\tilde{i}, \tilde{j}$ run over the fermionic ones. The resulting family of non-Gaussian states is then given by

$$|\psi(M, \alpha, \Gamma)\rangle = \mathcal{U}(M)\mathcal{U}_{\mathrm{NG}}(\alpha)|\Gamma\rangle \, , \tag{365}$$

where we have $M = (M_{\mathrm{b}}, z_{\mathrm{b}}, M_{\mathrm{f}})$, $\alpha = (\alpha^{\mathrm{b}}, \alpha^{\mathrm{f}}, \alpha^{\mathrm{bf}})$ and $\Gamma = (\Gamma_{\mathrm{b}}, z_{\mathrm{b}}, \Gamma_{\mathrm{f}})$. Note that this parametrizations certainly contains redundancies and careful analysis of the resulting family is desirable. This choice of non-Gaussian unitaries is motivated by the fact that, while going beyond the space of Gaussian transformation, they still allow for efficient computations. They satisfy the property $\mathcal{U}_{\mathrm{NG}}^\dagger \hat{\xi}^a \mathcal{U}_{\mathrm{NG}} = (\mathcal{U}_{\mathrm{G}})^a{}_b \hat{\xi}^b$, where $\mathcal{U}_{\mathrm{G}}$ is a Gaussian unitary combined with a linear transformation of the $\hat{\xi}^a$.

Moreover, these states can be understood as true generalizations of Gaussian states in the sense of a generalized Wick's theorem. As discussed in [32], the evaluation of $n$-function follows from a quadratic generating function, just like in Wick's theorem. However, what makes them truly different from Gaussian states is that this generating functional is different for different $n$, such that more interesting correlation structures (such as boson-fermion correlations, but even within one sector) can be captured. For this reason any $n$-point function can be efficiently computed for the states introduced in this section, generalizing the property that in the

case of Gaussian states follows from Wick's theorem. For this property to hold, the unitarity of $\mathcal{U}_{\mathrm{NG}}$ is essential. With this in mind, the parameters $\alpha$ should be considered real and not extended to the complex plane.

Therefore, the previous considerations about Kähler manifolds apply directly. Clearly, the manifold decomposes into equivalence classes (sheets) of states $\mathcal{U}|\phi(x)\rangle$ related by Gaussian transformations. In particular, the sheets constructed this way will only be Kähler where the reference state $|\phi(x)\rangle$ is the highest weight state of the representation, $i.e.$, the vacuum state $|\Gamma_{\mathrm{b}}, z_{\mathrm{b}}, \Gamma_{\mathrm{f}}\rangle = |0\rangle_{\mathrm{b}} \otimes |0\rangle_{\mathrm{f}}$, while the overall manifold (collection of sheets) will not be Kähler. Consequently, these states provide a natural playground for the concepts and methods introduced in this paper and vice versa a rigorous understanding of variational methods for non-Kähler manifolds is required to use generalized Gaussian states in practice.

# 6 Summary and discussion

We have presented a systematic geometric framework to use variational methods for the study of closed quantum systems. Our results build on extensive previous work ranging from the geometric formulation of the time dependent variational principle by Saraceno and Kramer [25] to the formulation of group theoretic coherent states due to Gilmore and Perelomov [29, 30].

The main contribution of the present work is to recognize the Kähler property as an important criterion in the classification of variational families, to extend existing methods to the non-Kähler case and thereby provide a systematic framework for such calculations. While section 2 served as less rigorous exposition, we wrote a concise review of the necessary mathematical background in section 3 to understand our main results in section 4.

**Real time evolution (4.1).** We gave explicit formulas for the Lagrangian evolution (88) and the McLachlan evolution (93), whose equivalence for Kähler manifolds was shown in proposition 6. We also compared their conservation laws when the two evolutions differ (non-Kähler case).

**Excitation spectra (4.2).** We rephrased the two approaches of computing excitation spectra from tangent spaces geometrically, namely projecting the Hamiltonian according to (119) vs. linearizing the equations of motion according to (122) à la Gross–Pitaevskii, and discuss their relations for Kähler and non-Kähler manifolds. For the latter approach, in (123) we introduced the local generator $K^{\mu}{}_{\nu}$ of time evolution.

**Spectral functions (4.3).** We formulated linear response theory in our geometric language to derive a new approximation of the spectral function in (156) using the eigenvectors of $K^{\mu}{}_{\nu}$, which can be applied to both Kähler and non-Kähler manifolds.

**Imaginary time evolution (4.4).** We gave a geometric derivation of projected imaginary time evolution (164), which makes its equivalence to gradient descent explicit and does not rely on the Kähler property.

In our application section 5, we considered commonly used families of states under the light of variational methods and Kähler geometry: We started with the well-known families of bosonic and fermionic Gaussian states (5.1), where we related for the first time the Kähler structures on tangent space $(\boldsymbol{g}, \boldsymbol{\omega}, \boldsymbol{J})$ of the family with the ones on the classical phase space $(g, \omega, J)$. We then reviewed group theoretic coherent states (5.2) à la Gilmore-Perelemov . Finally, we investigated the recently introduced families of generalized Gaussian states (5.3) through the lense of group theory and Kähler geometry.

Geometric formulations of quantum theory have a long tradition [24,26–28] in the mathematical physics community, but their usage for everyday applications in quantum many body physics and quantum optics has been limited for several reasons. Studying real and imaginary time evolution on high dimensional variational families requires extensive computational resources, which were not available thirty years ago. While it was often sufficient in the past to do a first order calculation based on standard families, such as coherent states, more complex models and more complicated physical questions (often inspired by experiments) often require new approaches, such as larger variational families. Finally, finding suitable variational families is always about striking a balance between choosing the family's dimension sufficiently large to capture interesting physics, but also sufficiently small to make calculations feasible. While the dimension of Hilbert space typically scales exponentially with the number of degrees of freedom $N$, most effective variational families scale polynomially (coherent states $\sim N$, Gaussian states $\sim N^2$, generalized Gaussian states $\sim N^2$).

We expect that in the near future the systematic application of existing and newly proposed variational families in quantum many body problems will contribute to a better understanding of the involved physics with the potential of making new predictions relevant for experimental studies. We believe that with the present paper we have laid the theoretical foundations for the exploration of general variational families and methods. A prominent example of these are generalized Gaussian states as proposed in [32] and reviewed in section 5.3. They present a new approach for the variational study of various systems, ranging from boson-fermion mixtures in Holstein models [21] to Su-Schrieer-Heeger models and Kondo models [32], but their geometric and mathematical structures have been largely unexplored. They form manifest non-Kähler manifolds and this makes our formalism particularly suited for their study. Even revisiting known models with enlarged variational families can reveal new properties. For example, moving from coherent to the larger family of bosonic Gaussian states revealed recently that the ground state of trapped Bose-Einstein condensates with attractive s-wave interaction exhibits features of a squeezed state [97,98].

The present paper focused exclusively on variational families of pure quantum states, used for the study of closed quantum systems, *i.e.*, at zero temperature. A natural extension of our formalism would be to also incorporate open quantum systems by allowing mixed states within our variational family. The approximate ground states $\psi_0$ minimizing the energy would be replaced by density operator $\rho_0$ minimizing the free energy. Non-equilibrium phenomena are often treated in their Markovian approximation, where time evolution is governed by master equations of the Gorini-Kossakowski-Sudarshan-Lindblad form [99,100]. Using this to derive meaningful variational equations is an important challenge, which has been only partially accomplished in the context of specific variational families [101,102]. We believe that the geometric perspective laid out in the current manuscript may also be helpful for developing variational methods to study the dynamics of open quantum systems.

# Acknowledgements

We thank Abhay Ashtekar, Pavlo Bulanchuk, Eugenio Bianchi, Jens Eisert, Marcos Rigol, Richard Schmidt, Jan Philip Solovej, Lev Vidmar, Albert Werner and Yao Wang for inspiring discussions. LH acknowledges support by VILLUM FONDEN via the QMATH center of excellence (grant no.10059) and by the Max Planck Harvard Research Center for Quantum Optics. TG, LH and IC thank Harvard University for the hospitality during several visits. TG, LH and IC are supported by the Deutsche Forschungsgemeinschaft (DFG, German Research Foundation) under Germany's Excellence Strategy – EXC-2111 – 39081486. TS acknowledges the Thousand-Youth-

Talent Program of China and NSFC 11974363. JH acknowledges funding through ERC Grant ERQUAF, ERC-2016-STG (Grant no.715861). ED acknowledges funding through Harvard-MIT CUA, AFOSR-MURI: Photonic Quantum Matter (award FA95501610323) and DARPA DRINQS program (award D18AC00014). IC acknowledges funding through ERC Grant QUENOCOBA, ERC-2016-ADG (Grant no.742102).

# A  Conventions and notation

In this appendix, we review the conventions and notation used in this manuscript. The goal of our formalism is to be largely self-explanatory with an easy conversion between abstract objects (vectors, tensors, operators) and their numerical representation (lists, matrices, arrays).

## A.1  Nomenclature

The following list contains most of the symbols and their meaning, used throughout this manuscript.

| Symbol | Meaning |
| --- | --- |
| $\lvert\psi\rangle\in\mathcal{H}$ | Hilbert space vector |
| $\lvert\psi(x)\rangle\in M$ | family of Hilbert space vectors $M\subset\mathcal{H}$ |
| $\lvert v_\mu\rangle$ | tangent space vector on $M\subset\mathcal{H}$ |
| $\omega_{\mu\nu},g_{\mu\nu},J^\mu{}_\nu$ | restricted Kähler structures on $M\subset\mathcal{H}$ |
| $\varepsilon$ | Expectation value $\varepsilon=\langle\psi\lvert\hat{H}\rvert\psi\rangle$ |
| $L$ | Lagrangian on $M$ |
| $\psi\in\mathcal{P}(\mathcal{H})$ | quantum state in projective Hilbert space |
| $\psi(x)\in\mathcal{M}$ | family of states $\mathcal{M}\subset\mathcal{H}$ |
| $\lvert V_\mu\rangle$ | tangent space vector on $\mathcal{M}\subset\mathcal{H}$ |
| $\mathcal{T}_\psi\mathcal{M}$ | tangent space of $\mathcal{M}$ at $\psi$ |
| $\mu,\nu,\delta,\gamma$ | tangent space indices |
| $\boldsymbol{\omega}_{\mu\nu},\boldsymbol{g}_{\mu\nu},\boldsymbol{J}^\mu{}_\nu$ | restricted Kähler structures on $\mathcal{M}\subset\mathcal{H}$ |
| $\mathcal{L}$ | Lagrangian on $\mathcal{L}$ |
| $E$ | Expectation value $E=\langle\psi\lvert\hat{H}\rvert\psi\rangle/\langle\psi\lvert\psi\rangle$ |
| $\mathcal{H}^\perp_\psi$ | vectors orthogonal to $\lvert\psi\rangle$ |
| $\mathbb{Q}_\psi$ | orthogonal projector onto $\mathcal{H}^\perp_\psi$ |
| $\mathcal{P}_\psi$ | orthogonal projector onto $\mathcal{T}_\psi\mathcal{M}$ |
| $A(x)=\langle\hat{A}\rangle(x)$ | expectation value of $\hat{A}$ for state $\psi(x)$ |
| $\{A,B\}$ | Poisson bracket on $\mathcal{M}$ for functions $A,B$ |
| $\mathcal{X}^\mu$ | real time evolution vector field |
| $\mathcal{F}^\mu$ | imaginary time evolution vector field |
| $\widetilde{P}^\mu{}_\nu$ | projector onto conservation laws subspace |
| $\widetilde{\mathcal{X}}^\mu,\widetilde{\mathcal{F}}^\mu$ | conservation laws preserving vector fields |
| $\widetilde{\mathcal{M}}$ | manifolds of constant conserved quantities |

| | |
|---:|:---|
| $\lvert\Psi\rangle = e^{\kappa+i\varphi}\lvert\psi\rangle$ | family with variable phase/normalization |
| $K^{\mu}{}_{\nu}$ | linearized time evolution flow |
| $\lambda_{\ell} = \pm i\omega_{\ell}$ | eigenvalues of $K^{\mu}{}_{\nu}$ |
| $\mathcal{E}^{\mu}(\lambda)$ | right-eigenvectors of $K^{\mu}{}_{\nu}$ |
| $\widetilde{\mathcal{E}}_{\mu}(\lambda)$ | left-eigenvectors of $K^{\mu}{}_{\nu}$ |
| $\mathcal{A}(\omega)$ | spectral function |
| $V$ | classical bosonic/fermionic phasespace |
| $a,b,c,d$ | classical phase space indices |
| $g_{ab},\omega_{ab},J^{a}{}_{b}$ | Gaussian Kähler structures |
| $G^{ab},\Omega^{ab}$ | inverse Kähler structures |
| $z^{a},C_{2}^{ab}$ | one-point and two-point function |
| $\Gamma^{ab}$ | bosonic/fermionic covariance matrix |
| $C_{n}^{a_{1}...a_{n}}$ | $n$-point function |
| $\widehat{K}$ | representation of generator $K$ |
| $\mathcal{S}(M)$ | squeezing transformation |
| $\mathcal{D}(z)$ | displacement transformation |
| $\mathcal{U}(M,z)$ | Gaussian transformation |
| $\Delta = -JJ_{0}$ | relative complex structure |
| $\lvert V_{a}\rangle \in \mathcal{D}_{(\Gamma,z)}$ | displacement tangent vector |
| $\lvert V_{ab}\rangle \in \mathcal{S}_{(\Gamma,z)}$ | squeezing tangent vector |
| $\mathcal{G},\mathfrak{g}$ | Lie group and Lie algebra |
| $i,j,k,l$ | Lie algebra indices |
| $\Xi_{i}$ | Lie algebra basis |
| $\hat{\Xi}_{i}$ | operator representation of Lie algebra |
| $\mathcal{K}_{ij}$ | Killing form |
| $\lvert\mathcal{V}_{i}\rangle$ | Lie algebra induced tangent space basis |
| $\phi,\lvert\phi\rangle$ | reference state and state vector |
| $\mathcal{M}_{\phi} \subset \mathcal{P}(\mathcal{H})$ | group theoretic coherent states |
| $H_{\phi},\mathfrak{h}_{\phi}$ | stabilizer group and algebra of $\phi$ |
| $\mathfrak{h}$ | real Cartan subalgebra |
| $I,J,K,L$ | Cartan subalgebra indices |
| $H_{I}$ | Cartan subalgebra basis |
| $\mathfrak{v}_{\alpha}$ | root spaces with roots $\alpha$ |
| $E_{\alpha}$ | root vectors with root $\alpha$ |
| $(\beta_{M})_{k} \in \mathfrak{g}^{*}$ | expectation value of $\hat{\Xi}_{k}$ |
| $\mathcal{O}_{\phi}$ | co-adjoint orbit of $\phi$ |
| $S_{\phi},\mathfrak{s}_{\phi}$ | stabilizer group and algebra of $\beta_{\mathbb{1}}$ |

## A.2 Abstract index notation

Throughout this paper, all equations containing indices follow the conventions of *abstract index notation*. This formalism is commonly used in the research field of general relativity and gravity, where differential geometry plays an important role, but we believe that it is also of great benefit when studying the geometry of variational manifolds.

The formalism is suitable to conveniently keep track of tensors built on a vector space. Given a finite dimensional real vector space $V$ with dual $V^*$, a $(r,s)$-tensor $T$ is a linear map

$$T : V^* \times \cdots \times V^* \times V \times \cdots \times V \to \mathbb{R} \,. \tag{366}$$

In particular, a $(1,0)$-tensor is a vector, a $(0,1)$-tensor is a dual vector and a $(1,1)$-tensor is a linear map. To keep track of the type of tensor, abstract index notation refers to the $(r,s)$-tensor $T$ as $T^{a_1 \cdots a_r}{}_{b_1 \cdots b_s}$, *i.e.*, we assign $r$ upper indices and $s$ lower indices. Typically, we choose the indices from some alphabet to indicate which vector space, we are referring to. For tangent space $\mathcal{T}_\psi \mathcal{M}$ of a variational manifold $\mathcal{M}$, we use Greek letters $\mu, \nu, \gamma, \delta$, while we use Latin letters $a, b, c, d$ to refer to the classical phase space $V$ of a bosonic or fermionic system. In the context of group theoretic states, we use $i, j, k, l$ to refer to the Lie algebra and $I, J, K, L$ to refer to its Cartan subalgebra.

The key advantage of abstract index notation in the context of variational manifolds is that it helps us to keep track of what types of tensors, we are dealing with and which contractions are allowed. Apart from vectors $X^a$ and dual vectors $w_a$, we are mostly dealing with tensors that have two indices, namely linear maps $J^a{}_b$, bilinear forms $g_{ab}$ and dual bilinear forms $\Omega^{ab}$

In the present paper, we often deal with linear maps and bilinear form, *i.e.*, tensors that have two indices. They are naturally represented as matrices, in particular, for numerical evaluation. For convenience, we will also use the notation, where tensors with suppressed indices are implied to be contracted, just as standard matrix multiplication works. Obviously, this means that only such expressions are allowed where the adjacent indices are given by one upper and one lower index.

## A.3 Special tensors and tensor operations

In the following, we review common matrix and tensor operations and emphasize how they are defined if we do not have a natural identification between a vector space and its dual space. This highlights that certain formulas involving matrix operations (such as computing determinants, traces, eigenvalues or transposes) are only well defined in certain cases, *i.e.*, if the respective matrix represents a linear map in some cases or bilinear form in other cases.

**Identity.** Every vector space $V$ comes with the canonical identity map $\delta^a{}_b$ satisfying $\delta^a{}_b X^a = X^a$. Note that the notation $\mathbb{1}^a{}_b$ would also be consistent, but we stayed with the commonly used Kronecker delta. There does not exist a canonical analogue as bilinear form, *e.g.*, a form $\delta_{ab}$ or $\delta^{ab}$ which only make sense with respect to a specific basis and are therefore not canonical, but rather a specific choice, such as a metric $g_{ab}$.

**Transformation rules.** An invertible linear map $M^a{}_b : V \to V$ of the vector space $V$ acts on a general $(r,s)$-tensor $T^{a_1 \cdots a_r}{}_{b_1 \cdots b_s}$ and transforms it to

$$M^{a_1}{}_{c_1} \cdots M^{a_r}{}_{c_r} (M^{-1})^{d_1}{}_{b_1} \cdots (M^{-1})^{d_s}{}_{b_s} T^{c_1 \cdots c_r}{}_{d_1 \cdots d_s} \,. \tag{367}$$

In particular, a vector $X^a$ transforms as $M^a{}_b X^b$, a dual vector $w_a$ as $w_b (M^{-1})^b{}_a$, a dual bilinear form $S^{ab}$ as $M^a{}_c B^{cd} (M^{\intercal})_d{}^b$, a bilinear form $s_{ab}$ as $(M^{-1\intercal})_a{}^c b_{cd} (M^{-1})^d{}_b$ and a linear map $K^a{}_b$ as $M^a{}_c K^d{}_d (M^{-1})^d{}_b$.

**Determinant.** The determinant $\det(M)$ is only well-defined for a linear map $M^a{}_b$. The determinant of a bilinear form $s_{ab}$ or $S^{ab}$ is ill defined, unless we have a reference object, such as a metric $g_{ab}$ or $G^{ab}$. Then, we can compute the determinant of the matrix of the linear maps $S^{ac}g_{cb}$ or $G^{ac}s_{cb}$.

**Trace.** The trace $\mathrm{tr}(M) = M^a{}_a$ is only defined for a linear map, not for bilinear forms $S^{ab}$ or $s_{ab}$, unless we again have a reference object, such as a metric.

**Eigenvalues.** Without additional structures, we can only defined eigenvalues for a linear map $M^a{}_b$, where an eigenvalue $\lambda$ associated to an eigenvector $X^a$ satisfies

$$M^a{}_b X^b = \lambda X^b. \tag{368}$$

This is well-known from linear algebra. A bilinear form $X^{ab}$ does not have intrinsic eigenvalues, but we can compute its eigenvalues relative to another bilinear form. Given a bilinear form $s_{ab}$ and a metric $G^{ab}$ or symplectic form $\Omega^{ab}$, we can define the metric or symplectic eigenvalues as the regular eigenvalues of the linear map $G^{ac}s_{cb}$ or $\Omega^{ac}s_{cb}$, respectively.

**Transpose.** The transpose of a linear map $M^a{}_b : V \to V$ is the map $(M^{\intercal})_a{}^b : V^* \to V^*$. We have the relation $(M^{\intercal})_a{}^b = M^b{}_a$, which means that the two represent the same tensor and typically one does not distinguish between the two in abstract index notation. However, for our shorthand notation, it is important to keep the order of indices in right order. From the perspective of abstract index notation, there is not really much point to use the transpose operation, but we will still write the respective expressions for convenience, so that they can be easily converted to matrix expressions, *e.g.*, for numerical implementations.

**Gradient.** Given a function $f(x)$ on some manifold $\mathcal{M}$, its gradient $(df)_\mu = \partial_\mu f$ is field of covectors, also known as 1-form. This means that the gradient alone does not define a tangent space direction, *e.g.*, to move in the direction of steepest ascent. Indeed, only if we have a metric $G^{\mu\nu}$, we can define typical gradient vector field $\mathcal{F}^\mu = G^{\mu\nu}(\partial_\nu f)$. The reason is that the gradient $df$ as dual vector encodes the linearized change $df(X) = df_\mu X^\mu$ of the function $f$, when performing a step in the direction $X^\mu$. Clearly, by increasing our step size, we can make this change arbitrarily large, so there is no "steepest" direction. Only if we have an absolute measure of our step size, *e.g.*, a norm $\|X\| = \sqrt{X^\mu g_{\mu\nu} X^\nu}$ induced by an inner product $g_{\mu\nu}$, we can find a unique direction, for which a step of fixed size maximizes the change.

## A.4 Common formulas

Given a triangle of Kähler structures $(G, \Omega, J)$ with inverses $(g, \omega, -J)$, we have the following relations. We list them both in abstract index notation and in shorthand notation.

$$-J^2 = \mathbb{1} \qquad \Leftrightarrow \qquad -J^a{}_c J^c{}_b = \delta^a{}_b \tag{369}$$

$$-(J^\intercal)^2 = \mathbb{1}^\intercal \qquad \Leftrightarrow \qquad -(J^\intercal)_a{}^c (J^\intercal)_c{}^b = \delta_a{}^b \tag{370}$$

$$-J^{-1} = J \qquad \Leftrightarrow \qquad -(J^{-1})^a{}_b = J^a{}_b \tag{371}$$

$$J\Omega J^\intercal = \Omega \qquad \Leftrightarrow \qquad J^a{}_c \Omega^{cd} (J^\intercal)_d{}^b = \Omega^{ab} \tag{372}$$

$$-\Omega J^\intercal = J\Omega \qquad \Leftrightarrow \qquad -\Omega^{ac}(J^\intercal)_c{}^b = J^a{}_c \Omega^{cb} \tag{373}$$

$$J G J^\intercal = G \qquad \Leftrightarrow \qquad J^a{}_c G^{cd}(J^\intercal)_d{}^b = G^{ab} \tag{374}$$

$$-G J^\intercal = JG \qquad \Leftrightarrow \qquad -G^{ac}(J^\intercal)_c{}^b = J^a{}_c G^{cb} \tag{375}$$

$$\Omega J^\intercal = G \qquad \Leftrightarrow \qquad \Omega^{ac}(J^\intercal)_c{}^b = G^{ab} \tag{376}$$

$$-J\Omega = G \qquad \Leftrightarrow \qquad -J^a{}_c \Omega^{cb} = G^{ab} \tag{377}$$

$$\Omega \omega = \mathbb{1} \qquad \Leftrightarrow \qquad \Omega^{ac}\omega_{cb} = \delta^a{}_b \tag{378}$$

$$\omega \Omega = \mathbb{1}^\intercal \qquad \Leftrightarrow \qquad \omega_{ac}\Omega^{cb} = \delta_a{}^b \tag{379}$$

$$Gg = \mathbb{1} \qquad \Leftrightarrow \qquad G^{ac}g_{cb} = \delta^a{}_b \tag{380}$$

$$gG = \mathbb{1}^\intercal \qquad \Leftrightarrow \qquad g_{ac}G^{cb} = \delta_a{}^b \tag{381}$$

$$-\omega G \omega = g \qquad \Leftrightarrow \qquad -\omega_{ac}G^{cd}\Omega_{db} = g_{ab} \tag{382}$$

$$-g \Omega g = \omega \qquad \Leftrightarrow \qquad -g_{ac}\Omega^{cd}G_{db} = \omega_{ab} \tag{383}$$

$$\Omega g = J \qquad \Leftrightarrow \qquad \Omega^{ac}g_{cb} = J^a{}_b \tag{384}$$

$$-G\omega = J \qquad \Leftrightarrow \qquad -G^{ac}\omega_{cb} = J^a{}_b \tag{385}$$

$$-\Omega^\intercal = \Omega \qquad \Leftrightarrow \qquad -\Omega^{ba} = \Omega^{ab} \tag{386}$$

$$G^\intercal = G \qquad \Leftrightarrow \qquad G^{ba} = G^{ab} \tag{387}$$

A symplectic group element $M^a{}_b \in \mathrm{Sp}(2N, \mathbb{R})$ and a symplectic algebra element $K^a{}_b \in \mathfrak{sp}(2N, \mathbb{R})$ are characterized by the following properties.

$$M\Omega M^\intercal = \Omega \qquad \Leftrightarrow \qquad M^a{}_c \Omega^{cd}(M^\intercal)_d{}^b = \Omega^{ab} \tag{388}$$

$$\Omega M^\intercal \omega = M^{-1} \qquad \Leftrightarrow \qquad \Omega^{ac}(M^\intercal)_c{}^d \omega_{db} = (M^{-1})^a{}_b \tag{389}$$

$$-\Omega K^\intercal = K\Omega \qquad \Leftrightarrow \qquad -\Omega^{ac}(K^\intercal)_c{}^b = K^a{}_c \Omega^b \tag{390}$$

An orthogonal group element $M^a{}_b \in \mathrm{O}(2N)$ and an orthogonal algebra element $K^a{}_b \in \mathfrak{so}(2N)$ are characterized by the following properties.

$$MGM^\intercal = G \qquad \Leftrightarrow \qquad M^a{}_c G^{cd}(M^\intercal)_d{}^b = G^{ab} \tag{391}$$

$$GM^\intercal G = M^{-1} \qquad \Leftrightarrow \qquad G^{ac}(M^\intercal)_c{}^d G_{db} = (M^{-1})^a{}_b \tag{392}$$

$$-GK^\intercal = KG \qquad \Leftrightarrow \qquad -G^{ac}(K^\intercal)_c{}^b = K^a{}_c G^b \tag{393}$$

# B Proofs

In this appendix, we present several technical proofs of selected propositions from the main text, whose proof would have interrupted the reading flow.

**Proposition 1.** *On a tangent space $\mathcal{T}_\psi\mathcal{M} \subset \mathcal{H}$ of a submanifold $\mathcal{M} \subset \mathcal{P}(\mathcal{H})$ we can always find an orthonormal basis $\{|V_\mu\rangle\}$, such that $\boldsymbol{g}_{\mu\nu} \equiv \mathbb{1}$ and the restricted complex structure is represented by the block matrix*

$$
\boldsymbol{J}^\mu{}_\nu \equiv \begin{pmatrix}
\begin{matrix} & 1 \\ -1 & \\ & & \ddots \end{matrix} & & \\
\hline
& \begin{matrix} & & c_1 & \\ -c_1 & & & \\ & & & c_2 \\ & -c_2 & & \\ & & & & \ddots \end{matrix} & \\
\hline
& & \begin{matrix} 0 & \\ & \ddots \end{matrix}
\end{pmatrix}
\tag{394}
$$

*with $0 < c_i < 1$. This standard form induces the decomposition of $\mathcal{T}_\psi\mathcal{M}$ into the three orthogonal parts*

$$
\mathcal{T}_\psi\mathcal{M} = \underbrace{\overline{\overline{\mathcal{T}_\psi\mathcal{M}}} \oplus \mathcal{I}_\psi\mathcal{M}}_{\overline{\mathcal{T}_\psi\mathcal{M}}} \oplus \mathcal{D}_\psi\mathcal{M} \,,
\tag{395}
$$

*where $\overline{\overline{\mathcal{T}_\psi\mathcal{M}}}$ is the largest Kähler subspace and $\overline{\mathcal{T}_\psi\mathcal{M}}$ is the largest space on which $J$ and $\omega$ are invertible.*

*Proof.* We focus on a single tangent space $\mathcal{T}_\psi\mathcal{M} \subset \mathcal{H}$ and refer to the Kähler structures on $\mathcal{H}$, rather than the restricted ones on $\mathcal{T}_\psi\mathcal{M}$, as $(g, \omega, J)$. To shorten notation, we define $A := \mathcal{T}_\psi\mathcal{M}$ and $B$ as its orthogonal complement in $\mathcal{H}$ with respect to $g$, so that $\mathcal{H} = A \oplus B$. We will refer to the restricted Kähler structures on $A$ or $B$ by $(g_A, \omega_A, J_A)$ and $(g_B, \omega_B, J_B)$, respectively. The relation $J = G\omega = -\Omega g$ implies $g = -\omega J$ or, equivalently, $g(v, w) = -\omega(v, Jw)$, and also $g(Jv, Jw) = g(v, w)$ for all $v, w \in \mathcal{H}$. From here, $g(Jv, w) = -g(v, Jw)$ follows and we can derive for $a, a' \in A$

$$
g_A(J_A a, a') = g(Ja, a') = -g_A(a, J_A a') \,,
\tag{396}
$$

which implies that $J_A$ is antisymmetric with respect to $g_A$ and thus is diagonalizable, has either vanishing or purely imaginary eigenvalues with the latter appearing in pairs $\pm c_i$. Furthermore, we can always choose an orthonormal basis, such that $g_A = \mathbb{1}$ and $J_A$ is represented by (394). Next, we show that $c_i \in (0, 1]$. We define the orthogonal projectors $\mathbb{P}_A : \mathcal{H} \to A$ and $\mathbb{P}_B : \mathcal{H} \to B$, such that

$$
J = \left( \begin{array}{c|c} J_A & J_{AB} \\ \hline J_{BA} & J_B \end{array} \right), \quad
\begin{array}{ll}
J_A : & A \to A, \quad a \mapsto \mathbb{P}_A(Ja) \,, \\
J_B : & B \to B, \quad b \mapsto \mathbb{P}_B(Jb) \,, \\
J_{AB} : & B \to A, \quad b \mapsto \mathbb{P}_A(Jb) \,, \\
J_{BA} : & A \to B, \quad a \mapsto \mathbb{P}_B(Ja) \,.
\end{array}
\tag{397}
$$

We write $J^2 - \mathbb{1} = 0$ in blocks to find

$$
\left( \begin{array}{c|c} J_A^2 + J_{AB}J_{BA} - \mathbb{1}_A & J_A J_{AB} + J_{AB} J_B \\ \hline J_B J_{BA} + J_{BA} J_A & J_B^2 + J_{BA} J_{AB} - \mathbb{1}_B \end{array} \right) = 0 \,.
\tag{398}
$$

We consider an eigenvector $a \in A$ of $J_A$ with $J_A a = \mathrm{i}ca$ for non-zero $c$, which implies $J_A^2 a = -c^2 a$. We compute

$$
\begin{aligned}
g(a, a) &= g(J_A a + J_{BA} a, J_A a + J_{BA} a) \\
&= g(J_A a, J_A a) + g(J_{BA} a, J_{BA} a) \\
&\geq g(J_A a, J_A a) = -g(a, J_A^2 a) = c^2 g(a, a)
\end{aligned}
\tag{399}
$$

where we used that $A$ and $B$ are orthogonal which eliminates crossing terms. This implies the inequality $c^2 \leq 1$ and thus, we can choose $c_i \in (0, 1]$ as in (66). $\qquad \square$

**Proposition 2.** *The Kähler property is equivalent to requiring that $\mathcal{T}_\psi \mathcal{M}$ is not just a real, but also a complex subspace, i.e., for all $|X\rangle \in \mathcal{T}_\psi \mathcal{M}$, we also have $i|X\rangle \in \mathcal{T}_\psi \mathcal{M}$. Therefore, the multiplication by $i$ commutes with the projector $\mathbb{P}_\psi$, i.e., $\mathbb{P}_\psi i = i \mathbb{P}_\psi$ and $\mathbb{P}_\psi$ is complex-linear.*

*Proof.* We want to show that $J_A^2 = -\mathbb{1}_A$ implies that for all $a \in A$, we also have $ia = Ja \in A$. Therefore, we need to show that $Ja = J_A a$, which is equivalent to $J_{BA} = 0$. For arbitrary $a \in A$, we compute

$$g(J_{BA}a, J_{BA}a) = g(Ja, J_{BA}a) = -g(a, JJ_{BA}a) = -g(a, J_{AB}J_{BA}a). \tag{400}$$

This expression vanishes if $J_A^2 = -\mathbb{1}_A$, because in that case $J_{AB}J_{BA} = J_A^2 - \mathbb{1}_A = 0$ follows from the first block in (398). Since $g$ is non-degenerate, this implies $J_{BA} = 0$. Similarly, we can use the last block in (398) to conclude $J_B^2 = -\mathbb{1}_B$, which implies $J_{BA} = 0$. With vanishing $J_{AB}$ and $J_{BA}$, $J$ is block diagonal and commutes with the projectors. In the language of complex vector spaces, this implies that $\mathbb{P}_\psi i = i \mathbb{P}_\psi$. $\qquad \square$

**Proposition 8.** *Given a variational manifold $\mathcal{M}$ we define (according to the Lagrangian action principle) the free projected real time evolution $|\psi(t)\rangle$ as governed by the free Hamiltonian $\hat{H}_0$ and the perturbed projected real time evolution $|\psi_\epsilon(t)\rangle$ as governed by the perturbed Hamiltonian $\hat{H}_\epsilon(t) = \hat{H}_0 + \epsilon \hat{A}(t)$, both with the same initial state $|\psi(0)\rangle$. Then, the propagated perturbation, defined according to (148), is given by*

$$\delta x^\mu(t) = -\int_{-\infty}^{t} dt' \, (d\Phi_{t-t'})^\mu{}_\nu \Omega^{\nu\rho} \, \partial_\rho A(t')\big|_{\psi(t')}, \tag{401}$$

*where $d\Phi_t$ is the linearized free evolution flow.*

*Proof.* Let us define the perturbed evolution flow $\Phi_t^\epsilon$ as the map that sends the coordinates of an initial state $x^\mu(0)$ to the coordinates $x^\mu(t)$ of the state time evolved under the projected perturbed real time evolution. It is governed by

$$\frac{d}{dt}\Phi_t^\epsilon(x) = \mathcal{X}_\epsilon(\Phi_t^\epsilon(x)) = \mathcal{X}_0(\Phi_t^\epsilon(x)) + \epsilon \mathcal{X}_A(\Phi_t^\epsilon(x)), \tag{402}$$

where $\mathcal{X}_0$ and $\mathcal{X}_A$ are the evolution vector fields associated to the Hamiltonians $\hat{H}_0$ and $\hat{A}$ respectively. We define the free evolution flow $\Phi_t^0$ analogously by just setting $\epsilon = 0$ in the previous expressions.

Let us now define the interaction picture flow $\widetilde{\Phi}_t^\epsilon = \Phi_{-t}^0 \circ \Phi_t^\epsilon$. It has the useful property that its

time evolution only depends on the perturbing vector field:

$$\frac{d}{dt}\widetilde{\Phi}_t^\epsilon(x) = -\mathcal{X}_0(\widetilde{\Phi}_t^\epsilon(x)) + d\Phi_{-t}^0 \mathcal{X}_\epsilon(\Phi_t^\epsilon(x)) \tag{403}$$

$$= -\mathcal{X}_0(\widetilde{\Phi}_t^\epsilon(x)) + d\Phi_{-t}^0 \mathcal{X}_0(\Phi_t^\epsilon(x))$$
$$+ \epsilon \, d\Phi_{-t}^0 \mathcal{X}_A(\Phi_t^\epsilon(x)) \tag{404}$$

$$= -\mathcal{X}_0(\widetilde{\Phi}_t^\epsilon(x)) + \frac{d}{dt'}\bigg|_{t'=0} \Phi_{-t}^0 \Phi_{t'}^0 \Phi_t^\epsilon(x)$$
$$+ \epsilon \, d\Phi_{-t}^0 \mathcal{X}_A(\Phi_t^\epsilon(x)) \tag{405}$$

$$= -\mathcal{X}_0(\widetilde{\Phi}_t^\epsilon(x)) + \frac{d}{dt'}\bigg|_{t'=0} \Phi_{t'-t}^0 \Phi_t^\epsilon(x)$$
$$+ \epsilon \, d\Phi_{-t}^0 \mathcal{X}_A(\Phi_t^\epsilon(x)) \tag{406}$$

$$= -\mathcal{X}_0(\widetilde{\Phi}_t^\epsilon(x)) + \mathcal{X}_0(\Phi_{-t}^0 \Phi_t^\epsilon(x))$$
$$+ \epsilon \, d\Phi_{-t}^0 \mathcal{X}_A(\Phi_t^\epsilon(x)) \tag{407}$$

$$= \epsilon \, d\Phi_{-t}^0 \mathcal{X}_A(\Phi_t^\epsilon(x)). \tag{408}$$

We are interested in the propagated perturbation

$$\delta x^\mu(t) = \frac{d}{d\epsilon}\bigg|_{\epsilon=0} \Phi_t^0 \widetilde{\Phi}_t^\epsilon(x) = d\Phi_t^0\left(\frac{d}{d\epsilon}\bigg|_{\epsilon=0} \widetilde{\Phi}_t^\epsilon(x)\right). \tag{409}$$

The quantity $\frac{d}{d\epsilon}\big|_{\epsilon=0} \widetilde{\Phi}_t^\epsilon(x)$ is for all times a vector of $\mathcal{T}_{\psi(0)}\mathcal{M}$ and its time evolution can be obtained by using (408) after having commuted derivatives:

$$\frac{d}{dt}\left[\frac{d}{d\epsilon}\bigg|_{\epsilon=0} \widetilde{\Phi}_t^\epsilon(x)\right] = \frac{d}{d\epsilon}\bigg|_{\epsilon=0}\left[\frac{d}{dt}\widetilde{\Phi}_t^\epsilon(x)\right] \tag{410}$$

$$= \frac{d}{d\epsilon}\bigg|_{\epsilon=0} \epsilon \, d\Phi_{-t}^0 \mathcal{X}_A(\Phi_t^\epsilon(x)) \tag{411}$$

$$= d\Phi_{-t}^0 \mathcal{X}_A(\Phi_t^0(x)). \tag{412}$$

The solution to this equation follows from integrating as

$$\frac{d}{d\epsilon}\bigg|_{\epsilon=0} \widetilde{\Phi}_t^\epsilon(x) = \int_{-\infty}^t dt' \, d\Phi_{-t'}^0 \mathcal{X}_A(\Phi_{t'}^0(x)). \tag{413}$$

Combining this with (409) and the expression (88) for the Lagrangian real time evolution vector field $\mathcal{X}_A(\Phi_{t'}^0(x))$ leads to the result (401). $\qquad\square$

## C Kähler manifolds

Kähler manifolds play a central role in this manuscript. For completeness, in this appendix we will discuss their definition and properties. More information can be found in [49]. A Kähler manifold is a manifold $\mathcal{M}$ equipped with a metric $\boldsymbol{g}_{\mu\nu}$ and a symplectic form $\boldsymbol{\omega}_{\mu\nu}$ that satisfy several properties. These include some local properties, that is that $J^\mu{}_\nu = -G^{\mu\sigma}\boldsymbol{\omega}_{\sigma\nu}$ verifies $J^2 = -\mathbb{1}$ at all points, and also some non-local properties (closedness of $\boldsymbol{\omega}$ and vanishing Nijenhuis tensor). The precise definition is as follows.

**Definition 7.** *A real manifold $\mathcal{M}$ is called Kähler if each tangent space is equipped with a positive definite metric $\boldsymbol{g}_{\mu\nu}$ and a compatible symplectic form $\boldsymbol{\omega}_{\mu\nu}$ as in definition 1, such that $J^\mu{}_\nu = -G^{\mu\sigma}\boldsymbol{\omega}_{\sigma\nu}$ with $J^2 = -\mathbb{1}$, and the following conditions are satisfied:*

- *Symplectic form $\boldsymbol{\omega}$ is closed ($d\boldsymbol{\omega} = 0$) with*

$$(d\boldsymbol{\omega})_{\mu\nu\sigma} = \tfrac{1}{6}\left(\partial_\mu\boldsymbol{\omega}_{\nu\sigma} + \partial_\nu\boldsymbol{\omega}_{\sigma\mu} + \partial_\sigma\boldsymbol{\omega}_{\mu\nu} - \partial_\mu\boldsymbol{\omega}_{\sigma\nu} - \partial_\nu\boldsymbol{\omega}_{\mu\sigma} - \partial_\sigma\boldsymbol{\omega}_{\nu\mu}\right). \qquad (414)$$

- *Nijenhuis tensor $N_J$ vanishes ($N_J = 0$) with*

$$(N_J)^\mu_{\nu\sigma} = \boldsymbol{J}^\lambda{}_\sigma\,\partial_\lambda\boldsymbol{J}^\mu{}_\nu - \boldsymbol{J}^\lambda{}_\nu\,\partial_\lambda\boldsymbol{J}^\mu{}_\sigma + \boldsymbol{J}^\mu{}_\lambda(\partial_\nu\boldsymbol{J}^\lambda{}_\sigma - \partial_\sigma\boldsymbol{J}^\lambda{}_\nu). \qquad (415)$$

In essence, a Kähler manifold is simultaneously a Riemannian, a symplectic and a complex manifold, such that the respective structures in every tangent space are compatible in the sense of definition 1.

For the purpose of the methods presented in this manuscript, it is of interest only whether the restricted Kähler structures on $\mathcal{M}$ satisfy the compatibility conditions from definition 1. If they do, the manifold is known as an almost-Hermitian manifold. We do not use the additional properties of $\boldsymbol{\omega}$ being closed or $N_J$ vanishing.

However, as shown in the following proposition, if an almost-Hermitian manifold $\mathcal{M}$ is also a submanifold of a Kähler manifold, the additional non-local conditions are automatically satisfied and $\mathcal{M}$ is itself a Kähler manifold. In the context of this manuscript we always deal with manifolds $\mathcal{M} \subset \mathcal{P}(\mathcal{H})$, where projective Hilbert space $\mathcal{P}(\mathcal{H})$ is known to be a Kähler manifold [73]. For this reason, for all the manifolds we encounter, the local compatibility conditions from definition 1 are sufficient conditions for the manifold to be Kähler and we will therefore refer to manifolds that satisfy them as Kähler.

**Proposition 9.** *Given a Kähler manifold $\tilde{\mathcal{M}}$ with compatible Kähler structures ($\tilde{\boldsymbol{g}}, \tilde{\boldsymbol{\omega}}, \tilde{\boldsymbol{J}}$), a sub manifold $\mathcal{M} \subset \tilde{\mathcal{M}}$ equipped with the restricted Kähler structures ($\boldsymbol{g}, \boldsymbol{\omega}, \boldsymbol{J} = -\boldsymbol{G}\boldsymbol{\omega}$) is itself a Kähler manifold provided that $\boldsymbol{J}^2 = -\mathbb{1}$.*

*Proof.* $\mathcal{M}$ satisfies all local Kähler conditions. We therefore only need to show that $\boldsymbol{\omega}$ is closed and $N_J = 0$. We consider local coordinates $\tilde{x}^{\tilde{\mu}}$ on $\tilde{\mathcal{M}}$, such that $x^{\tilde{\mu}} \equiv (x^\mu, x'^{\mu'})$ where changes in $x^\mu$ preserve the submanifold $\mathcal{M}$, while changes in $x'^{\mu'}$ are orthogonal to it. We can further choose $x^\mu$ and $x'^{\mu'}$ locally, such that the matrix representations of the Kähler structures ($\boldsymbol{\omega}, \boldsymbol{g}, \boldsymbol{J}$) with respect to the decomposition $\tilde{\mu} \equiv (\mu, \mu')$ are

$$\tilde{\boldsymbol{g}} \equiv \left(\begin{array}{c|c} \boldsymbol{g} & 0 \\ \hline 0 & \boldsymbol{g}' \end{array}\right), \quad \tilde{\boldsymbol{\omega}} \equiv \left(\begin{array}{c|c} \boldsymbol{\omega} & 0 \\ \hline 0 & \boldsymbol{\omega}' \end{array}\right), \quad \tilde{\boldsymbol{J}} \equiv \left(\begin{array}{c|c} \boldsymbol{J} & 0 \\ \hline 0 & \boldsymbol{J}' \end{array}\right), \qquad (416)$$

which is a consequence of $\boldsymbol{J}^2 = -\mathbb{1}$, as proven in proposition 1. Thus, this implies that $\tilde{\boldsymbol{J}} = \boldsymbol{J} \oplus \boldsymbol{J}'$ with respect to this decomposition $\mathcal{T}_\psi\tilde{\mathcal{M}} = \mathcal{T}_\psi\mathcal{M} \oplus (\mathcal{T}_\psi\mathcal{M})^\perp$.

- Symplectic form is closed. In the above basis, $(d\boldsymbol{\omega})_{\mu\nu\sigma}$ corresponds to a sub block of the array $(d\tilde{\boldsymbol{\omega}})_{\tilde{\mu}\tilde{\nu}\tilde{\sigma}}$. Consequently, $d\tilde{\boldsymbol{\omega}} = 0$ implies $d\boldsymbol{\omega} = 0$.

- We restrict $\tilde{N}_{\tilde{J}}$ on $\tilde{\mathcal{M}}$ to $\mathcal{M}$ to find

$$(\tilde{N}_{\tilde{J}})^\mu_{\nu\sigma} = \boldsymbol{J}^{\tilde{\lambda}}{}_\sigma\,\partial_{\tilde{\lambda}}\boldsymbol{J}^\mu{}_\nu - \boldsymbol{J}^{\tilde{\lambda}}{}_\nu\,\partial_{\tilde{\lambda}}\boldsymbol{J}^\mu{}_\sigma + \boldsymbol{J}^\mu{}_{\tilde{\lambda}}(\partial_\nu\boldsymbol{J}^{\tilde{\lambda}}{}_\sigma - \partial_\sigma\boldsymbol{J}^{\tilde{\lambda}}{}_\nu) \qquad (417)$$

which is not obviously equal to $(N_J)^\mu_{\nu\sigma}$ due to the contraction over $\tilde{\lambda}$, which takes the full manifold into account. However, our previous considerations showed that $\tilde{\boldsymbol{J}} = \boldsymbol{J} \oplus \boldsymbol{J}'$. This implies that $\boldsymbol{J}^{\tilde{\lambda}}{}_\mu = \boldsymbol{J}^\lambda{}_\mu$, which proves the equality. Consequently $N_{\tilde{J}} = 0$ implies $N_J = 0$.

We therefore conclude that any submanifold $\mathcal{M}$ of a Kähler manifold $\tilde{\mathcal{M}}$ with $\boldsymbol{J}^2 = -\mathbb{1}$ everywhere is again a Kähler manifold. Note that this implies in particular that $\mathcal{M}$ is also a complex and a symplectic manifold. $\qquad\square$

# D   Normalized states as principal bundle

We introduced projective Hilbert space $\mathcal{P}(\mathcal{H})$ as the space of distinguishable quantum states, where normalization and phases of Hilbert space vectors can be ignored. However, in practice we usually parametrizing a variational manifold $\mathcal{M} \subset \mathcal{P}(\mathcal{H})$ by a set normalized states $|\psi(x)\rangle$ which depend on some real parameters $x^i$. It is therefore useful to introduce the manifold of normalized states

$$\mathcal{N}(\mathcal{H}) = \left\{ |\psi\rangle \in \mathcal{H} \,\middle|\, \langle\psi|\psi\rangle = 1 \right\}. \tag{418}$$

This manifold will play an important when we are interested in relative phases between states. Mathematically, $\mathcal{N}(\mathcal{H})$ is a principal fiber bundle over the base manifold $\mathcal{P}(\mathcal{H})$. Given a quantum state $\psi \in \mathcal{P}(\mathcal{H})$, we have a corresponding fiber $e^{i\varphi}|\psi\rangle \in \mathcal{N}(\mathcal{H})$ of normalized Hilbert state vector representing this state. We also have a natural U(1) group action onto such fibers given by multiplication with a complex phase, *i.e.*, $|\psi\rangle \to e^{i\varphi}|\psi\rangle$ with $e^{i\varphi} \in U(1)$.

We can now ask if we can use any structures of the Hilbert space to define a natural notion of parallel transport, *i.e.*, how to choose the complex phases when changing the quantum state continuously. This question is well-studied in the context of gauge theories and amounts to choosing a natural notion of *horizontal tangent spaces*, which encode how to move naturally through the principal fiber bundle. Interestingly, $\mathcal{N}(\mathcal{H})$ is equipped with a natural notion of moving horizontally, namely by requiring that a horizontal curve $|\psi(t)\rangle$ satisfies

$$\langle\psi(t)|\tfrac{d}{dt}|\psi(t)\rangle = 0, \tag{419}$$

*i.e.*, we require that the tangent vector $\frac{d}{dt}|\psi(t)\rangle$ is always orthogonal to the state $|\psi(t)\rangle$. The tangent space of normalized states is given by

$$\mathcal{T}_{|\psi\rangle}\mathcal{N}(\mathcal{H}) = \left\{ |\varphi\rangle \in \mathcal{H} \,\middle|\, \mathrm{Re}\,\langle\psi|\phi\rangle = 0 \right\}. \tag{420}$$

Locally, we can decompose this space into

$$\mathcal{T}_{|\psi\rangle}\mathcal{N}(\mathcal{H}) = \mathrm{span}_{\mathbb{R}}(|\psi\rangle) \oplus \mathcal{H}_{\psi}^{\perp}, \tag{421}$$

where the former the former is vertical space along the fiber and the latter is our natural choice of horizontal subspace.

We can understand the local choice of complex phase $e^{i\varphi}$ as pure gauge and U(1) as the corresponding gauge group. This implies that our description of normalized states $\mathcal{N}(\mathcal{H})$ is equivalent to electromagnetism on the base manifold $\mathcal{P}(\mathcal{H})$. In particular, we can compute the gauge field $A_\mu$ and its field strength tensor $F_{\mu\nu}$.

**Example 27.** *We consider the Bloch sphere with*

$$|\psi(x)\rangle = \cos\left(\tfrac{\theta}{2}\right)|0\rangle + e^{i\phi}\sin\left(\tfrac{\theta}{2}\right)|0\rangle. \tag{422}$$

*We can compute the gauge field $A_i$ as*

$$A_\mu = \mathrm{Im}\,\langle\psi(x)|\partial_\mu|\psi(x)\rangle \equiv (0, \sin^2\left(\tfrac{\theta}{2}\right)). \tag{423}$$

*As differential form, we therefore have $A = \sin^2\left(\tfrac{\theta}{2}\right)d\phi$. We find the field strength as its differential*

$$F = dA = 2\sin\left(\tfrac{\theta}{2}\right)d\theta \wedge d\phi. \tag{424}$$

*We can compute the change of phase $\Delta\varphi$, also called holonomy, if we move horizontally in $\mathcal{N}(\mathcal{H})$, such that our path describes a circle at constant latitude $\theta$ when projected onto the Bloch sphere. We find*

$$\Delta\varphi = \int_0^{2\pi} d\phi \, A = 2\pi \sin^2\left(\frac{\theta}{2}\right). \tag{425}$$

*We can also use Hamiltonian evolution by*

$$\hat{H} = E_0 + \omega \, \sigma_z \tag{426}$$

*to compute the time evolution. The Hamiltonian vector field is given by*

$$\dot{x}^\mu = \mathcal{X}^\mu = (0, -2\omega), \tag{427}$$

*such that the solution of the equation of motions is just $\phi(t) = -\omega t$. The quantum state will return to its original state at $t = \frac{\pi}{\omega}$, where $\phi(t) = -2\pi$. However, we will pick up a relative phase given by the integral*

$$\Delta\varphi = \int_0^{\pi/2} (A_\mu \dot{x}^\mu - E) dt = -\frac{\pi(E_0 + \omega)}{\omega}, \tag{428}$$

*where we used $E = E_0 + \omega \cos\theta$.*

In general, we conclude that the change of complex phase after returning to the same quantum state through time evolution is given by

$$\Delta\varphi = \int_0^T (A_\mu \dot{x}^\mu - E(x)), \tag{429}$$

where $E(x) = E(x_0)$ is constant for time-independent Hamiltonians.

If we go to variational families $\mathcal{M} \subset \mathcal{P}(\mathcal{H})$, we can still use (429) to compute the holonomy associated to projected time evolution. This is important if we want to compute spectral functions from real time evolution on $\mathcal{M}$ rather than via the linearization around a stationary point.

In practice, we therefore see that the only required additional structure on a variational family $\mathcal{M}$ is the computation of the gauge field $A_\mu$. Once this is computed, we can derive the change of relative phases from integration over $A_\mu$ and the energy expectation value, which is constant for time-independent Hamiltonians. Once, we have taken the relative phase in the time evolution into account, we can also use it to compute inner products between different state vectors, while ensuring the correct complex phase.

# E  Calculation of the pseudo-inverse

In section 4.1, we have described how calculating the Lagrangian real time evolution amounts to solving the differential equation of motion

$$\frac{dx^\mu}{dt} \equiv \mathcal{X}^\mu = -\mathbf{\Omega}^{\mu\nu}(\partial_\nu E). \tag{430}$$

In practical applications, this equation will be integrated numerically, requiring to calculate its right hand side at each time step.

The numerical complexity of computing this quantity lies in the cost of evaluating the gradient $\partial_\mu E$ (which will depend of the specific structure of the energy function $E(x)$ under consideration) and in the cost of computing the matrix $\mathbf{\Omega}$ and contracting it with such gradient vector. As discussed in sections 3.3 and 3.4, the matrix $\mathbf{\Omega}$ is a certain pseudo-inverse of the symplectic form $\boldsymbol{\omega}$, namely the one where we invert $\boldsymbol{\omega}$ on the orthogonal complement of its kernel.

In this appendix, we will present a method to correctly evaluate this pseudo-inverse $\mathbf{\Omega}^{\mu\nu}$ and simultaneously contract it with the gradient $\partial_\nu E$ to optimize the numerical cost of the operation. The need to compute a pseudo-inverse rather than a regular inverse lies in the fact that $\boldsymbol{\omega}$ may be degenerate and thus not invertible in the regular sense. In other words, $\boldsymbol{\omega}$ may have a non-trivial kernel

$$\ker \boldsymbol{\omega} = \{v^\mu \in \mathbb{R}^{2N} \mid \boldsymbol{\omega}_{\mu\nu} v^\nu = 0\} \,. \tag{431}$$

We can distinguish two types of vectors inside $\ker \boldsymbol{\omega}$:

1. Vectors $v^\mu$ that also lie in $\ker \boldsymbol{g}$, *i.e.*, $\boldsymbol{g}_{\mu\nu} v^\nu = 0$.

2. Vectors $v^\mu$ that have a non-vanishing length with respect to $\boldsymbol{g}$, *i.e.*, $\|v\|^2 = v^\mu \boldsymbol{g}_{\mu\nu} v^\nu > 0$

The vectors of the first type arise due to redundancies in the parametrization of the variational manifold $\mathcal{M}$. In this case, not all vectors $|V_\mu\rangle$ will correspond to linearly independent directions in tangent space, so that it is possible to find such non-trivial linear combinations of them that vanish, *i.e.*, $v^\mu |V_\mu\rangle = 0$ for $v^\mu$ of the first type. Consequently, such $v^\mu$ will have vanishing matrix elements in $\boldsymbol{g}$ and $\boldsymbol{\omega}$. The vectors of the second type, instead, represent physical directions in tangent space and may arise if $\mathcal{M}$ is not a Kähler manifold, as explained in section 3.4. In particular, such directions always exist in odd dimensional manifolds $\mathcal{M}$.

The vector $\mathcal{X}^\mu$, we wish to compute in (430), must solve

$$\boldsymbol{\omega}_{\mu\nu} \mathcal{X}^\nu = -\partial_\mu E \,. \tag{432}$$

If $\boldsymbol{\omega}$ is non-degenerate and thus invertible, this solution is unique and given by $\mathcal{X}^\mu = -\mathbf{\Omega}^{\mu\nu}(\partial_\nu E)$, where $\mathbf{\Omega}$ is defined as the regular inverse of $\boldsymbol{\omega}$. If, however, $\boldsymbol{\omega}$ is degenerate, then equation (432) will generally be ill defined and not admit a unique solution. The set of possible solutions of equation (432) has the form $\mathcal{X}_0 + \ker \boldsymbol{\omega}$, where $\mathcal{X}_0^\mu$ is one possible solution of the equation. To find a meaningful solution, we need to consider the following:

(a) If two solutions differ by an element in $\ker \boldsymbol{\omega}$ of the first type discussed above (*i.e.*, arising from an overparametrization of the manifold), any of them could be picked without influencing the physical results of the calculation. Indeed, one would obtain the same physical vector, just parametrized in two of the many possible redundant ways.

(b) If, instead, two solutions differ by vectors of $\ker \boldsymbol{\omega}$ of the second type (*i.e.*, physical vectors), then it is necessary to specify which solution should be picked. This essentially amounts to specifying how to compute the pseudo-inverse of $\boldsymbol{\omega}$.

(c) Finally, it may be ill defined because $\partial_\mu E$ may not vanish on $\ker \boldsymbol{\omega}$, *i.e.*, there may be $v^\mu$ with $v^\mu \boldsymbol{\omega}_{\mu\nu} = 0$, but $v^\mu \partial_\mu E \neq 0$, which implies that (432) cannot be solved.

In section 3.3, we explain how issues (a) and (b) can be addressed by imposing that we pick a solution that is orthogonal with respect to $\boldsymbol{g}$ to $\ker \boldsymbol{\omega}$. In other words, we pick the

solution $\mathcal{X} \in \mathcal{X}_0 + \ker \boldsymbol{\omega}$, such that $\mathcal{X}^\mu \boldsymbol{g}_{\mu\nu} \nu^\nu = 0$ for all $\nu^\mu \in \ker \boldsymbol{\omega}$. This corresponds to choosing to move only in the submanifold in which $\boldsymbol{J}$ can be put in the form discussed in Proposition 1 without the diagonal zero block. This solution can also be identified[41] as the element $\mathcal{X} \in \mathcal{X}_0 + \ker \boldsymbol{\omega}$ with minimal length $\|\mathcal{X}\|^2 = \mathcal{X}^\mu \boldsymbol{g}_{\mu\nu} \mathcal{X}^\nu$. Thus, the problem of correctly computing $\mathcal{X}^\mu$ in (430) is equivalent to finding a solution of (432) that also minimizes the quantity $\mathcal{X}^\mu \boldsymbol{g}_{\mu\nu} \mathcal{X}^\nu$. In summary, we have recast the pseudo-inversion as the linearly constrained quadratic minimization problem (*quadratic program*):

$$\text{Minimize} \quad \mathcal{X}^\mu \boldsymbol{g}_{\mu\nu} \mathcal{X}^\nu \quad \text{such that} \quad \boldsymbol{\omega}_{\mu\nu} \mathcal{X}^\nu + \partial_\mu E = 0 \,.$$

It is known [103] that such problems can be solved by introducing the Lagrange multipliers $\lambda^\mu \in \mathbb{R}^{2N}$ and finding a stationary point of the Lagrangian function

$$f(\mathcal{X}, \lambda) = \mathcal{X}^\mu \boldsymbol{g}_{\mu\nu} \mathcal{X}^\nu + \lambda^\mu (\boldsymbol{\omega}_{\mu\nu} \mathcal{X}^\nu + \partial_\mu E) \,. \tag{433}$$

A stationary point of $f(\mathcal{X}, \lambda)$, in turn, is given by a solution of the equation

$$\begin{pmatrix} \frac{\partial f}{\partial \mathcal{X}} \\ \frac{\partial f}{\partial \lambda} \end{pmatrix} = \underbrace{\begin{pmatrix} 2\boldsymbol{g} & \boldsymbol{\omega}^\mathsf{T} \\ \boldsymbol{\omega} & 0 \end{pmatrix}}_{A} \begin{pmatrix} \mathcal{X} \\ \lambda \end{pmatrix} + \underbrace{\begin{pmatrix} 0 \\ \partial E \end{pmatrix}}_{B} = 0 \,. \tag{434}$$

This is a $4N$-dimensional linear problem of the form $Ax + B = 0$ which, for large system sizes, can be efficiently tackled numerically. While we have succeeded in reducing the complicated pseudo-inverse problem to the linear problem (434), this problem still contains in itself all the intrinsic redundancies that characterize the pseudo-inverse problem. Here, however, these redundancies are re-expressed in such a way that they can be easily dealt with. More specifically, redundant solutions of (434) can be produced either by shifting $\mathcal{X}$ by an element of $\ker \boldsymbol{g}$ or by shifting $\lambda$ by an element of $\ker \boldsymbol{\omega}$. In the first case, we are just shifting between different redundant parametrizations of the same physical state. In the second case, we are not creating any physical differences, as we are just modifying the Lagrange multipliers which play no physical role in the theory. So any solution of (434) returned to us by our numerical solution method corresponds to a physically acceptable value for $\mathcal{X}^\mu$ that can be used in the integration scheme of our choice for (430).

Finally, we also need to address (c), *i.e.*, that (432) and thus also (434) may actually not have any solutions if $\partial E$ has some overlap with the kernel of $\boldsymbol{\omega}$. In this case, the best we can do is to minimize $\|Ax + B\|$, which is equivalent to projecting out the components of $B$ in $\ker A$. Indeed, $(Ax + B)$ can be split into two orthogonal components $(Ax + B)_\parallel$ and $(Ax + B)_\perp$, which are in $\ker A$ and $(\ker A)^\perp$ with respect to the flat metric of equation (434). Clearly, $(Ax+B)_\parallel = B_\parallel$ is independent of $x$, while $(Ax + B)_\perp$ can be made to vanish exactly as $A$ is invertible in the orthogonal complement to its kernel. Therefore minimizing $\|Ax + B\|$ is equivalent to solving $Ax + B = 0$ after having removed the component $B_\parallel$ of $B$. This can be done efficiently by multiplying the whole equation by $A^\mathsf{T} = A$, that is solving

$$A(Ax + B) = A^2 x + AB = 0 \,. \tag{435}$$

Such equation indeed is solved if and only if $(Ax+B)_\perp = 0$. Another advantage of (435) is that now the coefficient matrix $A^2$ is non-negative definite and the problem can thus be efficiently

---

[41]For this, we observe that since $\boldsymbol{g}$ is non-negative the quantity $\|w\|^2 = w^\mu \boldsymbol{g}_{\mu\nu} w^\nu$ admits a global minimum at $w_0$, defined by the stationarity condition $2 w_0^\mu \boldsymbol{g}_{\mu\nu} \delta w^\nu = 0$, for all possible variations $\delta w$ of $w$ around $w_0$. Such minimum may not be unique because $\boldsymbol{g}$ might be degenerate. However, such arbitrariness corresponds precisely to variations along directions reflecting redundant parametrizations which, as we just explained, do not change the physical result. In our case, the possible variations around any point inside the set of solutions $\mathcal{X}_0 + \ker \boldsymbol{\omega}$ correspond to all vectors $v \in \ker \boldsymbol{\omega}$. Therefore, the solution vector of minimum length is identified by $\mathcal{X}^\mu \boldsymbol{g}_{\mu\nu} \nu^\nu = 0$ for every $\nu^\mu \in \ker \boldsymbol{\omega}$, *i.e.*, it is also orthogonal to $\ker \boldsymbol{\omega}$.

tackled with conjugate gradient methods. Such methods typically converge to an approximate solution more efficiently than relying on performing a full singular value decomposition.

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
