# Peer review of "Geometry of variational methods: dynamics of closed quantum systems"

_SciPost Physics, doi:SciPost Phys. 9, 048 (2020)_

## Round 1 · Referee Report · Thomas Klein Kvorning · 2020-7-2

Strengths

1. The paper presents a subject which I believe, without the article, it would be hard to access for the intended reader in a pedagogical and accessible manner. The authors have taken to use examples for newly introduced concepts.

2. The article covers a broad range of applications of variational principles and could become a standard reference manual for the subject.

Weaknesses

1. The paper is quite long. In part, this is unavoidable for such a broad subject as the article covers, but in some aspects, you wish, as a reader, that it was more to the point. Specifically, by waiting to introduce the projective Hilbert space until page nine, roughly the same material gets presented two times, so after reading ten pages, you have gotten pretty short into the story of the paper.

Report

The authors present the subject of phrasing quantum variational methods in a differential geometric language in a very pedagogical manner. As is evident from the article, this clarifies many subtleties, and I believe it is beneficial for a better understanding of variational methods.

Doing this is hardly new, phrasing the variational manifold in a language of differential geometry using the symplectic and metric tensors is very old indeed. Also, the different methods for approximating the spectrum, the spectral function, and the time-evolution have previously been presented elsewhere (as the authors indicate), but some have not been rewritten in the language of differential geometry, as here.

The real novelty here lies in (1) a careful treatment of the non-Kähler case. E.g., that the two different methods for time-evolution, mentioned in the article differs and that they have quite different behavior when it comes to symmetries.

(2) Using the fact that there are a maximal Kähler subspace and a direct way to get it, Preposition 1 to projecting a time-evolution to a Kähler subspace and thus keeping the Kähler properties, i.e., "Kählerization" (the preposition itself is previously known).

The article does live up to the general acceptance criteria for SciPost, and I think it lives up to one of the SciPost Physics expectations.

The novelties can not be considered a groundbreaking theoretical discovery, neither is it a breakthrough on a previously-identified and long-standing research stumbling block. But it does open up a new pathway in an existing research direction; e.g., it could open up for non-Kähler variational families who, without this article, may not have been considered.

I, therefore, recommend it for publication in SciPost Physics after some minor improvements.

Requested changes

Apart from the general comments given in the other sections of the report, I only have one real issue I think should be corrected, numbered 1) below. The other are small comments or suggestions of typo corrections.

1. There are several Propositions in the article, which I believe, at least in part, are previously known. E.g., at least one of the directions of the equivalence in Proposition 2 follows from the fact that complex projective spaces are Kähler and that complex submanifolds of Kähler spaces are in turn Kähler. Since the article is aimed at a quite broad audience, I do not think one can expect the reader to know what is "well-known facts". A reader could get the wrong impression and think there are more novelties than there are. I think this should be resolved with small comments and or references.

2. In a paper of this length, there are quite many equations, so most readers can not keep track of the equation indexes. By referencing the equations just by number, you get halted in your reading by having to go back. This could be avoided by simply reminding the reader, in words, what equation is referenced. This would make this pedagogical paper even easier to read. There are a few of these examples, but to be specific, I can mention the referencing of (33) on page 10. There one could write "using the definition of the metric and symplectic form from the Hilbert space inner product, (33)" and one would not have to go back to check whether (33) is what just the definition or something else.

3. After proposition 1 it is written "Proposition 1 is also known in the context of classifying real subspaces of complex Hilbert spaces." There should be a reference.

4. On page 10. In the sentence "The fact that the variational parameters are in general real has to be correctly taken into account when projecting time evolution...", the phrase "projecting time evolution" is confusing. I suggest writing "...when projecting the time evolution to the variational manifold..." instead. But the sentence could probably be improved further; it is a bit hard to read.

5. In (5) \psi^\prime should be a function of z_1 and z_2.

6. II C ends with a paragraph explaining that the section is a simplification, and the topic of the section will be treated in full later. As a reader, I would like to have this at the end of the section. It should come first, such that you know what you are expected to get out of the section.

7. In the second paragraph in III, should it not be "equipped with a so called Kähler structure. " and not "equipped with so called Kähler structures. "?

8. In the first paragraph, after Definition 1, it should be "forms", not "form".

9. In (42), removing the origin from C is unnecessary since the vectors are assumed to be non-zero in the above sentence.

10. Above (52) "does not span Hilbert space" should be "does not span the Hilbert space".

11. In Definition 3, it should be said that \psi represents a state in M. It is not valid for a general \psi. This might be evident to most readers but would still make the definition a bit easier to read.

12. Missing subscript on c in (99).

13. Grammar error above (121), "is" should be removed.

14. I guess footnote 15 is meant to clarify why the spectrum of \partial_\nu \chi^\mu makes sense. But I got more confused by it, why introduce a connection? My first line of thought was: As you state, one needs a connection to define a derivative of a vector field since it is a map from one tangent space to another. But even with such a connection, the spectrum of a matrix of that map in a particular coordinate system has no meaning since one can independently change the basis in the different tangent spaces. Why not write something like: Usually, defining a derivative of a vector field requires a way to relate tangent spaces at adjacent points. The derivative would have no well-defined spectrum since it is a map between two different vector spaces. However, at a stationary point, it is a map from a tangent space to itself and thus does not need a connection, and there are well-defined eigenvectors and eigenvalues.

15. What is meant by naive gradient descent on page 26? The one defined by the coordinate dependent flat metric? In any case, there should be an argument or example backing up the statement. And what is the claim? Does g approximate the energy Hessian better than a random metric?

16. \mathcal E is used for two different objects. It is pretty clear which one is meant, but maybe it is a good idea to change.

17. Proposition 1 and 2 have been renamed to proposition 2 and 3 in appendix B. It makes future referencing of them confusing.

  • validity: top
  • significance: high
  • originality: good
  • clarity: high
  • formatting: excellent
  • grammar: excellent

Author:  Lucas Hackl  on 2020-07-13  [id 882]

(in reply to Report 1 by Thomas Klein Kvorning on 2020-07-02)
Category:
correction

We thank the referee for the careful reading of our manuscript and his suggestions for improvement which we were very happy to implement. Please find a revised version of the manuscript attached, where we marked the respective changes in red. Let us add some comments regarding some of these changes. We will also resubmit the revised version to arXiv (without markup).

1: We agree with the referee and added respective comments/citations in front of any proposition which we consider "standard material".

3: We are not aware of a reference on the classification of real subspaces of complex Hilbert spaces, but it follows from the proof of proposition 1 that different types of real subspaces can be classified based on the spectrum of the restricted J. We adjusted the wording accordingly.

6: Section II functions as a prelude and summary of the more technical sections that follow. Readers can start with section II to get an overview and then read sections III and IV on the specific topics they are interested in. Alternatively, a more technically minded reader may skip directly to section III. Based on the referee's feedback, this was not clear in the previous version. We therefore added further remarks at the beginning of sections II and III.

7: We use the word Kähler structure as a general term referring to the three relevant structures (a metric, a symplectic form, a complex structure), which form the triangle of Kähler structures. We are aware that there exist other conventions, where one uses singular ("a manifold having a Kähler structure if it is equipped with a metric, symplectic form and complex structure"), but we decided to use plural for the "triangle of Kähler structures".

14: We agree with the referee that the covariant derivative of a vector field requires a connection. However, once such a connection is chosen we can unambiguously define $K^\mu_\nu=\nabla_\nu \mathcal{X}^\mu$, whose spectrum will be basis-independent, but will depend on the chosen connection. What we point out in the footnote is that at points $\mathcal{X}^\mu=0$, the dependence of $K^\mu_\nu$ on the connection (encoded in $\Gamma^\mu_{\nu\rho}$) drops out, so that the resulting spectrum is even independent of the connection. Of course, the reason for this effect is that $K^\mu_\nu$ is a generator of the diffeomorphism $\Phi$ at one of its fixed points. We followed the referee's suggestion to reformulate the footnote slightly.

15: The statement is that for practical calculations the convergence properties of performing gradient descent with respect to the Fubini-Study metric (equivalent to imaginary time evolution) are better than minimizing the energy expectation value directly with respect to a given parametrization without taking the geometry into account. This is what we called naive gradient descent, by which we mean to use the flat metric with respect to the given coordinates. We do not have a rigorous proof, but anecdotal evidence, i.e., even for simple systems and variational families, one finds slower convergence and may get stuck in local minima. As this was not clear from the manuscript, we clarified our statement accordingly.

16: We changed $\mathcal{E}$ to $\varepsilon$.

Finally, we added an appendix E which gives a more constructive description on how to compute the pseudo-inverse $\Omega$ in the case when $\omega$ is not invertible. We further remark that eq. (8) may be partially ill-defined if $\omega$ is non-invertible, which requires a projection as implemented by defining the pseudo-inverse in the suggested way.

Attachment:

PAPER__Geometry_of_variational_methods__dynamics_of_closed_q_towxm7i.pdf

---

## Round 2 · Referee Report · Anonymous · 2020-9-8

Strengths

1-A thorough exposition of the geometry of variational methods in quantum mechanics with a focus on the difference between Kähler and non-Kähler manifolds of variational wave functions.

2-Simple examples help illustrate basic concepts.

Weaknesses

1-The paper is long, maybe unnecessarily long, and not always concise.

2-It is not always clear what the original contribution of this work are.

Report

I broadly agree with the first report on this paper. The manuscripts provides a link to, and an exposition of, mathematical concepts that may be useful for various applications of variational methods in studies of closed quantum systems. The paper attempts to give a detailed and comprehensive discussion of these methods, in a way accessible to a relatively broad expertise. For these reason I believe the paper could have impact and should be considered for publication in SciPost Physics.

As a consequence of the aim to be comprehensive, maybe, the paper is quite long. I think that in principle the length is not a problem, but for such a long paper to be more likely to have an impact, one would have wanted the writing to be a bit more concise and focussed. For example, there are several sentences and paragraphs that discuss what will be done in the next section or in some other part of the paper, without actually adding anything to the story at that point. This has the effect on the reader (at least this reader) that they loose focus and it sometimes becomes hard to come back to the paper and find the information one is after.

I do not think I will insist on the paper being shortened, but I would encourage the authors to consider attempting to focus the writing a little bit. Since how that would be done is to a large extend matter of style, I don't think I should give specific instructions. What would be useful for a reader that maybe wants to use the result of the paper rather than reading every word, is to have a more detailed guide to the reader for where and how to find the key results. I also find that the summary and discussion could be used more efficiently to help the reader understand the key points of the paper. At the moment the summary reads more like the introduction (especially due to the chose of tense in some parts) and doesn't actually summarise very much. It may be useful to extend this section to ease the use of the paper. Again, I would probably not insist on these changes, as they are to some extend a matter of style, but I feel it would improve the readability of the paper.

Towards the end of the paper there is a discussion of the Generalized Gaussian states, and it is said that one of the main motivation of this work is to understand the nontrivial manifolds that arise from these states. There is also several references to earlier work on these states and it is not always clear what is fully new, what is described in a new way and what was discussed before. It would be useful to clarify this.

In addition to this I give a list of more detailed points for consideration in the Requested changes section.

Requested changes

Some of these points are more remarks than requested changes.

1-There are a few places, such as in the last sentence of paragraph 2 on page 2, which reads "Recently such methods have been used..." and then gives examples, where references are partially or completely missing. In this particular paragraph, for example, reference should be give to these recent works that are mentioned. Another example is start of IIB.3 "A standard approach..." Such a phrase seems to suggest that a reference would be appropriate.

2-First paragraph of section II reads "Readers may skip directly to section III." I suppose that this is an example of a guid to the reader I mentioned in my main report, except that here it is not useful. What readers may skip directly and for what reason should they do that?

3-I find the first four paragraphs of section III to be quite repetitive. Maybe one place to shorten?

4-There are several places where there is an additional article or a missing article. For example in second paragraph of III.A: "...where $|\psi(z)\rangle \in \mathcal{H}$ is a holomorphic in $z^\mu$..."

5-The definition of $\omega_{\mu\nu} $ given just below Eq. (8) and that in footnote 2, do not seem consistent. They seem to differ by an $i$.

6-Below Eq. (20) a $|\bar{v}_\mu\rangle$ is used. I may have missed it, but I don't think the bar notation has been defined.

7-In the paragraph below Example 2, should $\mathcal{P}(\mathcal{P})$ maybe be $\mathcal{P}(\mathcal{H})$?

8-Above Eq. (26) where the authors state that some things can be shown to be equivalent, is this a known result (and therefore maybe needing a reference) or is it something they can show but don't think is needed to expand on in the paper?

9-At the start of III should $\psi$ be $|\psi\rangle$.

10-Not all figures and tables seem to be referenced in the main text (for example Fig. 2). It would make sense to refer to the figures in the correct place. Also, I feel that sometimes the captions could be more detailed, as often not everything that is shown on the figures is defined or explained (for example caption to Fig. 3).

11-Should definition 1 read "...with inverse $\Omega^{\mu\nu}$..."?

12-Is there an superfluous dot at the end of Eq. (41)?

13-Start of IV: "Given a system $\mathcal{H}$" Here the concept of a system is being mixed with Hilbert space, since $\mathcal{H}$ is everywhere else used for that.

14-First sentence in the paragraph after Eq. (87): "In the case of the Lagrangian action principle...takes the following form." Nothing follows this, only the next sentence "A similar derivation..." What form does it take?

15-Last paragraph on page 16. "full Hilbert space, i.e., $\mathcal{M} = \mathcal{P}(\mathcal{H})$." Wasn't $\mathcal{P}(\mathcal{H})$ the projected Hilbert space?

16-Above proposition 5, again a sentence that ends in "take the following form" without any form following.

17-Below Eq. (98), should it be $\check{J}$ that "clearly satisfies $\check{J}^2=-1$?

18-IVB.2 starts with "A common alternative is..." A common alternative to what?

19-Above Eq. (19) the authors use $\mathcal{F}^i$ while elsewhere it is $\mathcal{F}^\mu$.

20-At the top of page 32, should the "associated tangent vector" be $|\delta\Gamma\rangle$ instead of $|\Delta\Gamma\rangle$?

21-In Eq. (265) one of these matrices should probably be $\delta\Gamma_2$.

22-In example 20, should some of the $q$ and $p$ have subscript 2 instead of 1. And also, is there a reason the order in $\xi$ is $q,p,q,p$ instead of $q,q,p,p$ as earlier in the paper.

23-In Eq. (292) would it make sense to show explicitly that the derivative is taken at $x=0$?

24-Proposition 13 says "The restricted Kähler structures are..." The restricted Kähler structures of what? It would be preferable to make the propositions self-contained.

25-In proposition 14, is is needed to write that $\mathcal{G}$ is a Lie group?

26-End of example 23. What balance between the properties of $\mathcal{M}_\phi$ and its dimension needs to be struck and for what purpose?

27-Example 24. "The representation of the representation on $\mathcal{H}_-$"?

28-First sentence in Example 25 seems unnecessary since the whole section has been discussing this.

29-Appendix A.3 starts with "In many areas of physics, ...". Then there is a heading and discussion goes somewhere else. The connection of this first statement to what follows is not clear.

30-In couple of places there are additional articles "we a reference", "we a corresponding" ...

  • validity: top
  • significance: good
  • originality: good
  • clarity: good
  • formatting: good
  • grammar: excellent

Author:  Lucas Hackl  on 2020-09-21  [id 980]

(in reply to Report 1 on 2020-09-08)
Category:
correction

We thank the referee for the careful reading of our manuscript and his/her suggestions for improvement which we were very happy to implement. Please find a revised version of the manuscript attached, where we marked the respective changes in red. Let us add some comments regarding some of these changes. We will also resubmit the revised version to arXiv (without markup) and SciPost.

Let us comment more specifically on the referee’s general comments: - “For example, there are several sentences and paragraphs that discuss what will be done in the next section or in some other part of the paper, without actually adding anything to the story at that point.” We believe that the referee particularly refers to the introductory paragraphs of section II, III and IV, which we rewrote to be more of a guide to the reader than a summary. - “What would be useful for a reader that maybe wants to use the result of the paper rather than reading every word, is to have a more detailed guide to the reader for where and how to find the key results.” “I also find that the summary and discussion could be used more efficiently to help the reader understand the key points of the paper. At the moment the summary reads more like the introduction (especially due to the chose of tense in some parts) and doesn't actually summarise very much.” We already partially addressed this by giving more instructions to the reader in the more compact summary paragraphs to the main sections, but we significantly rewrote the “Summary and Discussion” section to follow the suggestions of the referee. We now use past tense for the summary and highlight what we believe to be the main contributions of our paper. In particular, we try to be more explicit regarding our contributions. We believe that the main overall contribution is to highlight the Kähler property as an important criterion for variational families (which has been only implicitly used in the past by typically restricting to the Kähler case straight-away without often saying so) and the presentation of a systematic geometric framework. - “Towards the end of the paper there is a discussion of the Generalized Gaussian states, and it is said that one of the main motivations of this work is to understand the nontrivial manifolds that arise from these states. There is also several references to earlier work on these states and it is not always clear what is fully new, what is described in a new way and what was discussed before. It would be useful to clarify this.” This is an important point, which we addressed in section V.C. Generalized Gaussian states were introduced as an ansatz for many-body wave functions, but their group theoretic and geometric properties were not fully understood. The present paper defines them in group-theoretic language and highlights the fact that they are non-Kähler manifolds, which makes them a prime example for the application of the non-Kähler methods presented in the paper.

We also addressed the numbered list of items and could resolve most of them directly. Let us briefly comment on the following items: 3. As discussed previously, we rewrote this paragraph to serve as a useful guide to the reader and significantly shortened it to avoid repetition. 8. This is shown in the paper and we now explicitly refer to the place where we do it. 9. The letter \psi without \ket{...} is correct. We use the letter \psi without \ket{...} if we refer to the element in projective Hilbert space, i.e., the equivalence class of states as defined in (41) of section III. While we make sure to stay consistent with this convention, we did not want to be overly pedantic by emphasizing this point too much. 22. We made the numbering of the variables (q1,...,qN,p1,...,pN) consistent throughout the draft. 23. G needs to be a compact Lie group, which we stated in the proposition, so we did not change this. 26. We removed this sentence from the example and moved it instead to the discussion section. There, we emphasize that a good variational family must strike a balance between being large enough to capture the relevant physics, but small enough to decrease computational complexity (compared to the exponentially large full Hilbert space) to be able to do calculations.

Attachment:

PAPER__Geometry_of_variational_methods__dynamics_of_closed_q_wTqiUcK.pdf

---

## Round 2 · Author Response

Resubmission based on referee's suggestions.

---

## Round 2 · List of Changes

Detailed list of changes (including a pdf with changes marked in red) can be found in our reply to the referee.

Resubmission 2004.01015v3 on 22 September 2020

---

## Round 3 · List of Changes

Detailed list of changes (including a pdf with changes marked in red) can be found in our reply to the referee.

---

## Editorial Decision

published